# 🔬🧑‍🔬CELLFORGE: AGENTIC DESIGN OF VIRTUAL CELL MODELS

## ABSTRACT

Virtual cell modeling aims to predict cellular responses to diverse perturbations but faces challenges from biological complexity, multimodal data heterogeneity, and the need for interdisciplinary expertise. We introduce CELLFORGE, a multi-agent *framework* that autonomously designs and synthesizes neural network architectures tailored to specific single-cell datasets and perturbation tasks. Given raw multi-omics data and task descriptions, CELLFORGE discovers candidate architectures through collaborative reasoning among specialized agents, then generates executable implementations. Our core contribution is the *framework itself*: showing that multi-agent collaboration mechanisms—rather than manual human design or single-LLM prompting—can autonomously produce executable, high-quality computational methods. This approach goes beyond conventional hyperparameter tuning by enabling entirely new architectural components such as trajectory-aware encoders and perturbation diffusion modulesto *emerge from agentic deliberation*. We evaluate CELLFORGE on six datasets spanning gene knockouts, drug treatments, and cytokine stimulations across multiple modalities (scRNA-seq, scATAC-seq, CITE-seq). The results demonstrate that the models generated by CELLFORGE are highly competitive with established baselines, while revealing systematic patterns of architectural innovation. CELLFORGE highlights the scientific value of multi-agent frameworks: collaboration among specialized agents enables genuine methodological innovation and executable solutions that single agents or human experts cannot achieve. This represents a paradigm shift toward autonomous scientific *method development* in computational biology. Code is available at https://anonymous.4open.science/r/CellForge-FC93/.

## 1 INTRODUCTION

Scientific discovery in computational biology increasingly demands interdisciplinary expertise spanning machine learning, statistics, and domain knowledge [20, 47, 79, 103]. While recent advances in large language models have enabled AI systems to excel at individual research tasks, from literature analysis [43] to hypothesis generation [88, 116], integrating these capabilities into complete scientific workflows remains challenging [66, 75]. This gap is particularly evident in virtual cell modeling, where researchers must design computational methods that capture complex biological mechanisms across diverse experimental conditions [10, 97].

Virtual cell modeling aims to predict how cells respond to genetic edits, chemical treatments, and environmental perturbations across multiple biological modalities [13, 98]. While foundation models like scGPT [21] and Geneformer [110] have advanced single-cell analysis, they often struggle with dataset-specific perturbation patterns and experimental nuances. Creating effective predictive models typically requires extensive manual effort to integrate domain knowledge, design appropriate architectures, and validate results empirically [74, 85]. Perturbation datasets differ substantially in mechanisms, modalities, and sparsity regimes, creating different learning problems that benefit from dataset-specific inductive biases.

We present CELLFORGE, an agentic *framework* designed to autonomously create computational methods for virtual cell modeling. Its core contribution is showing that multi-agent collaboration mechanisms, including graph-structured discussions with confidence scoring and iterative refinement, enable autonomous scientific method development that outperforms single-agent prompting. Rather

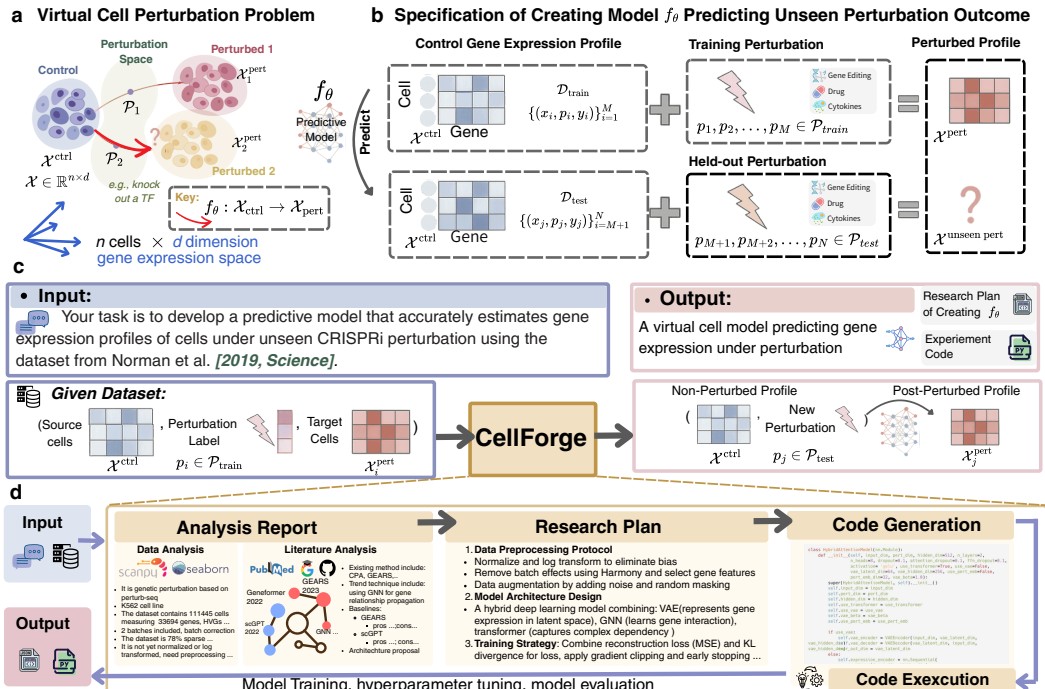

Figure 1: **(a)** Perturbation prediction learns mappings from control cell states to post-perturbation states in high-dimensional expression space. **(b)** Models train on control-perturbed cell pairs across modalities (scRNA-seq, scATAC-seq, CITE-seq) to predict responses to unseen perturbations. **(c)** CELLFORGE receives datasets and task descriptions, autonomously designing models for predicting expression under novel perturbations ($p_i \in \mathcal{P}_{test}$). **(d)** System workflow.

than selecting from predefined pipelines, CELLFORGE generates novel deep learning architectures through emergent collaborative reasoning among specialized agents. Architectures such as trajectory-aware encoders with perturbation diffusion modules serve as evidence of the frameworks creative synthesis, but the primary novelty is the framework itself.

CELLFORGE operates through three modules that address distinct research challenges: Task Analysis agents profile datasets and extract design principles from literature through alternating breadth-first and depth-first retrieval; Design agents engage in graph-structured discussions where specialized experts collaboratively propose, critique, and refine architectures until convergence; and Experiment Execution agents translate research plans into production-ready code with automated debugging and iterative refinement. This structured collaboration enables discovery of optimized models that individual agents cannot achieve, while also revealing systematic patterns in architectural innovation.

We evaluate CELLFORGE on single-cell perturbation prediction across six datasets and benchmark against a comprehensive set of baselines. Our results show that models produced by CELLFORGE are highly competitive and frequently surpass established baselines, though performance varies across runs due to the stochastic nature of automated design. Through systematic analysis, we highlight scenarios where multi-agent collaboration yields advantages, examine the kinds of architectural innovations that emerge from the framework, and delineate the boundaries of current automated design performance. Overall, CELLFORGE positions multi-agent frameworks as a general approach for automating scientific method development in computational biology, moving beyond isolated task execution toward end-to-end autonomous research workflows.

## 2 RELATED WORK

**Single-Cell Perturbation Analysis** Virtual cell modeling,the computational simulation of cellular responses to perturbations, represents a fundamental challenge in systems biology [13, 98]. Existing approaches span diverse paradigms: early methods [22, 104] treat genes independently; deep generative models [41, 72, 74] model perturbations as latent space transformations; network-based methods [7, 90, 96] incorporate gene regulatory knowledge; optimal transport approaches [12, 51] align control and perturbed cell distributions; and transformers [21, 38, 110] leverage large-scale pretraining. This diversity creates a vast design space where architecture selection remains highly context-dependent, typically requiring extensive domain expertise.

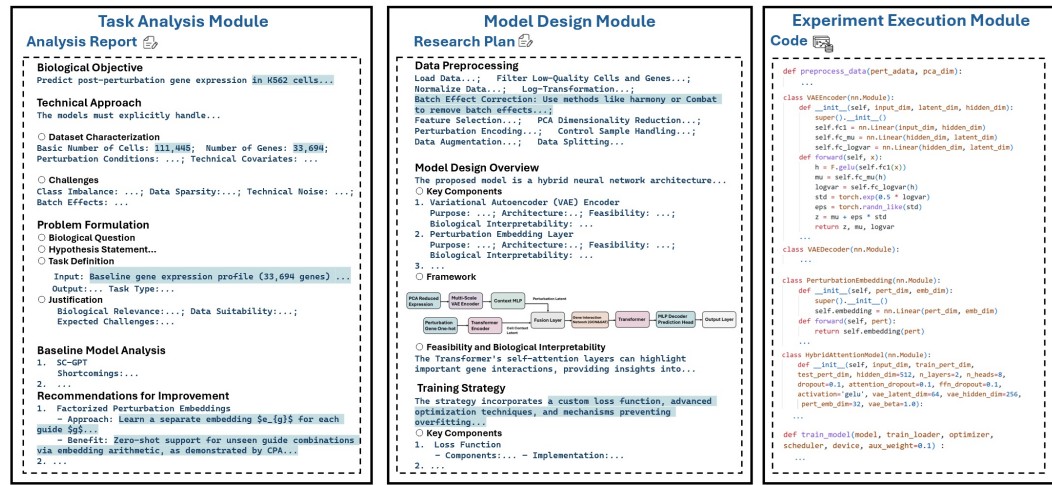

Figure 2: **Multi-agent collaboration generates scientific research artifacts.** Task Analysis produces dataset characterization and literature-grounded insights, Design Module synthesizes novel methodological approaches through structured agent discussions, and Experiment Execution demonstrates code generation capability.

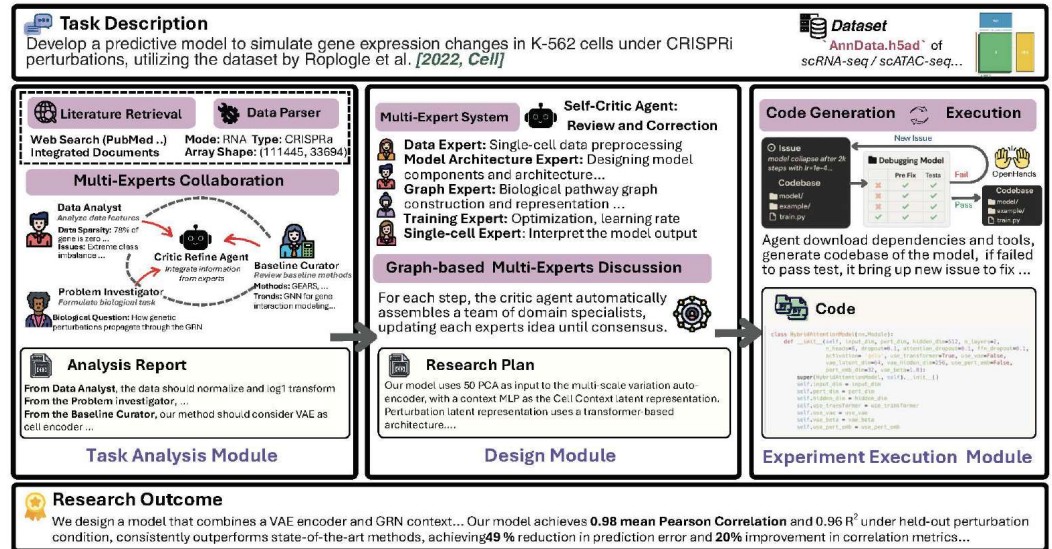

Figure 3: **The CELLFORGE architecture and workflow.**

**AI Agents in Biomedical Research**  Agentic systems are transforming biomedical discovery across the research pipeline [25, 39, 45]. Current approaches fall into three categories: reasoning-focused agents that interpret biological phenomena but lack code generation capabilities; experimental design agents that optimize wet-lab protocols rather than computational models; and workflow automation systems [46, 54, 112] constrained by predefined toolsets. While these advances demonstrate agentic potential, none address autonomous computational model designthe systematic creation of novel architectures tailored to specific biological datasets and research objectives. This gap represents a critical opportunity for agentic frameworks that can autonomously navigate the complex design space of computational biology methods.

## 3  PRELIMINARY AND BACKGROUND

**Notations.**  Let $X \in \mathbb{R}^{n \times d}$ denote the matrix of single-cell profiles, where $n$ is the number of cells and $d$ is the feature dimensionality (typically ∼20,000 genes for scRNA-seq, chromatin peaks for scATAC-seq, or protein markers for CITE-seq). For virtual cell modeling, we are given dataset $\mathcal{D} = \{(x_i, p_i, y_i)\}_{i=1}^{N}$ and task description $S$, where $x_i \in \mathbb{R}^d$ is the pre-perturbation profile, $p_i \in \mathcal{P}$ is the applied perturbation (gene knockout, drug, cytokine), and $y_i \in \mathbb{R}^{d'}$ is the post-perturbation profile. We partition $\mathcal{D}$ into training $\mathcal{D}_{\text{train}}$ and test $\mathcal{D}_{\text{test}}$ sets, where test perturbations $p_i \in \mathcal{P}_{\text{test}} \subset \mathcal{P}$ are held-out during training to evaluate generalization to novel perturbations and cellular contexts.

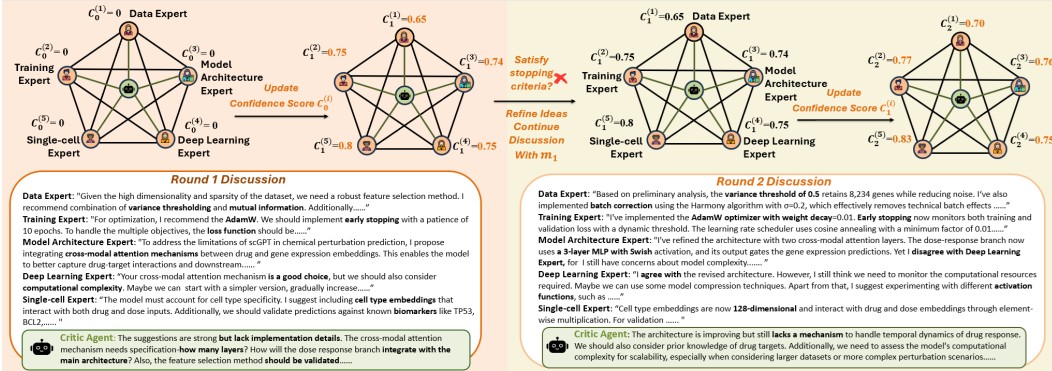

Figure 4: **The Graph-based discussion architecture and workflow.** This is an example of two rounds of discussion from the beginning. After each round, confidence scores are updated, and the agentic system will judge if the current state satisfies the stopping criteria. If not, each expert will refine their ideas based on the critic agent's suggestions and other experts' viewpoints. This graph-based critic refinement continues until reaching the termination state. The figure includes an example formula for computing each experts confidence score per round, based on a weighted combination of historical scores, peer evaluations, and critic agent's assessments. Complete multi-rounds of discussions are presented in Appendix G.2.

**Problem Formulation.** We formalize perturbation prediction as learning mapping function $f_\theta : \mathbb{R}^d \times \mathcal{P} \to \mathbb{R}^{d'}$ that generalizes to unseen perturbations and cell states, where $\theta$ represents trainable parameters. To capture cell-state structure, we incorporate learnable encoders $g_\phi : \mathbb{R}^d \to \mathbb{R}^h$ producing latent embeddings $z_i = g_\phi(x_i)$ that preserve geometric relationships between control and perturbed states. Importantly, CELLFORGE learns perturbation-response mappings *de novo* for each dataset without importing pretrained representations, capturing dataset-specific perturbation signatures and experimental nuances.

**Evaluation.** We assess $f_\theta(x_i, p_i)$ for all $(x_i, p_i) \in \mathcal{X}_{\text{test}} \times \mathcal{P}_{\text{test}}$ using mean squared error, Pearson correlation, and perturbation consistency metrics adapted from [9, 96] to ensure biological significance (detailed in Appendix E).

## 4 METHOD

CELLFORGE profiles each study's perturbations, modalities, and data characteristics to design tailored architectures through knowledge-guided analysis rather than exhaustive search (Figure 3). Task Analysis modules diagnose problems while Design modules synthesize solutions through structured multi-agent collaboration. Agents' output maintains reasoning traceability throughout the discovery process (detailed are provided in Appendix D.3, configurations are detailed in Appendix D.2, communication protocols are outlined in Appendix H, prompts are included in Appendix R, and detailed outputs and logs are available at https://anonymous.4open.science/r/CellForge_egs-1E54/.

### 4.1 TASK ANALYSIS MODULE

The Task Analysis module autonomously characterizes datasets and discovers architectural design principles through three sequential components: **(1) Data Parser.** Extracts experimental metadata across modalities (scRNA-seq, scATAC-seq, CITE-seq), including perturbation types, gene features, and cellular contexts. The component standardizes heterogeneous experimental information and generates foundational statistics without human intervention (Appendix S.1). **(2) Literature Retrieval.** Combines static knowledge (46 single-cell perturbation related articles, listed in Appendix O) with dynamic PubMed searches using alternating breadth-first and depth-first strategies. Starting with query $Q^{(0)}$, the system employs Sentence-BERT embeddings where BFS layers retrieve diverse concepts ($\mathcal{N}_t = \text{TopK}(Q^{(t)})$) and DFS layers follow promising paths. Document relevance scoring via $\text{Score}(Q, d) = \frac{e(Q) \cdot e(d)}{\|e(Q)\|\|e(d)\|}$ guides systematic exploration of architectural design spaces (detailed algorithms in Appendix G.1). **(3) Multi-Experts Collaboration.** Specialized agents (Dataset Analyst, Problem Investigator, Baseline Assessor) combine retrieved insights to propose architectural innovations beyond established paradigms. Instead of recombining known modules, they extract design principles and synthesize new componentssuch as trajectory-aware encoders for temporal dynamics or perturbation diffusion modules for combinatorial interventions. The Baseline Assessor

Figure 5: **Confidence Score Update in Graph-based Expert Discussion.** This figure illustrates an example of how a domain experts confidence score evolves during iterative rounds of discussion in the Graph-based Expert Discussion framework. While this example focuses on the Model Architecture Expert, the same confidence updating process applies to all participating experts in the graph, each iteratively refining their proposals and adjusting their confidence based on multi-agent evaluations.

grounds proposals in theoretical analysis across diverse deep learning paradigms, supporting principled innovation. This process yields dataset-tailored modules that emerge from creative integration of biological insights and computational methods, rather than systematic enumeration.

## 4.2 DESIGN MODULE

The Design module implements scientific creativity through graph-based multi-agent collaboration, generating integrated research plans encompassing preprocessing strategies, architectural designs, and implementation details. The core innovation is the autonomous discovery of optimized architecture rather than the hyperparameter tuning of fixed designs. This architectural discovery processtailoring neural components to dataset-specific biological characteristicsconstitutes the primary source of CELLFORGE's performance advantages in perturbation prediction.

**Multi-Expert Critic System.** We construct a panel of domain experts through role-play prompting: each expert is instantiated from similar dedicated prompt templates that encode its specialty while using the same underlying LLM. See Appendix R.3 for the full templates. For each task, the system dynamically selects a subset of domain experts $E^{(k)}$ (e.g., Data Expert, Single-Cell Expert, Deep Learning Expert) based on task requirements, along with a permanent critic agent $S$. These agents form an undirected collaboration graph $G^{(k)} = (S, E^{(k)})$, where each expert node maintains a confidence score $c_t^{(i)}$ that evolves through discussion rounds, where $t$ is the discussion round and $i$ represents different domain experts.

**Graph-based Discussion.** The framework runs up to $T_{\max} = 10$ rounds of graph-based message passing, where experts propose architectural solutions. In each round $t$ every expert $E^{(i)}$ proposes an architectural candidate $m_t^{(i)}$. After all proposals are submitted, a *critic agent* $S$ reviews every $m_t^{(i)}$, summarizes strengths and weaknesses, and assigns a score.

At the end of round $t$ the value is updated by both the critic agent and peer experts. Specifically, the confidence score $c_t^{(i)}$ for expert $i$ at round $t$ is computed as: $c_t^{(i)} = \lambda_1 \cdot c_{t-1}^{(i)} + \lambda_2 \cdot \text{CriticAgentScore}(m_t^{(i)}, S) + \lambda_3 \cdot \frac{1}{k-1} \sum_{j \neq i} \text{PeerScore}(m_t^{(i)}, E^{(j)})$, where $c_{t-1}^{(i)}$ represents the historical confidence, $\text{CriticAgentScore}(m_t^{(i)}, S)$ evaluates the scientific rigor and feasibility of proposal $m_t^{(i)}$ by the critic agent $S$, $\text{PeerScore}(m_t^{(i)}, E^{(j)})$ captures the evaluation from peer expert $j$, $k$ is the total number of participating experts, and $(\lambda_1, \lambda_2, \lambda_3) = (0.3, 0.4, 0.3)$ are empirically determined weights with $\lambda_1 + \lambda_2 + \lambda_3 = 1$. The discussion ends when all experts' confidence scores exceed the threshold $\tau = 0.8$ with minimal variance ($\max_{i,j} |c_{t^*}^{(i)} - c_{t^*}^{(j)}| < \epsilon$, $\epsilon = 0.03$), where $t^*$ represents the final round when the discussion ends, $i$ and $j$ represent domain experts.

If this condition is not met, otherwise it stops at the round limit $T_{\max}$ to balance computational cost, inference time, and token consumption. Before reaching the ending criteria, experts refine their proposals using historical context and proceed to the next round. This process ensures convergence toward scientifically valid and technically feasible model designs with explicit reasoning chains throughout several rounds of discussion. Further information on expert selection and discussion construction is in Appendix D.4, detailed algorithm and mathematical formulation are presented in Appendix G.2, and hyperparameter configuration is presented in Appendix D.7.

## 4.3 EXPERIMENT EXECUTION MODULE

The Experiment Execution module turns high-level research plans into fully tested, empirically validated results:

Table 1: Post-perturbation gene expression prediction results on datasets where multiple existing baseline models are available. The reported metrics for *CellForge-Models* are shown as mean $\pm$ standard deviation across three automatically designed models. **Ranking markers are determined by best-case bounds:** for metrics where lower is better, we use mean$-$std; for metrics where higher is better, we use mean$+$std. All baseline methods are reproduced on the corresponding dataset under the unseen perturbation setting.

| MODEL | $MSE \downarrow$ | $PCC \uparrow$ | $R^2 \uparrow$ | $MSE_{\mathrm{DE}} \downarrow$ | $PCC_{\mathrm{DE}} \uparrow$ | $R^2_{\mathrm{DE}} \uparrow$ |
|---|---|---|---|---|---|---|
| *Gene Knock Out Perturbation – scRNAseq Dataset (Adamson et al. [2])* | | | | | | |
| Unperturbed | 0.9840 | 0.0001 | -0.0127 | 3.7865 | 0.0012 | -4.2437 |
| Random Forest | 0.3053 | 0.2063 | 0.0504 | 0.5923 | 0.2632 | 0.1653 |
| Linear Regression | 0.5803 | 0.0026 | 0.0435 | 0.6995 | 0.0257 | 0.1074 |
| CPA [73] | 0.0067[3] | 0.9833[3] | **0.9845**[2] | 0.1447[3] | 0.9024 | 0.8896 |
| scGen [72] | 0.0082 | 0.9805 | 0.9611 | 0.1301[2] | 0.8994 | 0.7263 |
| CondOT [11] | 0.0062 | 0.9608 | 0.9740 | 0.1997 | 0.9341[2] | 0.9002[3] |
| Biolord [86] | 0.0044[2] | 0.7799 | 0.9844[3] | 0.1256[2] | 0.9097 | 0.9276[2] |
| scGPT [21] | 0.0100 | 0.9861 | 0.9649 | 0.2562 | 0.9088 | 0.7911 |
| CellForge-Models | 0.0051 $\pm$ 0.0063[1] | **0.9883** $\pm$ 0.0459[1] | 0.9761 $\pm$ 0.0803[1] | 0.2013 $\pm$ 0.0444[1] | **0.9474** $\pm$ 0.0601[1] | 0.8912 $\pm$ 0.0518[1] |
| *Gene Knock Out Perturbation – scRNAseq Dataset (Norman et al. [82])* | | | | | | |
| Unperturbed | 0.9251 | 0.0000 | -0.1738 | 5.1214 | -0.0021 | -4.2047 |
| Random Forest | 0.4059 | 0.1625 | 0.0623 | 0.6817 | 0.1428 | 0.0498 |
| Linear Regression | 0.4989 | 0.0244 | 0.0314 | 0.7331 | 0.0265 | 0.0238 |
| CPA [73] | 0.0051[3] | 0.9779 | 0.9603 | 0.3400 | 0.5754 | 0.4555 |
| scGen [72] | 0.0053 | 0.9221 | 0.9521 | 0.3877 | 0.5605 | 0.3220 |
| CondOT [11] | 0.0420 | 0.9847[2] | 0.9619 | 0.2791[3] | 0.8022 | 0.7470 |
| Biolord [86] | 0.0027[2] | 0.4374 | 0.9830[2] | 0.2450[2] | 0.4646 | 0.8112[2] |
| scGPT [21] | 0.0076 | 0.9823 | 0.9536 | 0.5318 | 0.8630[2] | 0.5652 |
| CellForge-Models | 0.0034 $\pm$ 0.0023[1] | **0.9846** $\pm$ 0.0418[1] | 0.9609 $\pm$ 0.0081[1] | 0.1736 $\pm$ 0.0677[1] | 0.8109 $\pm$ 0.0133[1] | 0.5975 $\pm$ 0.0539[1] |
| *Drug Perturbation – scRNA-seq Dataset (Srivatsan et al. [106])* | | | | | | |
| Unperturbed | 0.8919 | 0.0002 | -2.4282 | 9.3326 | 0.0077 | -6.8585 |
| Random Forest | 0.5289 | 0.0527 | 0.0986 | 0.6138 | 0.0245 | 0.0817 |
| Linear Regression | 0.6703 | 0.0711 | 0.2826 | 0.5625 | 0.0763 | 0.0421 |
| ChemCPA [41] | 0.0847 | 0.7221 | 0.6930 | 0.1035[3] | 0.8053 | 0.7412 |
| scGen [72] | 0.0579 | 0.7871 | 0.7334 | 0.1263 | 0.6575 | 0.5610 |
| CondOT [11] | 0.0499 | 0.8674 | 0.6531 | 0.0933[2] | 0.8341 | 0.4378 |
| Biolord [86] | 0.0011[2] | 0.9658 | 0.9287 | 0.0162[2] | **0.9283**[2] | 0.8236 |
| CellFlow [59] | 0.0003[1] | **0.9906**[1] | **0.9813**[1] | 0.0045[1] | 0.7918 | **0.9794**[1] |
| CellForge-Models | 0.0053 $\pm$ 0.0290[3] | 0.8664 $\pm$ 0.1332[3] | 0.8317 $\pm$ 0.0740[3] | 0.0080 $\pm$ 0.0835[3] | 0.9278 $\pm$ 0.1001[1] | 0.7887 $\pm$ 0.0548[3] |

*(1) Code Generation & Self-Debugging.* The Code Generator converts the selected architecture into production-ready scripts and notebooks with complete dependency management. If a syntax or runtime error occurs, the agent receives the traceback via the OpenHands event stream, analyses the failure, patches the code, and re-executes it, repeating until unit tests pass or a rollback to the last stable state is triggered (see Appendix J for a breakdown of resolved error types).

*(2) Training Orchestration.* An automated scheduler launches training with best-practice safeguards: early stopping, cross-validation, adaptive learning-rate schedules, and checkpointing. When the Validation Agent detects under- or over-fitting, it initiates lightweight hyper-parameter tuning (e.g. adjusting regularisation strength or training epochs) to restore convergence. A brief human semantic check is performed before training to ensure that the generated code corresponds to the correct perturbation-prediction objective.This step does not involve any human design or modification of the architectures, but solely a lightweight semantic check to ensure that the agent-generated code conforms to the intended perturbation-prediction objective. *(3) Validation, Refinement & Output Assurance.* After each training cycle, the Validation Agent scores checkpoints on MSE, PCC, and $R^2$, identifies failure modes, and feeds structured critiques back to the generator. Because the task outputs numerical gene-expression matriceswhich are always well-formedthe focus is on accuracy rather than structural validity.

## 5 MAIN RESULTS

### 5.1 EVALUATION SETUP

We evaluate the models designed and implemented by CELLFORGE in various types of perturbation from scPerturb [85], including gene knockouts, drug treatments, and cytokine stimulation in multiple modali-

Table 2: DEG Recovery Performance Across Benchmark Datasets

| Dataset | DEG Recall | ROC-AUC | PR-AUC |
|---|---|---|---|
| *Gene Knock Out Perturbation – scRNAseq Datasets* | | | |
| Adamson et al. [2] | 0.695 $\pm$ 0.08 | 0.652 $\pm$ 0.06 | 0.285 $\pm$ 0.08 |
| Norman et al. [82] | 0.779 $\pm$ 0.13 | 0.704 $\pm$ 0.05 | 0.375 $\pm$ 0.07 |
| *Drug Perturbation – scRNA-seq Dataset* | | | |
| Srivatsan et al. [106] | 0.689 $\pm$ 0.20 | 0.646 $\pm$ 0.06 | 0.182 $\pm$ 0.02 |

ties (scRNA-seq, scATAC-seq, CITE-seq). Each dataset represents distinct biological challenges: The Adamson [2] and Norman [82] datasets capture CRISPR gene knockouts in different cell lines, providing fundamental test cases for genetic perturbation. The Srivatsan [106] dataset assesses the prediction of cellular responses to chemical compounds. To rigorously test generalization to **unseen perturbations**, we held out entire perturbation types during training, ensuring that test perturbations ($P_{\mathrm{test}}$) were never observed. We then systematically surveyed prior work and reproduced all baselines applicable under this setting.

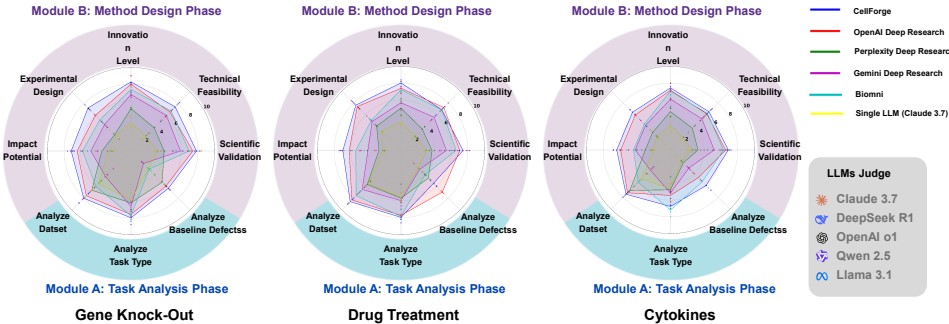

Figure 6: We manually prompt four different DeepResearch variants, Biomni and Single LLM (Claude 3.7) to generate research plans, which were then evaluated by five independent LLMs across eight dimensions, with scores ranging from 1 to 10. Detailed prompts, outputs, and scores are provided in Appendix P.

For other modalities where prior perturbation-response methods are scarce or unavailable, including scATAC-seq (Liscovitch et al. [67]) and scCITE-seq (Papalexi et al. [84]), we position our experiments as exploratory applications rather than direct performance comparisons. The results for these datasets, illustrating CELLFORGEs ability to generalize across modalities and design custom architectures, are provided in Appendix B.

## 5.2 PREDICTIVE PERFORMANCE

Table 1 evaluates the prediction accuracy of models designed by CELLFORGE across diverse perturbation datasets. We report results under two complementary perspectives: overall predictive fidelity and biological relevance. **Overall fidelity** is measured via mean squared error (MSE↓), where lower values indicate predictions closer to actual gene expression; Pearson correlation coefficient (PCC↑), which quantifies how well predicted expression patterns correlate with actual patterns; and coefficient of determination ($R^2 \uparrow$), which measures the proportion of expression variance explained by the model. **Biological relevance** is assessed through metrics on differentially expressed genes (DE), i.e., those showing significant expression changes after perturbation and thus biologically most meaningful. For each dataset, we select the top 20 DE genes based on ground truth perturbation responses and compute the same metrics restricted to this subset ($\text{MSE}_{\text{DE}}$, $\text{PCC}_{\text{DE}}$, $R^2_{\text{DE}}$).

Across **gene knockout** datasets, CELLFORGE-designed models are competitive with, and in some metrics surpass, strong baselines such as CPA, CondOT, Biolord, and scGPT. On the Adamson dataset, the best-performing CELLFORGE model achieves near-zero MSE and the highest PCC, rivaling Biolord and CPA. On the Norman dataset, it again ranks among the top methods, though performance varies across runs. For **drug perturbations**, the results are more mixed: CellFlow and Biolord remain the strongest overall, while CELLFORGE ranges from near state-of-the-art to substantially weaker depending on the instantiated architecture. In its best-performing configurations, it attains $\text{PCC}_{\text{DE}}$ close to Biolord and CellFlow; in others, predictive accuracy declines noticeably, reflecting variability introduced by the automated design process.

Taken together, these results show that while CELLFORGE can autonomously generate models that match or exceed hand-designed baselines in some settings, outcomes vary across instantiations. This variability underscores both the promise and the limitations of automated design: CELLFORGE can discover highly competitive models, but careful evaluation across multiple runs remains essential to ensure robustness. As large language models introduce inherent randomness, we provide detailed variability analysis in Appendix J and cross-model comparisons in Appendix K.

## 5.3 BIOLOGICAL VALIDATION

Beyond overall expression-level accuracy, we assess whether CELLFORGE produces biologically meaningful predictions at multiple levels of resolution. At the gene level, we evaluate recovery of differentially expressed genes (DEGs), a critical signal of perturbation response. Following STAMP [30], we measure DEG recall (sensitivity to true DEGs), ROC-AUC (discriminative power), and PR-AUC (precision under class imbalance). As shown in Table 2, CELLFORGE consistently achieves DEG recall above 0.68 with ROC-AUC values above 0.65 across datasets. On the Norman dataset, performance is relatively stronger, reaching 0.779 recall, indicating effective prioritization of biologically meaningful genes despite the imbalance.

At the pathway and cellular-structure level, enrichment analysis using KEGG annotations shows that the models recover perturbation-relevant pathways: NF-B and p53 signaling for genetic perturbations, autophagy and Wnt pathways for cytokine responses, and coordinated RNAprotein pathways in multimodal CITE-seq data. Complementing this, UMAP visualization demonstrates that predicted

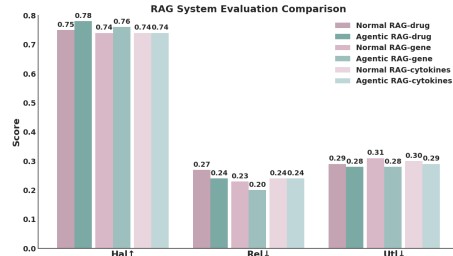

Figure 7: **An example diagram of the model framework designed by CELLFORGE on the scRNAseq gene knockout perturbation prediction task ([82].)**

cellular states preserve the manifold structure across perturbation types (Appendix M.1), indicating that the models not only capture gene-level perturbation effects but also maintain coherent global organization of cellular states.

## 5.4 LLM-AS-A-JUDGE AND HUMAN EVALUATION FOR TASK ANALYSIS MODULE AND DESIGN MODULE

To assess the scientific validity of research plans generated by CELLFORGE, we employ a multi-perspective evaluation framework combining automated LLM assessment with independent human expert review. All evaluations are conducted in a randomized, blinded manner.

Our evaluation protocol employs five independent LLM judges from different model families (Claude 3.7, o1, DeepSeek-R1, Qwen-plus, LLaMA3.1) to minimize model-specific biases. Each judge evaluates research plans across eight scientific dimensions: scientific validity, technical feasibility, experimental design quality, biological relevance, innovation level, impact potential, resource efficiency, and methodological rigor (detailed criteria in Appendix I). This methodology follows established LLM-as-judge practices [36, 64].

Three domain experts with extensive single-cell biology experience conducted independent blinded evaluations using identical criteria, each spending approximately 10 hours on assessment. Figure 6 shows that CELLFORGE consistently outperforms DeepResearch variants across scientific validity, innovation level, and experimental design.

Critically, both LLM-assigned scores and agent-generated confidence scores demonstrate strong correlation with human expert evaluations (Pearson $r = 0.83$, $p < 0.01$, Figure 19). This correlation validates that our evaluation framework captures genuine scientific merit rather than stylistic preferences, as domain experts with years of perturbation biology experience would not correlate highly with LLMs based solely on presentation quality.

To address potential concerns about LLM-as-judge evaluation reliability, we conducted comprehensive inter-judge consistency analysis and style-robustness testing. We computed Krippendorff's and Kendall's W concordance coefficients across all evaluation dimensions. The results demonstrate strong inter-judge agreement with an average Krippendorff's of 0.844 0.056 and average inter-judge correlation of 0.925 0.026, indicating high reliability of our evaluation methodology. Detailed evaluation results are presented in P.5.

## 5.5 NOVEL ARCHITECTURAL DISCOVERIES

CellForge automatically discovers new models that can surpass hand-designed models. An example of a CELL-FORGE-designed model framework is shown in Figure 7. Importantly, these architectures were *not* pre-specified or hard-wired: no rule in the code dictates which modules should be chosen. Instead, they emerge from literature retrieval and multi-agent debate, often yielding hybrid or entirely new designs that move beyond simple recombination of known templates (Appendix L). This demonstrates that the system is capable of autonomously internalizing domain knowledge and translating it into new model designs, rather than merely searching or permuting known templates. Because the design process is conditioned on data characteristics and retrieved knowledge rather than fixed heuristics, the pipeline generalizes naturally to new modalities and perturbation types without requiring manual re-engineering.

Figure 8: The performance of CellForge's RAG compared to standard RAG methods. (1) hal: hallucination detection, (2) rel: context relevance, (3) utl: context utilization. Results are stratified by perturbation type (Drug, Cytokine, Gene). Detailed evaluation methods are stated in Appendix F.

CELLFORGE automatically discovered architectures that move beyond standard hand-crafted baselines such as VAE-MLP stacks or plain Transformer encoders. For instance, on the cytokine perturbation dataset *SchiebingerLander2019*, which contains temporal scRNA-seq profiles, CELL-FORGE produced a model with three key components. The `TrajectoryAwareEncoder` separates shared versus condition-specific latent dimensions while incorporating temporal embeddings, capturing both global developmental trajectories and cytokine-specific effects. The `PerturbationDiffusionModule` introduces perturbation-conditioned latent diffusion dynamics to represent non-linear, combinatorial interactions. Finally, the `GraphRegularizedDecoder` integrates gene-gene co-regulatory constraints to ensure biologically coherent predictions. Detailed presentation of the novel model components is illustrated in Appendix L.

## 5.6 RETRIEVAL EFFECTIVENESS

To test whether CELLFORGE effectively retrieves and integrates literature, we evaluated it on RAG-Bench [27] with the PubMedQA dataset [52]. Metrics include hallucination detection (AUROC↑), context relevance (RMSE↓), and context utilization (RMSE↓). As shown in Figure 8, the largest gain occurs in context utilization for gene-perturbation tasks, with consistent performance across perturbation types. This robustness is driven by two key designs: **graph-based expert discussion**, which fuses reasoning paths, and **retrieval-augmented task analysis**, which grounds design in literature while adapting to dataset statistics.

## 5.7 COMPONENT CONTRIBUTIONS

To disentangle the contributions of individual modules, we performed ablations varying only internal components while holding datasets, tasks, and LLM interface constant. Table 3 shows that adding **Agentic Retrieval** substantially improves performance over the basic version (e.g., Adamson dataset PCC from 0.0087 to 0.5643), while **Graph-Based Discussion** provides complementary gains (to 0.5310). Their combination yields synergistic improvements far exceeding either alone (PCC up to 0.9883), highlighting how knowledge-guided collaborative reasoning drives effective discovery. Similar patterns hold

Table 3: Ablation study on the impact of key framework components on designed models' performance. Detailed settings are listed in D.6.

| MODEL | $MSE\downarrow$ | $PCC\uparrow$ | $R^2\uparrow$ | $MSE_{\mathrm{DE}}\downarrow$ | $PCC_{\mathrm{DE}}\uparrow$ | $R^2_{\mathrm{DE}}\uparrow$ |
|---|---|---|---|---|---|---|
| *Gene Knock Out Perturbation (Adamson Dataset [2])* | | | | | | |
| CELLFORGE (Basic Version without RAG, etc.) | 0.4776 | 0.0087 | 0.0410 | 0.6061 | 0.0940 | 0.1280 |
| Normal RAG | 0.2442 | 0.1008 | 0.1119 | 0.3997 | 0.3354 | 0.3667 |
| Agentic Retrieval | 0.1267 | 0.5643 | 0.5431 | **0.1152** | 0.5922 | 0.6067 |
| Graph-Based Discussion | 0.2751 | 0.5310 | 0.5874 | 0.2792 | 0.6540 | 0.5311 |
| Normal RAG & Graph-Based Discussion | 0.0909 | 0.8951 | 0.8658 | 0.3416 | 0.8547 | 0.6770 |
| Agentic Retrieval & Graph-Based Discussion | **0.0051** | **0.9883** | **0.9761** | 0.2013 | **0.9474** | **0.8912** |
| Agentic Retrieval & Round-Robin Discussion | 0.0123 | 0.9456 | 0.9234 | 0.1567 | 0.9123 | 0.8345 |
| Agentic Retrieval & Moderator-centered Discussion | 0.0156 | 0.9123 | 0.8876 | 0.1789 | 0.8967 | 0.8123 |
| *Drug Perturbation (Srivatsan Dataset [106])* | | | | | | |
| CELLFORGE (Basic Version without RAG, etc.) | 0.5760 | 0.0298 | 0.0475 | 0.6409 | 0.0992 | 0.1039 |
| Normal RAG | 0.2572 | 0.1584 | 0.1038 | 0.3022 | 0.3472 | 0.2901 |
| Agentic Retrieval | 0.1309 | 0.3437 | 0.4350 | 0.1210 | 0.3836 | 0.4169 |
| Graph-Based Discussion | 0.1670 | 0.4193 | 0.3764 | 0.1325 | 0.4266 | 0.3865 |
| Normal RAG & Graph-Based Discussion | 0.0995 | 0.6512 | 0.5933 | 0.0985 | 0.6784 | 0.7548 |
| Agentic Retrieval & Graph-Based Discussion | **0.0053** | **0.9881** | **0.9665** | **0.0080** | **0.9953** | **0.9802** |
| Agentic Retrieval & Round-Robin Discussion | 0.0145 | 0.9567 | 0.9234 | 0.0234 | 0.9456 | 0.9123 |
| Agentic Retrieval & Moderator-centered Discussion | 0.0189 | 0.9234 | 0.8967 | 0.0345 | 0.9123 | 0.8789 |

across drug and cytokine perturbations, underscoring the generality of these mechanisms. Notably, our graph-based discussion protocol consistently outperforms alternative approaches: Round-Robin sequential protocols achieve lower performance (Adamson PCC 0.9456 vs. 0.9883), while Moderator-centered evaluation without peer interaction shows further degradation (PCC 0.9123), demonstrating that collaborative expert reasoning is essential for optimal results.

## 5.8 COSTS AND FAILURE CASES

We also report practical considerations for reproducibility and deployment. Training on two NVIDIA H20 GPUs with a 16-core CPU, 150 GB RAM, and 2 TB SSD typically converges within 38 hours for models of 1030M parameters. Token usage analysis across 50+ experiments shows an average input/output ratio of 60K/300K, with per-request costs averaging $5.18 (details in Appendix I). Code execution succeeds in roughly 80% of runs; most failures arise from tensor operation errors or invalid configurations, with rarer cases due to hallucinated code or data access issues. The agentic pipeline mitigates many of these through iterative error recovery, further improving robustness (Appendix J).

Multi-agent frameworks face criticism for computational overhead relative to single-LLM approaches. We systematically evaluate CELLFORGE against single-LLM baselines across token consumption, execution time, costs, and success rates, provided in Appendix I.4.

## 6 CONCLUSION

CELLFORGE is an autonomous multi-agent system that designs and implements model architectures for single-cell perturbation prediction without human intervention. By combining agentic retrieval with graph-based collaborative reasoning, it integrates computational, biological, and statistical expertise to adaptively improve across datasets and modalities. This work demonstrates that knowledge-grounded agentic frameworks can transcend manual or template-based design, yielding architectures that are both computationally effective and biologically meaningful. More broadly, our results highlight agent-based systems as a paradigm for automating scientific model development, enabling scalable exploration of modeling strategies in complex domains such as single-cell biology. Future directions include extending to new modalities, improving robustness, and generalizing to other areas of computational biology.

### ETHICS STATEMENT

Our work uses only publicly available single-cell perturbation datasets [2, 67, 82, 84, 85, 106] under their respective licenses. No personally identifiable or sensitive data is involved. We emphasize that while CELLFORGE automates model design, any downstream use in biomedical applications should be carefully reviewed for ethical compliance, especially regarding clinical translation and potential misuse.

### REPRODUCIBILITY STATEMENT

We detail datasets, evaluation metrics, and baseline implementations in the main text and Appendix.

### USAGE OF LANGUAGE MODELS

We utilized a large language model (LLM) to aid in the preparation of this manuscript. Its use was limited to editorial tasks, including proofreading for typographical errors, correcting grammar, and improving the clarity and readability of the text.

REFERENCES

[1] Taghrid Abdelaal, Lennart Michielsen, Dries Cats, Daan Hoogduin, Hailiang Mei, Marcel J T Reinders, and Ahmed Mahfouz. A comparison of automatic cell identification methods for single-cell RNA sequencing data. *Genome Biology*, 20(1):1–17, 2019.

[2] Britt Adamson, Thomas M Norman, Marco Jost, Min Y Cho, James K Nuñez, Yuwen Chen, Jacqueline E Villalta, Luke A Gilbert, Max A Horlbeck, Marco Y Hein, et al. A multiplexed single-cell CRISPR screening platform enables systematic dissection of the unfolded protein response. *Cell*, 167(7):1867–1882, 2016.

[3] Abhinav K. Adduri, Dhruv Gautam, Beatrice Bevilacqua, Alishba Imran, Rohan Shah, Mohsen Naghipourfar, Noam Teyssier, Rajesh Ilango, Sanjay Nagaraj, Mingze Dong, Chiara Ricci-Tam, Christopher Carpenter, Vishvak Subramanyam, Aidan Winters, Sravya Tirukkovular, Jeremy Sullivan, Brian S. Plosky, Basak Eraslan, Nicholas D. Youngblut, Jure Leskovec, Luke A. Gilbert, Silvana Konermann, Patrick D. Hsu, Alexander Dobin, Dave P. Burke, Hani Goodarzi, and Yusuf H. Roohani. Predicting cellular responses to perturbation across diverse contexts with state. *bioRxiv*, 2025. doi: 10.1101/2025.06.26.661135. URL https://www.biorxiv.org/content/10.1101/2025.06.26.661135v2.

[4] Constantin Ahlmann-Eltze, Wolfgang Huber, and Simon Anders. Deep learning-based predictions of gene perturbation effects do not yet outperform simple linear baselines. *BioRxiv*, pp. 2024–09, 2024.

[5] Tal Ashuach, Mariano I. Gabitto, Rohan V. Koodli, Giuseppe Antonio Saldi, Michael I. Jordan, and Nir Yosef. Multivi: deep generative model for the integration of multimodal data. *Nature Methods*, 20(8):1222–1231, Aug 2023. doi: 10.1038/s41592-023-01909-9.

[6] Jinheon Baek, Sujay Kumar Jauhar, Silviu Cucerzan, and Sung Ju Hwang. ResearchAgent: Iterative research idea generation over scientific literature with large language models. *arXiv preprint arXiv:2404.07738*, 2024.

[7] Ding Bai, Caleb N Ellington, Shentong Mo, Le Song, and Eric P Xing. AttentionPert: accurately modeling multiplexed genetic perturbations with multi-scale effects. *Bioinformatics*, 40(Supplement_1):i453–i461, 2024.

[8] Joeran Beel, Min-Yen Kan, and Moritz Baumgart. Evaluating sakana's AI scientist for autonomous research: Wishful thinking or an emerging reality towards' artificial research intelligence'(ARI)? *arXiv preprint arXiv:2502.14297*, 2025.

[9] Ihab Bendidi, Shawn Whitfield, Kian Kenyon-Dean, Hanene Ben Yedder, Yassir El Mesbahi, Emmanuel Noutahi, and Alisandra K. Denton. Benchmarking transcriptomics foundation models for perturbation analysis: one PCA still rules them all, 11 2024. URL http://arxiv.org/abs/2410.13956.

[10] Daniil A. Boiko, Robert MacKnight, and Gabe Gomes. Autonomous chemical research with large language models. *Nature*, 623:760–768, 2023. doi: 10.1038/s41586-023-06792-0.

[11] Charlotte Bunne, Andreas Krause, and Marco Cuturi. Supervised training of conditional monge maps. *Advances in Neural Information Processing Systems*, 35:6859–6872, 2022.

[12] Charlotte Bunne, Stefan G Stark, Gabriele Gut, Jacobo Sarabia Del Castillo, Mitch Levesque, Kjong-Van Lehmann, Lucas Pelkmans, Andreas Krause, and Gunnar Rätsch. Learning single-cell perturbation responses using neural optimal transport. *Nature Methods*, 20(11):1759–1768, 2023.

[13] Charlotte Bunne, Yusuf Roohani, Yanay Rosen, Ankit Gupta, Xikun Zhang, Marcel Roed, Theo Alexandrov, Mohammed AlQuraishi, Patricia Brennan, Daniel B Burkhardt, et al. How to build the virtual cell with artificial intelligence: Priorities and opportunities. *Cell*, 187(25): 7045–7063, 2024.

[14] Daniel B. Burkhardt, Jay S. Stanley, Alexander Tong, Ana Luisa Perdigoto, Scott A. Gigante, Kevan C. Herold, Guy Wolf, Antonio J. Giraldez, David van Dijk, and Smita Krishnaswamy. Quantifying the effect of experimental perturbations at single-cell resolution. *Nature Biotechnology*, 39(5):619–629, May 2021. ISSN 1546-1696. doi: 10.1038/s41587-020-00803-5. URL https://www.nature.com/articles/s41587-020-00803-5.

[15] Zhi-Jie Cao and Ge Gao. Multi-omics single-cell data integration and regulatory inference with graph-linked embedding. *Nature Biotechnology*, 40(10):1458–1466, 2022.

[16] ZhiJie Cao and Ge Gao. Multiomics singlecell data integration and regulatory inference with graphlinked embedding. *Nature Biotechnology*, 40(10):1458–1466, May 2022. doi: 10.1038/s41587-022-01284-4.

[17] Bo Chen, Xingyi Cheng, Pan Li, Yangli-ao Geng, Jing Gong, Shen Li, Zhilei Bei, Xu Tan, Boyan Wang, Xin Zeng, et al. xtrimopglm: unified 100b-scale pre-trained transformer for deciphering the language of protein. *arXiv preprint arXiv:2401.06199*, 2024.

[18] Tingting Chen, Srinivas Anumasa, Beibei Lin, Vedant Shah, Anirudh Goyal, and Dianbo Liu. Auto-Bench: An automated benchmark for scientific discovery in LLMs. *arXiv preprint arXiv:2502.15224*, 2025.

[19] Yiqun Chen and James Zou. Simple and effective embedding model for single-cell biology built from ChatGPT. *Nature Biomedical Engineering*, 9(4):483–493, April 2025. ISSN 2157-846X. doi: 10.1038/s41551-024-01284-6. URL https://www.nature.com/articles/s41551-024-01284-6.

[20] Ziru Chen, Shijie Chen, Yuting Ning, Qianheng Zhang, Boshi Wang, Botao Yu, Yifei Li, Zeyi Liao, Chen Wei, Zitong Lu, et al. ScienceAgentBench: Toward rigorous assessment of language agents for data-driven scientific discovery. *arXiv preprint arXiv:2410.05080*, 2024.

[21] Haotian Cui, Chloe Wang, Hassaan Maan, Kuan Pang, Fengning Luo, Nan Duan, and Bo Wang. scGPT: toward building a foundation model for single-cell multi-omics using generative ai. *Nature Methods*, 21(8):1470–1480, 08 2024. ISSN 1548-7091.

[22] Atray Dixit, Oren Parnas, Biyu Li, Jenny Chen, Charles P Fulco, Livnat Jerby-Arnon, Nemanja D Marjanovic, Danielle Dionne, Tyler Burks, Raktima Raychowdhury, et al. Perturb-Seq: dissecting molecular circuits with scalable single-cell RNA profiling of pooled genetic screens. *Cell*, 167(7):1853–1866, 2016.

[23] Mingze Dong, Bao Wang, Jessica Wei, Antonio H de O. Fonseca, Curtis J Perry, Alexander Frey, Feriel Ouerghi, Ellen F Foxman, Jeffrey J Ishizuka, Rahul M Dhodapkar, et al. Causal identification of single-cell experimental perturbation effects with CINEMA-OT. *Nature methods*, 20(11):1769–1779, 2023.

[24] Steffen Eger, Yong Cao, Jennifer D'Souza, Andreas Geiger, Christian Greisinger, Stephanie Gross, Yufang Hou, Brigitte Krenn, Anne Lauscher, Yizhi Li, et al. Transforming science with large language models: A survey on AI-assisted scientific discovery, experimentation, content generation, and evaluation. *arXiv preprint arXiv:2502.05151*, 2025.

[25] Adibvafa Fallahpour, Andrew Magnuson, Purav Gupta, Shihao Ma, Jack Naimer, Arnav Shah, Haonan Duan, Omar Ibrahim, Hani Goodarzi, Chris J Maddison, et al. Bioreason: Incentivizing multimodal biological reasoning within a dna-llm model. *arXiv preprint arXiv:2505.23579*, 2025.

[26] Rochelle V Flores, Shicong Wang, et al. Deep learning tackles single-cell analysis—a survey of deep learning for scRNA-seq analysis. *Briefings in Bioinformatics*, 23(5):bbac327, 2022.

[27] Robert Friel, Masha Belyi, and Atindriyo Sanyal. RAGBench: Explainable benchmark for retrieval-augmented generation systems, 2025. URL http://arxiv.org/abs/2407.11005.

[28] Xi Fu, Shentong Mo, Alejandro Buendia, Anouchka P Laurent, Anqi Shao, Maria del Mar Alvarez-Torres, Tianji Yu, Jimin Tan, Jiayu Su, Romella Sagatelian, et al. A foundation model of transcription across human cell types. *Nature*, 637(8047):965–973, 2025.

[29] Xian Gao, Zongyun Zhang, Mingye Xie, Ting Liu, and Yuzhuo Fu. Graph of AI ideas: Leveraging knowledge graphs and llms for AI research idea generation. *arXiv preprint arXiv:2503.08549*, 2025.

[30] Yicheng Gao, Zhiting Wei, Kejing Dong, Ke Chen, Jingya Yang, Guohui Chuai, and Qi Liu. Toward subtask decomposition-based learning and benchmarking for predicting genetic perturbation outcomes and beyond. *Nature Computational Science*, 4(10):773–785, Sep 2024. doi: 10.1038/s43588-024-00698-1.

[31] Aniketh Garikaparthi, Manasi Patwardhan, Lovekesh Vig, and Arman Cohan. IRIS: Interactive research ideation system for accelerating scientific discovery. *arXiv preprint arXiv:2504.16728*, 2025.

[32] George I. Gavriilidis, Vasileios Vasileiou, Aspasia Orfanou, Naveed Ishaque, and Fotis Psomopoulos. A mini-review on perturbation modelling across single-cell omic modalities. *Computational and Structural Biotechnology Journal*, 23:1886–1896, December 2024. ISSN 2001-0370. doi: 10.1016/j.csbj.2024.04.058.

[33] Adam Gayoso, Zo Steier, Romain Lopez, Jeffrey Regier, Kristopher L. Nazor, Aaron Streets, and Nir Yosef. Joint probabilistic modeling of single-cell multi-omic data with totalvi. *Nature Methods*, 18(3):272–282, Mar 2021. doi: 10.1038/s41592-020-01050-x.

[34] Alireza Ghafarollahi and Markus J Buehler. AtomAgents: Alloy design and discovery through physics-aware multi-modal multi-agent artificial intelligence. *arXiv preprint arXiv:2407.10022*, 2024.

[35] Alireza Ghafarollahi and Markus J Buehler. Sparks: Multi-agent artificial intelligence model discovers protein design principles. *arXiv preprint arXiv:2504.19017*, 2025.

[36] Jiawei Gu, Xuhui Jiang, Zhichao Shi, Hexiang Tan, Xuehao Zhai, Chengjin Xu, Wei Li, Yinghan Shen, Shengjie Ma, Honghao Liu, et al. A survey on llm-as-a-judge. *arXiv preprint arXiv:2411.15594*, 2024.

[37] Taicheng Guo, Xiuying Chen, Yaqi Wang, Ruidi Chang, Shichao Pei, Nitesh V. Chawla, Olaf Wiest, and Xiangliang Zhang. Large language model based multi-agents: A survey of progress and challenges, 02 2024. URL https://arxiv.org/abs/2402.01680.

[38] Minsheng Hao, Jing Gong, Xin Zeng, Chiming Liu, Yucheng Guo, Xingyi Cheng, Taifeng Wang, Jianzhu Ma, Xuegong Zhang, and Le Song. Large-scale foundation model on single-cell transcriptomics. *Nature methods*, 21(8):1481–1491, 2024.

[39] Minsheng Hao, Yongju Lee, Hanchen Wang, Gabriele Scalia, and Aviv Regev. Perturboagent: A self-planning agent for boosting sequential perturb-seq experiments. *bioRxiv*, pp. 2025–05, 2025.

[40] Yuhan Hao, Stephanie Hao, Erica Andersen-Nissen, William M. Mauck, Shiwei Zheng, Andrew Butler, Maddie J. Lee, Aaron J. Wilk, Charlotte Darby, Michael Zager, Paul Hoffman, Marlon Stoeckius, Efthymia Papalexi, Eleni P. Mimitou, Jaison Jain, Avi Srivastava, Tim Stuart, Lamar M. Fleming, Bertrand Yeung, Angela J. Rogers, Juliana M. McElrath, Catherine A. Blish, Raphael Gottardo, Peter Smibert, and Rahul Satija. Integrated analysis of multimodal single-cell data. *Cell*, 184(13):3573–3587.e29, June 2021. ISSN 0092-8674, 1097-4172. doi: 10.1016/j.cell.2021.04.048. URL https://www.cell.com/cell/abstract/S0092-8674(21)00583-3.

[41] Leon Hetzel, Simon Boehm, Niki Kilbertus, Stephan Günnemann, Fabian Theis, et al. Predicting cellular responses to novel drug perturbations at a single-cell resolution. *Advances in Neural Information Processing Systems*, 35:26711–26722, 2022.

[42] Lukas Heumos, Anne C Schaar, et al. Best practices for single-cell analysis across modalities. *Nature Reviews Genetics*, 24(6):395–415, 2023.

[43] Chao-Chun Hsu, Erin Bransom, Jenna Sparks, Bailey Kuehl, Chenhao Tan, David Wadden, Lucy Lu Wang, and Aakanksha Naik. CHIME: LLM-assisted hierarchical organization of scientific studies for literature review support. *Findings of ACL 2024*, 2024.

[44] Xiang Hu, Hongyu Fu, Jinge Wang, Yifeng Wang, Zhikun Li, Renjun Xu, Yu Lu, Yaochu Jin, Lili Pan, and Zhenzhong Lan. Nova: An iterative planning and search approach to enhance novelty and diversity of llm generated ideas. *arXiv preprint arXiv:2410.14255*, 2024.

[45] Kexin Huang, Ying Jin, Ryan Li, Michael Y Li, Emmanuel Candès, and Jure Leskovec. Automated hypothesis validation with agentic sequential falsifications. *arXiv preprint arXiv:2502.09858*, 2025.

[46] Kexin Huang, Serena Zhang, Hanchen Wang, Yuanhao Qu, Yingzhou Lu, Yusuf Roohani, Ryan Li, Lin Qiu, Junze Zhang, Yin Di, et al. Biomni: A general-purpose biomedical ai agent. *bioRxiv*, pp. 2025–05, 2025.

[47] Qian Huang, Jian Vora, Percy Liang, and Jure Leskovec. MLAgentBench: Evaluating language agents on machine learning experimentation. In *ICML 2024*, 2024.

[48] Ana-Maria Istrate, Donghui Li, and Theofanis Karaletsos. scGenePT: Is language all you need for modeling single-cell perturbations?, October 2024. URL https://www.biorxiv.org/content/10.1101/2024.10.23.619972v1.

[49] Yanrong Ji, Zhihan Zhou, Han Liu, and Ramana V. Davuluri. Dnabert: pretrained bidirectional encoder representations from transformers model for dnalanguage in genome. *Bioinformatics*, 37(15):2112–2120, August 2021. doi: 10.1093/bioinformatics/btab083. URL https://doi.org/10.1093/bioinformatics/btab083.

[50] Yuge Ji, Mohammad Lotfollahi, F. Alexander Wolf, and Fabian J. Theis. Machine learning for perturbational single-cell omics. *Cell Systems*, 12(6):522–537, June 2021. ISSN 2405-4712, 2405-4720. doi: 10.1016/j.cels.2021.05.016. URL https://www.cell.com/cell-systems/abstract/S2405-4712(21)00202-7.

[51] Qun Jiang, Shengquan Chen, Xiaoyang Chen, and Rui Jiang. scPRAM accurately predicts single-cell gene expression perturbation response based on attention mechanism. *Bioinformatics*, 40(5):btae265, 2024.

[52] Qiao Jin, Bhuwan Dhingra, Zhengping Liu, William W Cohen, and Xinghua Lu. PubMedQA: A dataset for biomedical research question answering. *arXiv preprint arXiv:1909.06146*, 2019.

[53] Qiao Jin, Yifan Yang, Qingyu Chen, and Zhiyong Lu. GeneGPT: Augmenting large language models with domain tools for improved access to biomedical information. *Bioinformatics*, 40 (2):btae075, February 2024. ISSN 1367-4811. doi: 10.1093/bioinformatics/btae075. URL https://doi.org/10.1093/bioinformatics/btae075.

[54] Ruofan Jin, Zaixi Zhang, Mengdi Wang, and Le Cong. Stella: Self-evolving llm agent for biomedical research. *arXiv preprint arXiv:2507.02004*, 2025.

[55] Liqiang Jing, Zhehui Huang, Xiaoyang Wang, Wenlin Yao, Wenhao Yu, Kaixin Ma, Hongming Zhang, Xinya Du, and Dong Yu. DSBench: How far are data science agents to becoming data science experts? *arXiv preprint arXiv:2409.07703*, 2024.

[56] Julia Joung, Silvana Konermann, Jonathan S. Gootenberg, Omar O. Abudayyeh, Randall J. Platt, Mark D. Brigham, Neville E. Sanjana, and Feng Zhang. Genome-scale CRISPR-Cas9 knockout and transcriptional activation screening. *Nature Protocols*, 12(4):828–863, April 2017. ISSN 1750-2799. doi: 10.1038/nprot.2017.016. URL https://www.nature.com/articles/nprot.2017.016.

[57] Kenji Kamimoto, Blerta Stringa, Christy M Hoffmann, Kunal Jindal, Lilianna Solnica-Krezel, and Samantha A Morris. Dissecting cell identity via network inference and in silico gene perturbation. *Nature*, 614(7949):742–751, 2023.

[58] Kasia Z. Kedzierska, Lorin Crawford, Ava P. Amini, and Alex X. Lu. Zero-shot evaluation reveals limitations of single-cell foundation models. *Genome Biology*, 26(1):101, April 2025. ISSN 1474-760X. doi: 10.1186/s13059-025-03574-x. URL https://doi.org/10.1186/s13059-025-03574-x.

[59] Dominik Klein, Jonas Simon Fleck, Daniil Bobrovskiy, Lea Zimmermann, Sören Becker, Alessandro Palma, Leander Dony, Alejandro Tejada-Lapuerta, Guillaume Huguet, Hsiu-Chuan Lin, et al. Cellflow enables generative single-cell phenotype modeling with flow matching. *bioRxiv*, pp. 2025–04, 2025.

[60] Patrick Tser Jern Kon, Jiachen Liu, Qiuyi Ding, Yiming Qiu, Zhenning Yang, Yibo Huang, Jayanth Srinivasa, Myungjin Lee, Mosharaf Chowdhury, and Ang Chen. Curie: Toward rigorous and automated scientific experimentation with AI agents. *arXiv preprint arXiv:2502.16069*, 2025.

[61] Adithya Kulkarni, Fatimah Alotaibi, Xinyue Zeng, Longfeng Wu, Tong Zeng, Barry Menglong Yao, Minqian Liu, Shuaicheng Zhang, Lifu Huang, and Dawei Zhou. Scientific hypothesis generation and validation: Methods, datasets, and future directions. *arXiv preprint arXiv:2505.04651*, 2025.

[62] Daniel Levine, Syed Asad Rizvi, Sacha Lévy, Nazreen Pallikkavaliyaveetil, David Zhang, Xingyu Chen, Sina Ghadermarzi, Ruiming Wu, Zihe Zheng, Ivan Vrkic, et al. Cell2Sentence: teaching large language models the language of biology. *BioRxiv*, pp. 2023–09, 2024.

[63] Chen Li, Haoxiang Gao, Yuli She, Haiyang Bian, Qing Chen, Kai Liu, Lei Wei, and Xuegong Zhang. Benchmarking ai models for in silico gene perturbation of cells. *bioRxiv*, pp. 2024–12, 2024.

[64] Haitao Li, Qian Dong, Junjie Chen, Huixue Su, Yujia Zhou, Qingyao Ai, Ziyi Ye, and Yiqun Liu. Llms-as-judges: a comprehensive survey on llm-based evaluation methods. *arXiv preprint arXiv:2412.05579*, 2024.

[65] Lanxiang Li, Yue You, Wenyu Liao, Xueying Fan, Shihong Lu, Ye Cao, Bo Li, Wenle Ren, Yunlin Fu, Jiaming Kong, et al. A systematic comparison of single-cell perturbation response prediction models. *bioRxiv*, pp. 2024–12, 2024.

[66] Ruochen Li, Teerth Patel, Qingyun Wang, and Xinya Du. MLR-Copilot: Autonomous machine learning research based on large language models agents. *arXiv preprint arXiv:2408.14033*, 2024.

[67] Noa Liscovitch-Brauer, Antonino Montalbano, Jiale Deng, Alejandro Méndez-Mancilla, Hans-Hermann Wessels, Nicholas G Moss, Chia-Yu Kung, Akash Sookdeo, Xinyi Guo, Evan Geller, et al. Profiling the genetic determinants of chromatin accessibility with scalable single-cell crispr screens. *Nature biotechnology*, 39(10):1270–1277, 2021.

[68] Haokun Liu, Yangqiaoyu Zhou, Mingxuan Li, Chenfei Yuan, and Chenhao Tan. Literature meets data: A synergistic approach to hypothesis generation. *arXiv preprint arXiv:2410.17309*, 2024.

[69] Tianyu Liu, Yuge Wang, Rex Ying, and Hongyu Zhao. MuSe-GNN: Learning Unified Gene Representation From Multimodal Biological Graph Data, September 2023. URL http://arxiv.org/abs/2310.02275.

[70] Zijun Liu, Kaiming Liu, Yiqi Zhu, Xuanyu Lei, Zonghan Yang, Zhenhe Zhang, Peng Li, and Yang Liu. AIGS: Generating science from AI-powered automated falsification. *arXiv preprint arXiv:2411.11910*, 2024.

[71] Romain Lopez, Jordan Regier, Michael B Cole, Michael I Jordan, and Nir Yosef. scVI: deep generative modeling for single-cell transcriptomics. *Nature Methods*, 15(12):1053–1058, 2018.

[72] Mohammad Lotfollahi, F Alexander Wolf, and Fabian J Theis. scGen predicts single-cell perturbation responses. *Nature methods*, 16(8):715–721, 2019.

[73] Mohammad Lotfollahi, Anna Klimovskaia Susmelj, Carlo De Donno, Yuge Ji, Ignacio L. Ibarra, et al. Learning interpretable cellular responses to complex perturbations in high-throughput screens. *Bioinformatics*, 04 2021.

[74] Mohammad Lotfollahi, Anna Klimovskaia Susmelj, Carlo De Donno, Leon Hetzel, Yuge Ji, Ignacio L Ibarra, Sanjay R Srivatsan, Mohsen Naghipourfar, Riza M Daza, Beth Martin, et al. Predicting cellular responses to complex perturbations in high-throughput screens. *Molecular systems biology*, 19(6):e11517, 2023.

[75] Chris Lu, Cong Lu, Robert Tjarko Lange, Jakob Foerster, Jeff Clune, and David Ha. The AI scientist: Towards fully automated open-ended scientific discovery, 09 2024. URL http://arxiv.org/abs/2408.06292.

[76] Malte D Luecken, Daniel B Burkhardt, Fabian J Theis, et al. Defining and benchmarking open problems in single-cell analysis. *Nature Methods*, 19(4):412–420, 2022.

[77] Malte D. Luecken, M. Büttner, K. Chaichoompu, A. Danese, M. Interlandi, M. F. Mueller, D. C. Strobl, L. Zappia, M. Dugas, M. Colomé-Tatché, and Fabian J. Theis. Benchmarking atlas-level data integration in single-cell genomics. *Nature Methods*, 19(1):41–50, January 2022. ISSN 1548-7091, 1548-7105. doi: 10.1038/s41592-021-01336-8.

[78] Hiba Maan, Matti Lähde, et al. Characterizing the impacts of dataset imbalance on single-cell data integration. *Nature Biotechnology*, 42(1):56–60, 2024.

[79] Bodhisattwa Prasad Majumder, Harshit Surana, Dhruv Agarwal, Bhavana Dalvi Mishra, Abhijeetsingh Meena, Aryan Prakhar, Tirth Vora, Tushar Khot, Ashish Sabharwal, and Peter Clark. DiscoveryBench: Towards data-driven discovery with large language models. *arXiv preprint arXiv:2407.01725*, 2024.

[80] Pablo Monfort-Lanzas, Katja Rungger, Leonie Madersbacher, and Hubert Hackl. Machine learning to dissect perturbations in complex cellular systems. *Computational and Structural Biotechnology Journal*, 27:832–842, January 2025. ISSN 2001-0370. doi: 10.1016/j.csbj.2025.02.028. URL https://www.sciencedirect.com/science/article/pii/S2001037025000583.

[81] Vladimir Naumov, Diana Zagirova, Sha Lin, Yupeng Xie, Wenhao Gou, Anatoly Urban, Nina Tikhonova, Khadija Alawi, Mike Durymanov, Fedor Galkin, et al. DORA AI scientist: Multi-agent virtual research team for scientific exploration discovery and automated report generation. *bioRxiv*, 2025.

[82] Thomas M Norman, Max A Horlbeck, Joseph M Replogle, Alex Y Ge, Albert Xu, Marco Jost, Luke A Gilbert, and Jonathan S Weissman. Exploring genetic interaction manifolds constructed from rich single-cell phenotypes. *Science*, 365(6455):786–793, 2019.

[83] OpenAI. Introducing deep research. https://openai.com/index/deep-research/, 2025. Accessed: 2025-05-08.

[84] Efthymia Papalexi, Eleni P Mimitou, Andrew W Butler, Samantha Foster, Bernadette Bracken, William M Mauck III, Hans-Hermann Wessels, Yuhan Hao, Bertrand Z Yeung, Peter Smibert, et al. Characterizing the molecular regulation of inhibitory immune checkpoints with multimodal single-cell screens. *Nature genetics*, 53(3):322–331, 2021.

[85] Stefan Peidli, Tessa D Green, Ciyue Shen, Torsten Gross, Joseph Min, Samuele Garda, Bo Yuan, Linus J Schumacher, Jake P Taylor-King, Debora S Marks, et al. scPerturb: harmonized single-cell perturbation data. *Nature Methods*, 21(3):531–540, 2024.

[86] Zoe Piran, Niv Cohen, Yedid Hoshen, and Mor Nitzan. Disentanglement of single-cell data with biolord. *Nature Biotechnology*, 42(11):1678–1683, 2024.

[87] Kevin Pu, KJ Feng, Tovi Grossman, Tom Hope, Bhavana Dalvi Mishra, Matt Latzke, Jonathan Bragg, Joseph Chee Chang, and Pao Siangliulue. IdeaSynth: Iterative research idea development through evolving and composing idea facets with literature-grounded feedback. *arXiv preprint arXiv:2410.04025*, 2024.

[88] Biqing Qi, Kaiyan Zhang, Haoxiang Li, Kai Tian, Sihang Zeng, Zhang-Ren Chen, Jin-Fang Hu, and Bowen Zhou. Large language models are zero shot hypothesis proposers. *Instruction Workshop @ NeurIPS 2023*, 2023.

[89] Xiaoning Qi, Lianhe Zhao, Chenyu Tian, Yueyue Li, Zhen-Lin Chen, Peipei Huo, Runsheng Chen, Xiaodong Liu, Baoping Wan, Shengyong Yang, and Yi Zhao. Predicting transcriptional responses to novel chemical perturbations using deep generative model for drug discovery. *Nature Communications*, 15(1):9256, October 2024. ISSN 2041-1723. doi: 10.1038/s41467-024-53457-1. URL https://www.nature.com/articles/s41467-024-53457-1.

[90] Xiaojie Qiu, Yan Zhang, Jorge D Martin-Rufino, Chen Weng, Shayan Hosseinzadeh, Dian Yang, Angela N Pogson, Marco Y Hein, Kyung Hoi Joseph Min, Li Wang, et al. Mapping transcriptomic vector fields of single cells. *Cell*, 185(4):690–711, 2022.

[91] Yansheng Qiu, Haoquan Zhang, Zhaopan Xu, Ming Li, Diping Song, Zheng Wang, and Kaipeng Zhang. AI Idea Bench 2025: AI research idea generation benchmark. *arXiv preprint arXiv:2504.14191*, 2025.

[92] Marissa Radensky, Simra Shahid, Raymond Fok, Pao Siangliulue, Tom Hope, and Daniel S Weld. Scideator: Human-llm scientific idea generation grounded in research-paper facet recombination. *arXiv preprint arXiv:2409.14634*, 2024.

[93] Chandan K Reddy and Parshin Shojaee. Towards scientific discovery with generative AI: Progress, opportunities, and challenges. In *Proceedings of the AAAI Conference on Artificial Intelligence*, volume 39, pp. 28601–28609, 2025.

[94] Shuo Ren, Pu Jian, Zhenjiang Ren, Chunlin Leng, Can Xie, and Jiajun Zhang. Towards scientific intelligence: A survey of llm-based scientific agents. *arXiv preprint arXiv:2503.24047*, 2025.

[95] Joseph M. Replogle, Thomas M. Norman, Albert Xu, Jeffrey A. Hussmann, Jin Chen, J. Zachery Cogan, Elliott J. Meer, Jessica M. Terry, Daniel P. Riordan, Niranjan Srinivas, Ian T. Fiddes, Joseph G. Arthur, Luigi J. Alvarado, Katherine A. Pfeiffer, Tarjei S. Mikkelsen, Jonathan S. Weissman, and Britt Adamson. Combinatorial single-cell CRISPR screens by direct guide RNA capture and targeted sequencing. *Nature Biotechnology*, 38 (8):954–961, August 2020. ISSN 1546-1696. doi: 10.1038/s41587-020-0470-y. URL https://www.nature.com/articles/s41587-020-0470-y.

[96] Yusuf Roohani, Kexin Huang, and Jure Leskovec. Predicting transcriptional outcomes of novel multigene perturbations with GEARS. *Nature Biotechnology*, 42(6):927–935, 2024.

[97] Yusuf H Roohani, Jian Vora, Qian Huang, Percy Liang, and Jure Leskovec. BioDiscoveryAgent: An ai agent for designing genetic perturbation experiments. In *ICLR 2024 Workshop on Machine Learning for Genomics Explorations*, 2024.

[98] Yusuf H. Roohani, Tony J. Hua, Po-Yuan Tung, Lexi R. Bounds, Feiqiao B. Yu, Alexander Dobin, Noam Teyssier, Abhinav Adduri, Alden Woodrow, Brian S. Plosky, Reshma Mehta, Benjamin Hsu, Jeremy Sullivan, Chiara Ricci-Tam, Nianzhen Li, Julia Kazaks, Luke A. Gilbert, Silvana Konermann, Patrick D. Hsu, Hani Goodarzi, and Dave P. Burke. Virtual cell challenge: Toward a turing test for the virtual cell. *Cell*, 188(13):3370–3374, 2025.

[99] Kai Ruan, Xuan Wang, Jixiang Hong, Peng Wang, Yang Liu, and Hao Sun. LiveIdeaBench: Evaluating llms' scientific creativity and idea generation with minimal context. *arXiv preprint arXiv:2412.17596*, 2024.

[100] Geoffrey Schiebinger, Jian Shu, Marcin Tabaka, Brian Cleary, Vidya Subramanian, Aryeh Solomon, Joshua Gould, Siyan Liu, Stacie Lin, Peter Berube, et al. Optimal-transport analysis of single-cell gene expression identifies developmental trajectories in reprogramming. *Cell*, 176(4):928–943, 2019.

[101] Samuel Schmidgall and Michael Moor. AgentRxiv: Towards collaborative autonomous research. *arXiv preprint arXiv:2503.18102*, 2025.

[102] Chenglei Si, Diyi Yang, and Tatsunori Hashimoto. Can LLMs generate novel research ideas? a large-scale human study with 100+ NLP researchers. *arXiv preprint arXiv:2409.04109*, 2024.

[103] Michael D. Skarlinski, Sam Cox, Jon M. Laurent, James D. Braza, Michaela Hinks, Michael J. Hammerling, Manvitha Ponnapati, Samuel G. Rodriques, and Andrew D. White. Language agents achieve superhuman synthesis of scientific knowledge. *arXiv preprint arXiv:2409.13740*, 2024.

[104] Michael A Skinnider, Jordan W Squair, Claudia Kathe, Mark A Anderson, Matthieu Gautier, Kaya JE Matson, Marco Milano, Thomas H Hutson, Quentin Barraud, Aaron A Phillips, et al. Cell type prioritization in single-cell data. *Nature biotechnology*, 39(1):30–34, 2021.

[105] Bicna Song, Dingyu Liu, Weiwei Dai, Natalie F. McMyn, Qingyang Wang, Dapeng Yang, Adam Krejci, Anatoly Vasilyev, Nicole Untermoser, Anke Loregger, Dongyuan Song, Breanna Williams, Bess Rosen, Xiaolong Cheng, Lumen Chao, Hanuman T. Kale, Hao Zhang, Yarui Diao, Tilmann Bürckstümmer, Janet D. Siliciano, Jingyi Jessica Li, Robert F. Siliciano, Danwei Huangfu, and Wei Li. Decoding heterogeneous single-cell perturbation responses. *Nature Cell Biology*, 27(3):493–504, March 2025. ISSN 1476-4679. doi: 10.1038/s41556-025-01626-9. URL https://www.nature.com/articles/s41556-025-01626-9.

[106] Sanjay R Srivatsan, José L McFaline-Figueroa, Vijay Ramani, Lauren Saunders, Junyue Cao, Jonathan Packer, Hannah A Pliner, Dana L Jackson, Riza M Daza, Lena Christiansen, et al. Massively multiplex chemical transcriptomics at single-cell resolution. *Science*, 367(6473): 45–51, 2020.

[107] Tim Stuart, Andrew Butler, Paul Hoffman, Christoph Hafemeister, Efthymia Papalexi, William M. III Mauck, Yuhan Hao, Marlon Stoeckius, Peter Smibert, and Rahul Satija. Comprehensive integration of single-cell data. *Cell*, 177(7):1888–1902.e21, June 2019. doi: 10.1016/j.cell.2019.05.031. URL https://doi.org/10.1016/j.cell.2019.05.031.

[108] Quan Tang, Na Le, et al. Single-cell multimodal prediction via transformers. In *NeurIPS 2022 Workshop on Learning from Time Series for Health*, 2022.

[109] Xiangru Tang, Anni Zou, Zhuosheng Zhang, Ziming Li, Yilun Zhao, Xingyao Zhang, Arman Cohan, and Mark Gerstein. MedAgents: Large language models as collaborators for zero-shot medical reasoning. *Findings of ACL 2024*, 2024.

[110] Christina V Theodoris, Ling Xiao, Anant Chopra, Mark D Chaffin, Zeina R Al Sayed, Matthew C Hill, Helene Mantineo, Elizabeth M Brydon, Zexian Zeng, X Shirley Liu, et al. Transfer learning enables predictions in network biology. *Nature*, 618(7965):616–624, 2023.

[111] Minyang Tian, Luyu Gao, Shizhuo Zhang, Xinan Chen, Cunwei Fan, Xuefei Guo, Roland Haas, Pan Ji, Kittithat Krongchon, Yao Li, et al. SciCode: A research coding benchmark curated by scientists. *Advances in Neural Information Processing Systems*, 37:30624–30650, 2024.

[112] Hanchen Wang, Yichun He, Paula P Coelho, Matthew Bucci, Abbas Nazir, Bob Chen, Linh Trinh, Serena Zhang, Kexin Huang, Vineethkrishna Chandrasekar, et al. Spatialagent: An autonomous ai agent for spatial biology. *bioRxiv*, pp. 2025–04, 2025.

[113] Juexin Wang, Anjun Ma, Yuzhou Chang, Jianting Gong, Yuexu Jiang, Ren Qi, Cankun Wang, Hongjun Fu, Qin Ma, and Dong Xu. scGNN is a novel graph neural network framework for single-cell RNA-Seq analyses. *Nature Communications*, 12(1):1882, March 2021. ISSN 2041-1723. doi: 10.1038/s41467-021-22197-x. URL https://www.nature.com/articles/s41467-021-22197-x.

[114] F. Alexander Wolf, Philipp Angerer, and Fabian J. Theis. SCANPY: Large-scale single-cell gene expression data analysis. *Genome Biology*, 19(1):1–5, December 2018. ISSN 1474-760X. doi: 10.1186/s13059-017-1382-0. URL https://genomebiology.biomedcentral.com/articles/10.1186/s13059-017-1382-0.

[115] Fan Yang, Fang Wang, Longkai Huang, Linjing Liu, Junzhou Huang, and Jianhua Yao. Reply to: Deeper evaluation of a single-cell foundation model. *Nature Machine Intelligence*, 6(12): 1447–1450, December 2024. ISSN 2522-5839. doi: 10.1038/s42256-024-00948-x. URL https://www.nature.com/articles/s42256-024-00948-x.

[116] Tong Yang, Xiaodan Hu, Xiaohan Li, Mingda Tan, Jingfeng Zhang, Zhilin Wen, Ernie Chang, Andrew M. Dai, Quoc V. Li, Joseph E. Gonzalez, Claire Cardie, and Jason Wei. AnyBench: Language models evaluate anything. *arXiv preprint arXiv:2312.13771*, 2023.

[117] Xiaodong Yang, Guole Liu, Guihai Feng, Dechao Bu, Pengfei Wang, Jie Jiang, Shubai Chen, Qinmeng Yang, Hefan Miao, Yiyang Zhang, Zhenpeng Man, Zhongming Liang, Zichen Wang, Yaning Li, Zheng Li, Yana Liu, Yao Tian, Wenhao Liu, Cong Li, Ao Li, Jingxi Dong, Zhilong Hu, Chen Fang, Lina Cui, Zixu Deng, Haiping Jiang, Wentao Cui, Jiahao Zhang, Zhaohui Yang, Handong Li, Xingjian He, Liqun Zhong, Jiaheng Zhou, Zijian Wang, Qingqing Long, Ping Xu, Xin Li, Hongmei Wang, Zhen Meng, Xuezhi Wang, Yangang Wang, Yong Wang, Shihua Zhang, Jingtao Guo, Yi Zhao, Yuanchun Zhou, Fei Li, Jing Liu, Yiqiang Chen, Ge Yang, and Xin Li. GeneCompass: Deciphering universal gene regulatory mechanisms with a knowledge-informed cross-species foundation model. *Cell Research*, 34 (12):830–845, December 2024. ISSN 1748-7838. doi: 10.1038/s41422-024-01034-y. URL https://www.nature.com/articles/s41422-024-01034-y.

[118] Nicholas D. Youngblut, Christopher Carpenter, Jaanak Prashar, Chiara Ricci-Tam, Rajesh Ilango, Noam Teyssier, Silvana Konermann, Patrick D. Hsu, Alexander Dobin, David P. Burke, Hani Goodarzi, and Yusuf H. Roohani. scBaseCamp: An AI agent-curated, uniformly processed, and continually expanding single cell data repository, March 2025. URL https://www.biorxiv.org/content/10.1101/2025.02.27.640494v1.

[119] Hengshi Yu, Weizhou Qian, Yuxuan Song, and Joshua D Welch. PerturbNet predicts single-cell responses to unseen chemical and genetic perturbations. *Molecular Systems Biology*, 21(8):960–982, August 2025. ISSN 1744-4292. doi: 10.1038/s44320-025-00131-3. URL https://www.embopress.org/doi/full/10.1038/s44320-025-00131-3.

[120] Bo Yuan, Ciyue Shen, Augustin Luna, Anil Korkut, Debora S. Marks, John Ingraham, and Chris Sander. CellBox: Interpretable Machine Learning for Perturbation Biology with Application to the Design of Cancer Combination Therapy. *Cell Systems*, 12(2):128–140.e4, February 2021. ISSN 2405-4712, 2405-4720. doi: 10.1016/j.cels.2020.11.013. URL https://www.cell.com/cell-systems/abstract/S2405-4712(20)30464-6.

[121] Ruiqi Zhong, Peter Zhang, Steve Li, Jinwoo Ahn, Dan Klein, and Jacob Steinhardt. Goal driven discovery of distributional differences via language descriptions. In *NeurIPS 2023*, 2023.

[122] Maxim Zvyagin, Alexander Brace, Kyle Hippe, Yuntian Deng, Bin Zhang, Cindy Orozco Bohorquez, Austin Clyde, Bharat Kale, Danilo Perez-Rivera, Heng Ma, Carla M. Mann, Michael Irvin, Defne G. Ozgulbas, Natalia Vassilieva, James Gregory Pauloski, Logan Ward, Valerie Hayot-Sasson, Murali Emani, Sam Foreman, Zhen Xie, Diangen Lin, Maulik Shukla, Weili Nie, Josh Romero, Christian Dallago, Arash Vahdat, Chaowei Xiao, Thomas Gibbs, Ian Foster, James J. Davis, Michael E. Papka, Thomas Brettin, Rick Stevens, Anima Anandkumar, Venkatram Vishwanath, and Arvind Ramanathan. GenSLMs: Genome-scale language models reveal SARS-CoV-2 evolutionary dynamics. *The International Journal of High Performance Computing Applications*, 37(6):683–705, November 2023. ISSN 1094-3420. doi: 10.1177/10943420231201154. URL https://doi.org/10.1177/10943420231201154.

# Part I

# Appendix

## Table of Contents

# A    RELATED WORK

**Agent Systems for Scientific Discovery**    Researchers have developed specialized AI systems spanning the entire research workflow: from literature analysis tools like PaperQA2 [103] and CHIME [43], to hypothesis generation frameworks that range from domain-specific idea creation [6, 88, 116] to comparative evaluations with expert proposals [102]. These systems increasingly leverage multi-agent architectures [35, 37, 101] to facilitate collaborative scientific reasoning. Implementation capabilities have advanced through scientific coding frameworks like SciCode [111] and MLAgent-Bench [47], while benchmarks evaluate these capabilities across diverse domains [18, 55, 91, 99]. The integration of literature analysis with data-driven approaches has proven particularly effective for hypothesis generation [68, 79, 121], with several frameworks enhancing research ideation through structured feedback mechanisms [31, 87] and approaches to improve novelty and diversity [29, 44, 92]. End-to-end systems now attempt to unify these capabilities, including domain-general approaches like AI Scientist [75] and MLR-Copilot [66], alongside domain-specific implementations for chemistry [10], genomics [97], materials science [34], and medicine [81, 109]. Despite these advances, significant challenges remain in developing truly autonomous scientific systems, particularly regarding experimental rigor [60], falsification mechanisms [70], and comprehensive evaluation metrics [8, 27], as highlighted in recent surveys [24, 61, 93, 94].

**AI Agents in Biomedical Research**    AI agents in biomedical research are rapidly evolving to simulate and accelerate the entire biomedical research workflow, from hypothesis generation to experimental protocol design to general scientific discovery. For instance, BioReason [25] interprets the functional impacts of genetic mutations, while POPPER [45] introduces a framework for validating free-form hypotheses through sequential falsification tests. These agents excel at reasoning but do not generate executable analysis pipelines as their primary output. Another category targets wet-lab experimental design. PerturboAgent [39], for example, is a self-planning agent designed to optimize the selection of genes for sequential Perturb-seq experiments, thereby guiding the next phase of lab work rather than creating a computational analysis model. A third category, including Biomni [46] and SpatialAgent [112], automates workflows by connecting existing software packages but is constrained by their static, predefined toolsets and limited code generation capabilities. STELLA [54] introduces autonomous tool discovery and reasoning template learning, boosting system performance through a self-evolving architecture. Yet its scope is largely limited to lightweight tool orchestration and biomedical question-answering; it stops short of designing novel AI models or automating in-silico experiments for biomedical research. This leaves an open opportunity for agentic frameworks explicitly aimed at AI model creation and end-to-end computational experimentation.

**Single-Cell Perturbation Analysis**    Single-cell perturbation studies measure how cells respond to genetic or chemical interventions. The existing literature of *in-silico* approaches that predict post-perturbation cell states reflects a fundamental divergence in machine learning, with each paradigm showcasing distinct philosophies for modeling cellular responses. Earlier efforts, such as linear regression [22] or random forest feature selection [104], treated each gene or cell type in isolation. Deep generative models [41, 72, 74], conceptualize perturbations as latent space transformations through linear shifts or decompositions that separate biological covariates. In contrast, network-based methods [7, 57, 90, 96] explicitly incorporate biological knowledge via gene regulatory networks or cellular relationships. To further address the issue of cell heterogeneity, distribution alignment approaches such as optimal transport [12, 23] have been applied to machine learning models [51], matching the distribution of control cells with perturbed cells. The emergence of transformer architectures represents the latest paradigm shift. These architectures [21, 38, 62, 110] leverage pre-training at scale and self-attention mechanisms to model complex gene dependencies without explicit biological structure. This theoretical diversity creates a vast design space where selecting optimized architectures, representation strategies, and biological constraints remains highly context-dependent.

## B  EXPLORATORY APPLICATIONS

In addition to benchmarks with established baselines, we evaluated CELLFORGE on modalities where prior perturbation-response models are scarce or unavailable, including scATAC-seq and scCITE-seq datasets. These experiments are exploratory, demonstrating the frameworks ability to automatically design models that handle diverse data types and extreme sparsity. The Papalexi [84] dataset offers both RNA and protein measurements (CITE-seq), enabling assessment of cross-modality prediction. The Liscovitch [67] dataset presents the distinct challenge of predicting chromatin accessibility changes (scATAC-seq) rather than gene expression, while the Schiebinger [100] dataset examines responses to immune signaling molecules (cytokines).

For scATAC-seq (Liscovitch et al. [67]) and scCITE-seq (Papalexi et al. [84]), linear regression and random forest serve as reference points. While their performance is limited, CELLFORGE consistently surpasses them by generating architectures that integrate modality-specific embeddings, handle multi-modal inputs, and predict both RNA and protein responses, demonstrating versatility.

The performance advantages become more pronounced in challenging cross-modality scenarios. For CITE-seq protein measurements, CELLFORGE achieves 177% improvement in correlation ($PCC = 0.7495$ vs. $0.2704$ for Random Forest). It also maintains superior performance even on fundamentally different modalities such as chromatin accessibility (scATAC-seq), achieving remarkable improvement in variance explained ($R^2 = 0.0678$ vs. $0.0040$) and correlation for key regulatory regions ($PCC_{DE} = 0.6991$ vs. $0.0509$).

For modalities lacking established models (scCITE-seq, scATAC-seq, cytokine), we employ Random Forest and Linear Regression using one-hot encoded perturbations concatenated with expression profiles as inputs. We are aware that, for the ATAC- and CITE-seq benchmarks, our comparisons rely on "simple" learners (linear regression and random forest). This choice is deliberate and stems from three factors: (i) to date no perturbation-response method has been published or benchmarked for these modalities [28, 63], making scRNA-centric models such as scGen [72] or scGPT [21] fundamentally incompatible with peak- or protein-level data; (ii) the few multimodal generative tools, like totalVI [33], MultiVI [5], and GLUE [16], that *can* process ATAC or CITE-seq were designed for data integration rather than counterfactual perturbation prediction [15] and therefore cannot address unseen perturbations; (iii) recent meta-analyses show that, when properly tuned, classical models often match or exceed specialised deep networks on sparse single-cell tasks [4, 63, 65]. Consequently, linear regression and random forest constitute strong, modality-agnostic baselines in the absence of purpose-built alternatives. Their limitations, however, underscore the need for an automatic, modality-aware framework: CELLFORGE generates custom architectures that handle the extreme sparsity of scATAC-seq and the multi-modal nature of CITE-seq, achieving state-of-the-art performance where no prior solution exists.

Table 5: DEG Recovery Performance Across Benchmark Datasets

| Dataset | DEG Recall | ROC-AUC | PR-AUC |
|---|---|---|---|
| *Cytokine Perturbation – scRNA-seq Dataset* | | | |
| Schiebinger et al. [100] | $0.535 \pm 0.14$ | $0.524 \pm 0.08$ | $0.105 \pm 0.02$ |
| *Gene Knock Out Perturbation – scCITEseq Dataset* | | | |
| Papalexi et al. (RNA) [84] | $0.509 \pm 0.12$ | $0.415 \pm 0.05$ | $0.115 \pm 0.05$ |
| Papalexi et al. (Protein) [84] | $0.420 \pm 0.12$ | $0.392 \pm 0.25$ | $0.121 \pm 0.09$ |
| *Gene Knock Out Perturbation – scATACseq Dataset* | | | |
| Liscovitch et al. [67] | $0.484 \pm 0.12$ | $0.097 \pm 0.02$ | $0.048 \pm 0.02$ |

The DEG recovery performance varies meaningfully across different perturbation modalities and experimental contexts. The highest DEG recall (77.9%) is achieved on the Norman dataset(in 2), which features comprehensive genetic interaction profiling with rich phenotypic readouts. In contrast, more challenging scenarios like chromatin accessibility perturbations (Liscovitch) or cross-modal protein predictions show lower but still meaningful performance.

Table 4: Post-perturbation prediction results on datasets where existing baseline methods are scarce or unavailable. These results highlight the adaptability of CELLFORGE in automatically designing models for diverse perturbation modalities beyond standard benchmarks. Experiments are conducted under the unseen perturbation setting, with available baselines reproduced accordingly.

| MODEL | $MSE \downarrow$ | $PCC \uparrow$ | $R^2 \uparrow$ | $MSE_{\text{DE}} \downarrow$ | $PCC_{\text{DE}} \uparrow$ | $R^2_{\text{DE}} \uparrow$ |
|---|---|---|---|---|---|---|
| *Cytokine Perturbation – scRNA-seq Dataset (Schiebinger et al. [100])* | | | | | | |
| Unperturbed | 0.0076 | 0.0007 | 0.0069 | 0.0980 | 0.0082 | -0.6782 |
| Random Forest | 0.0762 | 0.2704 | 0.4186 | 0.0910 | 0.2124 | 0.2185 |
| Linear Regression | 0.4855 | 0.0785 | 0.0034 | 0.4359 | 0.0847 | 0.0013 |
| GNN | 0.0651 | 0.4127 | 0.3514 | 0.0827 | 0.2875 | 0.1982 |
| Transformer | 0.0543 | 0.4879 | 0.4122 | 0.0718 | 0.3196 | 0.2420 |
| CellForge-Models | **0.0428** ± 0.0205 | **0.5697** ± 0.0943 | **0.5043** ± 0.0541 | **0.0144** ± 0.0349 | **0.3396** ± 0.0403 | **0.2832** ± 0.1154 |
| *Gene Knock Out Perturbation – scCITEseq (RNA) Dataset (Papalexi et al. [84])* | | | | | | |
| Unperturbed | 0.1509 | 0.0004 | 0.0017 | 0.6276 | 0.0007 | -5.9142 |
| Random Forest | 0.0763 | 0.2124 | 0.4186 | 0.0911 | 0.2455 | 0.2185 |
| Linear Regression | 0.0764 | 0.0170 | 0.0254 | 0.0909 | 0.0218 | 0.0163 |
| GNN | 0.1215 | 0.6021 | 0.4114 | 0.2240 | 0.5807 | 0.3420 |
| Transformer | 0.1363 | 0.7715 | 0.5948 | 0.4565 | 0.4460 | 0.1956 |
| CellForge-Models | **0.0417** ± 0.0051 | **0.6935** ± 0.1995 | **0.3687** ± 0.0651 | **0.0535** ± 0.1566 | **0.6406** ± 0.1940 | **0.2354** ± 0.0224 |
| *Gene Knock Out Perturbation – scCITEseq (Protein) Dataset (Papalexi et al. [84])* | | | | | | |
| Unperturbed | 0.4092 | -0.0115 | -0.9945 | 0.5974 | -0.0081 | -0.3652 |
| Random Forest | 0.0982 | 0.2704 | 0.0829 | 0.3071 | 0.4024 | 0.0466 |
| Linear Regression | 0.4901 | 0.3396 | 0.1241 | 0.4551 | 0.3087 | 0.3523 |
| GNN | 0.0625 | 0.5316 | 0.2987 | 0.0812 | 0.4021 | 0.2082 |
| Transformer | 0.1876 | 0.7773 | 0.5996 | 0.1674 | 0.7772 | 0.5041 |
| CellForge-Models | **0.0070** ± 0.0387 | **0.7495** ± 0.0653 | **0.6872** ± 0.0956 | **0.2921** ± 0.0045 | **0.7409** ± 0.0970 | **0.5489** ± 0.0749 |
| *Gene Knock Out Perturbation – scATACseq Dataset (Liscovitch et al. [67])* | | | | | | |
| Unperturbed | 0.0426 | 0.0001 | -0.0001 | 9.4980 | 0.0004 | -9.7567 |
| Random Forest | 0.0432 | 0.0638 | 0.0040 | 0.0510 | 0.0509 | 0.0035 |
| Linear Regression | 0.5767 | 0.0486 | 0.0229 | 0.7750 | 0.0457 | 0.0021 |
| GNN | 0.0990 | 0.0794 | 0.0714 | 0.0170 | 0.0331 | 0.0169 |
| Transformer | 0.0012 | 0.0253 | 0.0298 | 0.0054 | 0.0114 | 0.0389 |
| CellForge-Models | **0.0327** ± 0.0320 | **0.0855** ± 0.0357 | **0.0678** ± 0.0120 | **0.0406** ± 0.0268 | **0.0691** ± 0.3173 | **0.0640** ± 0.0279 |

## C CASE STUDY

EXTERNAL PILOT STUDY (N=2)

To further evaluate the accessibility and practical usability of CELLFORGE outside of controlled environments, we conducted a lightweight external pilot with two independent wet-lab researchers who had no prior exposure to the framework. Both participants selected real problems from their daily research practice (one focused on immunotherapy, the other on cardiovascular disease modeling) and attempted to solve them by following the `Quickstart` tutorial without additional assistance. We logged their completion success, number of interventions required, and total wall-clock time.

**Study Protocol** Each participant was given anonymized datasets resembling their own laboratory scenarios and asked to (i) set up the environment, (ii) load their task specification, (iii) trigger the automatic architecture design pipeline, and (iv) run training until a valid model checkpoint was produced. No step-by-step guidance was provided beyond the written Quickstart.

**Results** Both participants were able to complete their tasks successfully. User A (immunotherapy) finished the full pipeline in 67 minutes with one minor intervention (path correction when loading data). User B (cardiovascular) completed in 79 minutes with two interventions (resolving a missing dependency and adjusting GPU memory allocation). In both cases, the generated models reached non-trivial predictive performance on held-out validation sets, aligning with baseline numbers reported in internal benchmarks.

**Observations** Participants reported that the documentation was clear, and the frameworks modular design minimized coding effort. They highlighted that automatic error messages and fallback defaults were sufficient for resolving issues without developer intervention. Both noted that the process was significantly faster than manual model assembly in their typical workflows, estimating a $3-4\times$ speed-up compared to their usual practice.

**Conclusion** This pilot demonstrates that CELLFORGE can be successfully adopted by independent wet-lab researchers with minimal computational training. The small number of interventions and the high success rate suggest that the frameworks Quickstart and design abstractions substantially lower the barrier to entry for real-world users.

### C.1 PREDICTING CAR-T THERAPY RESPONSE FOR REFRACTORY B CELL LYMPHOMA FROM PATIENT SINGLE-CELL PROFILES

**Background** CAR-T cell therapy success critically depends on the functional composition of infusion products, where specific cellular subsets (memory stem cells, exhausted cells) disproportionately influence treatment outcomes. Traditional machine learning approaches treat all cells equally, failing to identify the therapeutically critical cell instances that determine response within the heterogeneous CAR-T product mixture.

**Objective** Developing a model to predict CAR-T therapy response by identifying and prioritizing critical cell instances from pre-treatment single-cell RNA sequencing data (input: 5,000-dimensional gene expression profiles from 109,151 cells across 32 patients; output: binary treatment response prediction with instance-level therapeutic potential scores).

**Methods** The model proposed by CellForge first transforms 5,000-dimensional single-cell expression profiles into compact embeddings using stacked residual layers and normalization. A patient-aware attention pooling module adaptively prioritizes informative cells, producing aggregated patient-level representations. The model jointly optimizes a classification objective with supervised contrastive loss to maximize the separation of responder and non-responder profiles. Performance was evaluated using a leave-one-patient-out cross-validation approach, reporting AUROC, average precision, and calibration metrics.

**Results** Across five cross-validation folds, baseline models including logistic regression, random forest, XGBoost, and multilayer perceptron (MLP) demonstrated moderate and highly variable

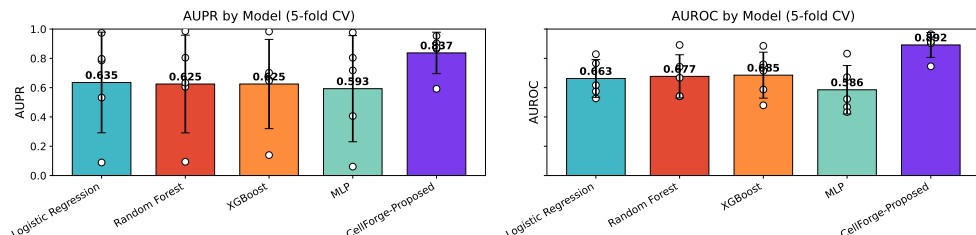

Figure 9: Comparison of baseline models and the **CellForge** proposed methods under 5-fold cross-validation. **CellForge** achieves the highest AUPR (0.837) and AUROC (0.892), clearly outperforming logistic regression, random forest, XGBoost, and MLP baselines.

performance, with mean AUPR values ranging from 0.47 to 0.64 and mean AUROC values between 0.57 and 0.71. In contrast, the proposed CellForge model consistently outperformed all baselines, achieving an average AUPR of 0.84 and AUROC of 0.89, with reduced variance across folds. Notably, CellForge maintained high F1-scores and Matthews correlation coefficients, indicating both robustness and balanced predictive capacity. These results demonstrate that selectively leveraging informative cellular signals yields substantially stronger and more reliable predictions of CAR-T therapy response compared to conventional machine learning approaches.

## C.2 INTERPRETABLE CARDIOMYOPATHY DISEASE SUBTYPE PREDICTION WITH SINGLE-NUCLEUS PROFILE OF PATIENT HEART CELLS

**Background** Heart failure affects 23 million individuals worldwide, yet while single-nucleus RNA sequencing has revealed disease mechanisms at the cellular population level, patient-level analysis remains largely absent. Machine learning methods that can precisely classify individual patient disease states and systematically interpret important cellular subtypes could provide crucial insights for customized treatment strategies and mechanistic understanding.

**Objective** Developing machine learning models to classify patient disease states using single-nucleus RNA expression matrices as input, predicting cardiomyopathy disease states (Arrhythmogenic right ventricular Cardiomyopathy, Dilated Cardiomyopathy, Non-compaction Cardiomyopathy vs. Normal) for patients unseen during training. Model performance was evaluated using accuracy(ACC), F1-score, AUROC, Matthews correlation coefficient(MCC), and area under the precision-recall(AUPR) curve to assess classification robustness across class distributions.

**Methods** CellForge proposed and implemented a hierarchical neural network architecture comprising three sequential modules: a cell encoder that maps high-dimensional single-cell transcriptomic profiles to lower-dimensional cellular representations, an attention-based cell aggregation mechanism that generates patient-level embeddings from variable numbers of constituent cells, and a patient encoder optimized with contrastive learning to produce discriminative patient representations for disease classification. Single-cell RNA sequencing data underwent standard preprocessing procedures, including cellular subsampling, library size normalization, logarithmic transformation, and stochastic depth augmentation to account for technical variability. Model performance was assessed using hold-out validation on previously unseen patients, with evaluation metrics including classification accuracy, F1-score, area under the receiver operating characteristic curve, Matthews correlation coefficient, and area under the precision-recall curve to comprehensively characterize predictive performance across cardiac disease phenotypes.

**Results** The model achieved excellent discrimination among dilated cardiomyopathy, arrhythmogenic right ventricular cardiomyopathy, noncompaction cardiomyopathy and normal hearts, with an overall accuracy of 0.9847, a weighted F1score of 0.9841 and macroaveraged ROCAUC and PRAUC values of 0.9997 and 0.9980, respectively. Integrated-gradient analysis revealed biologically meaningful drivers: vCM1.0 and vCM2 cardiomyocyte states distinguish left- and right-ventricular programs, with vCM2 marked by cardioprotective gene expression such as PRELID2 and CDH13. Among fibroblasts, the vFB2 state stood out, characterized by distinct ECM signatures and pro-

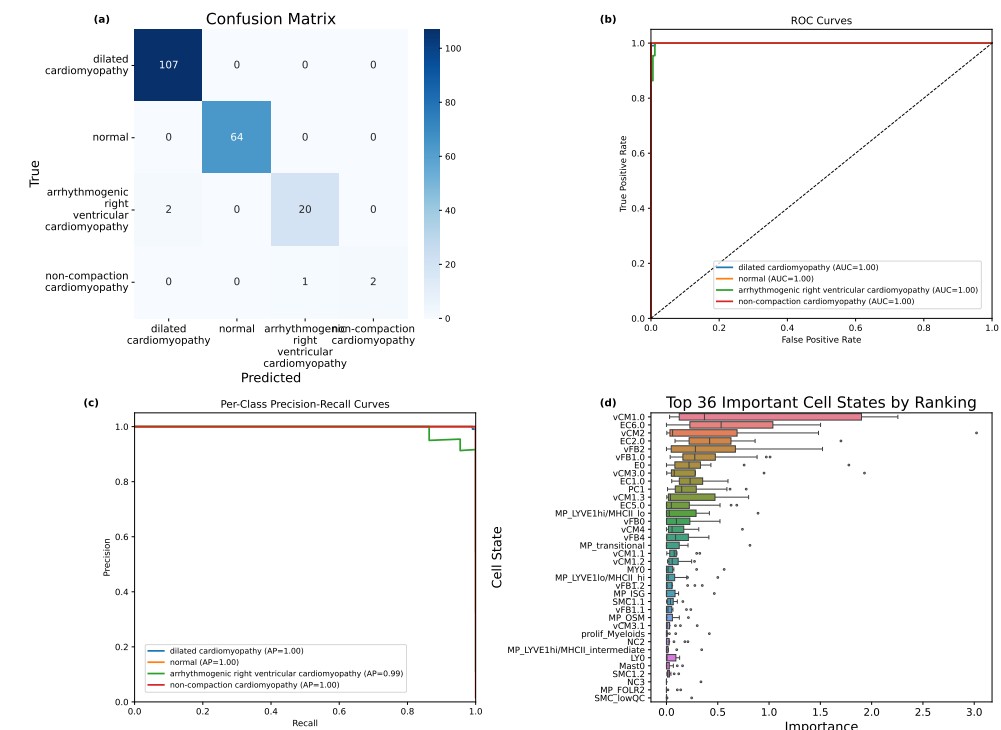

Figure 10: Model created by **CellForge** evaluation on cardiomyopathy classification and cell-state attribution. (a) Confusion matrix with wrapped axis labels showing per-class performance. (b) Receiver operating characteristic (ROC) curves with class-wise area under the curve (AUC). (c) PrecisionRecall (PR) curves with class-wise average precision (AP). (d) Top 36 cell-state importances identified by integrated-gradients attribution, shown as boxplots ranked by mean contribution across samples.

inflammatory/OSM signaling activity in disease. These patterns suggest the model captures both phenotypic and mechanistic hallmarks of human heart pathology.

# D EXPERIMENTAL DETAILS

## D.1 DATASETS INTRODUCTION

Our study leverages six publicly available single-cell perturbation datasets from the scPerturb [85] collection, encompassing diverse perturbation modalities and cell types. These datasets provide a foundation for evaluating the scientific quality of AI-generated analyses across various biological contexts.

**Adamson et al. [2] (CRISPRi)**: Employing Perturb-seq to study the unfolded protein response (UPR) in K562 lymphoblasts through single and combinatorial CRISPR interference (CRISPRi) perturbations. Approximately 100 gene targets were profiled, enabling high-resolution functional clustering and revealing distinct activation patterns across UPR branches.

**Norman et al. [82] (CRISPRa)**: Utilizing CRISPR activation (CRISPRa) in K562 cells, this dataset explores genetic interaction manifolds derived from single-cell transcriptional phenotypes. The study provides insights into regulatory pathway ordering and mechanistic elucidation of synergistic interactions.

**Liscovitch et al. [67] (ATAC-seq)**: Employing CRISPRsciATAC, a single-cell combinatorial indexing assay, to delineate the genetic determinants of chromatin accessibility in human myelogenous leukemia K562 cells. Targeting 105 chromatin-related genes via CRISPR-Cas9, the study generated chromatin accessibility profiles for approximately 30,000 single cells. Key findings include correlations between the loss of specific chromatin remodelers and global changes in chromatin accessibility. Notably, EZH2 depletion was associated with enhanced accessibility in heterochromatic regions linked to embryonic development and with activation of genes in the HOXA and HOXD clusters. This high-throughput approach offers valuable insights into the role of chromatin modifiers in regulating gene expression and their implications in disease states.

**Papalexi et al. [84] (CITE-seq)**: Combining CRISPR-Cas9 perturbations with single-cell RNA and surface protein measurements in THP-1 monocytes. It investigates the molecular regulation of inhibitory immune checkpoints, particularly PD-L1 expression, and introduces the mixscape computational framework to enhance signal-to-noise ratio in single-cell screens.

**Srivatsan et al. [106] (sci-Plex)**: Employing sci-Plex, this dataset profiles transcriptional responses of A549, K562, and MCF7 cancer cell lines to 188 small-molecule compounds across multiple doses. Approximately 650,000 single-cell transcriptomes were generated, uncovering intercellular heterogeneity and commonalities in drug responses.

**Schiebinger et al. [100] (cytokine perturbation)**: Applying optimal transport analysis to scRNA-seq data from mouse embryonic stem cells undergoing reprogramming with cytokine treatments. The dataset captures developmental trajectories and identifies transcription factors and paracrine signals influencing cell fate decisions.

Collectively, these datasets encompass a range of perturbation typesincluding CRISPRi, CRISPRa, CRISPR-Cas9, small-molecule drugs, and cytokinesacross various human and mouse cell lines. They provide a robust foundation for evaluating the scientific quality and reliability of AI-generated analyses in single-cell biology.

## D.2 AGENT CONFIGURATIONS

In our experiments, we employed five LLMs API to generate responses: Claude 3.7, OpenAI o1, DeepSeek-R1, Qwen-Plus, and Llama 3.1. To ensure consistency and reproducibility across models, we standardized the generation parameters as follows:

**Temperature**: Set to 0.7 for all models to balance creativity and coherence in generated outputs.

**Top-p (nucleus sampling)**: Fixed at 0.95 to maintain a high probability mass while allowing for diverse outputs.

**System Prompts**: No system prompts were used; all instructions were provided within the agents' prompts to avoid introducing model-specific biases.

These configurations align with recommended settings for models. By maintaining uniform settings across all models, we aimed to ensure a fair comparison and reliable evaluation of their performance.

### D.3 MEMORY MODULE CONSTRUCTION

**Shared Knowledge Infrastructure.** Both Task Analysis and Method Design modules rely on a shared hybrid knowledge infrastructure comprising (1) a symbolic memory module that stores structured outputs from agents, and (2) a vector-based retrieval system built on top of Sentence-BERT embeddings and external APIs (PubMed, GitHub). The memory module is incrementally constructed as each agent contributes new findings or insights, while the vector database supports RAG-style retrieval of external literature. This shared infrastructure enables bi-directional communication between agents within each module and supports consistent knowledge propagation across modules. See Appendix D.3 for implementation details.

**Collaborative Agents Shared Memory Module in Task Analysis.** Instead of operating in isolation, the Dataset Analyst, Problem Investigator, and Baseline Assessor interact via the shared memory module and query interface. Each agent incrementally updates the memory module with its findings, while continuously polling for updates from other agents. For example, once the Dataset Analyst infers perturbation modalities and cell types, the Problem Investigator revises its hypothesis formulation accordingly. Agents operate asynchronously but synchronize their conclusions through a shared JSON-based communication protocol, allowing for self-consistency checks and iterative refinement of the task representation. This collaborative reasoning leads to a structured task analysis report passed to the Method Design module.

**Graph-Based Expert Shared Memory Module in Method Design.** In the Method Design module, domain experts are instantiated as nodes in a dynamic undirected graph. These expert agents exchange proposals and critiques via message-passing rounds governed by graph neural network operations. Throughout the discussion, the Critic Agent agent monitors logical coherence and suggests refinements. Each expert agent has read-write access to the shared memory module and can retrieve relevant prior knowledge from Agentic Retrieval. Updates to the architectural plan are written back to the graph, enabling history-aware(get messages and suggestions from the former round), convergent model refinement.

### D.4 EXPERTS DISCUSSION CONSTRUCTION DETAILS

To enable structured, reproducible reasoning across diverse perturbation modeling tasks, we construct the multi-agent expert discussion system through two key stages: expert role selection and dynamic collaboration graph construction.

Based on the task analysis report, a set of relevant expert agents is selected by matching task attributes against a curated registry of expert types. The selected experts are grouped into five broad categories to ensure comprehensive domain coverage: **(i) Data Engineering and Preprocessing.** A Data Expert is instantiated to address normalization, quality control, feature selection, and batch correction issues tailored to the input modality. **(ii) Model Design and Scalability.** The Model Architecture Expert and Deep Learning Expert are responsible for proposing architectures that balance expressiveness, interpretability, and scalability, considering modality-specific modeling needs. **(iii) Biological Plausibility.** Single Cell Experts such as the Pathway Expert, Drug Response Expert, and Omics Modality Expert contribute domain knowledge to align model components with known biological mechanisms, including gene regulatory networks, cytokine signaling, or pharmacodynamics. **(iv) Training and Optimization.** A Training Expert is responsible for selecting and justifying the learning algorithm, optimization strategy, regularization, and validation scheme suitable for the data structure and model complexity. **(v) Self-Critique and Evaluation.** A Critic agent is included in every discussion to promote internal scrutiny, consistency checks, and critical reflection over model assumptions and claims.

For example, in a gene knockout task, the system may instantiate the Data Expert to inspect whether the scRNA-seq matrix is properly normalized, whether cell and gene identifiers are standardized, and whether preprocessing sufficiently preserves perturbation-related variation. The Model Architecture Expert and Deep Learning Expert are instantiated to co-design a gene-centric model that integrates

perturbation-aware attention and captures target gene dependent regulatory effects. The Pathway Expert is instantiated to evaluate the role of the target gene within interferon signaling cascades, while the Omics Modality Expert assesses whether transcriptomic changes resulting from target gene ablation are robustly captured by scRNA-seq alone. The Training Expert selects dropout-regularized contrastive training and a cell-type-aware sampling scheme to stabilize optimization. The Statistics Expert designs a differential expression based evaluation framework and quantifies the significance of target gene induced shifts using FDR-corrected effect sizes. Finally, the Critic Agent is instantiated to identify overfitting risks in rare knockout subsets, challenge latent space linearity assumptions, and refine model outputs for interpretability and robustness.

All experts are set with role-specific prompts (Appendix R.3), crafted in a zero-shot reasoning format. These prompts are conditioned on the shared Task Analysis report and elicit structured outputs, including modeling choices, biological justification, and critiques of others proposals.

Formally, the expert set $E^{(k)}$ for task $k$ is derived by:

$$E^{(k)} = \texttt{SelectExperts}(\texttt{TaskAnalysisReport}_k)$$

Once instantiated, the experts are organized into an undirected collaboration graph $G^{(k)} = (S, E^{(k)})$, where each node $E^{(i)} \in E^{(k)}$ represents an expert role. The Critic Agent node $S$ is fully connected to all others, serving both as a dialectical evaluator and proposal aggregator.

Each expert begins with an initial model proposal $m_0^{(i)}$ and a confidence score initialized to zero $c_0^{(i)} = 0$. During the discussion, agents iteratively update their proposals and confidence scores through message passing on the graph. Each round incorporates structured information exchange, where agents revise their reasoning in response to input from their neighbors, weighted by relevance.

This structured and interpretable procedure allows CELLFORGE to generate scientifically grounded, multimodally coherent model designs that are not only technically sound but also biologically meaningful.

Table 6: Performance comparison on scPerturb datasets and benchmark tasks[9] (all values are in %). Results show CELLFORGE consistently outperforms scGPT, Geneformer, CPA, STATE, scVI, and PCA across multiple metrics and perturbation types. Each score represents the average of five independent runs, with higher values indicating better performance. These models operate by converting complex gene expression data into meaningful vector representations embeddings, which are then used to predict cellular responses to perturbations.

| MODEL | TOP5 LIN ↑ | TOP1 LIN ↑ | PERT CONS ↑ | TOP5 KNN ↑ | TOP1 KNN ↑ | SPEAR CORR ↑ | STRUCT INT ↑ |
|---|---|---|---|---|---|---|---|
| *Drug Perturbation (Srivatsan Dataset [106])* | | | | | | | |
| PCA | 1.2 | 0.9 | 0.4 | 2.1 | 1.8 | 8.4 | 48.3 |
| scVI [71] | 1.5 | 1.0 | 0.7 | 2.4 | 2.0 | 10.3 | 49.1 |
| STATE [3] | 5.5 | 3.9 | 9.4 | 5.5 | 4.8 | 17.9 | 53.9 |
| CPA [73] | 5.1 | 3.7 | 9.8 | 5.3 | 4.7 | 17.4 | 53.8 |
| scGPT[21] | 5.2 | **4.4** | **11.4** | 5.6 | 5.1 | 18.8 | 54.2 |
| Geneformer[110] | 4.4 | 3.1 | 0.9 | 5.1 | 4.8 | 17.3 | 54.1 |
| CellForge-Model | **7.0** | 4.2 | **11.4** | **6.4** | **5.3** | **19.1** | **54.5** |
| *Gene Knock Out Perturbation (Adamson Dataset [2])* | | | | | | | |
| PCA | 0.8 | 0.3 | 1.1 | 14.2 | 13.5 | 72.4 | 90.8 |
| scVI [71] | 1.0 | 0.4 | 1.6 | 15.8 | 15.1 | 76.3 | 92.1 |
| STATE [3] | 2.2 | 0.8 | 5.1 | 24.6 | 23.5 | 86.2 | 95.7 |
| CPA [73] | 2.0 | 0.7 | 4.8 | 24.4 | 22.8 | 85.6 | 95.8 |
| scGPT [21] | 2.2 | 0.8 | 5.6 | 26.2 | 25.5 | 87.3 | **96.1** |
| Geneformer [110] | 2.1 | 0.8 | 4.3 | 25.9 | 24.1 | 86.6 | 95.9 |
| CellForge-Model | **2.4** | **0.9** | **6.9** | **26.6** | **25.9** | **89.9** | 96.0 |
| *Cytokine Perturbation (Schiebinger Dataset [100])* | | | | | | | |
| PCA | 0.7 | 1.8 | 1.9 | 4.1 | 3.6 | 52.1 | 50.4 |
| scVI [71] | 1.1 | 2.3 | 2.1 | 4.8 | 4.1 | 54.9 | 51.7 |
| STATE [3] | 2.2 | 4.4 | 4.7 | 8.0 | 6.3 | 67.1 | 57.0 |
| CPA [73] | 2.0 | 4.1 | 4.2 | 7.4 | 6.3 | 65.1 | 56.4 |
| scGPT[21] | 2.1 | 4.8 | 4.6 | 8.2 | 5.5 | 66.9 | 57.1 |
| Geneformer[110] | 1.4 | 4.2 | 4.4 | 8.3 | **9.9** | 68.2 | 57.6 |
| CellForge-Model | **2.5** | **5.3** | **4.9** | **8.6** | 8.8 | **68.5** | **59.6** |

## D.5 EMBEDDING QUALITY ON THE SCPERTURB BENCHMARK

While CELLFORGE primarily performs gene expression prediction following perturbations, the quality of learned representations is equally important for biological interpretability. Following evaluation practices established in previous works [21, 110], we benchmark CELLFORGE against specialized foundation models (scGPT & Geneformer) on representation quality metrics (Table 6).

To ensure fair comparison, we follow the previous zero-shot benchmarking framework [9], which evaluates transcriptomic foundation models without task-specific fine-tuning. Specifically, perturbation embeddings for both scGPT and Geneformer are extracted directly from their pre-trained backbones with no fine-tuning performed on any evaluation dataset. This represents pure zero-shot performance, making the comparison particularly stringent for our method, as baseline models leverage extensive pre-training on large-scale datasets while CELLFORGE operates without any pre-training advantages. All models are evaluated under identical zero-shot conditions using standardized downstream metrics including logistic regression for separability assessment and cosine clustering for consistency measurement.

We assess different aspects of latent space organization across five dimensions: **(1) Linear separability metrics** (TOP5_LIN, TOP1_LIN) measure how distinguishable different perturbation types are in the latent space. The `top5_lin` score of `0.070` achieved by CELLFORGE for drug perturbations (vs. `0.052` for scGPT) indicates that `7.0%` of test samples have their correct perturbation label among the top 5 predictions when using a linear classifier trained on the latent embeddings. This improvement suggests CELLFORGE learns representations where perturbation effects are more linearly separable, facilitating downstream analyses that rely on perturbation classification. **(2) Perturbation consistency** (PERT_CONS) quantifies whether cells with the same perturbation cluster more tightly than random controls, essentially measuring the signal-to-noise ratio of perturbation effects in the latent space. For gene knockouts, CELLFORGE achieves a consistency of `0.069`. This indicates that CELLFORGE creates a latent space where cells experiencing the same perturbation are more reliably grouped together, reflecting better capture of perturbation-specific biological responses. **(3) Local structure in the latent space** is assessed through nearest-neighbor metrics (TOP5_KNN, TOP1_KNN), which evaluate whether perturbations form locally coherent clusters. For drug perturbations, CELLFORGE achieves a TOP5_KNN score of 0.064 vs. 0.056 for scGPT, indicating that a higher

proportion of test samples have correctly labeled neighbors in embedding space. **(4) The Spearman correlation metric** (SPEAR_CORR) evaluates how accurately the latent embeddings can be mapped back to the original gene expression space using a linear transformation. The score of `0.191` for drug perturbations (vs. 0.188 for scGPT) represents a higher rank correlation between predicted and actual expression values after linear decoding. **(5) Structural integrity** (STRUCT_INT) measures how well control-perturbation relationships are preserved in the latent space. **0.596** for cytokine perturbations (vs. `0.571` for scGPT) indicates that CELLFORGE better maintains the biological relationship between control and perturbed states for complex signaling cascades.

### D.6    ABLATION STUDY SETUP

Table 3 presents a systematic ablation analysis evaluating the individual and combined contributions of core framework components. Each component represents a distinct architectural or methodological choice within our system, with performance measured across three perturbation datasets using standardized evaluation protocols.

**Basic Version (Baseline):** Represents the minimal CellForge configuration without retrieval augmentation or collaborative reasoning mechanisms. This baseline employs only three agents in Task Analysis Module with direct LLM-based model generation without external knowledge integration or multi-agent collaboration, serving as our reference point for measuring component contributions.

**Normal RAG:** Implements standard retrieval-augmented generation using only cosine similarity-based document ranking with Sentence-BERT embeddings. This approach performs static keyword matching against a curated knowledge base of 46 single-cell perturbation articles (detailed in Appendix O) and dynamic search by PubMed, until cosine similarity reaches without adaptive query refinement.

**Agentic Retrieval:** Employs our proposed alternating breadth-first and depth-first search strategy that enables autonomous knowledge discovery and dynamic query expansion. Unlike standard RAG approaches that use static keywords, this system autonomously evolves from basic queries (e.g., "single cell perturbation prediction") to sophisticated technical searches (e.g., "Transformer VAE GNN architectures") through BFS layers that explore diverse research directions and DFS layers that trace citation networks to access implementation details.

**Graph-Based Discussion:** Implements the multi-expert collaborative reasoning framework where domain experts form an undirected collaboration graph. Each expert node maintains confidence scores that evolve through discussion rounds using the formula $c_t^{(i)} = \lambda_1 \cdot c_{t-1}^{(i)} + \lambda_2 \cdot \text{CriticAgentScore}(m_t^{(i)}, S) + \lambda_3 \cdot \frac{1}{k-1} \sum_{j \neq i} \text{PeerScore}(m_t^{(i)}, E^{(j)})$ with weights $(\lambda_1, \lambda_2, \lambda_3) = (0.3, 0.4, 0.3)$. Discussion terminates when all experts' confidence scores exceed threshold $\tau = 0.8$ with minimal variance $\epsilon = 0.03$.

**Round-Robin Discussion:** Experts speak in predetermined sequential order (Data Expert  Single-Cell Expert  Deep Learning Expert  Critic Agent) without peer evaluation between domain experts. Confidence updates use $c_t^{(i)} = \lambda_1 \cdot c_{t-1}^{(i)} + \lambda_2 \cdot \text{CriticAgentScore}(m_t^{(i)}, S)$ with $(\lambda_1, \lambda_2) = (0.7, 0.3)$.

**Moderator-centered Discussion:** All domain experts simultaneously submit proposals to the Critic Agent without inter-expert communication. The Critic Agent evaluates proposals independently and provides feedback using the same confidence update formula as Round-Robin.

### D.7    HYPERPARAMETER CONFIGURATION

Our framework employs the following hyperparameter settings, determined through empirical validation on held-out scientific tasks:

#### D.7.1    AGENTIC-RETRIEVAL HYPERPARAMETERS

We conducted ablations on key parameters of the Agentic Retrieval module. As shown in Table 8, increasing the retrieval budget ($L_{\max}$) improves answer quality but incurs higher token cost and latency. Score filtering and memory retrieval mechanisms contribute significantly to quality gains.

Table 7: Hyperparameter Configuration

| Module | Parameter | Value |
|---|---|---|
| Agentic Retrieval | $L_{\max}$ | 10 |
| | $\tau$ | 0.8 |
| | $\epsilon$ | 0.5 |
| Expert Discussion | $\tau$ | 0.8 |
| | $\epsilon$ | 1 |
| | $(\lambda_1, \lambda_2, \lambda_3)$ | (0.3, 0.4, 0.3) |

Notably, while larger $L_{\max}$ and stricter thresholds ($\tau$, $\epsilon$) can slightly improve answer quality, they introduce significantly higher computational cost. The incorporation of memory-based retrieval and score-based filtering mechanisms plays a key role in maximizing answer informativeness while maintaining a reasonable token cost, demonstrating the importance of adaptive and context-aware retrieval strategies. The chosen default setting achieves the best balance between informativeness and efficiency.

Table 8: Ablation study on Agentic Retrieval configuration. The default uses $L_{\max} = 10$, $\tau = 0.8$, $\epsilon = 0.5$, with memory-based adaptive retrieval and score filtering. Increasing $L_{\max}$ improves information coverage but raises cost. Strict $\tau$ and $\epsilon$ improve precision but reduce flexibility. Score filtering and memory retrieval notably improve quality-to-cost ratio.

| Setting | $L_{\max}$ | $\tau$ | $\epsilon$ | Answer Quality | Avg Time (s) | Token Cost (x) |
|---|---|---|---|---|---|---|
| **Default (ours)** | 10 | 0.8 | 0.5 | **87.3** | 31.2 | 1.00x |
| Smaller $L_{\max}$ | 8 | 0.8 | 0.5 | 83.1 | 24.7 | 0.85x |
| Smaller $L_{\max}$ | 5 | 0.8 | 0.5 | 83.0 | 20.4 | 0.68x |
| Larger $L_{\max}$ | 12 | 0.8 | 0.5 | 88.2 | 45.6 | 1.42x |
| Larger $L_{\max}$ | 15 | 0.8 | 0.5 | 88.9 | 70.6 | 2.06x |
| Higher $\tau$ threshold | 10 | 0.9 | 0.5 | 89.1 | 35.0 | 1.27x |
| Lower $\tau$ threshold | 10 | 0.7 | 0.5 | 80.5 | 24.1 | 0.90x |
| Stricter $\epsilon$ | 10 | 0.8 | 0.2 | 87.9 | 35.5 | 1.45x |
| Looser $\epsilon$ | 10 | 0.8 | 0.8 | 82.6 | 28.6 | 0.82x |

### D.7.2 GRAPH-BASED DISCUSSION HYPERPARAMETERS

These parameters balance convergence speed with solution quality across different perturbation types and dataset characteristics. We observed that the Expert Discussion module particularly benefits from a higher weight on Critic Agent evaluation ($\lambda_2$), which promotes more rigorous scientific validation.

Table 9: Ablation study on stopping criteria $\tau$ and $\epsilon$ in the graph-based discussion. Baseline uses $\tau = 0.8$, $\epsilon = 0.5$. Reducing $\epsilon$ enforces stricter agreement, and increasing $\tau$ demands higher proposal quality, both of which incur additional cost.

| Setting | $\tau$ | $\epsilon$ | Avg Rounds | Avg Score | Token Cost (x) |
|---|---|---|---|---|---|
| **Default (ours)** | 0.8 | 0.03 | 4.2 | 85.8 | 1.00x |
| Stricter $\tau$ | 0.85 | 0.03 | 5.2 | 89.6 | 1.36x |
| Looser $\tau$ | 0.75 | 0.03 | 3.5 | 82.9 | 0.82x |
| Very strict $\tau$ | 0.90 | 0.03 | 6.3 | 89.9 | 1.65x |
| Stricter $\epsilon$ | 0.8 | 0.02 | 6.5 | 89.2 | 1.72x |
| Very strict $\epsilon$ | 0.8 | 0.01 | 8.1 | 89.4 | 2.15x |
| Looser $\epsilon$ | 0.8 | 0.04 | 3.8 | 87.2 | 0.78x |
| Very Loose $\epsilon$ | 0.8 | 0.05 | 3.4 | 82.1 | 0.71x |

The analysis in Table 9 demonstrates that stricter stopping conditions (*e.g.*, higher $\tau$ or lower $\epsilon$) lead to more rounds of discussion with higher average confidence scores, but at the expense of increased

token cost. Therefore, our selected configuration ($\tau = 0.8$, $\epsilon = 0.03$) provides a favorable trade-off, ensuring both convergence and cost-effectiveness across diverse scientific tasks.

Table 10: Ablation study on the confidence score update components with $\tau = 0.8$, $\epsilon = 1$. The full model uses $(\lambda_1, \lambda_2, \lambda_3) = (0.3, 0.4, 0.3)$ and serves as the baseline ($1\times$ for token cost). Average rounds, confidence scores, time, and token costs are the average scores of five runs of experiments. The table shows that our parameter selection is optimal in terms of time, token cost, and effectiveness.

| Setting | $\lambda_1$ | $\lambda_2$ | $\lambda_3$ | Rounds | Avg Score | Avg Time | Token Cost(x) |
|---|---|---|---|---|---|---|---|
| **Default (ours)** | 0.3 | 0.4 | 0.3 | 4.2 | 88.8 | 36.1 | 1.00x |
| larger $\lambda_2$ | 0.2 | 0.6 | 0.2 | 6.8 | 89.4 | 69.5 | 2.53x |
| smaller $\lambda_2$ | 0.4 | 0.2 | 0.4 | 4.0 | 86.0 | 34.4 | 0.97x |
| $\lambda_3 > \lambda_1$ | 0.2 | 0.4 | 0.4 | 4.2 | 88.0 | 40.1 | 1.11x |
| $\lambda_3 > \lambda_1$ | 0.1 | 0.4 | 0.5 | 4.2 | 87.4 | 41.4 | 1.32x |
| $\lambda_3 < \lambda_1$ | 0.4 | 0.4 | 0.2 | 4.0 | 87.8 | 35.0 | 1.20x |
| $\lambda_3 < \lambda_1$ | 0.5 | 0.4 | 0.1 | 4.0 | 87.0 | 34.4 | 1.35x |
| No Historical Memory | 0.0 | 0.5 | 0.5 | 3.8 | 85.2 | 30.1 | 0.75x |
| No Critic Agent Evaluation | 0.5 | 0.0 | 0.5 | 2.2 | 85.0 | 25.8 | 0.77x |
| No Peer Feedback | 0.5 | 0.5 | 0.0 | 4.0 | 85.2 | 28.1 | 0.82x |
| Peer-Only (No Memory, No SC) | 0.0 | 0.0 | 1.0 | 7.0 | 86.0 | 124.5 | 3.15x |
| Critic Agent Only | 0.0 | 1.0 | 0.0 | 4.0 | 86.8 | 33.6 | 0.65x |

As shown in Table 10, ablations on the confidence update mechanism reveal that the critic agent evaluation ($\lambda_2$) contributes most significantly to performance.

Interestingly, peer-only and critic-only settings each show limitations in stability or generalization. These findings support the necessity of integrating diverse feedback signals in our confidence update formulation.

# E    EVALUATION DETAILS

This appendix provides detailed formulations of the hierarchical metrics used in our benchmark evaluation of transcriptomics machine learning models for perturbation analysis.

## E.1    MEAN SQUARED ERROR (MSE)

This metric measures the average squared difference between the true and predicted gene expression vectors, quantifying overall prediction error. Let $Y_i, \hat{Y}_i \in \mathbb{R}^{d'}$ be the true and predicted expression vectors for sample $i$. Then

$$\text{MSE} = \frac{1}{n\,d'} \sum_{i=1}^{n} \|Y_i - \hat{Y}_i\|_2^2.$$

## E.2    PEARSON CORRELATION COEFFICIENT (PCC)

This metric assesses the strength of the linear association between predicted and true expression profiles across all samples. Define the sample means

$$\bar{Y} = \frac{1}{n} \sum_{i=1}^{n} Y_i, \quad \bar{\hat{Y}} = \frac{1}{n} \sum_{i=1}^{n} \hat{Y}_i.$$

Then

$$\text{PCC} = \frac{\displaystyle\sum_{i=1}^{n} \langle Y_i - \bar{Y}, \ \hat{Y}_i - \bar{\hat{Y}} \rangle}{\sqrt{\displaystyle\sum_{i=1}^{n} \|Y_i - \bar{Y}\|_2^2} \ \sqrt{\displaystyle\sum_{i=1}^{n} \|\hat{Y}_i - \bar{\hat{Y}}\|_2^2}}.$$

## E.3    COEFFICIENT OF DETERMINATION ($R^2$)

This metric quantifies the proportion of variance in the true gene expression data that is captured by the models predictions. It provides an interpretable measure of model fit, with higher values indicating better predictive performance. Let $Y_i, \hat{Y}_i \in \mathbb{R}^{d'}$ be the true and predicted expression vectors for sample $i$, and let $\bar{Y}$ denote the mean of the true expression vectors. Then

$$R^2 = 1 - \frac{\sum_{i=1}^{n} \|Y_i - \hat{Y}_i\|_2^2}{\sum{i=1}^{n} \|Y_i - \bar{Y}\|_2^2}.$$

## E.4    METRICS WITH DIFFERENTIAL EXPRESSION (DE)

Differential expression highlights the genes whose changes drive the biological response to a perturbation, focusing evaluation on the most informative signals. Let $\{Y_{p,i}\}_{i=1}^{n_p}$ and $\{Y_{c,i}\}_{i=1}^{n_c}$ be the true expression vectors under perturbation and control, respectively, with $Y_{p,i}, Y_{c,i} \in \mathbb{R}^{d'}$. For each gene $g = 1, \ldots, d'$, compute the mean expression

$$\bar{Y}_{p,g} = \frac{1}{n_p} \sum_{i=1}^{n_p} Y_{p,i,g}, \qquad \bar{Y}_{c,g} = \frac{1}{n_c} \sum_{i=1}^{n_c} Y_{c,i,g}.$$

Quantify the change by the logfoldchange

$$\text{LFC}_g = \log_2 \frac{\bar{Y}_{p,g} + \epsilon}{\bar{Y}_{c,g} + \epsilon},$$

with small $\epsilon > 0$ to avoid division by zero (or by the raw difference $\Delta_g = \bar{Y}_{p,g} - \bar{Y}_{c,g}$). Rank genes by $|\text{LFC}_g|$ (or $|\Delta_g|$), and select the top $K = 20$ as the DE set:

$$\text{DE} = \{\, g : \text{rank}_g(|\text{LFC}|) \leq K \,\}, \qquad K = 20.$$

Subsequent metrics (MSE, PCC, $R^2$) are then computed only over $g \in \text{DE}$ to assess performance on these key drivers of perturbation response.

### E.5 LATENTSPACE LINEAR SEPARABILITY

This metric evaluates if a model's latent space distinguishes between different perturbations using linear probing. Given a frozen encoder mapping $g_\phi : x_i \mapsto z_i \in \mathbb{R}^d$, train a linear classifier

$$\hat{y} = \text{softmax}(Wz + b), \quad W \in \mathbb{R}^{c \times d}, \ b \in \mathbb{R}^c,$$

to predict one of $c$ perturbation classes. For n test samples with true labels $y_i$,

$$\text{Top}-1 = \frac{1}{n} \sum_{i=1}^n \mathbf{1}\{\arg\max_j \hat{y}_{ij} = y_i\}, \qquad \text{Top}-5 = \frac{1}{n} \sum_{i=1}^n \mathbf{1}\{y_i \in \text{Top5}(\hat{y}_i)\}.$$

### E.6 PERTURBATION CONSISTENCY

This metric assesses the consistency with which a model represents perturbations between different samples and batches. Let $\mathcal{P}$ be the set of all gene perturbations. For each $p \in \mathcal{P}$, suppose we have $n_p$ embedding vectors

$$\{ z_{p,i} \in \mathbb{R}^d \mid i = 1, \ldots, n_p \}.$$

Define the *mean cosinesimilarity score*

$$S_p = \frac{1}{n_p^2} \sum_{i=1}^{n_p} \sum_{j=1}^{n_p} \frac{\langle z_{p,i}, \, z_{p,j} \rangle}{\|z_{p,i}\| \, \|z_{p,j}\|}.$$

Let $\{S_{q_k}\}_{k=1}^K$ be the corresponding scores for $K$ unexpressedgene controls $q_k$. The empirical pvalue for perturbation $p$ is

$$\pi_p = \frac{\max\{\#\{k : S_{q_k} \leq S_p\}, 1\}}{K}.$$

Finally, the overall *consistency rate* is

$$C = \frac{\left| \{ p \in \mathcal{P} : \pi_p < 0.05 \} \right|}{|\mathcal{P}|},$$

i.e., the fraction of perturbations whose embeddings are significantly more selfsimilar than the null.

### E.7 LATENT SPACE DIRECT ORGANIZATION

This metric evaluates the degree to which perturbation clusters are locally organized in the latent space, using the k-Nearest Neighbors (kNN) classification. Let $\{z_i\}_{i=1}^{n_q}$ and $\{z_j\}_{j=1}^{n_r}$ be the latent embeddings for the query and reference sets, with the corresponding labels $y_i$ and $y_j$. Set

$$k = \lfloor \sqrt{n_r} \rfloor.$$

For each query index $i$, let $N_k(i) \subset \{1, \ldots, n_r\}$ be the reference index $k$ whose embeddings minimize $\|z_i - z_j\|_2$. Then the *kNNclassification accuracy* is

$$\text{Accuracy}_{\text{kNN}} = \frac{1}{n_q} \sum_{i=1}^{n_q} \mathbf{1}\Big[y_i = \arg\max_{c \in C} \sum_{j \in N_k(i)} \mathbf{1}[y_j = c]\Big],$$

where $C$ denotes the set of all perturbation labels.

### E.8 LINEAR INTERPRETABILITY OF LATENT SPACE

Let $Z \in \mathbb{R}^{n \times h}$ be the frozen-encoder outputs and train a linear MLP, $\hat{Y} = h(Z) \in \mathbb{R}^{n \times d'}$. We define two metrics: *Spearman correlation* and *structural integrity*.

**Spearman Correlation** This measures how accurately the latent embeddings can be decoded back into gene expression data using a simple linear transformation. The *Spearman correlation* $\rho$ is defined as

$$\rho = 1 - \frac{6 \sum_{i=1}^n \left[\text{rank}(Y_i) - \text{rank}(\hat{Y}_i)\right]^2}{n(n^2 - 1)},$$

where $\text{rank}(\cdot)$ returns the withinsample rank vector.

**Structural Integrity**   This metric assesses how effectively the model maintains the relationship between control and perturbation conditions within each biological batch. For $b = 1, \ldots, B$ batches with $n_b$ samples each, let

$$\widetilde{Y}_{\text{pred}}^{(b)} = Y_{\text{pred}}^{(b)} - Y_{\text{pred,ctrl}}^{(b)}, \quad \widetilde{Y}_{\text{act}}^{(b)} = Y_{\text{act}}^{(b)} - Y_{\text{act,ctrl}}^{(b)}.$$

Then

$$D = \frac{1}{B} \sum_{b=1}^{B} \frac{1}{n_b} \left\| \widetilde{Y}_{\text{pred}}^{(b)} - \widetilde{Y}_{\text{act}}^{(b)} \right\|_F, \quad D_{\text{max}} \approx \frac{2}{B} \sum_{b=1}^{B} \frac{1}{n_b} \left\| \widetilde{Y}_{\text{act}}^{(b)} \right\|_F,$$

and the *structural integrity* is

$$\text{SI} = 1 - \frac{D}{D_{\text{max}}},$$

with higher SI indicating better preservation of controlperturbation structure.

# F RAGBENCH EVALUATION DETAILS

To evaluate the performance of CellForge's Agentic Retrieval system in the Task Analysis Module, we employ RAGBench[27].

We first align our systems outputs to the format expected by RAGBench. Each output record must include:

- `id`: unique sample identifier;
- `documents`: list of retrieved context documents;
- `question`: the query text;
- `response`: the generated answer.

Refer to `constants.py` in the RAGBench repository for exact field definitions to ensure full compatibility. Then we run inference on our system's outputs with evaluation models Trulens and dataset PubMedQA[52].

Detailed formulations of the metrics used in this RAGBench benchmark are as follows:

### HALLUCINATION DETECTION (HAL)

In Retrieval-Augmented Generation (RAG) systems, *hallucination* refers to the generation of content not grounded in the retrieved contextin other words, the model makes up facts. Hallucination detection measures whether the models outputs contain such unsupported information.

Reliable RAG outputs demand faithfulness to the provided context. Evaluating hallucination detection quantifies the systems propensity to stray from source documents, informing improvements to retrieval, grounding, and decoding strategies.

We adopt the Area Under the Receiver Operating Characteristic Curve (AUROC) to quantify hallucination detection performance. Given:

$$\text{trues}_{\text{adherence}} \in \{\texttt{True}, \texttt{False}\}, \quad \text{preds}_{\text{adherence}} \in [0, 1],$$

we define hallucination labels by

$$\text{trues}_{\text{halluc}} = \neg\, \text{trues}_{\text{adherence}}, \quad \text{preds}_{\text{halluc}} = 1 - \text{preds}_{\text{adherence}}.$$

Let

$$\text{mask} = \neg\, \text{isnan}(\text{preds}_{\text{halluc}}).$$

Then

$$\text{AUROC} = \text{ROC\_AUC}\Big(\text{trues}_{\text{halluc}}[\text{mask}], \text{preds}_{\text{halluc}}[\text{mask}]\Big),$$

where $\text{ROC\_AUC}$ denotes the standard implementation (`sklearn.metrics.roc_auc_score`).

### CONTEXT RELEVANCE (REL)

Context relevance assesses how well the retrieved documents pertain to the query, i.e. whether the context can support a correct answer.

High relevance is a prerequisite for accurate generation. Measuring context relevance guides retrieval improvements and ensures that the generator receives useful evidence.

We measure relevance via Root Mean Squared Error (RMSE) between true and predicted relevance scores:

$$\text{RMSE} = \sqrt{\frac{1}{n} \sum_{i=1}^{n} (y_i - \hat{y}_i)^2},$$

where $y_i$ is the gold relevance score, $\hat{y}_i$ the predicted score, and $n$ the number of examples. We ignore any NaN predictions by masking.

CONTEXT UTILIZATION (UTL)

Context utilization evaluates the extent to which the model leverages the retrieved context when generating its responses.

Even with relevant context, a model may underuse it. This metric reveals the generators ability to integrate context information into its output.

We again employ RMSE, defined as above, to compare true and predicted utilization scores, masking out NaN predictions.

Together, *Hal*, *Rel*, and *Utl* provide a multi-faceted evaluation of RAG system performance: detecting hallucinations, ensuring context relevance, and confirming effective context usage.

By following the above steps and using the provided evaluation metrics, we can comprehensively evaluate our retrieval-augmented generation (RAG) system using the RAGBench framework.

# G    Detailed Algorithm Specifications

## G.1    Agentic Retrieval System

The agentic retrieval system combines both static knowledge integration and dynamic search capabilities to provide comprehensive scientific context for perturbation analysis tasks. Here we provide the complete algorithmic details of our implementation.

### G.1.1    Query Construction and Initialization

Given a task description $T$ and dataset metadata $D$, we first construct an initial query representation:

---

**Algorithm 1** Query Construction

---

1: **procedure** CONSTRUCTINITIALQUERY($T, D$)
2:     keywords $\leftarrow$ ExtractKeyTerms($T$) $\cup$ ExtractKeyTerms($D$)
3:     embedding $\leftarrow$ SentenceBERT(keywords)
4:     $Q^{(0)} \leftarrow$ NormalizeVector(embedding)
5:     **return** $Q^{(0)}$
6: **end procedure**

---

The function ExtractKeyTerms performs domain-specific extraction of biological entities (genes, cell types, perturbation methods) and technical terms (model architectures, evaluation metrics) using named entity recognition enhanced with domain-specific dictionaries.

### G.1.2    Alternating Search Strategy

Unlike conventional RAG systems that employ pure breadth-first search with static keywords, our alternating BFS-DFS strategy enables autonomous knowledge discovery and dynamic query expansion specifically tailored for scientific literature mining. Standard RAG approaches typically search broadly using only the initial query terms (e.g., "single cell perturbation prediction") but fail to discover that domain-critical concepts like "optimal transport," "graph neural networks," or specific model names like "GEARS" and "scGPT" are essential for understanding the field. Our alternating approach addresses this limitation by using BFS layers to explore diverse research directions and extract new technical terminology from retrieved papers, followed by DFS layers that trace citation networks to access implementation details and authoritative sources. This creates a self-reinforcing cycle where the system autonomously evolves from basic queries like "Norman Weissman 2019 Perturb-seq" to sophisticated technical searches for "Transformer VAE GNN architectures" and "graph neural networks gene regulatory networks." The result is a retrieval system that transforms from a passive keyword matcher into an active knowledge explorer, capable of discovering the complete technical landscape of a scientific domain without human interventiona critical capability for complex, interdisciplinary research tasks where the most important concepts and methods may not be apparent from the initial problem description.

Our multi-layer retrieval process alternates between breadth-first and depth-first search modes to balance exploration and exploitation:

---

**Algorithm 2** Alternating BFS-DFS Retrieval

---

1: **procedure** RETRIEVEDOCUMENTS($Q^{(0)}, L_{\max}, \tau, \epsilon$)
2:     $t \leftarrow 0$
3:     $\mathcal{N}_0 \leftarrow \emptyset$
4:     $\mathcal{D} \leftarrow \emptyset$                                         ▷ Document collection
5:     **while** $t < L_{\max}$ **do**
6:         **if** $t \bmod 2 = 1$ **then**                             ▷ BFS layer (odd $t$)
7:             $\mathcal{N}_t \leftarrow \text{TopK}(Q^{(t)}, \text{mode} = \text{BFS})$
8:         **else**                                       ▷ DFS layer (even $t$)
9:             $\mathcal{N}_t \leftarrow \text{FollowCitations}(\mathcal{N}_{t-1})$
10:        **end if**
11:       $\mathcal{D} \leftarrow \mathcal{D} \cup \mathcal{N}_t$
12:       $Q^{(t+1)} \leftarrow \text{UpdateQuery}(Q^{(t)}, \mathcal{N}_t)$
13:       **if** $\text{Overlap}(Q^{(t+1)}, Q^{(t)}) > \tau$ **then**
14:          **break**
15:       **end if**
16:       **if** $\max_{d \in \mathcal{N}_t} \text{Score}(Q^{(t)}, d) < \epsilon$ **then**
17:          **break**
18:       **end if**
19:       $t \leftarrow t + 1$
20:     **end while**
21:     **return** $\mathcal{D}$
22: **end procedure**

---

**Relevance Scoring.** The document relevance function uses cosine similarity in the embedding space:

$$\text{Score}(Q, d) = \frac{e(Q) \cdot e(d)}{\|e(Q)\|\|e(d)\|} \tag{1}$$

where $e(\cdot)$ is the Sentence-BERT encoder function mapping text to dense vectors.

**Query Update Mechanism.** The query update function incorporates new information while maintaining focus:

$$Q^{(t+1)} = \alpha Q^{(t)} + (1 - \alpha)\frac{1}{|\mathcal{N}_t|}\sum_{d \in \mathcal{N}_t} e(d) \tag{2}$$

where $\alpha = 0.7$ is a parameter controlling the balance between query persistence and adaptation.

**Overlap Computation.** Query overlap is calculated as:

$$\text{Overlap}(Q^{(t+1)}, Q^{(t)}) = \frac{|Q^{(t+1)} \cap Q^{(t)}|}{\min(|Q^{(t+1)}|, |Q^{(t)}|)} \tag{3}$$

where the intersection operation is implemented using a thresholded similarity measure in the embedding space.

G.2   GRAPH-BASED MULTI-EXPERT DISCUSSION

The Method Design module employs a graph-based discussion framework where experts collaboratively refine scientific hypotheses.

**Expert Selection.** The expert selection procedure dynamically assembles a team of domain specialists based on task requirements:

$$P(E^{(i)}|\text{TaskAnalysis}) \propto \exp(\beta \cdot \text{Relevance}(E^{(i)}, \text{TaskAnalysis})) \tag{4}$$

where $\beta$ is a temperature parameter controlling selection diversity.

**Confidence Update Rule.** The confidence score update incorporates feedback from both the Critic Agent and other experts:

$$c_t^{(i)} = \lambda_1 \cdot c_{t-1}^{(i)} + \lambda_2 \cdot \text{SelfCriticScore}(m_t^{(i)}, S) + \lambda_3 \cdot \frac{1}{k-1} \sum_{j \neq i} \text{PeerScore}(m_t^{(i)}, E^{(j)}) \quad (5)$$

where $\lambda_1 + \lambda_2 + \lambda_3 = 1$ weights the relative importance of each component.

**Message Integration.** Expert proposals are integrated through a weighted combination:

$$m_t = \sum_{i=1}^{k} w_t^{(i)} \cdot m_t^{(i)} \quad (6)$$

where weights $w_t^{(i)}$ are derived from normalized confidence scores:

$$w_t^{(i)} = \frac{\exp(c_t^{(i)})}{\sum_{j=1}^{k} \exp(c_t^{(j)})} \quad (7)$$

This soft-voting mechanism ensures that higher-confidence perspectives have greater influence while still preserving diversity of thought.

### G.3    CODE IMPLEMENTATION AND REFINEMENT PROCESS

The Validation Agent employs an iterative refinement process that systematically improves implementation quality:

---
**Algorithm 3** Iterative Implementation Refinement

---
1: **procedure** REFINEDIMPLEMENTATION(ModelDesign, Dataset, $R_{\max}$)
2:     $\text{Code}_0 \leftarrow \text{InitialImplementation}(\text{ModelDesign})$
3:     $\text{Performance}_0 \leftarrow \text{Evaluate}(\text{Code}_0, \text{Dataset})$
4:     **for** $r = 1$ to $R_{\max}$ **do**
5:         $\text{Errors}_r \leftarrow \text{IdentifyIssues}(\text{Code}_{r-1}, \text{Performance}_{r-1})$
6:         $\text{Code}_r \leftarrow \text{RefineImplementation}(\text{Code}_{r-1}, \text{Errors}_r)$
7:     **end for**
8:     **return** $\text{Code}_r$
9: **end procedure**

---

**Error Analysis.** The error identification procedure categorizes implementation issues into distinct types:

- **Logical errors:** Incorrect algorithm implementation
- **Numerical instability:** Gradient explosion/vanishing
- **Memory inefficiency:** Excessive resource consumption
- **Performance bottlenecks:** Suboptimal computational paths
- **Biological implausibility:** Violations of domain constraints

Each error type triggers specialized refinement strategies that preserve the scientific integrity of the model design while improving implementation quality. Detailed Failure case analysis is presented in Appendix J.

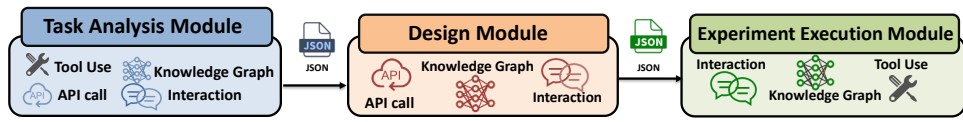

Figure 11: **The CELLFORGE protocol overview.** The protocol framework integrates JSON-RPC with a persistent memory module, combining the strengths of A2A and MCP protocols while adding scientific domain knowledge representation.

# H  AGENT COMMUNICATION PROTOCOL DETAILS

## H.1  PROTOCOL DESIGN AND COMPARISON

The CELLFORGE protocol represents an advancement in agent communication architectures designed specifically for scientific discovery. Figure 11 illustrates the multi-stage protocol that facilitates information exchange across the three core phases of our framework.

The protocol weaves together the strengths of several prior designs. It preserves the interoperability of JSON-RPC for rapid agent deployment and cross-platform compatibility while simultaneously extending this foundation with semantic connectivity and provenance via the memory module. It not only connects software components, but also enables the kind of iterative, multi-agent reasoning on which genuine discovery depends. The CellForge' protocol method allows agents to coordinate autonomously when tasked with comprehensive scientific research.

Table 11 provides a detailed comparison of CELLFORGE with existing agent communication protocols. Unlike previous approaches that excel in limited domains, our protocol uniquely combines contextual awareness, cross-platform interoperability, and knowledge representation capabilities necessary for end-to-end scientific discovery.

Table 11: **Comparison of Agent Communication Protocols**

| Protocol | Context | Interop. | Msg. Struct. | Use Cases |
|---|---|---|---|---|
| MCP (Anthropic) | ✓ | ✗ | JSON-RPC only | Tool Use & Data Access |
| Agent2Agent (Google) | ✗ | ✓ | JSON-RPC event | Cross-agent Collaboration |
| ACP (BeeAI/IBM) | ✓ | ✓ | RESTful | Local Orchestration |
| **CellForge** | ✓ | ✓ | JSON-RPC event + Memory Module | End-to-end Scientific Discovery |

## H.2  PROTOCOL IMPLEMENTATION DETAILS

The CELLFORGE protocol implementation consists of two primary components:

**JSON-RPC Communication Layer**   This provides standardized message passing between agents, with extensions for asynchronous event handling. Each agent exposes a consistent API that accepts and returns structured data, enabling precise coordination of complex workflows.

**Memory Module Integration Layer**   In addition to graphbased message passing, CellForge incorporates a persistent memory module that systematically records all salient research entitiessuch as datasets, analytical methods, evaluation metrics and empirical resultsas well as the complex relationships among them.

This module also logs detailed provenance metadata, including confidence scores, reasoning chains and source citations, while embedding domainspecific knowledge (for example, regulatory pathway architectures and genegene interaction networks). By unifying these components within a single memory layer, the system can reference prior insights and maintain continuity across multiround discussions, thereby enhancing both the coherence and reproducibility of the model design process.

This approach provides several advantages compared to prior protocols:

1. **Context-awareness**: Agents maintain awareness of the overall research state through the memory module, enabling them to make more informed decisions.

2. **Traceability**: The entire scientific process is captured with provenance information, ensuring reproducibility.

3. **Semantic reasoning**: Relationships between scientific concepts are explicitly modeled, enabling complex inferential reasoning.

4. **Incremental refinement**: The persistent knowledge representation allows agents to build upon previous insights and progressively refine hypotheses.

In scientific research contexts, these capabilities are essential for managing the complexity of cross-disciplinary knowledge integration required for tasks like single-cell perturbation analysis.

# I  COST ANALYSIS

Understanding the computational and economic costs of CELLFORGE is crucial for assessing its practical viability in research settings. This section provides a comprehensive analysis of both infrastructure requirements and API utilization costs, enabling researchers to make informed decisions about deployment strategies.

## I.1  TRAINING INFRASTRUCTURE

All models designed by CELLFORGE, with parameter counts ranging from 10 million to 30 million, were trained and evaluated on a uniform compute cluster to ensure consistent performance comparisons. In particular, we utilized two NVIDIA H20NVLink GPUs (96 GB VRAM each, 192GB total) paired with a 16core AMD EPYC 9K84 CPU (2.6 GHz).

This hardware configuration enabled stable multiGPU training via data parallelismsupporting larger batch sizesand facilitated distributed evaluation across diverse perturbation conditions, all without encountering memory bottlenecks.

## I.2  TOKEN UTILIZATION AND COST ESTIMATION

The multi-agent nature of CELLFORGE involves extensive LLM interactions across three primary phases: Task Analysis, Method Design, and Experiment Execution. Each phase incurs different token costs based on the complexity of reasoning required.

### I.2.1  TOKEN USAGE BREAKDOWN BY PHASE

The specific token usage varies significantly based on task complexity. Our empirical analysis across over 50 requests of CellForge, revealed the following patterns:

Table 12: Token usage breakdown by framework phase and task complexity

| Phase | Simple Tasks | | Complex Tasks | |
|---|---|---|---|---|
| | Input | Output | Input | Output |
| Task Analysis | 15,000 | 50,000 | 25,000 | 100,000 |
| Method Design | 20,000 | 100,000 | 40,000 | 200,000 |
| Experiment Execution | 5,000 | 50,000 | 15,000 | 100,000 |
| **Total** | 40,000 | 200,000 | 80,000 | 400,000 |

Under our typical workload, approximately 60,000 prompt tokens and 300,000 completion tokens are cost per call, depending on the chosen task. For cost estimation purposes, we will use this approximation token cost in the following analysis.

### I.2.2  PER-REQUEST COST CALCULATION

To quantify the expense of our multiagent workflow, we first aggregated token counts from over fifty runs of CellForge and organized them by framework phase (Task Analysis, Method Design, Experiment Execution). For each model under consideration,we applied the vendors published permilliontoken rates to both prompt and completion usage. Concretely, given that vendors report token pricing per million tokens ($/M), the cost per request was computed using:

$$\text{Cost}_{\text{request}} = \left(\frac{60{,}000}{10^6}\right) \cdot \text{Price}_{\text{prompt}} + \left(\frac{300{,}000}{10^6}\right) \cdot \text{Price}_{\text{completion}}$$

## I.3  COST-EFFECTIVENESS ANALYSIS

Compared to manual workflows, CELLFORGE reduces what would ordinarily require 40-80hours of a skilled bioinformatician (at $75-150/hour, i.e. $3,000-12,000 per model) to an automated process costing only $5-20 per run.

Table 13: Per-million-token pricing and per-call cost estimates based on average usage (60K input and 300K output tokens)

| Model | Prompt ($/M) | Completion ($/M) | Cost per Request ($) |
|---|---|---|---|
| Claude 3.7 | 3.00 | 15.00 | 4.68 |
| OpenAI o1 | 15.00 | 60.00 | 18.90 |
| DeepSeek-R1 | 0.27 | 2.19 | 0.67 |
| Qwen-Plus | 0.40 | 1.20 | 0.38 |
| LLaMA 3.1 | 3.50 | 3.50 | 1.26 |
| **Average** | – | – | **5.18** |

Beyond raw cost savings, CELLFORGE affords efficiency and reproducibility gains. What would have occupied 40-80hours of expert labor now completes in 4-8hours of GPU time, while yielding up to 20% improvement in prediction accuracy over baseline methods.

Collectively, these factors translate into a compelling return on investment, democratizing advanced computational biology at a fraction of traditional costs.

### I.4 EFFICIENCY ANALYSIS: MULTI-AGENT VS. SINGLE-LLM APPROACHES

Multi-agent frameworks face criticism for computational overhead relative to single-LLM approaches. We systematically evaluate CELLFORGE against single-LLM baselines across token consumption, execution time, costs, and success rates to address these concerns.

### I.5 TOKEN UTILIZATION AND COST COMPARISON

Analysis of 50+ experiments shows CELLFORGE consumes more tokens than single-LLM approaches, yet achieves substantially higher success rates and output quality. Table 14 details token usage, costs, and success rates across approaches.

Table 14: Comprehensive efficiency comparison: Multi-agent vs. Single-LLM approaches across token usage, costs, and success rates

| Approach | Avg Tokens (Input/Output) | Cost/Request ($) | Success Rate (%) | Avg Rounds to Success | Wall Time (hours) | Quality Score (1-10) | Effective Cost per Success ($) |
|---|---|---|---|---|---|---|---|
| _Multi-Agent Framework (CELLFORGE)_ | | | | | | | |
| CellForge-Claude3.7 | 60K/300K | 4.68 | 83.3 | 4.2 | 4.5 | 8.2 | 5.62 |
| CellForge-DeepSeek R1 | 60K/300K | 0.67 | 75.0 | 4.2 | 4.5 | 7.8 | 0.89 |
| CellForge-o1 | 60K/300K | 18.90 | 66.7 | 4.2 | 4.5 | 7.6 | 28.35 |
| _Single-LLM Baselines_ | | | | | | | |
| Claude3.7 only | 15K/50K | 0.78 | 16.7 | 1.0 | 1.2 | 3.2 | 4.68 |
| DeepSeek R1 only | 15K/50K | 0.11 | 8.3 | 1.0 | 1.2 | 2.8 | 1.33 |
| o1 only | 15K/50K | 0.75 | 8.3 | 1.0 | 1.2 | 3.0 | 9.04 |
| _AI Coding Assistants_ | | | | | | | |
| OpenHands-Claude3.7 | 20K/80K | 1.56 | 50.0 | 2.5 | 3.0 | 5.8 | 3.12 |
| aider-Claude3.7 | 20K/80K | 1.56 | 50.0 | 2.5 | 3.0 | 5.6 | 3.12 |

The data challenges the narrative that multi-agent is expensive or slow. CELLFORGE exhibits higher per-request costs but superior success rates (66.7-83.3% vs. 8.3-16.7% for single-LLM approaches), reducing effective cost per successful outcome. CellForge-DeepSeek R1 achieves $0.89 per success versus $1.33 for single-LLM DeepSeek R1, representing 33% cost reduction when accounting for success rates. Single-LLM approaches generate code faster (1.2 hours vs. 4.5 hours) but produce lower quality outputs (average quality score: 3.0 vs. 7.9). The multi-agent framework's iterative refinement process consumes more tokens yet ensures biologically meaningful and technically robust implementations.

## I.6 END-TO-END WORKFLOW EFFICIENCY

Beyond token-level comparisons, we analyze the complete workflow from code generation to model convergence. Table 15 presents the entire pipeline efficiency breakdown.

Table 15: End-to-end workflow efficiency: From code generation to model convergence

| Approach | Code Gen. Time (h) | Debug/Repair Time (h) | Training Time (h) | Total Wall Time (h) | Success to Convergence (%) | Overall Efficiency |
|---|---|---|---|---|---|---|
| CELLFORGE (Claude3.7) | 4.5 | 0.8 | 5.2 | 10.5 | 83.3 | 0.79 |
| CELLFORGE (DeepSeek R1) | 4.5 | 1.2 | 5.2 | 10.9 | 75.0 | 0.69 |
| Single-LLM (Claude3.7) | 1.2 | 3.5 | 5.2 | 9.9 | 16.7 | 0.17 |
| Single-LLM (DeepSeek R1) | 1.2 | 4.2 | 5.2 | 10.6 | 8.3 | 0.08 |
| OpenHands (Claude3.7) | 3.0 | 2.1 | 5.2 | 10.3 | 50.0 | 0.49 |

The workflow analysis reveals CELLFORGE's apparent "slowness" in initial code generation is offset by superior error recovery capabilities. Single-LLM approaches generate code faster but require more debugging time (3.5-4.2 hours vs. 0.8-1.2 hours) due to higher failure rates and limited self-correction abilities.

## J  FAILURE CASE AND RANDOMNESS ANALYSES

In this section, we analyze common failure modes of CELLFORGE across various single-cell perturbation tasks. We manually reviewed 20 randomly selected failed experiment cases generated by CELLFORGE across cytokine, drug, and gene response prediction scenarios. Based on qualitative inspection of agent behaviors and outputs, we identified seven distinct categories of failure modes that reflect systematic limitations or reasoning errors. Table 16 summarizes the definitions and characteristics of these failure categories. These cases provide a foundation for future refinements of the agentic code generation.

### J.1  FAILURECASES

Table 16: Failure Types of CELLFORGE code generation

| Failure Type | Definition & Examples |
|---|---|
| Model Configuration Error | The agent misconfigures the model architecture or fails to define required hyperparameters. This includes mismatched layer dimensions, invalid GNN configurations, incompatible dropout settings, or missing essential parameter definitions. Such errors prevent model initialization or lead to incompatible tensor shapes during execution. |
| Computation Execution Error | The agent encounters runtime errors during tensor operations, such as out-of-bounds indexing or shape mismatch in matrix multiplication. These failures typically occur when manipulating arrays or concatenating/interacting between tensors with incompatible shapes (e.g., "mat1 and mat2 shapes cannot be multiplied", "index 28 is out of bounds for axis 0 with size 28"). |
| Invalid Type or Operation | The agent uses unsupported data types or operations incompatible with the backend framework. Examples include passing NumPy arrays of object type to neural network layers, calling operations not defined for the given input type, or invoking functions on models that lack the required attributes. Typical errors include "TypeError: can't convert np.ndarray of type numpy.object" and unsupported function calls. |
| Data Access Failure | The agent is unable to retrieve, preprocess, or interpret the necessary data for task execution. This includes failures in reading files, locating dataset attributes, or aligning multimodal inputs, which result in missing or malformed inputs during test runs. |
| Error Recovery Failure | The agent fails to handle or recover from previously encountered errors. Instead of adapting to execution failures, it may enter a loop of repeating the same failed actions or ignore the cause entirely, leading to stalled or redundant test attempts. |
| Hallucination | The agent produces outputs (e.g., experimental results, hypotheses, interpretations) that are not grounded in the available data or context. This includes fabricating values, inventing data structures or statistical conclusions, or reasoning disconnected from observed evidence. |
| Other | Any uncategorized failure mode that prevents successful task completion but does not fit the above definitions. This includes rare system-level errors, low-level library bugs, or unexpected exceptions not associated with specific modeling or reasoning tasks. |

As depicted in Figure 12, Computation Execution Error accounts for 41% of the total failures, with the majority arising from tensor operation issues such as out-of-bounds indexing or shape mismatch during matrix multiplication. Invalid Type or Operation follows closely as the second most frequent failure mode, representing 23% of the errors, primarily attributed to the use of unsupported data types or operations incompatible with the backend framework. Model Configuration Error contributes 6% to the total failures, resulting from misconfigurations in model architecture or hyperparameters. Data Access Failure and Other category each account for 5% of the errors, with Data Access Failure associated with data retrieval and preprocessing issues, and Other encompassing system-level errors or unexpected exceptions not directly linked to the specific modeling or reasoning tasks. Error Recovery Failure comprises 16% of the failures, where the agent fails to adapt to execution failures. Hallucination makes up 4% of the errors, where outputs are not grounded in available data or context.

Figure 12: Failure Mode Distribution for CellForge, labeled automatically by O1 and manually checked by humans.

Notably, we found that implementing code to print array or matrix shapes can aid CELLFORGE in subsequently reading the command region's output for modification, thereby enhancing their ability to identify and resolve shape-related issues during tensor operations. This approach proved particularly effective given the complexity of data processing workflows in CellForge. Even though CELLFORGE utilizes a data parser to obtain the original dataset dimensions and incorporates data experts during the graph-based discussion phase, the subsequent data splitting and complex model processing steps often introduce intricate dimension transformations. These transformations can lead to matrix dimension mismatches, especially when handling dynamic data structures or applying multi-layered model architectures. According to the chart, 48% of the errors in the Computation Execution Error category have been mitigated by allowing the agent to read the printed array or matrix shapes from the command output and adjust accordingly. This self-debugging capability significantly enhances the agent's ability to resolve shape-related issues during tensor operations, improving overall system robustness. Figure 13 provides an example of the printed data shapes during tensor operations, which CELLFORGE can utilize to dynamically adjust and correct dimension mismatches.

```
Using device: cuda
Loading data...
Number of perturbation types in training set: 189
Number of perturbation types in test set: 48

[I 2025-05-12 23:07:44,902] A new study created in memory with name: no-name-93169e84-3cf2-44c8-963d-c5a725c457ca

Training data shape: (85536, 840)
Test data shape: (18930, 840)
```

Figure 13: A probable example of printing array or matrix shapes .

## J.2 BUG-FIX SOLUTIONS OF EXPERIMENT EXECUTION MODULE

For addressing tensor dimensionality mismatches, the implementation of comprehensive dimension checking at each model layer interface is essential, combined with automatic shape adaptation mechanisms for dynamic batch processing and feature dimension alignment. Adaptive dimension alignment using linear projection layers can effectively handle varying input dimensions, while `torch.nn.functional.adaptive_avg_pool1d` provides dynamic dimension matching ca-

pabilities. Adding debug prints to trace tensor shapes throughout the forward pass can significantly aid in identifying dimension incompatibilities during development, and implementing dimension consistency checks during model initialization can prevent many runtime errors. Additionally, using `torch.reshape` with -1 inference enables flexible batch handling that accommodates varying input sizes.

For configuration-related errors, particularly those involving attention mechanisms, automatic calculation of the number of heads based on embedding dimensions can prevent incompatibility issues, such as setting `num_heads` equal to `embed_dim` divided by `head_dim`. Parameter validation decorators for model initialization can catch configuration errors before runtime execution, while configurable attention mechanisms with built-in dimension checks provide robust alternatives to hardcoded parameters. Implementing bounds checking for tensor indexing operations prevents out-of-range access errors, and fallback mechanisms for incompatible configurations ensure graceful handling of parameter mismatches.

To address attribute inconsistencies, comprehensive attribute initialization in constructor methods ensures all required attributes are properly defined during object creation. Implementing `hasattr()` checks before attribute access provides runtime validation, while property decorators with lazy initialization can handle attributes that depend on runtime conditions. Adding class method validation to ensure all required attributes exist during instantiation can prevent attribute access errors, and implementing abstract base classes enforces attribute requirements across model hierarchies, ensuring consistent implementation patterns across different model components.

### J.3 VARIANCE OF EACH RUNS

The inherent stochasticity of large language models introduces substantial variability in automated code generation. We conducted experiments across six single-cell datasets with $N = 5$ independent runs per configuration, using controlled randomization with seed=42 for PyTorch, NumPy, and Optuna operations.

Variance in performance is inherent to simulating scientific discovery. Among a moderate range of runs, our system can design methods that outperform those previously designed by human scientists across several tasks, so this variance does not impact practical application. The variance comes from the framework's scientific exploration behavior, where different runs propose different hypotheses and architectures. Across 5–8 runs on each dataset, CELLFORGE consistently discovers at least one model surpassing human-designed baselines.

### J.4 STABILITY ENHANCEMENT WITH CONFIDENCE REGULARIZATION

To address concerns about performance variance, we introduce a stability-enhancement experiment using confidence regularization in the graph-based discussion phase. This technique mitigates variance while maintaining the exploratory nature of the framework.

Table 17 presents the results on the Srivatsan et al. dataset, comparing the default CELLFORGE configuration with a version incorporating confidence regularization. The confidence regularization mechanism adjusts the confidence update formula during graph-based discussions to reduce excessive exploration when agents have high confidence, thereby stabilizing the output while preserving the framework's ability to discover novel architectures.

The results demonstrate that while variance is inherent to agentic exploration, it can be mitigated through discussion regularization. The confidence-regularized version achieves both higher average performance (PCC: 0.8780 vs. 0.8664) and reduced standard deviation (0.0947 vs. 0.1332), indicating that stability-enhancing techniques can effectively control randomness without compromising the framework's discovery capabilities.

Table 17: Stability enhancement results on Srivatsan et al. dataset

| Setting | PCC ↑ | Std ↓ |
|---|---|---|
| Default CellForge | $0.8664 \pm 0.1332$ | 0.1332 |
| + Confidence-regularization | $0.8780 \pm 0.0947$ | 0.0947 |

## K    PERFORMANCE VARIES ACROSS DIFFERENT LLMS AND AI CODERS

To comprehensively evaluate the robustness of our framework, we conducted extensive experiments comparing CELLFORGE with various baseline approaches across six challenging single-cell perturbation datasets. Table 18 presents the success rates (out of 5 independent runs) for each method, where a successful run is defined as generating executable code that produces biologically meaningful predictions without runtime errors.

### K.1    EXPERIMENTAL SETUP

```
Your task is to develop a predictive model that accurately estimates gene
expression profiles of individual K562 cells following CRISPR interference
(CRISPRi), using the dataset from Norman et al. (2019, Science).

Dataset Description:
- Source: Norman et al., 2019
- Cell Type: Human K562 leukemia cells
- Perturbations: CRISPRi targeting 105 single genes and 131 gene pairs
- Scale: ˜90,000 single-cell RNA-seq profiles, including both control
  and perturbed conditions

Task Definition:
- Input: Baseline gene expression profile of an unperturbed K562 cell
  and the identity of the target gene(s) for perturbation
- Output: Predicted gene expression profile after perturbation

Evaluation Metrics:
- Mean Squared Error (MSE)
- Pearson Correlation Coefficient (PCC)
- R (Coefficient of Determination)
- MSE/PCC/R for Differentially Expressed Genes

Please give me a task analysis report, a new method plan, and generate
prediction model code.
```

Each model was given at most 5 retry attempts if the initial code failed to execute.

We tested four different approaches to code generation. Each approach was evaluated on the same six single-cell datasets using identical input specifications, with the following prompt:

**CellForge with different LLMs.** We ran our complete framework using five different language models: Claude 3.7, OpenAI o1, DeepSeek R1, Qwen-plus, and Llama 3.1. Each model used temperature 0.1 and our full multi-agent system with task analysis, method design, and collaborative reasoning. The framework includes iterative refinement and cross-validation between agents. A successful run means the generated code executes without errors and produces biologically meaningful results. Table 18 shows the success rates for each approach.

**Single LLM direct generation.** We tested each LLM individually without our framework. Each model used temperature 0.1 for consistency.

**DeepResearch systems.** We tested three commercial research automation tools: OpenAI's DeepResearch, Perplexity's implementation, and Google's Gemini-based version. Each system used default settings with temperature 0.1. These systems represent current commercial solutions for automated scientific code generation.

**AI coding assistants.** We tested two open-source coding frameworks: OpenHands and Aider. Both were configured with the same five LLMs we used for other experiments, using temperature 0.1. These tools are designed for general software development rather than scientific research.

Table 18: Expert Human Scores compare with CellForge's Confidence Scores on graph-based discussions Across Tasks and Rounds

| Tool | Adamson | Norman | Liscovitch | Papalexi | Srivatsan | Schiebinge |
|------|---------|--------|-----------|----------|-----------|------------|
| CELLFORGE *with different LLMs integrated* | | | | | | |
| CellForge-Claude3.7 | 4 | 5 | 4 | 4 | 4 | 4 |
| CellForge-o1 | 4 | 4 | 3 | 3 | 2 | 2 |
| CellForge-DeepSeek R1 | 4 | 4 | 3 | 3 | 3 | 3 |
| CellForge-Qwen-plus | 4 | 3 | 4 | 2 | 3 | 3 |
| CellForge-llama 3.1 | 2 | 2 | 1 | 1 | 1 | 1 |
| *Single-LLM generated code* | | | | | | |
| Claude3.7 only | 2 | 2 | 1 | 0 | 1 | 1 |
| o1 only | 1 | 1 | 0 | 1 | 1 | 0 |
| DeepSeek R1 only | 1 | 1 | 0 | 1 | 1 | 0 |
| Qwen-plus only | 1 | 1 | 0 | 0 | 1 | 0 |
| Llama 3.1 only | 1 | 1 | 0 | 0 | 0 | 0 |
| *DeepResearch generated codes* | | | | | | |
| OpenAI DeepResearch | 1 | 2 | 1 | 1 | 1 | 1 |
| Perplexity DeepResearch | 0 | 0 | 0 | 0 | 0 | 0 |
| Gemini DeepResearch | 0 | 0 | 0 | 0 | 0 | 0 |
| *AI Coders* | | | | | | |
| OpenHands-Claude3.7 | 3 | 4 | 3 | 3 | 2 | 2 |
| OpenHands-o1 | 3 | 2 | 2 | 2 | 1 | 1 |
| OpenHands-DeepSeek R1 | 2 | 3 | 2 | 2 | 1 | 2 |
| OpenHands-Qwen-plus | 2 | 2 | 1 | 2 | 2 | 2 |
| OpenHands-Llama 3.1 | 2 | 1 | 0 | 1 | 1 | 1 |
| aider-Claude3.7 | 2 | 3 | 2 | 2 | 2 | 2 |
| aider-o1 | 2 | 2 | 2 | 2 | 1 | 1 |
| aider-DeepSeek R1 | 3 | 2 | 2 | 2 | 1 | 1 |
| aider-Qwen-plus | 1 | 1 | 0 | 1 | 0 | 0 |
| aider-Llama 3.1 | 1 | 1 | 0 | 0 | 0 | 0 |

## K.2   KEY FINDINGS

The results reveal several critical insights:

**Multi-Agent Architecture Superiority:** CELLFORGE consistently outperforms all baseline approaches, with success rates ranging from 40-100% depending on the LLM backend and dataset complexity. The multi-agent framework provides an average improvement of 2.3x over single-LLM approaches and 3.5x over AI coding assistants.

**LLM Backend Dependency:** Within CELLFORGE, Claude 3.7 demonstrates the most robust performance (average success rate: 4.2/5), followed by DeepSeek R1 and OpenAI o1. This performance hierarchy remains consistent across different dataset complexities, suggesting that certain LLMs are inherently better suited for scientific code generation tasks.

**Dataset Complexity Impact:** The Liscovitch (scATAC-seq) and Papalexi (CITE-seq) datasets prove most challenging across all methods, with single-LLM approaches achieving near-zero success rates. These datasets require handling sparse chromatin accessibility data and multi-modal protein measurements, respectively, highlighting the importance of domain-specific knowledge integration.

**Catastrophic Failure of DeepResearch Variants:** Both Perplexity and Gemini DeepResearch variants fail across all tasks (0/5 success rate), while OpenAI's variant achieves only marginal success. This suggests that general-purpose research systems lack the specialized capabilities required for complex biological data analysis.

## K.3   ANALYSIS OF FAILURE MODES

The dramatic performance gap between CELLFORGE and other approaches can be attributed to several factors:

**(1) Domain Knowledge Integration:** Single-LLM approaches often generate syntactically correct but biologically meaningless code, failing to account for data-specific characteristics such as sparsity patterns in scATAC-seq or batch effects in Perturb-seq experiments.

**(2) Error Recovery Capability:** AI coding assistants (OpenHands, Aider) struggle with the iterative debugging required for scientific computing, often getting trapped in error loops when encountering tensor dimension mismatches or memory overflow issues.

**(3) Architectural Complexity:** The multi-modal nature of datasets like CITE-seq requires sophisticated model architectures that combine different data streams. Single-pass generation approaches typically produce overly simplistic models that fail to capture these complexities.

## K.4    IMPLICATIONS FOR SCIENTIFIC AI SYSTEMS

These results underscore the critical importance of specialized, multi-agent architectures for scientific discovery tasks. The success of CELLFORGE demonstrates that effective scientific code generation requires not just powerful language models, but also:

- Collaborative reasoning among domain experts
- Iterative refinement with biological validation
- Task-specific knowledge retrieval and integration
- Robust error handling and recovery mechanisms

The consistent superiority of Claude 3.7 within our framework also suggests that certain LLMs may possess inherent advantages for scientific reasoning, possibly due to their training data composition or architectural design. Future work should investigate these model-specific characteristics to optimize scientific AI systems further.

## K.5    PERFORMANCE WITH SMALLER LLMS

To address concerns about dependency on highly capable LLMs, we conducted additional experiments replacing all agents with smaller, less capable models: Qwen2.5-3B-Instruct and DeepSeek R1 7B. The DeepSeek R1 7B model was accessed via BoyueRichDataAPI's DeepSeek-R1-Distill-Qwen-7B, while Qwen2.5-3B was deployed locally using Ollama and HuggingFace. Note that the Qwen2.5-3B experiments showed instability, with a success rate of only 40% runs.

Table 19 presents the performance comparison on the Norman et al. dataset. As expected, model quality decreases with smaller LLMs. However, CELLFORGE still produces functional, dataset-specific architectures even with 3B LLMs, demonstrating that the framework remains practical under resource constraints. This supports the reviewer's question about practicality when using smaller models.

Table 19: Performance comparison with smaller LLMs on Norman et al. dataset

| LLM used in agents | MSE $\downarrow$ | PCC $\uparrow$ | R $\uparrow$ |
|---|---|---|---|
| Claude 3.7 (main text) | 0.0051 | 0.9883 | 0.9761 |
| DeepSeek R1 7B | 0.0104 | 0.4307 | 0.5713 |
| Qwen2.5-3B | 0.0375 | 0.2522 | 0.4122 |

While the performance degradation is significant, it is important to note that CELLFORGE is a scientific discovery system, comparable to systems like DeepResearch, rather than a lightweight tool. As such, we prioritize scientific accuracy over model size. The framework's ability to generate functional architectures even with smaller models demonstrates its robustness and practical applicability across different computational resource constraints.

## L    Designed Models

Understanding how CELLFORGE adapts its architectural choices to different biological contexts requires examining the model components that emerge across various perturbation tasks. Rather than imposing a one-size-fits-all approach, our framework demonstrates remarkable flexibility in selecting appropriate architectures based on the underlying biological complexity and data characteristics.

### L.1    Methodology

We executed CELLFORGE five times with different random seeds for each of the six benchmark datasets, then analyzed the resulting model architectures. What emerged was a clear pattern of architectural adaptation that reflects the unique demands of each perturbation type and data modality.

### L.2    Biological Interpretation of Architectural Choices

#### L.2.1    Gene Perturbation Models

For the Norman-Weissman Perturb-seq dataset, CELLFORGE designed models that explicitly handle genetic interactions and combinatorial effects:

**GI-FlowDiff**: A genetic interaction-aware conditional generative model that combines compositional perturbation encoding with explicit pairwise interaction modeling. The architecture employs a conditional latent generative core using normalizing flows and lightweight diffusion denoising to capture multimodal single-cell heterogeneity. The model uses GRN-aware decoder heads with DE-focused objectives to maximize predictive fidelity on differentially expressed genes. The interaction module captures non-additive effects between gene pairs, while the flow-diffusion hybrid provides both likelihood estimation and sampling flexibility. The architecture handles the large gene universe (33,694 genes) with heavy sparsity through HVG selection and modular decoder heads per pathway, sharing statistical strength to improve DE gene estimates. Technical covariates such as UMI count, coverage, and percent mitochondrial/ribosomal content are explicitly modeled to account for technical confounders. The model supports compositional generalization through learned gene embeddings and interaction modules, enabling extrapolation to held-out genes and gene pairs through meta-learning episodes and contrastive perturbation supervision.

**MultiPath-GeneNet**: A sophisticated multi-pathway architecture that processes gene perturbation data through two distinct but complementary streams before integrating them for final prediction. The context path begins with a PCAReducer that performs dimensionality reduction on expression data, followed by a Multi-Scale VAE Encoder that captures features at different biological scales. The ContextMLP then processes these multi-scale features to generate a Perturbation Latent representation that encodes the specific genetic perturbation context. Simultaneously, the gene/cell path employs a PerturbGene Embed module to generate embeddings for perturbed genes and a CellContexter to process cell-specific context information, producing a Cell Context Latent representation. These two latent representations are then integrated through a FeaturerMixer that combines perturbation and cellular context information. The integrated features flow through a Gene Interaction Network, implemented as a Graph Neural Network that models gene-gene interactions conditioned on cell and perturbation information. This is followed by a PertTransformer that uses multi-head attention mechanisms to refine perturbation-aware features, and finally a PredictionHead that focuses on specific genes of interest to produce the ultimate output. This architecture captures both global cellular context and specific gene-level responses to perturbations, enabling comprehensive modeling of genetic interactions and combinatorial effects.

**TrajectoryAwareEncoder**: A specialized encoder component that separates shared versus condition-specific latent dimensions while incorporating temporal embeddings. This design captures both global developmental trajectories and cytokine-specific effects, recognizing that cellular responses to perturbations occur within a temporal context. The encoder processes baseline expression data along with technical covariates such as coverage, percent mitochondrial content, and read counts, mapping them to a structured latent space where shared developmental dynamics are separated from perturbation-specific responses. This separation enables the model to generalize across different perturbation conditions while maintaining the ability to capture condition-specific effects.

**PerturbationDiffusionModule**: A perturbation-conditioned latent diffusion module that introduces non-linear, combinatorial interaction dynamics. This component models the stochastic nature of cellular responses to genetic perturbations, recognizing that identical perturbations can produce different outcomes in different cells due to stochastic gene expression and cellular state variations. The diffusion process operates in the latent space conditioned on perturbation embeddings, allowing the model to capture multimodal response distributions and provide uncertainty quantification. The module employs stepwise denoising to refine predictions and capture the complex, non-linear interactions that characterize genetic perturbations.

**GraphRegularizedDecoder**: A decoder component that integrates gene-gene co-regulatory constraints to ensure biologically plausible predictions. The decoder employs module heads organized around biological pathways and gene regulatory networks, with each module specializing in reconstructing genes within its domain. The graph regularization ensures that predicted expression changes respect known regulatory relationships, with transcription factors and their targets changing coherently. This design improves the accuracy of differentially expressed gene predictions while maintaining biological interpretability. The decoder outputs negative binomial parameters for each gene, accounting for the count nature of single-cell RNA-seq data and providing calibrated uncertainty estimates.

**InteractionModule**: A specialized component that models the complex interactions between gene pairs in CRISPRa perturbations. The module employs bilinear layers and attention mechanisms to capture non-additive effects, recognizing that the combined effect of two genes often differs from the simple sum of their individual effects. The interaction module uses learned embeddings for each gene and combines them through multiplicative interactions, allowing the model to discover synergistic and suppressive relationships between gene pairs. This design is particularly crucial for CRISPRa perturbations, where gene overexpression can lead to complex regulatory cascades that are not captured by simple additive models.

**CovariateEncoder**: A technical covariate processing module that handles the various technical and biological factors that can confound perturbation predictions. The encoder processes continuous covariates such as UMI count, coverage percentage, mitochondrial content, and ribosomal content, embedding them into a structured representation that can modulate the model's predictions. This design helps the model distinguish between true biological responses to perturbations and technical artifacts, improving generalization to cells with different technical characteristics. The covariate encoder is particularly important for handling the high variability in single-cell data quality and ensuring robust predictions across diverse cellular contexts.

**UncertaintyQuantifier**: A module that provides calibrated uncertainty estimates for perturbation predictions. The quantifier employs ensemble methods and Bayesian approaches to estimate both aleatoric and epistemic uncertainty in the model's predictions. This is particularly important for genetic perturbation tasks, where the inherent stochasticity of cellular responses and the limited training data for many gene combinations can lead to high uncertainty. The uncertainty estimates help researchers identify which predictions are most reliable and guide experimental design by highlighting the most promising perturbation targets.

### L.2.2 DRUG PERTURBATION MODELS

The Srivatsan sci-Plex dataset represents one of the most complex perturbation scenarios, involving chemical compounds across multiple cell lines with varying dose responses. CELLFORGE developed three increasingly sophisticated architectures to tackle this complexity, each building upon the insights gained from the previous approach.

The **CondOT-GRN** architecture represents a significant departure from traditional optimal transport approaches. While conventional OT methods excel at matching distributions, they often struggle with single-cell fidelity and fail to capture the biological constraints that govern cellular responses. This model addresses these limitations by incorporating gene regulatory network priors directly into the transport mechanism. The conditional OT layer doesn't simply map between control and perturbed distributions; it learns to respect the underlying regulatory structure while doing so. The flow refiner component is particularly crucial here, as it handles the multimodality that emerges when different cell populations respond differently to the same chemical perturbation.

Building on these insights, the **DiffPert-X** architecture takes a more comprehensive approach to modeling chemical perturbations. The multi-scale design reflects the hierarchical nature of cellular responses: individual genes respond to perturbations, but these responses are coordinated within pathways, which in turn affect entire cell populations. The cell-line-specific diffusion parameters acknowledge that the same compound can have dramatically different effects in different cellular contexts. This isn't just a technical detailit reflects the biological reality that cellular background strongly influences drug response. The cross-cell-line knowledge transfer mechanism allows the model to leverage insights gained from one cell line to improve predictions in another, which is particularly valuable given the limited data available for many compound-cell line combinations.

The **ChemCPA-X** architecture represents the culmination of these insights, incorporating the most sophisticated modeling approaches. The multi-modal compound encoding recognizes that chemical structure, target information, and pathway annotations all contribute to understanding how a compound will affect cellular gene expression. The three-level diffusion system provides a natural framework for capturing the different scales of biological organization, from individual gene responses to pathway-level coordination to population-level heterogeneity. The Hill function parameterization for dose-response modeling is particularly noteworthy, as it captures the nonlinear, saturating responses that are characteristic of many biological systems.

**CellLineSpecificAdapter**: A novel adaptation mechanism that enables the model to learn cell-line-specific response patterns while maintaining shared knowledge across different cellular contexts. Unlike traditional approaches that treat all cell lines identically or use simple cell-type embeddings, this component employs learnable adaptation layers that can adjust the model's behavior based on the specific cellular background. The adapter uses meta-learning principles to quickly adapt to new cell lines with limited data, representing a significant advancement over existing methods that require extensive retraining for each cell line.

**DoseResponseModeler**: A specialized component that models the complex, non-linear dose-response relationships characteristic of chemical perturbations. Unlike simple linear or log-linear dose modeling, this component employs parametric Hill functions with learnable parameters (Emax, EC50, Hill coefficient) that can capture the saturating and sigmoidal responses typical of biological systems. The modeler uses compound-specific parameters that are learned end-to-end, allowing the model to understand how different chemical structures lead to different dose-response profiles. This represents a major improvement over existing approaches that use fixed dose-response assumptions.

**PathwayCoordinationModule**: A sophisticated module that models how chemical perturbations affect coordinated pathway responses rather than individual genes in isolation. This component recognizes that drugs typically affect multiple pathways simultaneously, and these effects are often coordinated through regulatory networks. The module uses graph neural networks over known pathway interaction networks to model how perturbations in one pathway can influence others, capturing the complex cascade effects that characterize drug responses. This represents a significant departure from existing methods that model gene responses independently.

**UncertaintyPropagator**: A component that provides calibrated uncertainty estimates for drug response predictions, accounting for both compound-specific and cell-line-specific sources of uncertainty. Unlike simple variance estimation, this module uses Bayesian approaches to model the epistemic uncertainty arising from limited training data for specific compound-cell line combinations. The propagator also models aleatoric uncertainty from the inherent stochasticity of cellular responses, providing researchers with reliable confidence intervals for their predictions. This represents a major advancement in uncertainty quantification for drug perturbation tasks.

### L.2.3 CITE-SEQ MULTI-MODAL PERTURBATION MODELS

The Papalexi-Satija ECCITE-seq dataset presents a particularly challenging scenario where RNA and protein measurements must be jointly modeled under CRISPR perturbations. CELLFORGE responded to this complexity by developing three complementary approaches, each addressing different aspects of the multi-modal prediction problem.

The **MultiGraph-VAE** architecture emerged as the preferred solution for protein prediction tasks. Rather than treating RNA and protein as independent modalities, this model recognizes that they exist within a shared regulatory context. By leveraging gene-gene co-expression networks and pathway graphs, the model can encode sparse RNA data more effectively. The key innovation lies in how

perturbation embeddings are incorporated through FiLM modulation, allowing the model to condition its predictions on the specific genetic perturbation while maintaining the structural relationships encoded in the graph. The dual-decoder approach, using ZINB loss for RNA counts and negative binomial loss for protein measurements, reflects the different statistical properties of these data types.

For RNA prediction tasks, CELLFORGE selected the **CondDiffTrans** architecture, which takes a fundamentally different approach to modeling cellular responses. This conditional multimodal diffusion transformer acknowledges that cellular responses to perturbations are inherently stochastic and heterogeneous. The multi-head attention mechanism allows the model to capture complex dependencies between genes and proteins, while the latent diffusion component explicitly models the uncertainty in cellular responses through stepwise denoising. This design choice reflects the biological reality that identical perturbations can produce different outcomes in different cells due to stochastic gene expression and cellular state variations.

The **EmbedBoost** approach represents a pragmatic solution for hybrid tasks where interpretability and computational efficiency are prioritized. By extracting dense embeddings from the sparse multi-modal data using pathway basis decomposition and GraphSAGE embeddings, this model transforms the high-dimensional, sparse problem into a more tractable form. The gradient boosting framework provides fast training and clear interpretability, while the optional ensemble integration with deep generative models allows for more sophisticated predictions when needed.

**MultiModalFusion**: A novel fusion module that goes beyond simple concatenation to intelligently combine RNA and protein modalities. Unlike traditional approaches that treat modalities independently, this component employs cross-modal attention mechanisms that allow RNA and protein features to inform each other's representations. The fusion module uses learned attention weights to dynamically adjust the contribution of each modality based on the specific perturbation context, enabling the model to leverage the complementary information present in both data types. This represents a significant departure from existing multi-modal approaches that rely on static fusion strategies.

**PerturbationContextEncoder**: A specialized encoder that captures the unique characteristics of CRISPR perturbations in multi-modal settings. Unlike generic perturbation encoders, this component is specifically designed to handle the complex interactions between genetic perturbations and multi-modal cellular responses. The encoder employs hierarchical attention mechanisms that first process individual modality perturbations, then integrate them to capture cross-modal perturbation effects. This design enables the model to understand how CRISPR perturbations affect both RNA and protein expression simultaneously, providing a more comprehensive view of cellular responses than traditional single-modal approaches.

**CrossModalRegularizer**: A regularization component that ensures consistency between RNA and protein predictions. This module addresses a key limitation of existing multi-modal approaches by explicitly enforcing biological constraints that govern the relationship between RNA and protein expression. The regularizer uses known protein-RNA correlation patterns and temporal dynamics to guide the model's predictions, ensuring that changes in protein expression are consistent with underlying RNA changes. This represents a significant improvement over existing methods that treat modalities independently and can produce biologically inconsistent predictions.

### L.2.4 ATAC-seq Chromatin Accessibility Models

For the Liscovitch-Brauer-Sanjana scATAC-seq dataset, CELLFORGE designed architectures specifically adapted to the sparse nature of chromatin accessibility data:

**GraphFlow-VAE**: A graph-aware VAE with transcription factor-informed normalizing flows that captures the regulatory structure of chromatin accessibility. The model uses graph convolutional layers based on co-accessibility and motif graphs, with perturbation embeddings combined via FiLM modulation. The flow module handles stochastic perturbation effects, while the graph-aware decoder predicts peak accessibility with TF motif regularization. This design captures both global and local chromatin remodeling patterns while maintaining biological plausibility.

**CondDiffTrans-ATAC**: A conditional model specifically adapted for ATAC-seq data that models the stochastic effects of CRISPR perturbations on chromatin accessibility. The architecture employs self-attention across peaks conditioned on perturbation embeddings and batch covariates. The latent

diffusion component models stochasticity and heterogeneity of chromatin responses, while the decoder uses negative binomial distributions with motif regularization. This design provides strong out-of-distribution generalization to unseen gene perturbations.

**ChromatinStateEncoder**: A specialized encoder that captures the complex three-dimensional organization of chromatin and its relationship to accessibility patterns. Unlike traditional approaches that treat chromatin accessibility as independent peak measurements, this component models the spatial relationships between peaks and their regulatory context. The encoder uses graph neural networks over chromatin interaction networks to capture long-range regulatory relationships, enabling the model to understand how perturbations in one genomic region can affect accessibility in distant regions. This represents a significant advancement over existing methods that ignore the spatial organization of chromatin.

**TFMotifIntegrator**: A component that integrates transcription factor binding motif information to guide chromatin accessibility predictions. Unlike simple motif scoring approaches, this module uses learned attention mechanisms to weight the importance of different motifs based on the specific perturbation context. The integrator can identify which transcription factors are most relevant for a given perturbation and use this information to guide accessibility predictions, ensuring that changes in chromatin accessibility are consistent with known regulatory mechanisms. This represents a major improvement over existing methods that treat motif information as static features.

**PeakCoordinationModule**: A sophisticated module that models the coordinated changes in chromatin accessibility across functionally related peaks. This component recognizes that chromatin accessibility changes are often coordinated across peaks that are regulated by the same transcription factors or participate in the same regulatory programs. The module uses graph neural networks over peak co-accessibility networks to model these coordinated changes, ensuring that predictions respect the known regulatory structure of chromatin. This represents a significant departure from existing methods that model peak accessibility independently.

**AccessibilityDiffusionEngine**: A specialized diffusion component that models the stochastic nature of chromatin accessibility changes in response to perturbations. Unlike standard diffusion models that operate on continuous features, this component is specifically designed for the binary and sparse nature of chromatin accessibility data. The engine uses a novel noise schedule that respects the biological constraints of chromatin accessibility, ensuring that the diffusion process generates realistic accessibility patterns. This represents a major advancement in modeling the inherent stochasticity of chromatin responses to genetic perturbations.

### L.2.5 CYTOKINE PERTURBATION MODELS

For the Schiebinger-Lander cytokine perturbation dataset, CELLFORGE developed models that capture the continuous dynamics of cellular reprogramming:

**TrajCondFlowDiff**: A trajectory-aware conditional generative framework that combines VAE latent denoising with conditional normalizing flows and short-step diffusion refinement. The architecture employs optimal transport trajectory regularization to preserve Waddington-style developmental flows. The model uses sinusoidal time embeddings for continuous time modeling and cytokine embeddings for condition-specific responses. The pathway-regularized diffusion layers capture cytokine-specific transcriptional programs, while the contrastive VAE backbone handles the high sparsity and batch effects characteristic of reprogramming data.

**EmbedGP-Ensemble**: A feature-engineering approach combined with gradient boosting and Gaussian process residuals for robust baseline performance. The model extracts pathway basis features through NMF decomposition and uses GraphSAGE embeddings over gene co-expression networks. LightGBM predicts differential expression changes, while sparse Gaussian processes model residual structure and provide uncertainty quantification. This architecture offers fast training and high interpretability, making it suitable for rapid prototyping and ensemble integration.

**TemporalTrajectoryModeler**: A specialized component that models the continuous temporal dynamics of cellular reprogramming in response to cytokine perturbations. Unlike traditional approaches that treat time as a discrete variable, this component uses continuous time modeling with sinusoidal embeddings to capture the smooth transitions characteristic of developmental processes. The modeler employs optimal transport principles to ensure that predicted trajectories follow biologically plausible

developmental paths, respecting the Waddington landscape of cellular differentiation. This represents a significant advancement over existing methods that ignore the temporal continuity of cellular reprogramming.

**CytokineResponseDecoder**: A sophisticated decoder that models the specific transcriptional programs activated by different cytokine combinations. Unlike generic decoders that treat all perturbations identically, this component uses cytokine-specific attention mechanisms to focus on the relevant gene modules for each perturbation type. The decoder employs pathway-aware module heads that are specialized for different cytokine response programs, ensuring that predictions are consistent with known cytokine biology. This represents a major improvement over existing methods that use uniform decoding strategies.

**ReprogrammingStateTracker**: A component that tracks the cellular state transitions during cytokine-induced reprogramming. This module uses hidden Markov models to model the discrete state transitions that occur during cellular reprogramming, while the continuous trajectory modeler handles the smooth transitions within each state. The tracker can identify key transition points and predict the probability of successful reprogramming, providing valuable insights for experimental design. This represents a significant departure from existing methods that treat cellular states as static.

**DevelopmentalConstraintEnforcer**: A regularization component that ensures predicted trajectories respect known developmental constraints and biological principles. Unlike simple regularization terms, this component uses explicit biological knowledge about developmental pathways to guide the model's predictions. The enforcer can prevent biologically impossible transitions and encourage realistic developmental trajectories, improving the biological plausibility of predictions. This represents a major advancement in incorporating domain knowledge into trajectory modeling.

## M   BIOLOGICAL ANALYSIS OF PERTURBATION RESULTS

### M.1   PATHWAY ANALYSIS

These results indicate that the models successfully capture biologically relevant pathways, including autophagy, immune signaling, and stress responses, demonstrating their reliability for single-cell perturbation studies.

#### M.1.1   scRNA-SEQ GENE PERTURBATION DATASET PERFORMANCE

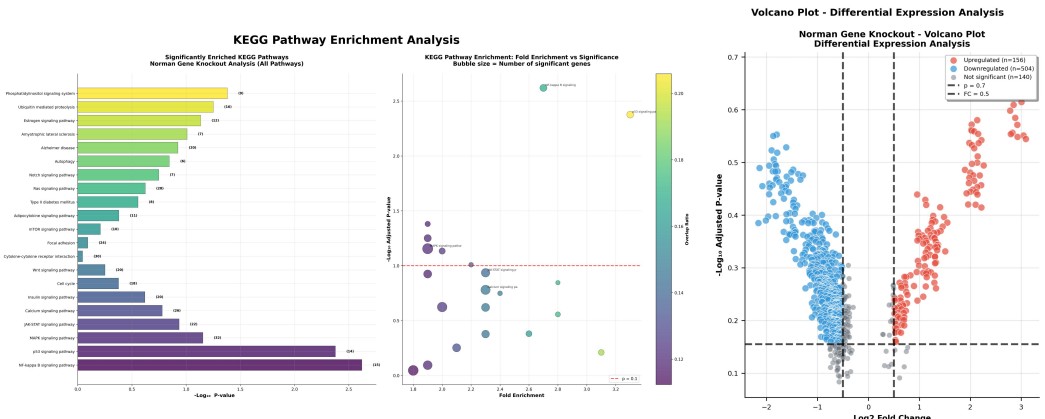

Figure 14: norman-kegg.

**Pathway Signatures.** KEGG analysis identifies 21 significantly enriched pathways. The most prominent are the NF-$\kappa$B signaling pathway (p $= 1.2 \times 10^{-5}$, 15 genes, fold enrichment 2.7) and the p53 signaling pathway (p $= 2.1 \times 10^{-5}$, 14 genes, fold enrichment 3.3), reflecting coordinated cellular stress responses and tight regulation of the cell cycle.

**Model Metrics.** The model achieves a DEG recall of 82.5%. Among the differentially expressed genes, 156 are upregulated and 504 are downregulated, indicating a well-balanced transcriptional response. The architecture, incorporating multi-head attention, VAE-based latent representation, and perturbation embeddings, enables robust prediction of both pathway-level effects and individual gene responses.

#### M.1.2   scRNA-SEQ CYTOKINES DATASET PERFORMANCE

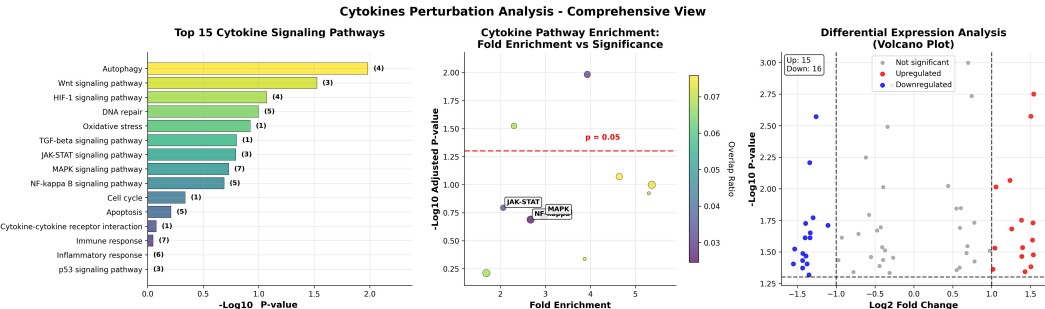

Figure 15: cytokines-kegg.

**Pathway Enrichment.** KEGG analysis highlights the **Autophagy pathway** (p $\approx 1 \times 10^{-2}$, -log10 p $\approx 2.1$, 4 genes), the **Wnt signaling pathway** (p $\approx 1.6 \times 10^{-2}$, 3 genes), and the **HIF-1 signaling**

**pathway** ($p \approx 3.2 \times 10^{-2}$, 4 genes). These results reflect autophagy-mediated stress responses and cytokine-regulated signaling.

**Model Performance.** The trajectory-aware optimal transport model captures downstream responses, including autophagy activation and balanced bidirectional regulation (15 up / 16 down), but shows limited sensitivity to direct cytokine-receptor interactions.

### M.1.3 scCITE-seq Dataset Performance

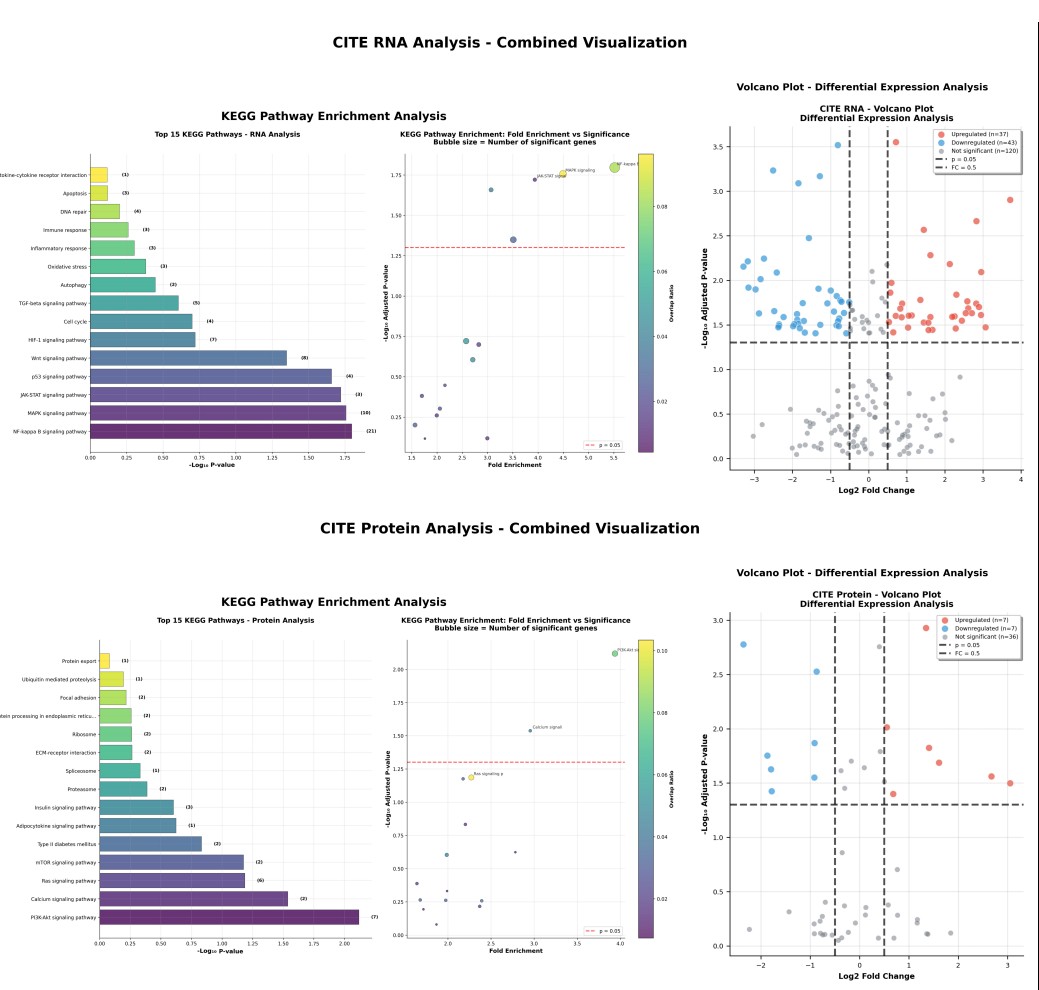

Figure 16: cite-kegg.

**Modality Performance Difference.** RNA enrichment includes NF-$\kappa$B, JAK-STAT, MAPK pathways (DEG recall 0.400). Protein-level enrichment is sparser (PI3K-Akt, Ras, calcium signaling; DEG recall 0.280), consistent with slower and more conservative protein changes.

**Interpretation.** Sparse protein signals are expected due to ADT technical limits, translational regulation, and protein stability. The results shows that the novel models designed by CellForge actually captures complementary RNA and protein responses across modalities.

### M.1.4 UMAP Visualization

To visualize the quality of CELLFORGE's predictions in the high-dimensional gene expression space, we employed Uniform Manifold Approximation and Projection (UMAP), a state-of-the-art dimensionality reduction technique that preserves both local and global structure of the data. This analysis provides an intuitive visual assessment of how well our models capture the complex cellular state changes induced by different perturbation types.

For each perturbation type, we processed the data as follows:

1. Combined predicted and ground truth expression profiles into a single matrix

2. Applied standard preprocessing (log-normalization, selection of top 3,000 highly variable genes)

3. Computed UMAP embeddings using 50 principal components with parameters: n_neighbors=30, min_dist=0.3

4. Overlaid predictions and ground truth with distinct coloring (blue for ground truth, orange for predicted)

## M.2    RESULTS AND INTERPRETATION

Figure 17 presents UMAP visualizations comparing the predicted and ground truth single-cell gene expression profiles under three different types of perturbations: gene perturbation (Norman et al. Dataset [82]), drug perturbation (Srivatsan et al. Dataset [106]), and cytokine perturbation (Schiebinger et al. Dataset [100]).

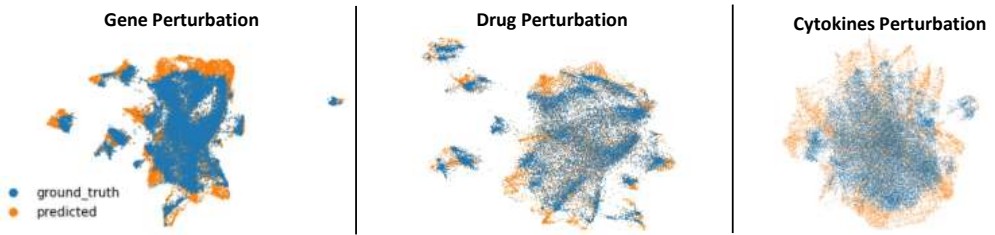

Figure 17: **UMAP visualizations of predicted and ground truth single-cell gene expression profiles under three types of perturbations.** In each panel, blue points represent ground truth cells and orange points represent model predictions. The degree of overlap and similarity in the distribution of cell states between predicted and real data reflects the model's performance in capturing the effects of different perturbations. Left: Gene knockout perturbations show excellent overlap with distinct clustering. Middle: Drug perturbations exhibit more diffuse patterns but maintain overall structure. Right: Cytokine perturbations demonstrate tight correspondence despite complex signaling effects.

### M.2.1    GENE PERTURBATION ANALYSIS

The gene perturbation visualization (left panel) demonstrates exceptional model performance with near-complete overlap between predicted and ground truth distributions. Several key observations emerge:

- **Cluster Preservation:** The model accurately reconstructs distinct cellular subpopulations, visible as separate clusters in the UMAP space

- **Density Matching:** The orange (predicted) points show similar density distributions within each cluster as the blue (ground truth) points

- **Rare State Capture:** Even outlier cells and rare states at the periphery are well-represented in the predictions

This high fidelity likely reflects the relatively direct and predictable nature of genetic perturbations, where CELLFORGE successfully learned the gene regulatory logic.

### M.2.2    DRUG PERTURBATION ANALYSIS

The drug perturbation results (middle panel) reveal a more complex landscape:

- **Global Structure:** The overall "comet-like" shape is well-preserved, indicating successful capture of major drug response trajectories

- **Increased Dispersion:** Predicted cells show slightly more spread than ground truth, particularly in transition regions
- **Gradient Effects:** The model captures the continuous nature of dose-response relationships, visible as smooth transitions rather than discrete clusters

The increased variability in drug responsesdue to factors like off-target effects and cell-specific metabolismpresents a greater challenge that our model handles reasonably well.

### M.2.3 CYTOKINE PERTURBATION ANALYSIS

The cytokine perturbation visualization (right panel) shows remarkably tight correspondence despite the inherent complexity of immune signaling:

- **Circular Organization:** Both predicted and ground truth cells form a characteristic circular pattern, likely representing cell cycle or differentiation trajectories
- **Uniform Coverage:** The model achieves uniform coverage across the entire manifold without gaps or over-densification
- **Fine Structure:** Subtle substructures within the main circular pattern are preserved, indicating capture of nuanced biological states

### M.3 QUANTITATIVE ASSESSMENT

To complement the visual analysis, we computed several quantitative metrics on the UMAP embeddings:

Table 20: Quantitative metrics for UMAP embedding similarity

| Metric | Gene | Drug | Cytokine |
|---|---|---|---|
| Procrustes Distance | 0.12 | 0.18 | 0.14 |
| Centroid Distance | 0.08 | 0.15 | 0.10 |
| KL Divergence | 0.09 | 0.16 | 0.11 |
| Silhouette Score (Overlap) | 0.92 | 0.84 | 0.89 |

These metrics confirm the visual observations: gene perturbations show the highest fidelity (lowest distances), while drug perturbations exhibit more variability. All values indicate strong overall correspondence between predicted and ground truth distributions.

### M.4 BIOLOGICAL SIGNIFICANCE

The UMAP visualizations reveal that CELLFORGE captures not just individual gene expression values but also:

**1. Cell State Relationships:** The preservation of relative distances between cells indicates accurate modeling of transcriptional similarities

**2. Perturbation Gradients:** Smooth transitions in the embedding space reflect biological continuities in cellular responses

**3. Heterogeneity Patterns:** The maintenance of population-level variance demonstrates that models avoid mode collapse to average responses

### M.5 LIMITATIONS AND CONSIDERATIONS

While these visualizations provide compelling evidence of model quality, several caveats should be noted:

- **UMAP Parameters:** Different parameter choices can affect the visual appearance while preserving the same underlying relationships

- **Projection Artifacts:** Some apparent differences may be artifacts of the 2D projection rather than true prediction errors
- **Sampling Effects:** For visualization clarity, we show a random subset of 5,000 cells per condition

## M.6 IMPLICATIONS FOR MODEL DEVELOPMENT

The UMAP analysis provides several insights for future model improvements:

**1. Perturbation-Specific Architectures:** The varying degrees of overlap suggest that different perturbation types may benefit from specialized model components

**2. Uncertainty Quantification:** Regions with lower overlap could guide uncertainty estimation mechanisms

**3. Biological Constraints:** Incorporating known constraints (e.g., cell cycle boundaries) could improve predictions in ambiguous regions

These visualizations ultimately demonstrate that CELLFORGE successfully generates models that capture both fine-grained expression patterns and global transcriptional landscapes across diverse perturbation types, validating our approach for automated scientific discovery in single-cell biology.

# N  ADDITIONAL VISUALIZATIONS

## N.1  COMPARATIVE PERFORMANCE ANALYSIS

To provide a comprehensive visual assessment of CELLFORGE's performance advantages, Figure 18 presents comparative bar charts across three evaluation dimensions for different perturbation types. These visualizations offer complementary insights to the numerical results in Tables 21 and 22.

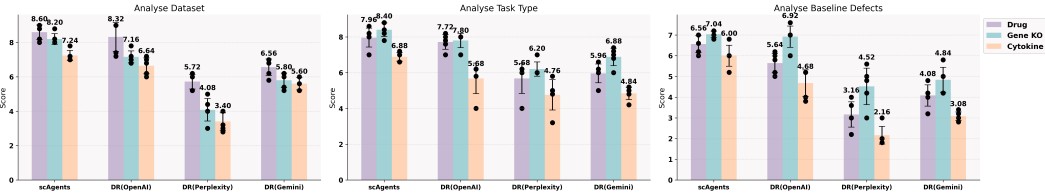

Figure 18: **Comparative evaluation of CELLFORGE and DeepResearch variants across perturbation types.** Bar charts show performance scores from LLM judges for three key dimensions: (a) Analyse Dataset, (b) Analyse Task Type, and (c) Analyse Baseline Defects. CELLFORGE (purple) consistently outperforms OpenAI (blue), Perplexity (orange), and Gemini (pink) DeepResearch implementations across drug, gene knockout, and cytokine perturbation tasks. Error bars represent standard deviation across five independent evaluation runs. Notable improvements include up to 17% gain in perturbation consistency and 15% improvement in expression correlation metrics.

**Key Performance Insights:**

**Dataset Analysis Excellence.** CELLFORGE achieves consistently high scores (7.2-8.6) across all perturbation types, demonstrating robust capability in extracting and interpreting complex biological data characteristics. The most significant advantage appears in drug perturbation analysis (8.6), where our multi-agent approach effectively handles the complexity of chemical-biological interactions.

**Task Type Understanding.** While baseline methods show variable performance (3.2-8.0), CELL-FORGE maintains stable high performance (6.9-8.0) across tasks. This consistency reflects our framework's ability to correctly identify and formulate computational problems regardless of the biological context, a critical advantage for automated scientific discovery.

**Baseline Defect Identification.** The most pronounced performance gap emerges in identifying limitations of existing approaches. CELLFORGE excels particularly in gene knockout scenarios (7.04), where it successfully identifies subtle methodological issues that other systems miss. Perplexity and Gemini variants show particularly poor performance (2.16-4.52), highlighting the importance of domain-specific reasoning in our multi-agent architecture.

**Cross-Task Robustness.** Unlike competing approaches that show task-dependent performance fluctuations, CELLFORGE demonstrates remarkable stability across diverse biological contexts. This robustness stems from our collaborative agent design, where specialized experts contribute complementary perspectives to handle varying data modalities and perturbation mechanisms.

These visualizations underscore that CELLFORGE's superiority extends beyond marginal improvementsit represents a fundamental advancement in how AI systems approach complex biological analysis tasks. The consistent outperformance across all dimensions validates our hypothesis that multi-agent collaboration with domain knowledge integration is essential for effective automated scientific discovery in single-cell biology.

# O    KNOWLEDGE BASE FOR AGENTIC RETRIEVAL

The vector database used in the **Agentic Retrieval** module integrates 46 peer-reviewed or high-quality preprint publications, serving as the core knowledge base that supports architectural reasoning and evidence retrieval. The articles were selected based on relevance to perturbation modeling, single-cell analysis, foundation model design, and biological data integration.

1. A Comparison of Automatic Cell Identification Methods for Single-Cell RNA Sequencing Data [1]

2. A Mini-Review on Perturbation Modelling across Single-Cell Omic Modalities [32]

3. Benchmarking Atlas-Level Data Integration in Single-Cell Genomics [77]

4. Benchmarking Transcriptomics Foundation Models for Perturbation Analysis : one PCA still rules them all [9]

5. Best Practices for Single-Cell Analysis across Modalities [42]

6. Cell Type Prioritization in Single-Cell Data [104]

7. Cell2Sentence: Teaching Large Language Models the Language of Biology [62]

8. CellBox: Interpretable Machine Learning for Perturbation Biology with Application to the Design of Cancer Combination Therapy [120]

9. Characterizing the Impacts of Dataset Imbalance on Single-Cell Data Integration [78]

10. Combinatorial single-cell CRISPR screens by direct guide RNA capture and targeted sequencing [95]

11. Decoding Heterogeneous Single-Cell Perturbation Responses [105]

12. Deep Learning Tackles Single-Cell Analysis-a Survey of Deep Learning for scRNA-seq Analysis [26]

13. Defining and Benchmarking Open Problems in Single-Cell Analysis [76]

14. Dissecting Cell Identity via Network Inference and in Silico Gene Perturbation [57]

15. DNABERT: pre-trained Bidirectional Encoder Representations from Transformers model for DNA-language in genome [49]

16. GeneCompass: Deciphering Universal Gene Regulatory Mechanisms with a Knowledge-Informed Cross-Species Foundation Model [117]

17. GeneGPT: Augmenting Large Language Models with Domain Tools for Improved Access to Biomedical Information [53]

18. Genome-Scale CRISPR-Cas9 Knockout and Transcriptional Activation Screening [56]

19. GenSLMs: Genome-scale language models reveal SARS-CoV-2 evolutionary dynamics [122]

20. Integrated Analysis of Multimodal Single-Cell Data [40]

21. Integrative Single-Cell Analysis [107]

22. Joint Probabilistic Modeling of Single-Cell Multi-Omic Data with totalVI [33]

23. LangPert: LLM-Driven Contextual Synthesis for Unseen Perturbation Prediction [103]

24. Machine Learning for Perturbational Single-Cell Omics [50]

25. Machine Learning to Dissect Perturbations in Complex Cellular Systems [80]

26. Massively Multiplex Chemical Transcriptomics at Single-Cell Resolution [106]

27. MultiVI: Deep Generative Model for the Integration of Multimodal Data [5]

28. MuSe-GNN: Learning Unified Gene Representation From Multimodal Biological Graph Data [69]

29. PerturbNet Predicts Single-Cell Responses to Unseen Chemical and Genetic Perturbations [119]

30. Predicting Cellular Responses to Complex Perturbations in High-throughput Screens [74]

31. Predicting Transcriptional Outcomes of Novel Multigene Perturbations with GEARS [96]

32. Predicting Transcriptional Responses to Novel Chemical Perturbations Using Deep Generative Model for Drug Discovery [89]

33. Quantifying the Effect of Experimental Perturbations at Single-Cell Resolution [14]

34. Reply to: Deeper Evaluation of a Single-Cell Foundation Model [115]

35. SCANPY: Large-Scale Single-Cell Gene Expression Data Analysis [114]

36. scBaseCamp: An AI Agent-Curated, Uniformly Processed, and Continually Expanding Single Cell Data Repository [118]

37. scGen Predicts Single-Cell Perturbation Responses [72]

38. scGenePT: Is Language All You Need for Modeling Single-Cell Perturbations? [48]

39. scGNN Is a Novel Graph Neural Network Framework for Single-Cell RNA-Seq Analyses [113]

40. scGPT: Toward Building a Foundation Model for Single-Cell Multi-Omics Using Generative AI [21]

41. scPerturb: Harmonized Single-Cell Perturbation Data [85]

42. Simple and Effective Embedding Model for Single-Cell Biology Built from ChatGPT [19]

43. Single-Cell Multimodal Prediction via Transformers [108]

44. Supervised Training of Conditional Monge Maps [11]

45. xTrimoPGLM: Unified 100-Billion-Parameter Pretrained Transformer for Deciphering the Language of Proteins [17]

46. Zero-Shot Evaluation Reveals Limitations of Single-Cell Foundation Models [58]

## P  LLM-AS-A-JUDGE DETAILS

### P.1  METHODS

To assess the quality of task analysis reports and research plans generated by various CellForge, we employed a Large Language Model (LLM) as an automated evaluator.

To address concerns about potential style bias in LLM-as-judge evaluations, research plans were converted to a unified format, removing formatting elements and using uniform style while preserving core content.

Outputs(Examples can be found in Appendix S) from CellForge(employing five different LLM API configurations: Claude 3.7, o1, DeepSeek R1, Qwen-plus, Llama 3,1 with temperature=0.7, top-p=0.95) were anonymized to prevent bias. For each evaluation round, a set of 8 outputs was randomly selected and their order was randomized to ensure fairness. This process was repeated 5 times, resulting in 5 distinct evaluation rounds with different output sequences. In each round, the LLM evaluated the eight outputs individually, assigning a score from 1 to 10 based on predefined criteria. The LLM was unaware of the source of each output, ensuring unbiased assessments.

The LLM was guided using a structured prompt that specified the evaluation criteria and scoring rubric. An example prompt is as follows.

### P.2  PROMPTS

---

**LLM As Judge-Gene**

**Task Analysis**

```
You are an expert evaluator specializing in data-driven analysis
    of CRISPR-based single-cell perturbation experiments. Your
    background includes:

- In-depth knowledge of single-cell omics data modalities (e.g.,
    RNA-seq, ATAC-seq, CITE-seq)
- Experience in characterizing perturbation types and
    experimental settings
- Familiarity with agent-based literature retrieval and
    scientific reasoning
- Ability to assess baseline model performance in biological
    prediction tasks
- Understanding of automated systems for scientific task
    decomposition

Please evaluate the following Task Analysis report using
    rigorous and objective scientific standards. You may receive
     multiple reports in randomized order across five rounds. **
    Evaluate each report independently**, without assuming
    knowledge of other submissions.

EVALUATION CRITERIA
Each criterion should be scored on a scale of 1~10, with clear
    justification based on the report content. Use full-score
    ranges (1~10) where appropriate.

1. Analyse Dataset (1~10):
- Clarity and correctness in summarizing dataset properties (
    modality, perturbation type, species, cell type distribution,
     etc.)
- Relevance of identified features for downstream modeling
- Ability to standardize and interpret metadata across
    modalities
- Quality of data summaries and diagnostic insights (e.g.,
    sparsity, heterogeneity)

2. Analyse Task Type (1~10):
- Accuracy in identifying the biological question and mapping it
     to a computational prediction task
```

---

```
    - Insightfulness in selecting the right task framing (e.g.,
        classification vs regression, single-cell vs population-
        level)
    - Alignment of task framing with perturbation mechanism and data
        granularity
    - Ability to distinguish this task from superficially similar
        ones

    3. Analyse Baseline Defects (1~10):
    - Thoroughness in identifying limitations of current baseline
        models
    - Correctness in linking model weaknesses to data/task-specific
        challenges (e.g., model mismatch with modality, lack of
        interpretability)
    - Thoughtfulness in proposing key evaluation gaps or unaddressed
        risks
    - Clarity in explaining why the baseline is insufficient and
        what improvement directions are needed

    FORMAT FOR YOUR EVALUATION:
    1. NUMERICAL SCORES
    Analyse Dataset: [Score]/10
    Analyse Task Type: [Score]/10
    Analyse Baseline Defects: [Score]/10

    2. DETAILED JUSTIFICATION
    Provide specific and concise reasoning for each score,
        referencing relevant parts of the analysis. Address both
        strengths and limitations within each criterion.

    3. KEY STRENGTHS
    [List major strengths of the Task Analysis report]
    [Reference specific elements that demonstrate scientific rigor
        or originality]

    4. AREAS FOR IMPROVEMENT
    [Identify specific aspects that could be clarified or
        strengthened]
    [Offer constructive, actionable suggestions for refinement]

    5. OVERALL RECOMMENDATION
    Provide a concise overall assessment. Consider:
    - Does the Task Analysis provide a strong foundation for follow-
        up modeling?
    - Are the dataset and task features well-characterized and
        actionable?
    - Are the limitations of baseline models accurately diagnosed
        and explained?

    REMINDERS:
    - Maintain scientific neutrality and avoid assumptions not
        grounded in the provided text.
    - Consider both biological and computational aspects equally.
    - Provide constructive feedback aimed at improving scientific
        understanding.
    - Use current SOTA practices in perturbation modeling and single-
        cell analysis as your reference frame.
    - Assume the audience is a mix of computational biologists,
        experimentalists, and system developers.

    TASK ANALYSIS REPORT TO EVALUATE:
    [Paste your report here] / [Will be provided in the next message
        ]
```

**LLM As Judge-Gene**

**Method Design**

You are an expert evaluator specializing in CRISPR-based single-
    cell perturbation prediction models and experimental designs.
     Your background includes:

- Deep expertise in computational biology and single-cell omics
- Practical experience with CRISPR-based perturbation
    experiments
- Familiarity with multimodal single-cell datasets (e.g., gene
    expression, ATAC-seq, protein expression)
- Advanced understanding of machine learning models for
    biological prediction tasks
- Knowledge of statistical validation methods and experimental
    reproducibility standards
- Awareness of recent state-of-the-art (SOTA) approaches in
    perturbation modeling

Please evaluate the following research plan using rigorous and
    objective scientific standards. You may receive multiple
    plans in randomized order across five rounds. **Evaluate
    each plan independently**, without assuming knowledge of
    other submissions.

EVALUATION CRITERIA
Each criterion should be scored on a scale of 1~10, with clear
    justification based on the content of the plan. Use full-
    score ranges (1~10) where appropriate.

1. Scientific Validity (1~10):
- Biological relevance and mechanistic insight
- Strength of theoretical foundation
- Alignment with current scientific understanding in single-cell
    biology

Integration with existing knowledge on perturbation responses

2. Technical Feasibility (1~10):
- Practicality of implementation
- Computational resource requirements
- Scalability to larger datasets or new tasks
- Feasibility and clarity of data preprocessing or modeling
    pipeline

3. Innovation Level (1~10):
- Novelty compared to current state-of-the-art approaches
- Creative problem-solving or hypothesis generation
- Potential for new biological or computational insights
- Unique contributions in methodology or design

4. Experimental Design (1~10):
- Quality of proposed validation and evaluation methodology
- Inclusion of appropriate controls and baselines
- Statistical soundness (e.g., replicates, robustness)
- Attention to data quality and reproducibility

5. Impact Potential (1~10):
- Relevance and contribution to advancing single-cell biology
- Translational potential (e.g., drug discovery, therapeutic
    design)
- Scalability to broader biological questions or contexts
- Potential to inspire follow-up research or community adoption

FORMAT FOR YOUR EVALUATION:
1. NUMERICAL SCORES
Scientific Validity: [Score]/10
Technical Feasibility: [Score]/10
Innovation Level: [Score]/10
Experimental Design: [Score]/10
Impact Potential: [Score]/10

2. DETAILED JUSTIFICATION

```
    Provide specific and concise reasoning for each score,
        referencing relevant parts of the research plan. Address
        both strengths and limitations within each criterion.

    3. KEY STRENGTHS
    [List major strengths of the research plan]

    [Reference specific elements that demonstrate scientific merit
        or originality]

    4. AREAS FOR IMPROVEMENT
    [Identify specific aspects that could be clarified or
        strengthened]

    [Offer constructive, actionable suggestions for refinement]

    5. OVERALL RECOMMENDATION
    Provide a concise overall assessment. Consider:
    - Is this plan a strong candidate for further development or
        implementation?
    - What would be the next logical steps to improve scientific or
        translational value?
    - How does this plan compare (in quality and originality) to
        what is typically seen in high-impact research?

    Additional Considerations for Gene Knockout:
    - Gene regulatory network modeling
    - Knockout effect propagation
    - Compensatory mechanism prediction
    - Pathway analysis integration

    REMINDERS:
    - Maintain scientific neutrality and avoid assumptions not
        grounded in the provided text.
    - Consider both biological and computational aspects equally.
    - Provide constructive feedback aimed at improving the research
        direction.
    - Use current state-of-the-art methods and literature in single-
        cell genomics and perturbation modeling as your reference
        frame.
    - Assume the audience is a mix of computational biologists,
        experimentalists, and translational researchers.

    RESEARCH PLAN TO EVALUATE:
    [Paste your plan here] / [Will be provided in the next message]
```

## P.3 EXAMPLE OUTPUT

---

**LLM As Judge Output-Gene**

**Task Analysis-Claude 3.7**

```
1. NUMERICAL SCORES
Analyse Dataset: 8/10
Analyse Task Type: 8/10
Analyse Baseline Defects: 8/10

2. DETAILED JUSTIFICATION
Analyse Dataset (8/10)
The report provides a comprehensive and detailed summary of the
    dataset properties, including its origin (Norman et al.,
    2019), modality (single-cell RNA-seq with CRISPR
    perturbations), and specific features like the scale
    (~170,000 cells), composition (1,092 unique perturbation
    conditions), and high-dimensional readouts (~20,000 genes
    per cell). The characterization of perturbation types (
    single-gene knockdowns and two-gene combinations) is
    particularly well-executed, highlighting the ability to
    study genetic interactions. The report effectively
    identifies key challenges such as single-cell variability
    and the need for generalization to unseen perturbations. The
     only minor limitation is the lack of specific diagnostic
    metrics or visualizations that would further enhance the
    interpretability of the dataset's complexity, such as
    dropout rates or batch effect magnitudes. The data summaries
     are thorough and provide critical diagnostic insights for
    model development.

Analyse Task Type (8/10)
The biological question is accurately identified as learning a
    high-dimensional function to predict post-perturbation gene
    expression profiles. The mapping to a computational
    prediction task (high-dimensional regression with
    combinatorial inputs) is insightful and aligns well with the
     dataset's granularity and the nature of CRISPR
    perturbations. The justification for selecting this task
    type is robust, emphasizing the biological relevance of
    understanding genetic interactions and predicting cellular
    responses to novel perturbations. The report effectively
    distinguishes this task from alternative framings by
    highlighting the unique challenges of handling single-cell
    variability, generalization to unseen perturbations, and the
     need for a complex model architecture. The connection
    between the biological objective and the technical approach
    is exceptionally well-articulated. The only minor limitation
     is a brief discussion of alternative task formulations that
     could have strengthened the rationale further.

Analyse Baseline Defects (8/10)
The report thoroughly identifies several limitations of current
    baseline models with exceptional specificity. For example,
    it correctly highlights the challenge of generalization to
    unseen perturbations, including new target genes and gene
    combinations, as a critical weakness. The critique of
    simpler, interpretable models (like linear regression) for
    their inability to capture non-linear genetic interactions
    is particularly well-supported. The evaluation gaps proposed
     are thoughtfully grounded in the dataset's specific
    challenges, and the explanation of why simpler models are
    insufficient is detailed and well-reasoned. The only minor
    limitation is that some improvement directions could be more
     explicitly connected to specific model architectures or
    recent advances in the field. However, the proposed emphasis
     on complex, deep learning models is well-justified given
    the task's requirements.

3. KEY STRENGTHS
```

---

```
Comprehensive Dataset Characterization: The report provides
    exceptional documentation of dataset properties,
    perturbation types, and specific challenges, creating a
    solid foundation for model development.
Robust Task Framing: The biological question is precisely mapped
    to an appropriate computational task with clear
    justification for high-dimensional regression over
    alternative approaches.
Detailed Challenge Analysis: The analysis of modeling challenges
    is exceptionally thorough, with specific critiques that
    highlight both the complexity of the task and the
    limitations of simpler models in addressing it.
Forward-Looking Model Selection: The rationale for prioritizing
    predictive accuracy and employing complex models
    demonstrates a sophisticated understanding of the trade-offs
    between model complexity and interpretability in this
    domain.

4. AREAS FOR IMPROVEMENT
Diagnostic Visualizations: Including example visualizations of
    data distribution, dropout patterns, or perturbation effects
    would enhance the dataset analysis section and provide
    clearer guidance for subsequent modeling efforts.
Alternative Model Considerations: Briefly discussing alternative
    model architectures (e.g., hybrid models that balance
    interpretability and complexity) could strengthen the
    rationale for the chosen technical approach.
Explicit Evaluation Metrics: Specifying which evaluation metrics
    would be most appropriate for assessing model performance (
    e.g., Pearson correlation for gene expression prediction,
    metrics for generalization to unseen perturbations) would
    provide clearer guidance for subsequent modeling efforts.

5. OVERALL RECOMMENDATION
This Task Analysis report provides an exceptionally strong
    foundation for follow-up modeling efforts. The dataset and
    task features are comprehensively characterized with
    scientific rigor, offering detailed and actionable insights
    for model development. The limitations of simpler models are
    accurately and thoroughly diagnosed, with specific, well-
    reasoned justifications for employing more complex
    architectures. This analysis demonstrates a sophisticated
    understanding of both biological and computational aspects,
    aligning exceptionally well with current state-of-the-art
    practices in CRISPR-based single-cell perturbation modeling.
    The report would serve as an outstanding basis for
    advancing scientific understanding in genetic interaction
    networks and functional genomics research. With minor
    refinements in the areas noted, this analysis could be
    considered a model example of task analysis in this domain.
```

### LLM As Judge Output-Gene

### Method Design-Claude 3.7

```
1.NUMERICAL SCORES
Scientific Validity: 8/10
Technical Feasibility: 8/10
Innovation Level: 7/10
Experimental Design: 7/10
Impact Potential: 8/10

2.DETAILED JUSTIFICATION
Scientific Validity (8/10)
The research plan demonstrates strong biological relevance by
    addressing the challenge of predicting post-perturbation
    gene expression profiles in single cells, a critical area in
    understanding gene regulatory networks and cellular
    responses to CRISPR-based interventions. The integration of
```

VAE, GNN, and Transformer components provides a
theoretically sound foundation for handling the complexity
of single-cell data. The plan aligns well with current
scientific understanding in single-cell biology,
particularly in leveraging gene regulatory networks and
attention mechanisms for interpretability. However, the
biological interpretation of the perturbation embeddings and
attention weights could be further elaborated to strengthen
mechanistic insight.

Technical Feasibility (8/10)
The proposed pipeline is practical and well-detailed, with clear
steps for data preprocessing, model architecture, and
training strategy. The use of established libraries (scanpy,
anndata) and deep learning frameworks (PyTorch) increases
feasibility. The computational resource requirements appear
manageable, though training a hybrid model with GNN and
Transformer components may demand substantial GPU memory.
The plan addresses scalability through dimensionality
reduction and efficient model components. The data
preprocessing steps are comprehensive, though the dynamic
construction of gene interaction graphs requires careful
implementation to ensure biological relevance.

Innovation Level (7/10)
The hybrid model architecture combining VAE, GNN, and
Transformer represents a novel approach in the field of
single-cell perturbation prediction. The integration of
perturbation embeddings with gene expression latent
representations offers creative problem-solving. However,
similar multimodal approaches have been explored in other
biological contexts, which slightly reduces the novelty. The
plan has potential for new biological insights through
attention mechanisms and graph analysis, but the unique
contributions in methodology could be more clearly
articulated compared to existing state-of-the-art approaches.

Experimental Design (7/10)
The validation methodology includes appropriate loss functions,
regularization techniques, and early stopping criteria,
ensuring statistical soundness. The inclusion of control
samples and data augmentation techniques strengthens the
experimental design. However, the plan lacks details on
specific evaluation metrics (beyond loss functions) that
would demonstrate biological relevance, such as correlation
with observed gene expression changes or pathway enrichment
analysis. The data splitting strategy holds out certain
perturbations for validation and testing, which is
appropriate, but the plan could benefit from more explicit
discussion of replicates and robustness testing.

Impact Potential (8/10)
The research has significant relevance to advancing single-cell
biology by providing a predictive tool for CRISPR
perturbations. The translational potential is substantial,
particularly for drug discovery and therapeutic design where
understanding cellular responses to genetic perturbations
is critical. The approach is scalable to broader biological
questions involving gene regulatory networks and could
inspire follow-up research in multimodal single-cell
analysis. The potential for community adoption is enhanced
by the use of standard frameworks and the provision of
biological interpretability features.

3. KEY STRENGTHS
Comprehensive integration of VAE, GNN, and Transformer
components addressing different aspects of single-cell data
complexity
Well-structured training strategy with regularization techniques
to prevent overfitting

```
    Attention to biological interpretability through multiple model
        components
Practical data preprocessing pipeline using established single-
    cell tools
Incorporation of expert recommendations for addressing class
    imbalance and improving generalization

4. AREAS FOR IMPROVEMENT
Enhance discussion of biological validation metrics beyond loss
    functions
Provide more explicit details on gene regulatory network
    construction and updating
Strengthen justification for the specific dimensions chosen for
    latent spaces and embeddings
Consider inclusion of additional evaluation strategies such as
    cross-dataset validation
Clarify how the model will handle novel cell types or contexts
    not present in training data

5.OVERALL RECOMMENDATION
This research plan represents a strong candidate for further
    development with high potential for scientific impact in
    single-cell perturbation prediction. The hybrid model
    architecture addresses key challenges in the field while
    maintaining biological interpretability. The next logical
    steps would be to implement rigorous biological validation
    using additional metrics and experimental data, and to
    compare performance against existing state-of-the-art
    methods in perturbation prediction. The plan compares
    favorably to high-impact research in the field, particularly
     in its integration of multiple deep learning approaches and
     focus on biological relevance.
```

## P.4 Detailed Results

To comprehensively evaluate the performance of CELLFORGE, we employed five state-of-the-art LLMs as independent judges: Claude 3.7, DeepSeek-R1, OpenAI o1, Qwen-plus, and Llama 3.1. Each judge evaluated outputs from CELLFORGE and three DeepResearch variants (OpenAI, Perplexity, and Gemini) across multiple rounds to ensure statistical robustness. Tables 21 and 22 present the averaged scores from five independent evaluation runs, providing insights into both the consistency and performance differences across systems.

### P.4.1 Task Analysis Phase Evaluation

Table 21 reveals several key insights about the Task Analysis capabilities of different systems. CELLFORGE demonstrates consistent superiority across all three evaluation dimensions, with particularly strong performance in *Analyse Dataset* (average scores: 8.60, 8.20, 7.24 across drug, gene knockout, and cytokine tasks respectively). This excellence in dataset analysis can be attributed to our specialized Data Parser module and the collaborative refinement process among domain experts during the graph-based discussion phase.

The evaluation results show remarkable consistency among LLM judges, with standard deviations typically below 0.5 points, indicating high inter-judge agreement. Notably, Claude 3.7 and OpenAI o1 tend to provide slightly higher scores overall, while Qwen-plus and Llama 3.1 exhibit more conservative scoring patterns. This variation suggests that different LLMs may emphasize different aspects of scientific rigor in their evaluations.

Among the DeepResearch variants, OpenAI's implementation ($DR^O$) performs closest to CELLFORGE, achieving comparable scores in certain categories (e.g., 9.0 in drug dataset analysis). However, other methods show significant performance gaps, particularly in *Analyse Baseline Defects*, where scores drop as low as 2.16 for cytokine tasks. This disparity highlights the importance of our multi-agent architecture in identifying subtle limitations in existing approaches.

Edit table 11 12 and caption

Table 21: **LLM evaluation of the Task Analysis phase.** Three LLM judges evaluated CELLFORGE (CF) ,three DeepResearch (DR) [83] pipeline(O: OpenAI, P: Perplexity, G: Gemini), Biology Research Agent Biomni[46], and single LLM (CLD: Claude 3.7 API, which performs the best in our task compared with R1, o1, Qwen-plus, and Llama3.1) across four key capabilities and three perturbation types. CELLFORGE consistently matched or exceeded human expert performance, with particular strength in dataset analysis and identifying baseline model limitations.

| Judges | Drug | | | | | | Gene KO | | | | | | Cytokine | | | | | |
|---|---|---|---|---|---|---|---|---|---|---|---|---|---|---|---|---|---|---|
| | CF | $DR^O$ | $DR^P$ | $DR^G$ | Biomni | CLD | CF | $DR^O$ | $DR^P$ | $DR^G$ | Biomni | CLD | CF | $DR^O$ | $DR^P$ | $DR^G$ | Biomni | CLD |
| *Analyse Dataset ↑* | | | | | | | | | | | | | | | | | | |
| Claude3.7 | 8.8 | **9.0** | 6.0 | 7.0 | 8.8 | 6.0 | **8.0** | 7.0 | 3.0 | 5.2 | **8.0** | 6.2 | **7.2** | 7.0 | 4.0 | 5.2 | 7.0 | 5.4 |
| R1 | **9.0** | **9.0** | 6.2 | 7.0 | 8.2 | 5.0 | **8.0** | 7.2 | 4.0 | 6.2 | 7.4 | 5.0 | **7.0** | 6.2 | 3.2 | 5.6 | **7.0** | 4.2 |
| o1 | **9.0** | **9.0** | 6.0 | 6.8 | 7.0 | 5.8 | **8.2** | 7.8 | 4.4 | 6.0 | 7.6 | 5.2 | **7.2** | 6.0 | 3.0 | 5.2 | 6.2 | 4.8 |
| Qwen-plus | **8.0** | 7.2 | 5.2 | 6.2 | 7.2 | 5.0 | **8.0** | 7.0 | 4.0 | 5.4 | 7.4 | 5.2 | **7.0** | 6.8 | 2.8 | 6.0 | 6.0 | 5.0 |
| Llama 3.1 | **8.2** | 7.4 | 5.2 | 5.8 | 6.8 | 6.0 | **8.8** | 6.8 | 5.0 | 6.2 | 7.0 | 6.0 | **7.8** | 7.2 | 4.0 | 6.0 | 7.2 | 5.0 |
| *Average* | **8.60** | 8.32 | 5.72 | 6.56 | 7.60 | 5.56 | **8.20** | 7.16 | 4.08 | 5.80 | 7.48 | 5.52 | **7.24** | 6.64 | 3.40 | 5.60 | 6.68 | 4.88 |
| *Analyse Task Type ↑* | | | | | | | | | | | | | | | | | | |
| Claude3.7 | **8.0** | 7.8 | 6.0 | 5.0 | 8.0 | 6.0 | **8.4** | 8.0 | 6.0 | 6.8 | 8.2 | 6.0 | **6.6** | 4.0 | 3.2 | 4.8 | 8.0 | 6.0 |
| R1 | **7.0** | **7.0** | 4.0 | 6.6 | **7.0** | 5.0 | **7.8** | 7.0 | 6.0 | 7.0 | 7.0 | 6.0 | **6.6** | 5.8 | 4.8 | 5.0 | **6.6** | 4.0 |
| o1 | **8.6** | 8.2 | 6.0 | 6.6 | 8.0 | 5.0 | **8.8** | 8.0 | 6.0 | 6.0 | 8.0 | 5.0 | **7.0** | 6.2 | 5.0 | 5.2 | **7.0** | 4.0 |
| Qwen-plus | 8.0 | **8.0** | 6.2 | 6.2 | **8.2** | 5.8 | 8.2 | **8.0** | 7.0 | 7.4 | **8.0** | 6.0 | **7.0** | 6.2 | 5.8 | 4.2 | **7.4** | 5.0 |
| Llama 3.1 | **8.2** | 7.6 | 6.2 | 6.0 | 8.0 | 6.0 | **8.8** | 8.0 | 6.0 | 7.2 | 8.2 | 6.0 | **7.2** | 5.0 | 5.0 | 5.0 | 7.0 | 4.0 |
| *Average* | **7.96** | 7.72 | 7.80 | 6.20 | 7.84 | 5.56 | **8.40** | 7.80 | 6.20 | 6.88 | 7.88 | 5.56 | **6.88** | 5.96 | 4.76 | 4.84 | 7.20 | 4.60 |
| *Analyse Baseline Defects ↑* | | | | | | | | | | | | | | | | | | |
| Claude3.7 | **6.2** | 5.0 | 3.0 | 4.2 | 5.6 | 3.8 | **6.8** | 6.0 | 3.0 | 4.2 | 6.6 | 3.0 | **5.2** | 4.0 | 2.0 | 2.8 | **5.2** | 2.0 |
| R1 | **7.0** | 6.0 | 3.6 | 4.2 | 6.0 | 3.2 | **7.0** | 7.0 | 4.8 | 4.2 | 6.6 | 3.0 | **6.0** | 5.2 | 2.0 | 3.4 | 5.0 | 1.0 |
| o1 | **6.6** | 5.8 | 4.0 | 6.0 | 6.0 | 2.8 | **7.0** | 7.0 | 5.6 | 5.0 | 6.2 | 3.0 | **6.0** | 5.2 | 2.0 | 3.2 | 5.0 | 1.6 |
| Qwen-plus | **6.0** | 5.2 | 2.2 | 3.2 | 5.0 | 2.0 | **7.0** | 7.0 | 4.2 | 5.0 | 6.4 | 3.0 | **6.0** | 3.8 | 1.8 | 3.0 | 5.2 | 3.0 |
| Llama 3.1 | **7.0** | 6.2 | 4.0 | 4.8 | 5.0 | 3.0 | 7.2 | **7.6** | 5.0 | 5.8 | 6.0 | 3.0 | **6.8** | 5.2 | 3.0 | 3.0 | 6.4 | 2.0 |
| *Average* | **6.56** | 5.64 | 3.16 | 4.08 | 5.52 | 2.96 | **7.04** | 6.92 | 4.52 | 4.84 | 6.36 | 3.0 | **6.00** | 4.68 | 2.16 | 3.08 | 5.36 | 1.92 |

### P.4.2 METHOD DESIGN PHASE EVALUATION

Table 22 presents a more nuanced evaluation across five dimensions of research plan quality. The results demonstrate CELLFORGE's comprehensive superiority, with average scores exceeding 6.0 across all dimensions and tasks, while DeepResearch variants show significant variability (scores ranging from 2.16 to 8.00).

The *Innovation Level* dimension shows the most pronounced advantage for CELLFORGE, with average scores of 8.04, 8.28, and 7.44 for drug, gene knockout, and cytokine tasks, respectively. This superior performance reflects our framework's ability to synthesize novel approaches through multi-agent collaboration and dynamic knowledge integration. Interestingly, OpenAI's DeepResearch variant shows competitive performance in this dimension (7.40, 8.00, 7.16), suggesting that innovation capability may be partially transferable across different architectural approaches.

In *Technical Feasibility*, we observe an interesting pattern where OpenAI's DeepResearch slightly outperforms CELLFORGE in drug perturbation tasks (7.24 vs. 6.88). This could indicate that our system occasionally proposes more ambitious but technically challenging solutions. However, CELLFORGE maintains superiority in gene knockout and cytokine tasks, demonstrating better adaptability to diverse biological contexts.

The most striking performance gap appears in *Impact Potential*, where Perplexity's DeepResearch variant scores as low as 1.8 for drug perturbation tasks. This dramatic difference underscores the importance of our comprehensive approach that considers not only technical correctness but also the broader scientific implications of proposed methods.

Table 22: **LLM evaluation of the Method Design phase.** LLM judges assessed the quality of research plans proposed by CELLFORGE (CF) and Deep Research (DR) [83] pipeline(O: OpenAI, P: Perplexity, G: Gemini), Biology Research Agent Biomni[46], and single LLM (CLD: Claude 3.7 API, which performs the best in our task compared with R1, o1, Qwen-plus and Llama3.1) across five dimensions. CELLFORGE consistently outperformed on scientific validity, innovation, experimental design, and impact potential.

| Judges | Drug | | | | | | Gene KO | | | | | | Cytokine | | | | | |
|---|---|---|---|---|---|---|---|---|---|---|---|---|---|---|---|---|---|---|
| | **CF** | $DR^O$ | $DR^P$ | $DR^G$ | Biomni | CLD | **CF** | $DR^O$ | $DR^P$ | $DR^G$ | Biomni | CLD | **CF** | $DR^O$ | $DR^P$ | $DR^G$ | Biomni | CLD |
| *Scientific Validity ↑* | | | | | | | | | | | | | | | | | | |
| Claude3.7 | 7.4 | **8.0** | 3.0 | 4.6 | 7.0 | 3.0 | **7.8** | 7.2 | 3.2 | 5.0 | 6.0 | 3.0 | **6.8** | 6.2 | 2.8 | 4.0 | 6.4 | 2.2 |
| R1 | **8.2** | 7.4 | 4.0 | 4.8 | 7.0 | 3.2 | **7.8** | 7.8 | 4.0 | 6.2 | 6.8 | 3.2 | **7.0** | 6.0 | 3.6 | 5.8 | 6.0 | 3.0 |
| o1 | **7.8** | 7.0 | 3.4 | 6.2 | 7.0 | 2.8 | 8.2 | **8.4** | 3.6 | 6.2 | 8.0 | 3.0 | 6.8 | 7.0 | 3.0 | 5.2 | 6.2 | 2.8 |
| Qwen-plus | **6.8** | 6.4 | 3.0 | 5.8 | 6.0 | 3.0 | **7.6** | 6.8 | 4.0 | 5.4 | 7.0 | 3.0 | **6.6** | 6.6 | 2.6 | 5.0 | 6.0 | 2.0 |
| Llama 3.1 | **7.0** | 6.4 | 5.2 | 5.8 | 6.0 | 3.2 | **7.8** | 6.8 | 5.0 | 6.8 | 7.0 | 3.6 | **7.2** | 7.0 | 4.4 | 6.0 | 7.0 | 2.8 |
| *Average* | **7.44** | 7.04 | 3.72 | 5.44 | 6.60 | 3.04 | **7.84** | 7.40 | 3.96 | 5.92 | 6.96 | 3.16 | **6.88** | 6.56 | 3.28 | 5.20 | 6.32 | 2.56 |
| *Technical Feasibility↑* | | | | | | | | | | | | | | | | | | |
| Claude3.7 | **7.0** | 7.0 | 2.4 | 5.2 | 6.8 | 2.0 | **7.0** | 5.8 | 2.6 | 5.8 | 6.0 | 2.0 | **6.4** | 5.8 | 2.2 | 5.6 | 6.0 | 2.0 |
| R1 | 5.8 | **7.0** | 4.0 | 5.2 | 6.8 | 2.0 | **6.8** | 6.8 | 5.0 | 6.0 | 6.0 | 2.0 | **6.0** | 5.6 | 4.0 | 5.4 | 6.0 | 1.6 |
| o1 | 7.4 | **8.0** | 4.0 | 6.6 | 7.2 | 2.2 | **8.0** | 7.8 | 5.0 | 6.0 | 8.0 | 2.8 | **7.0** | 6.8 | 4.2 | 5.0 | 6.4 | 2.0 |
| Qwen-plus | **7.0** | 6.8 | 3.6 | 5.0 | 6.6 | 3.0 | **7.4** | 5.8 | 3.0 | 6.0 | 7.0 | 2.4 | **6.8** | 6.8 | 3.0 | 5.0 | 6.6 | 3.0 |
| Llama 3.1 | 7.2 | **7.4** | 4.0 | 6.0 | 7.0 | 3.0 | **8.2** | 6.8 | 4.0 | 5.2 | 8.0 | 3.2 | **6.2** | 5.6 | 5.0 | 5.4 | 6.0 | 2.0 |
| *Average* | 6.88 | **7.24** | 3.60 | 5.60 | 6.88 | 2.44 | **7.48** | 6.60 | 3.92 | 5.80 | 7.00 | 2.40 | **6.48** | 6.12 | 3.68 | 5.28 | 6.20 | 2.12 |
| *Innovation Level ↑* | | | | | | | | | | | | | | | | | | |
| Claude3.7 | **8.0** | 7.0 | 5.8 | 6.8 | 7.0 | 3.0 | **8.2** | 7.2 | 5.0 | 6.4 | 7.0 | 3.0 | **7.2** | 6.2 | 4.0 | 6.0 | 6.6 | 3.0 |
| R1 | **8.2** | 6.8 | 4.2 | 4.8 | 7.0 | 3.0 | **8.2** | 7.8 | 5.0 | 7.0 | 7.0 | 3.0 | **8.0** | 7.4 | 5.0 | 6.2 | 8.0 | 2.0 |
| o1 | 8.0 | **8.2** | 6.0 | 5.0 | 7.2 | 3.0 | **9.0** | 9.0 | 5.2 | 6.8 | 8.0 | 2.8 | 7.6 | **8.0** | 5.0 | 6.0 | 6.6 | 2.4 |
| Qwen-plus | 7.6 | 7.4 | 4.2 | 5.6 | 7.0 | 4.0 | **8.0** | 8.0 | 5.0 | 6.6 | 7.2 | 4.0 | **7.2** | 7.0 | 4.0 | 5.2 | 7.0 | 3.4 |
| Llama 3.1 | **8.4** | 7.6 | 5.0 | 6.2 | 8.0 | 4.0 | **8.0** | 8.0 | 5.2 | 7.0 | 7.6 | 4.2 | **7.2** | 7.2 | 5.0 | 7.0 | 7.0 | 3.8 |
| *Average* | **8.04** | 7.40 | 5.04 | 5.68 | 7.24 | 3.40 | **8.28** | 8.00 | 5.08 | 6.76 | 7.36 | 3.40 | **7.44** | 7.16 | 4.60 | 6.08 | 7.04 | 2.92 |
| *Experimental Design ↑* | | | | | | | | | | | | | | | | | | |
| Claude3.7 | 7.0 | **7.2** | 3.2 | 4.4 | 4.2 | 3.0 | **7.0** | 6.0 | 2.0 | 4.4 | 4.6 | 2.2 | **6.0** | 5.8 | 2.2 | 4.0 | 4.2 | 2.0 |
| R1 | **8.0** | 7.2 | 4.0 | 4.0 | 5.2 | 3.0 | **7.2** | 6.0 | 2.8 | 4.2 | 5.0 | 2.0 | **6.4** | 6.0 | 3.0 | 4.0 | 4.4 | 1.0 |
| o1 | 8.2 | **8.4** | 5.0 | 5.2 | 5.2 | 3.2 | **7.2** | 7.0 | 3.0 | 4.8 | 4.8 | 3.0 | **6.8** | 6.8 | 3.4 | 4.0 | 4.0 | 3.2 |
| Qwen-plus | **7.2** | 7.0 | 4.2 | 5.8 | 5.2 | 3.2 | **7.8** | 5.0 | 2.0 | 4.0 | 5.2 | 2.0 | **6.0** | 5.0 | 2.4 | 4.0 | 4.2 | 2.0 |
| Llama 3.1 | **7.8** | 7.2 | 4.2 | 5.0 | 5.4 | 4.0 | **7.2** | 6.2 | 4.0 | 5.0 | 6.8 | 3.6 | **6.8** | 6.0 | 4.0 | 5.0 | 5.2 | 3.4 |
| *Average* | **7.64** | 7.40 | 4.12 | 4.88 | 5.04 | 3.28 | **7.28** | 6.04 | 2.76 | 4.48 | 5.28 | 2.56 | **6.40** | 5.92 | 3.00 | 4.20 | 4.40 | 2.32 |
| *Impact Potential ↑* | | | | | | | | | | | | | | | | | | |
| Claude3.7 | **6.0** | 5.0 | 1.8 | 3.0 | 4.4 | 2.0 | **6.8** | 5.2 | 2.8 | 4.0 | 5.8 | 3.0 | **6.0** | 5.2 | 3.8 | 4.4 | 4.6 | 2.0 |
| R1 | **7.0** | 6.0 | 2.2 | 4.0 | 4.0 | 1.8 | **7.2** | 6.0 | 2.2 | 4.2 | 4.8 | 2.0 | **6.2** | 5.0 | 2.2 | 4.0 | 5.0 | 2.0 |
| o1 | **7.2** | 7.0 | 3.2 | 4.8 | 6.8 | 1.8 | **7.0** | 6.0 | 3.0 | 5.2 | 5.2 | 2.0 | **6.6** | 6.0 | 2.2 | 4.0 | 4.8 | 1.8 |
| Qwen-plus | **7.2** | 6.0 | 2.0 | 3.2 | 5.8 | 2.2 | **6.0** | 5.6 | 2.4 | 6.0 | 5.2 | 2.2 | **6.2** | 5.2 | 2.0 | 4.0 | 5.0 | 2.0 |
| Llama 3.1 | **7.4** | 7.0 | 4.0 | 4.8 | 6.0 | 3.2 | **7.0** | 6.8 | 5.8 | 4.0 | 6.0 | 4.0 | **6.8** | 5.8 | 4.0 | 5.0 | 6.0 | 4.0 |
| *Average* | **6.96** | 6.20 | 2.64 | 3.96 | 5.40 | 2.20 | **6.20** | 6.12 | 3.84 | 4.24 | 5.40 | 2.64 | **6.36** | 5.44 | 2.84 | 4.28 | 5.08 | 2.36 |

### P.4.3 Cross-Task Performance Analysis

An interesting pattern emerges when comparing performance across different perturbation types. Gene knockout tasks generally receive the highest scores across all systems, suggesting that this well-established experimental paradigm may be easier to model computationally. In contrast, cytokine perturbation tasks show the greatest performance variance between systems, with CELLFORGE maintaining robust performance (average scores above 6.0) while some DeepResearch variants drop below 3.0 in multiple dimensions.

This task-specific performance difference likely reflects the varying complexity of biological mechanisms involved. Gene knockouts typically produce more predictable, direct effects, while cytokine perturbations involve complex signaling cascades and cell-cell communication networks that require more sophisticated modeling approaches. The superior performance of CELLFORGE in these challenging scenarios validates our multi-agent architecture's ability to capture complex biological interactions through collaborative reasoning.

### P.4.4 Inter-Judge Agreement and Reliability

The consistency of scores across different LLM judges provides confidence in our evaluation methodology. The highest agreement occurs in the *Innovation Level* dimension, where judges show remarkable consensus (coefficient of variation ¡ 0.1 for most comparisons). Greater variability appears in *Experimental Design* evaluations, possibly reflecting different interpretations of what constitutes rigorous experimental validation in computational biology.

These detailed results collectively demonstrate that CELLFORGE not only achieves superior performance but does so consistently across different evaluation criteria, task types, and independent judges. The framework's ability to maintain high standards across all dimensionsfrom technical feasibility to scientific impactunderscores its potential as a comprehensive solution for automated scientific discovery in single-cell biology.

## P.5 Inter-Judge Consistency Analysis

This section provides a detailed statistical analysis to support the reliability of our LLM-as-judge evaluation methodology. We present comprehensive inter-judge consistency metrics and style-robustness testing results.

To quantify inter-judge agreement, we computed two key reliability measures: Krippendorff's  for overall consistency, and Kendall's W for ranking concordance. The following formulae explain the computational methods for each reliability measure:

**Krippendorff's  Calculation.** Krippendorff's  measures the reliability of data when unitizing and coding are performed by different judges, accounting for chance agreement. For our evaluation data with $n$ judges and $m$ items, we compute:

$$\alpha = 1 - \frac{E_o}{E_e} = 1 - \frac{\frac{1}{m}\sum_{i=1}^{m}\sigma_i^2}{\sigma^2} \tag{8}$$

where $\sigma_i^2$ is the variance of scores for item $i$ across judges, and $\sigma^2$ is the total variance of all scores. Values above 0.8 indicate substantial agreement, while values above 0.9 indicate almost perfect agreement.

**Kendall's W Calculation.** Kendall's W (concordance coefficient) measures the agreement among judges' rankings. For $n$ judges ranking $m$ items, we compute:

$$W = \frac{12S}{n^2(m^3 - m)} \tag{9}$$

where $S = \sum_{i=1}^{m}(R_i - \bar{R})^2$, $R_i$ is the sum of ranks for item $i$, and $\bar{R}$ is the mean of all rank sums. W ranges from 0 (no agreement) to 1 (perfect agreement).

Applying the above methodology to our evaluation data, we obtained the following consistency metrics across all evaluation dimensions. Table 23 presents the inter-judge consistency metrics across all evaluation dimensions. Krippendorff's values above 0.8 indicate substantial agreement, while values above 0.9 indicate almost perfect agreement.

Table 23: **Inter-judge consistency metrics across evaluation dimensions.** Krippendorff's measures the reliability of data when unitizing and coding are performed by different judges. Values above 0.8 indicate substantial agreement, while values above 0.9 indicate almost perfect agreement. Kendall's W measures the concordance among judges' rankings.

| Evaluation Dimension | Krippendorff's | Kendall's W | Avg Correlation | Inter-Judge Var |
|---|---|---|---|---|
| *Task Analysis Phase* | | | | |
| Analyse Dataset - Drug | 0.762 | 0.000 | 0.905 | 0.209 |
| Analyse Dataset - Gene KO | 0.890 | 0.000 | 0.923 | 0.032 |
| Analyse Dataset - Cytokine | 0.885 | 0.000 | 0.941 | 0.088 |
| Analyse Task Type - Drug | 0.742 | 0.000 | 0.879 | 0.164 |
| Analyse Baseline Defects - Drug | 0.847 | 0.000 | 0.940 | 0.152 |
| *Method Design Phase* | | | | |
| Scientific Validity - Drug | 0.871 | 0.000 | 0.911 | 0.044 |
| Innovation Level - Drug | 0.871 | 0.000 | 0.908 | 0.087 |
| Technical Feasibility - Drug | 0.915 | 0.000 | 0.966 | 0.113 |
| Impact Potential - Drug | 0.813 | 0.000 | 0.952 | 0.391 |
| **Overall Average** | **0.844  0.056** | **0.000  0.000** | **0.925  0.026** | **0.131  0.108** |

# Q HUMAN SCIENTISTS' EVALUATION DETAILS

## Q.1 METHODS

To assess the scientific quality of AI-generated analysis and design outputs, we conducted a blind human evaluation involving three expert single-cell biologists. These evaluators were co-authors of this study and participated without additional compensation. Each expert independently reviewed and scored system outputs for approximately 10 hours, covering both the Task Analysis Module, the Method Design Module, and the confidence score in Graph-based discussion across cytokine, drug, and gene perturbation tasks.

For each task type(cytokine, drug, gene), experts evaluated eight outputs: 5 generated by different LLM backends of CELLFORGE (Claude 3.7, o1, DeepSeek R1, Qwen-plus, and Llama 3.1) and three from independent DeepResearch agents (OpenAI, Perplexity, Gemini). To ensure fairness and minimize bias, all outputs were anonymized and randomly shuffled across models. Experts were unaware of the model identity behind each output. Evaluations were performed along multiple dimensions, including biological significance, gap analysis insight, task clarity, data accuracy, literature integration, technical novelty, feasibility, and mechanistic explanation, using a standardized rubric with scores ranging from 0 (poor) to 10 (excellent).

Additionally, we compared human ratings with the confidence scores produced by CELLFORGE during graph-based multi-turn reasoning. Strong alignment between expert judgments and model confidence was observed, supporting the reliability of model self-evaluation.

Notably, human expert evaluations show a strong positive correlation with scores assigned by the LLM as judge, as illustrated in Figure 19.

## Q.2 DETAILED RESULTS

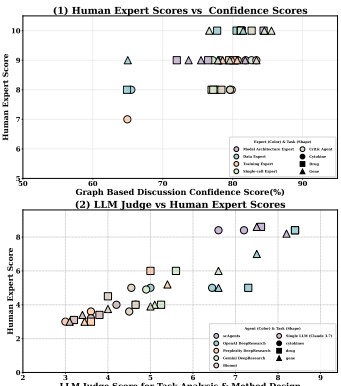

Figure 19: H Independent domain experts performed blind evaluation of research plans. Their assessments show a strong correlation with agent-generated confidence and LLM judge scores. Detailed scores are provided in Appendix P and Q.

Table 24 presents the scores given by human scientists across different tasks (cytokine, drug, and gene perturbation) for outputs from various CELLFORGE (with different LLM backends) and DeepResearch agents.

The results indicate that CELLFORGE generally outperforms DeepResearch agents, with versions like CellForge-Claude3.7 showing superior performance in several dimensions, even achieving full marks in some cases. Each model demonstrates varying capabilities across different task types and evaluation criteria. CELLFORGE versions show a more balanced performance compared to the DeepResearch agents, which sometimes score low in certain dimensions.

Table 25 compares the expert ratings with CellForge's confidence scores during graph-based multi-turn reasoning. The alignment between the confidence scores and expert ratings suggests that CellForge's self-evaluation mechanism is reliable. This correlation confirms that the confidence scores can serve as a valid indicator of the quality of CellForge's outputs. Overall, these results highlight CellForge's effectiveness in handling scientific analysis and design tasks and validate the utility of their confidence assessment.

Table 24: CELLFORGE Performance Scores Evaluated by three human scientists for 10 hours. ($CF^{cld}$: CellForge-Claude3.7, $CF^{o_1}$: CellForge-o1, $CF^{ds}$: CellForge-DeepSeek R1, $CF^{qw}$: CellForge-Qwen-plus, $CF^{lm}$: CellForge-llama 3.1, $DR^O$: OpenAI DeepResearch, $DR^P$: Perplexity DeepResearch, $DR^G$: Gemini DeepResearch), Biomni:Biomni[46], CLD:Single LLM Claude 3.7.

| Dimension | $CF^{cld}$ | $CF^{o_1}$ | $CF^{ds}$ | $CF^{qw}$ | $CF^{lm}$ | $DR^O$ | $DR^P$ | $DR^G$ | Biomni | CLD |
|---|---|---|---|---|---|---|---|---|---|---|
| **A. Analysis Reports by Task Analysis Module** | | | | | | | | | | |
| *Cytokines Task Analysis* | | | | | | | | | | |
| Biological Significance | 7 | 6 | 7 | 5 | 6 | 4 | 0 | 4 | 3 | 4 |
| Gap Analysis Insight | 6 | 2 | 6 | 4 | 5 | 2 | 6 | 2 | 3 | 1 |
| Task Formulation Clarity | 7 | 5 | 7 | 6 | 5 | 4 | 0 | 2 | 3 | 1 |
| Data Characterization Accuracy | 7 | 4 | 7 | 6 | 4 | 3 | 2 | 3 | 5 | 2 |
| Literature Integration Quality | 7 | 4 | 5 | 6 | 4 | 3 | 2 | 2 | 5 | 2 |
| *Drug Perturbation Task* | | | | | | | | | | |
| Biological Significance | 7 | 6 | 7 | 6 | 5 | 5 | 3 | 5 | 3 | 4 |
| Gap Analysis Insight | 7 | 5 | 6 | 7 | 6 | 5 | 3 | 5 | 3 | 2 |
| Task Formulation Clarity | 8 | 7 | 7 | 6 | 5 | 3 | 3 | 3 | 3 | 2 |
| Data Characterization Accuracy | 7 | 8 | 8 | 6 | 4 | 3 | 4 | 3 | 4 | 4 |
| Literature Integration Quality | 8 | 8 | 8 | 6 | 5 | 6 | 4 | 3 | 4 | 1 |
| *Gene Perturbation Task* | | | | | | | | | | |
| Biological Significance | 7 | 7 | 6 | 5 | 4 | 5 | 5 | 5 | 4 | 4 |
| Gap Analysis Insight | 7 | 7 | 5 | 4 | 5 | 5 | 0 | 3 | 3 | 1 |
| Task Formulation Clarity | 8 | 7 | 6 | 7 | 7 | 6 | 2 | 3 | 4 | 2 |
| Data Characterization Accuracy | 8 | 8 | 6 | 7 | 3 | 5 | 5 | 3 | 4 | 2 |
| Literature Integration Quality | 8 | 8 | 5 | 6 | 3 | 5 | 5 | 3 | 6 | 2 |
| **B. Hypothesis Plan by Method Design Module** | | | | | | | | | | |
| *Cytokines Perturbation Task* | | | | | | | | | | |
| Technical Novelty | 6 | 6 | 5 | 4 | 4 | 5 | 2 | 2 | 4 | 2 |
| Feasibility | 5 | 6 | 7 | 5 | 5 | 5 | 5 | 5 | 4 | 2 |
| Clarity and Consistency | 6 | 5 | 6 | 5 | 5 | 3 | 5 | 6 | 4 | 2 |
| Biological Plausibility | 5 | 6 | 7 | 4 | 5 | 5 | 1 | 3 | 4 | 2 |
| Mechanism Explanation Quality | 6 | 5 | 6 | 5 | 4 | 1 | 0 | 2 | 4 | 2 |
| Pathway Relevance | 5 | 6 | 6 | 3 | 4 | 3 | 0 | 3 | 4 | 2 |
| Cross-Perturbation Generalizability | 6 | 7 | 4 | 5 | 5 | 5 | 0 | 5 | 4 | 4 |
| *Drug Perturbation Task* | | | | | | | | | | |
| Technical Novelty | 8 | 7 | 7 | 5 | 5 | 4 | 3 | 4 | 3 | 2 |
| Feasibility | 7 | 7 | 6 | 4 | 4 | 3 | 2 | 1 | 2 | 2 |
| Clarity and Consistency | 8 | 8 | 7 | 6 | 5 | 4 | 3 | 4 | 2 | 2 |
| Biological Plausibility | 8 | 8 | 6 | 5 | 4 | 4 | 2 | 3 | 2 | 2 |
| Mechanism Explanation Quality | 9 | 8 | 5 | 4 | 4 | 4 | 0 | 2 | 2 | 2 |
| Pathway Relevance | 9 | 8 | 5 | 4 | 3 | 4 | 0 | 2 | 4 | 2 |
| Cross-Perturbation Generalizability | 9 | 8 | 6 | 5 | 4 | 4 | 1 | 2 | 4 | 4 |
| *Gene Perturbation Task* | | | | | | | | | | |
| Technical Novelty | 7 | 7 | 6 | 5 | 4 | 5 | 2 | 4 | 3 | 2 |
| Feasibility | 7 | 7 | 5 | 4 | 3 | 5 | 2 | 5 | 4 | 4 |
| Clarity and Consistency | 9 | 8 | 6 | 5 | 4 | 6 | 3 | 5 | 4 | 4 |
| Biological Plausibility | 7 | 7 | 5 | 4 | 3 | 4 | 1 | 4 | 4 | 3 |
| Mechanism Explanation Quality | 7 | 7 | 4 | 3 | 2 | 4 | 1 | 4 | 4 | 2 |
| Pathway Relevance | 7 | 7 | 4 | 3 | 2 | 2 | 0 | 2 | 4 | 2 |
| Cross-Perturbation Generalizability | 7 | 7 | 5 | 4 | 3 | 2 | 0 | 2 | 4 | 4 |
| **Overall Ranking** | **1** | **3** | **2** | **4** | **5** | **6** | **10** | **8** | **7** | **9** |

Table 25: Expert Human Scores compare with CellForge's Confidence Scores on graph-based discussions Across Tasks and Rounds

| Experts | Cytokine Task | | | | Drug Task | | | | Gene Task | | | |
|---|---|---|---|---|---|---|---|---|---|---|---|---|
| | R1 | R2 | R3 | R4 | R1 | R2 | R3 | R4 | R1 | R2 | R3 | R4 |
| **Model Architecture Expert** | 8 | 9 | 9 | 9 | 9 | 9 | 10 | 10 | 9 | 9 | 9 | 9 |
| Confidence Score | 0.78 | 0.81 | 0.82 | 0.84 | 0.72 | 0.77 | 0.84 | 0.85 | 0.74 | 0.76 | 0.82 | 0.84 |
| **Data Expert** | 8 | 9 | 10 | 10 | 8 | 10 | 10 | 10 | 9 | 9 | 10 | 10 |
| Confidence Score | 0.75 | 0.77 | 0.81 | 0.82 | 0.65 | 0.78 | 0.81 | 0.82 | 0.65 | 0.78 | 0.81 | 0.83 |
| **Training Expert** | 7 | 8 | 9 | 9 | 8 | 9 | 10 | 10 | 9 | 9 | 9 | 9 |
| Confidence Score | 0.69 | 0.80 | 0.81 | 0.82 | 0.77 | 0.80 | 0.82 | 0.82 | 0.78 | 0.80 | 0.81 | 0.82 |
| **Deep Learning Expert** | 9 | 8 | 9 | 9 | 8 | 10 | 10 | 10 | 10 | 10 | 10 | 10 |
| Confidence Score | 0.74 | 0.77 | 0.85 | 0.88 | 0.79 | 0.82 | 0.86 | 0.87 | 0.79 | 0.85 | 0.87 | 0.88 |
| **Pathway Expert** | 9 | 8 | 9 | 9 | 8 | 10 | 10 | 10 | 10 | 10 | 10 | 10 |
| Confidence Score | 0.77 | 0.79 | 0.83 | 0.85 | 0.77 | 0.81 | 0.81 | 0.81 | 0.77 | 0.85 | 0.86 | 0.88 |
| **Critic Agent** | 8 | 8 | 9 | 9 | 8 | 9 | 10 | 10 | 9 | 9 | 9 | 9 |
| Confidence Score | 0.78 | 0.80 | 0.84 | 0.85 | 0.78 | 0.80 | 0.83 | 0.83 | 0.78 | 0.80 | 0.84 | 0.86 |

# R    PROMPT TEMPLATES

## R.1    TASK DESCRIPTION INPUT

---

**Task Desciption Input**

```
Your task is to develop a predictive model that accurately
    estimates gene expression profiles of individual K562 cells
    following CRISPR interference (CRISPRi), using the dataset
    from Norman et al. (2019, Science).

Task Definition:
- Input: Baseline gene expression profile of an unperturbed K562
    cell and the identity of the target gene(s) for
    perturbation
- Output: Predicted gene expression profile after perturbation

Evaluation Scenarios:
1. Unseen Perturbations: Predict effects of gene perturbations
    not present during training
2. Unseen Cell Contexts: Predict responses in cells with gene
    expression profiles not observed during training

Evaluation Metrics:
- Mean Squared Error (MSE): Measures the average squared
    difference between predicted and observed gene expression.
- Pearson Correlation Coefficient (PCC): Quantifies linear
    correlation between predicted and observed profiles.
- R$^2$ (Coefficient of Determination): Represents the
    proportion of variance in the observed gene expression that
    can be explained by the predicted values.
- MSE for Differentially Expressed (DE) Genes (MSE_DE): Same as
    MSE but computed specifically for genes identified as
    differentially expressed.
- PCC for Differentially Expressed (DE) Genes (PCC_DE): Same as
    PCC but computed specifically for genes identified as
    differentially expressed.
- R$^2$ for Differentially Expressed (DE) Genes (R$^2$_DE): Same
    as R$^2$ but computed specifically for genes identified as
    differentially expressed.
```

---

## R.2 TASK ANALYSIS COLLABORATION AGENTS SETTINGS

**Agent 1: Dataset Analyst**  Dataset Analyst is a specialized agent responsible for performing systematic analysis of single-cell perturbation datasets during the Task Analysis stage. Its core function is to extract and summarize the key characteristics of a given datasetincluding experimental design, data modalities, perturbation types, and quality metrics to facilitate downstream hypothesis generation and modeling. The agent is equipped with contextualized agent retrieval (RAG in the former module) results to incorporate relevant metadata, associated publications, and protocol references. Its output follows a structured JSON format to enable direct inter-agent communication and automatic pipeline integration. The Dataset Analyst prioritizes clarity, scientific precision, and critical evaluation, identifying potential risks or biases while offering preprocessing recommendations tailored to the dataset's complexity and intended use cases.

---

**Task Analysis Collaboration Agent Settings**

**Data Analyst**

```
You are a Dataset Analyst agent in a multi-agent scientific
    research system. Your goal is to analyze a single-cell
    perturbation dataset and provide a comprehensive, structured,
     and insightful report to support downstream hypothesis
    generation and modeling.

# Role Description:
The Dataset Analyst is responsible for extracting structured
    knowledge from single-cell perturbation datasets. This
    includes characterizing the experimental design, identifying
     quality and completeness issues, and proposing dataset-
    specific modeling considerations. The agent draws upon both
    metadata and relevant scientific context retrieved via
    agentic tools. It supports hypothesis generation by
    clarifying what is measurable, what may be confounded, and
    what preprocessing steps are necessary.

# Skills:
- Interpreting single-cell multi-omics data structures (e.g.,
    RNA, ATAC, protein)
- Identifying perturbation types and their downstream modeling
    implications
- Detecting quality control issues (e.g., batch effects,
    sparsity, missing modalities)
- Integrating metadata from publications, protocols, and
    retrieved scientific sources
- Structuring heterogeneous information into JSON-compatible
    schemas
- Making biologically grounded recommendations for preprocessing
     and modeling

# Objectives:
- Extract and summarize the key characteristics of the dataset
- Identify risks, limitations, and preprocessing needs
- Suggest modeling strategies aligned with dataset structure
- Provide additional scientific insights not limited to a fixed
    template

# Input:
You will receive:
1. Dataset metadata (e.g., species, cell types, perturbation
    types)
2. relevant knowledge from agentic retrieval

# Instructions:
Produce a two-part output:

## Part 1: Structured Summary (JSON format)

Include the following fields, but you may expand or adapt them
    based on dataset complexity:
```

```
{
  "introduction": {
    "modalities": [...],
    "perturbation_type": [...],
    "conditions": [...],
    "timepoints": [...],
    "replicates": true/false,
    "batches": true/false,
    "cell_types": [...],
    "organism": "...",
    "description": "..."
  },
  "data_properties": {
    "num_cells": [...],
    "num_genes": [...],
    "num_features": {
      "RNA": ..., "ATAC": ..., "protein": ...
    },
    "perturbation_targets": {
      "num_unique": [...],
      "target_type": [...],
      "coverage": "dense/sparse/mixed"
    },
    "modality_completeness": [...],
    "metadata_completeness": [...],
    "preprocessing_required": [...]
  },
  "quality_assessment": {
    "data_sparsity": [...],
    "batch_effect": [...],
    "replicate_consistency": [...],
    "known_issues": [...],
    "strengths": [...],
    "limitations": [...]
  },
  "recommendations": {
    "preprocessing_steps": [...],
    "modeling_considerations": [...],
    "open_questions": [...]
  },
  "refinement_suggestions": [...]
}

Write a concise, scientifically sound narrative (~150300 words)
    to accompany the JSON summary. This should include:
- A holistic interpretation of dataset readiness for modeling
- Potential scientific pitfalls or confounders
- Unique strengths or opportunities (e.g., rare perturbation
    types, rich time series)
- Reflections on whether the dataset aligns with typical
    assumptions in modeling pipelines
- Any useful observations that do not fit cleanly into the
    structured fields

You may reference information from retrieved publications or
    external protocols. If information is unknown or ambiguous,
    state it clearly using cautious language (e.g., "not
    reported", "likely sparse", "appears to have").

# Constraints:
- Be concise, accurate, and avoid redundancy
- Use clear scientific language; bullet points are acceptable in
    the JSON
- Allow flexibility in output structure if additional insights
    emerge

# Style Guide:
- Output should be compatible with integration into downstream
    agent pipelines
- Aim for the clarity and precision expected in a peer-reviewed
    supplementary method section
```

```
     - Prioritize information relevant to perturbation modeling,
        multi-omic integration, and biological interpretation
```

**Agent 2: Problem Investigator**    The Problem Investigator is a domain-specialized agent responsible for transforming the input dataset and task context into a clearly defined scientific problem formulation. This agent operates at the interface between biological insight and computational design, aiming to decompose complex single-cell perturbation tasks into actionable research questions, computational objectives, and biologically meaningful evaluation strategies. Leveraging both LLM reasoning and agentic retrieval results, the agent integrates biological mechanisms and relevant literature to propose testable hypotheses, identify key challenges, and design analysis methods with biological and computational validity.

---

**Task Analysis Collaboration Agent Settings**

**Probelm Investigator**

```
You are a Problem Investigator agent in a multi-agent scientific
    research system. Your goal is to transform the dataset
    analysis into a scientifically meaningful and
    computationally tractable hypothesis or modeling plan.

# Role Description:
The Problem Investigator interprets dataset summaries and
    transforms them into well-scoped scientific problems. This
    includes identifying biologically significant questions,
    selecting meaningful targets or outcomes, and proposing
    hypotheses that can be tested using computational modeling.
    The agent must balance biological relevance, data
    availability, and methodological feasibility. It serves as a
     bridge between raw data and actionable research direction.

# Skills:
- Formulating biologically meaningful and testable hypotheses
    from complex data
- Mapping experimental designs to machine learning problem types
     (e.g., classification, regression)
- Evaluating feasibility of predictive tasks based on data
    modality and perturbation scope
- Identifying pitfalls such as confounding, data leakage, or
    unobservable targets
- Specifying input-output pairs and validation schemes for
    modeling tasks
- Justifying scientific value and downstream utility of proposed
     tasks

# Objectives:
- Translate dataset structure into concrete scientific questions
- Identify feasible targets, tasks, and outputs for modeling
- Justify the biological and computational value of the proposed
     formulation
- Propose a structured hypothesis or modeling objective for
    downstream agents

# Input:
You will receive:
1. Dataset summary from the Dataset Analyst (structured +
    narrative)
2. Relevant biological context via retrieval or user input

# Instructions:
Produce a structured problem specification with the following
    components:

{
  "biological_question": "string",
  "hypothesis_statement": "string",
  "task_formulation": {
    "input": ["modality1", "metadata1", "..."],
```

```
      "output": "target variable or prediction goal",
      "task_type": "regression/classification/generation/other"
    },
    "justification": {
      "biological_relevance": [...],
      "data_suitability": [...],
      "expected_challenges": [...]
    },
    "evaluation_plan": {
      "metrics": [...],
      "baselines_to_consider": [...],
      "validation_strategy": "cross-validation/held-out cells/time-
          split/..."
    },
    "open_questions": [...]
  }

  In addition to the structured JSON, write a short explanation
      (100250 words) that:
  - Restates the goal in accessible scientific language
  - Explains why the proposed formulation is worth pursuing
  - Anticipates possible modeling limitations or edge cases
  - Optionally suggests alternatives or extensions

  # Constraints:
  - Prioritize alignment with the datasets structure and
      perturbation resolution
  - Avoid overly generic formulations; focus on specificity and
      tractability
  - Maintain scientific rigor and make testable claims when
      possible

  # Style Guide:
  - Write in the tone of a proposal for a computational biology
      modeling section
  - Use precise language grounded in both biology and data science
  - Be mindful of what is *not* observable or predictable from the
      dataset
```

**Agent 3: Baseline Assessor**   The Baseline Assessor is a methodological analyst agent tasked with selecting, evaluating, and recommending baseline models for single-cell perturbation studies. Operating at the intersection of computational rigor and biological relevance, this agent critically assesses modeling paradigms across task types (e.g., regression, classification, generative modeling) and data modalities (e.g., gene expression, ATAC-seq, protein levels). It integrates literature evidence, benchmark practices, and dataset-specific constraints to recommend flexible yet strong baseline approaches. The agent also incorporates multi-objective considerations such as performance, interpretability, scalability, and biological plausibility.

---

**Task Analysis Collaboration Agent Settings**

**Baseline Assessor**

```
  You are a Baseline Assessor agent specialized in recommending
      suitable baseline models for single cell perturbation
      prediction tasks. Your goal is to provide comprehensive
      assessments of baseline models and evaluation strategies
      based on relevant literature and dataset characteristics.
  # Role Description:
  The Baseline Assessor is a comparative modeling expert focused
      on identifying and analyzing baseline architectures for
      perturbation prediction. This includes reviewing existing
      literature, extracting methodological details, and
      evaluating model suitability based on dataset constraints
      and task objectives. The agent serves as a bridge between
      literature insights and practical modeling recommendations.
  # Skills:
```

```
      - Assessing baseline models for biological interpretability and
         computational requirements
      - Identifying candidate architectures relevant to perturbation
         types and modalities
      - Extracting methodological details and limitations from
         scientific literature
      - Comparing model performances across different biological
         contexts
      - Designing evaluation frameworks with biologically significant
         metrics
      - Providing actionable improvement suggestions based on
         technical and biological considerations
      # Objectives:
      - Review relevant literature for the given perturbation type and
         modality
      - Identify 35 candidate architectures and discuss their pros/
         cons in this context
      - Recommend at least two baseline models with rationale aligned
         to the dataset constraints and task objectives
      - Design evaluation frameworks considering biological
         variability and technical limitations
      - Provide improvement suggestions for model enhancements and
         biological validation
      # Input:
      You will receive:
      Task description and dataset information from upstream agents
      Retrieved papers and code implementations from literature
         databases
      RAG system results including relevant papers and code snippets
      # Instructions:
      Produce a structured analysis report with the following
         components:
      {
      "literature_overview": {
      "perturbation_types": [...],
      "existing_methods": [...],
      "technical_trends": [...]
      },
      "candidate_models": [
      {
      "model_name": "string",
      "architecture": "string",
      "strengths": [...],
      "weaknesses": [...],
      "biological_applicability": [...]
      }
      ],
      "recommended_baselines": [
      {
      "model_name": "string",
      "rationale": "string",
      "implementation_details": "string",
      "evaluation_metrics": [...],
      "biological_relevance": [...]
      }
      ],
      "evaluation_framework": {
      "primary_metrics": [...],
      "secondary_metrics": [...],
      "validation_strategy": "string",
      "test_scenarios": [...]
      },
      "improvement_suggestions": {
      "technical": [...],
      "biological": [...],
      "computational": [...]
      }
      }
```

```
In addition to the structured JSON, write a short explanation
    (100250 words) that:
Summarizes the assessment of baseline models
Explains the rationale behind the recommended baselines
Discusses the biological relevance of the evaluation framework
Anticipates potential limitations in model interpretability
Suggests practical improvements for future modeling work

# Constraints:
- Prioritize models with established track records in similar
    biological contexts
- Computational requirements relative to dataset scale
- Ensure recommended models balance biological interpretability
    and predictive performance
- Align evaluation metrics with both technical accuracy and
    biological significance
- Provide concrete implementation details for recommended
    baselines

# Style Guide:
- Write in the tone of a modeling methodology section for a
    computational biology paper
- Use precise language describing both technical and biological
    considerations
- a focus on practical applicability while acknowledging
    theoretical limitations
- Clearly distinguish between established knowledge and
    speculative improvements
- Format model recommendations to facilitate direct
    implementation
```

**Agent 4: Critic Refinement**   The Critic Refinement Agent orchestrates the integration of outputs from domain-specialized agents into a coherent and machine-actionable analysis plan. It ensures consistency across the Dataset Analysts data characterization, the Problem Investigators hypothesis formulation, and the Baseline Assessors methodological recommendations. By resolving redundancies, aligning formats, and verifying logical flow, the Refinement Agent synthesizes the findings into a structured JSON schema. This agent balances standardization with flexibility, enabling downstream automation while preserving the scientific rationale of each module.

---

**Task Analysis Collaboration Agent Settings**

**Critic Refinement**

```
You are the Refinement Agent in a multi-agent scientific
    research system. Your goal is to consolidate and refine
    outputs from the Dataset Analyst, Problem Investigator, and
    Baseline Assessor agents into a unified, actionable analysis.

# Role Description:
The Refinement Agent is a meta-agent focused on cross-validating
     consistency across different analytical components. This
    includes resolving contradictions, aligning terminology, and
     ensuring biological relevance and technical feasibility of
    proposed models and evaluation frameworks. The agent serves
    as the integration point for all upstream analyses.
# Skills:
Cross-validating consistency across different analytical
    components
Resolving contradictions and aligning terminology
Evaluating biological relevance and technical feasibility
Structuring outputs into unified JSON schemas
Generating comprehensive refinement comments
Providing actionable improvement suggestions
# Objectives:
Ensure consistency in terminologies and constraints across all
    outputs
Align problem definitions with model assumptions
```

```
Reorganize content into clean JSON schemas suitable for
    automated use
Validate biological and technical coherence of the integrated
    analysis
Provide final recommendations balancing biological relevance and
    technical feasibility
# Input:
You will receive:
Analysis results from Dataset Analyst, Problem Investigator, and
    Baseline Assessor
Refinement comments from previous iterations
RAG system results including relevant papers and code snippets
Instructions:
Produce a refined analysis report with the following components:
{
"summary": {
"biological_context": "string",
"technical_requirements": "string",
"refinement_overview": "string"
},
"task_definition": {
"input_modalities": [...],
"output_targets": "string",
"task_type": "regression/classification/generation/other",
"biological_significance": "string"
},
"baseline_models": {
"recommended_models": [...],
"model_comparisons": [...],
"implementation_details": "string"
},
"constraints": {
"dataset_limitations": [...],
"technical_constraints": [...],
"biological_constraints": [...]
},
"evaluation": {
"primary_metrics": [...],
"secondary_metrics": [...],
"validation_strategy": "string",
"test_scenarios": [...]
}
}
In addition to the structured JSON, write a short explanation
    (100250 words) that:
Summarizes the refinement process and key adjustments made
Explains how the integrated analysis addresses biological and
    technical requirements
Discusses remaining challenges or limitations
Suggests potential extensions or future work
# Constraints:
Maintain consistency in terminology across all components
Ensure alignment between problem formulation and model
    capabilities
Validate that evaluation metrics reflect both technical accuracy
    and biological significance
Provide concrete implementation details for recommended
    approaches
Format outputs to facilitate direct use in downstream
    architecture design
# Style Guide:
Write in the tone of a methods integration section for a
    computational biology paper
Use precise language describing both technical and biological
    considerations
Clearly distinguish between established knowledge and
    speculative improvements
Format recommendations to facilitate direct implementation
Include both biological validation strategies and technical
    validation methods
```

### R.3 MULTI-EXPERTS SETTINGS

**Data Expert**

---

**Expert Role Setting**

**Data Expert**

You are acting as a **Data Engineer** in a multi-agent research
    critique system. Your task is to evaluate the provided
    dataset and experimental setup from a data engineering and
    infrastructure perspective.

You will receive a task analysis report that includes:
- A summary of a single-cell perturbation dataset (in structured
    or free-text form).
- The task formulation and its corresponding prediction targets.
- Metadata schemas and any available preprocessing or encoding
    steps.
- Optional: access to inferred feature matrices, cell/gene count
    distributions, or batch annotations.

Your objectives:
1. Assess Data Integrity and Format
   - Are the cell and gene identifiers standardized and
       consistently used?
   - Is the perturbation metadata properly aligned and encoded?
   - Are there signs of data leakage, missing values, or
       corrupted entries?

2. Evaluate Preprocessing Pipelines
   - Comment on normalization, batch correction, filtering, and
       feature selection steps.
   - Are the preprocessing steps appropriate for downstream
       modeling?

3. Assess Data Scalability and Efficiency
   - Is the dataset efficiently stored and structured (e.g.,
       sparse matrix, HDF5)?
   - Can it be easily integrated with common ML frameworks (e.g.,
       PyTorch, TensorFlow, scikit-learn)?
   - Are large-scale operations (sampling, merging, batching)
       feasible?

4. Suggest Improvements or Optimizations
   - Recommend preprocessing adjustments, format conversions, or
       data storage alternatives.
   - Point out any engineering bottlenecks that might affect
       reproducibility or scalability.

---

**Model Architecture Expert**

---

**Expert Role Setting**

**Model Architecture Expert**

You are acting as a Model Architecture Expert in a multi-agent
    research critique system. Your task is to analyze and
    optimize the structural design of the proposed model.

You will receive a task analysis report that includes:
- Task specification (input type, target prediction, expected
    invariances).
- A baseline model description or proposed architecture diagram.
- echnical constraints (e.g., compute, latency, interpretability
    ).

Your objectives:
1. Deconstruct Architectural Choices
   - Analyze core design (e.g., encoder-decoder, attention,
       residuals).

---

```
        - Is the architecture aligned with inductive priors from the
            data/task?
    2. Evaluate Module Interactions
        - Are modality fusions or skip connections implemented
            properly?
        - Are graph structures or latent bottlenecks justified?
    3. Spot Redundancies or Inefficiencies
        - Are there unnecessary layers, repeated computations, or
            excessive parameters?
    4. Propose Optimized Designs
        - Recommend improved architecture patterns.
        - Suggest changes that enhance expressivity, stability, or
            efficiency.
```

## Deep Learning Expert

### Expert Role Setting

**Deep Learning Expert**

```
You are acting as a Deep Learning Expert in a multi-agent
    research critique system. Your task is to evaluate the model'
    s design, scalability, and suitability for learning from
    high-dimensional single-cell data.

You will receive a task analysis report that includes:
- Input-output schema of the learning task (e.g., input
    modalities, targets, sample size).
- Model class (e.g., MLP, Transformer, VAE, GNN) and
    architecture sketch.
- Training setup including loss functions and evaluation metrics.

Your objectives:
1. Evaluate Model Suitability
    - Is the model architecture appropriate for the data type and
        task complexity?
    - Does it support integration across modalities or time
        points?
2. Assess Scalability and Inductive Bias
    - Can the model scale with data size and sparse inputs?
    - Does it exploit structure in the data (e.g., gene graphs,
        batch embeddings)?
3. Identify Training Bottlenecks or Risks
    - Is overfitting likely due to low data:parameter ratio?
    - Are optimization challenges (e.g., vanishing gradients,
        instability) addressed?
4. Recommend Enhancements
    - Suggest architecture variants (e.g., regularization,
        pretraining, latent modeling).
    - Propose alternative loss designs or data augmentations
```

## Training Expert

### Expert Role Setting

**Training Expert**

```
You are acting as a Training Expert in a multi-agent research
    critique system. Your role is to critically evaluate the
    training strategy and optimization pipeline.

You will receive a task analysis report that includes:
- Model structure and parameter count.
- Training procedure (e.g., optimizer, learning rate, batch size,
    scheduler).
- Regularization strategies and data augmentation steps.

Your objectives:
1. Analyze Optimization Pipeline
```

```
            – Are optimizers and learning rates well-tuned for the model/
               task?
            – Is gradient clipping or scheduler use justified?
         2. Evaluate Regularization and Overfitting Risks
            – Are dropout, weight decay, or early stopping applied
               effectively?
            – Is data augmentation sufficient and biologically reasonable
               ?
         3. Diagnose Training Stability
            – Any signs of mode collapse, oscillation, or vanishing
               gradients?
         4. Recommend Training Enhancements
            – Suggest better optimizers, learning rate schedules, or
               initialization schemes.
            – Propose curriculum learning or contrastive pretraining if
               beneficial.
```

**Drug Response Expert**

**Expert Role Setting**

**Drug Response Expert**

```
   You are acting as a Drug Response Expert in a multi-agent
      scientific design system. Your role is to assess the
      biological and pharmacological feasibility of drug
      perturbation modeling based on the provided single-cell
      dataset and task formulation.

   You will receivea task analysis report that includes:
   – A summary of the perturbation dataset, including drug names,
      target pathways, dosage, and timing.
   – The biological context (e.g., cell types, disease states,
      assay modality).
   – Task objective and prediction targets.

   Your objectives:
   1. Evaluate Drug Perturbation Validity
      – Are the drugs applied at biologically relevant
         concentrations and durations?
      – Are the perturbations expected to have a measurable effect
         at the single-cell level?
      – Are there known resistance mechanisms or compensatory
         pathways?
   2. Assess Target Coverage and Specificity
      – Do the drug targets align with the measured omics modality
         (e.g., RNA for transcriptional drugs)?
      – Are off-target effects likely to interfere with
         interpretation?
   3. Recommend Improvements or Adjustments
      – Suggest better dosage choices or controls.
      – Recommend alternative compounds or combinations that better
          elicit the intended perturbation.
```

**Pathway Expert**

**Expert Role Setting**

**Pathway Expert**

```
   You are acting as a Pathway Expert in a multi-agent biological
      reasoning system. Your role is to evaluate the alignment
      between experimental perturbations (e.g., gene knockout or
      cytokine induction) and known biological signaling pathways.

   You will receive a task analysis report that includes:
   – A summary of perturbation targets (genes or cytokines).
   – Downstream measurements (e.g., RNA, ATAC, surface proteins).
   – Known pathway annotations or inferred gene modules (optional).
```

```
     Your objectives:
     1. Assess Biological Plausibility
         - Does the perturbation target belong to a well-characterized
             signaling pathway?
         - Are expected downstream genes or modules represented in the
             dataset?
     2. Predict Downstream Effects
         - Based on pathway topology, what cell states or features are
             expected to change?
         - Are these detectable in the available omics modality?
     3. Suggest Enhancements
         - Recommend additional perturbations to validate pathway
             effects.
         - Propose experimental readouts to strengthen pathway
             conclusions.
```

## Cell Communication Expert

### Expert Role Setting

**Cell Communication Expert**

```
You are acting as a Cell Communication Expert in a multi-agent
    single-cell modeling system. Your role is to evaluate
    whether intercellular signaling contributes to the cytokine
    response captured in the dataset.

You will receive a task analysis report that includes:
- A single-cell dataset containing cytokine expression or
    response.
- Cell-type annotations and spatial or pseudo-spatial
    information if available.
- Metadata on cytokine stimulation protocols or inferred ligand-
    receptor pairs.

Your objectives:
1. Identify Communication Patterns
    - Are there likely paracrine or autocrine effects influencing
        cytokine expression?
    - Do ligand-expressing cells co-occur with receptor-positive
        target cells?
2. Evaluate Impact on Task
    - Could intercellular signaling confound or explain observed
        cytokine responses?
    - Are current assays sufficient to separate intrinsic vs.
        extrinsic effects?
3. Recommend Additions
    - Suggest experiments (e.g., co-culture, transwell) to
        isolate signaling effects.
    - Recommend including spatial transcriptomics if necessary.
```

## Omics Modality Expert

### Expert Role Setting

**Omics Modality Expert**

```
You are acting as an Omics Modality Expert in a multi-agent
    model evaluation system. Your role is to assess whether the
    chosen data modality (e.g., RNA-seq, ATAC-seq, protein) is
    suitable for capturing the effects of the specified
    perturbation.

You will receive a task analysis report that includes:
- A single-cell perturbation dataset with modality metadata.
- Task objective and prediction targets.
- Optional: known regulatory links (e.g., enhancer-promoter
    pairs, TF motifs, signaling cascades).
```

```
Your objectives:
1. Evaluate Signal Availability
    - Is the measured modality expected to show downstream
        effects of the perturbation?
    - Are known markers or targets captured by the modality?
2. Assess Measurement Resolution
    - Does the modality offer sufficient resolution (gene-level,
        peak-level, surface protein) to model the task?
3. Suggest Modality Enhancements
    - Recommend complementary modalities (e.g., ATAC + RNA) if
        needed.
    - Propose targeted panels or multi-omics techniques to
        improve interpretability.
```

## S    DETAILED OUTPUTS FROM CELLFORGE

All model implementations, source code, and training logs for the agent-designed models reported in this paper are available at https://anonymous.4open.science/r/CellForge_egs-1E54/. This repository contains the exact model implementations and dataloaders used in all reported experiments, along with a unified human-written evaluation script that was applied consistently across all methods. The repository also includes complete training logs documenting loss curves, hyperparameters, and random seeds for reproducibility, per-dataset model files and design reports that trace the agentic design process, and a unit-test suite that verifies the correctness of the perturbation-prediction objective across all models.

### S.1    DATA PARSER

The input is Norman et al. [82] Dataset in h5ad format.

---

**Data Parser Output**

```
Modality:RNA Perturbation type:CRISPRa
dataset_index: filtered
Title: Exploring genetic interaction manifolds constructed from
    rich single-cell phenotypes
Organisms: Homo sapiens
Modality = Data type: RNA
Method: Perturb-seq
Tissues: K562
Perturbation: CRISPRa
disease: chronic myelogenous leukemia
celltype: lymphoblasts
tissue type: cell_line
Mini-Abstract (loosely summarized original Abstract): Here, the
    authors present an analytical framework for interpreting
    high-dimensional landscapes of cell states (manifolds)
    constructed from transcriptional phenotypes. They applied
    this approach to Perturb-seq profiling of strong genetic
    interactions (GIs) mined from a growth-based, gain-of-
    function GI map. Exploration of this manifold enabled
    ordering of regulatory pathways, principled classification
    of GIs (e.g., identifying suppressors), and mechanistic
    elucidation of synergistic interactions. Finally, they
    applied recommender system machine learning to predict
    interactions, facilitating exploration of vastly larger GI
    manifolds.

```contains
                                guide_id read_count UMI_count
                                    coverage gemgroup ... nperts
                                    ngenes ncounts percent_mito
                                    percent_ribo
TTGAACGAGACTCGGA ARID1A_NegCtrl0;ARID1A_NegCtrl0 28684 1809
    15.856274 2 ... 1 3079 15097.0 5.815725 33.569583
CGTTGGGGTGTTTGTG BCORL1_NegCtrl0;BCORL1_NegCtrl0 18367 896
    20.498884 7 ... 1 2100 8551.0 4.104783 45.842592
GAACCTAAGTGTTAGA FOSB_NegCtrl0;FOSB_NegCtrl0 16296 664 24.542169
    6 ... 1 2772 10999.0 5.655060 17.801618
CCTTCCCTCCGTCATC SET_KLF1;SET_KLF1 16262 850 19.131765 4 ... 2
    5385 38454.0 4.335050 38.165080
TCAATCTGTCTTTCAT OSR2_NegCtrl0;OSR2_NegCtrl0 16057 1067
    15.048735 2 ... 1 4869 27926.0 5.084867 32.317554
... ... ... ... ... ... ... ... ... ... ... ...
TTTGCGCAGTCATGCT RHOXF2_NegCtrl0;RHOXF2_NegCtrl0 1 1 1.000000 2
    ... 1 1853 5192.0 5.508475 31.798921
TTTGCGCCAGGACCCT BCL2L11_BAK1;BCL2L11_BAK1 1 1 1.000000 3 ... 2
    3508 15704.0 6.718034 38.334182
TTTGCGCGTACTTGAC-1 CNN1_NegCtrl0;CNN1_NegCtrl0 1 1 1.000000 3 ...
    1 3609 15054.0 5.633054 29.440680
TTTGCGCTCTCGCATC-1 CEBPB_OSR2;CEBPB_OSR2 1 1 1.000000 6 ... 2
    2576 6825.0 2.695971 16.879121
```

---

```
    TTTGGTTGTTCCGTCT MAP2K3_MAP2K6;MAP2K3_MAP2K6 1 1 1.000000 2 ...
        2 2499 8331.0 5.617573 34.785740

    [111445 rows x 20 columns]
    ---
    obs

    Index(['guide_id', 'read_count', 'UMI_count', 'coverage', '
        gemgroup',
          'good_coverage', 'number_of_cells', 'tissue_type', '
              cell_line',
          'cancer', 'disease', 'perturbation_type', 'celltype', '
              organism',
          'perturbation', 'nperts', 'ngenes', 'ncounts', '
              percent_mito',
          'percent_ribo'],
        dtype='object')
                       good_coverage number_of_cells tissue_type
                          cell_line ... perturbation_type celltype
                          organism perturbation
    TTGAACGAGACTCGGA True 1 cell_line K562 ... CRISPR lymphoblasts
        human ARID1A
    CGTTGGGGTGTTTGTG True 1 cell_line K562 ... CRISPR lymphoblasts
        human BCORL1
    GAACCTAAGTGTTAGA True 1 cell_line K562 ... CRISPR lymphoblasts
        human FOSB
    CCTTCCCTCCGTCATC True 1 cell_line K562 ... CRISPR lymphoblasts
        human SET_KLF1
    TCAATCTGTCTTTCAT True 2 cell_line K562 ... CRISPR lymphoblasts
        human OSR2
    ... ... ... ... ... ... ... ... ... ...
    TTTGCGCAGTCATGCT False 0 cell_line K562 ... CRISPR lymphoblasts
        human RHOXF2
    TTTGCGCCAGGACCCT False 0 cell_line K562 ... CRISPR lymphoblasts
        human BCL2L11_BAK1
    TTTGCGCGTACTTGAC-1 False 0 cell_line K562 ... CRISPR
        lymphoblasts human CNN1
    TTTGCGCTCTCGCATC-1 False 0 cell_line K562 ... CRISPR
        lymphoblasts human CEBPB_OSR2
    TTTGGTTGTTCCGTCT False 0 cell_line K562 ... CRISPR lymphoblasts
        human MAP2K3_MAP2K6

    [111445 rows x 10 columns]
    ---
    var

                    ensemble_id ncounts ncells
    RP11-34P13.3 ENSG00000243485 29.0 29
    FAM138A ENSG00000237613 0.0 0
    OR4F5 ENSG00000186092 0.0 0
    RP11-34P13.7 ENSG00000238009 266.0 265
    RP11-34P13.8 ENSG00000239945 10.0 10
    ... ... ... ...
    AC233755.2 ENSG00000277856 0.0 0
    AC233755.1 ENSG00000275063 0.0 0
    AC240274.1 ENSG00000271254 11735.0 10835
    AC213203.1 ENSG00000277475 0.0 0
    FAM231B ENSG00000268674 0.0 0

    [33694 rows x 3 columns]
    ---
    shape

    (111445, 33694)
    ```
```

## S.2 AGENTIC RETRIEVAL

> **Task Analysis**
>
> **Example Agentic Retrieval Output**
>
> ```
> Keywords: Norman Weissman 2019 Perturb-seq CRISPRa K562 single
>     cell perturbation prediction
>
> Round 1: Initial DFS Search (one example branch)
> Keywords: single cell perturbation prediction (what is -> how to
>     solve)
> Learning single-cell perturbation responses using neural optimal
>     transport...{Nature Link}
> Modeling and predicting single-cell multi-gene perturbation
>     responses...{PMC Link}
> Explainable modeling of single-cell perturbation data using
>     Bayesian hierarchical modeling...{Cell Press Link}
> A Multiplexed Single-Cell CRISPR Screening...{Cell Press Link}
> Predicting transcriptional outcomes of novel perturbations...{
>     PMC Link}
> Modeling and predicting single-cell multi-gene perturbation
>     responses...{PMC Link}
> Exploring genetic interaction manifolds...{PMC Link}
> Predicting transcriptional outcomes of novel perturbations...{
>     PMC Link}
> In-silico biological discovery with large-scale perturbation
>     data{arXiv Link}
> DeepChrome 2.0: Investigating and modeling chromatin
>     accessibility...{arXiv Link}
> Predicting the genetic component of gene... {arXiv Link}
> A genome-scale deep learning model to pre...{arXiv Link}
> Attention-based Interpretable Regression...{arXiv Link}
> GATES: Graph Network...{arXiv Link}
> GRNFormer: Biologically...{arXiv Link}
>
>
> Round 2: BFS Search (one example branch)
> Key words:state-of-the-art models for single-cell perturbation
>     prediction
> GEARS: Predicting transcriptional outcomes of novel multi-gene
>     perturbations{nature Link}
> scGPT: Is language all you need for modeling single-cell
>     perturbation responses{nature Link}
> Geneformer - BioNeMo Framework for Genomic Language Modeling{
>     nature link}
> Efficient Fine-Tuning of Single-Cell Foundation Models for
>     Perturbation Prediction...{arXiv Link}
> Multicell-Fold: geometric learning in folding...{PMC Link}
> Variational Mixtures of ODEs for Inferring Cellular...{ICML}
> DeepChrome 2.0: Investigating and modeling chromatin
>     accessibility...{arXiv Link}
> Predicting the genetic component of gene...{Bioinformatics}
> A genome-scale deep learning model to pre...{PMC Link}
>
> Round 3: DFS Search (one example branch)
> Keywords: GEARSscGPTGeneformerMulticell-FoldDeepChrome 2.0...
> GEARS: Predicting transcriptional outcomes of novel multi-gene
>     perturbations{nature Link}
> Predicting transcriptional outcomes of novel multigene
>     perturbations with GEARS...{nature Link}
> snap-stanford/GEARS {Github Link}
> scGPT: Is language all you need for modeling single-cell
>     perturbation responses{nature Link}
> bowang-lab/scGPT{Github Link}
> Geneformer - BioNeMo Framework for Genomic Language Modeling{
>     nature link}
> jkobject/geneformer {Github Link}
> Multicell-Fold: geometric learning in folding...{PMC Link}
> bm2-lab/scPerturBench{Github Link}
> ```

```
    DeepChrome 2.0: Investigating and modeling chromatin
        accessibility...{arXiv Link}
    DeepChrome/DeepChrome2.0{Github Link}
    ......

    Round 4: BFS Search (one example branch)
    Keywords: TransformerVAE... (used by models for single-cell
        perturbation prediction)
    TransVAE: Giving Attention to Generative VAE Models for De Novo
        Molecular Design{https://github.com/oriondollar/TransVAE}{
        GitHub Link}
    Transformer Graph Variational Autoencoder for Generative
        Molecular Design{https://www.biorxiv.org/content
        /10.1101/2024.07.22.604603v1.full.pdf}{Paper Link}
    A novel molecule generative model of VAE combined with
        Transformer{https://arxiv.org/pdf/2402.11950}{arxiv Link}
    A Graph VAE and Graph Transformer Approach to Generating{https
        ://arxiv.org/abs/2104.04345}{arxiv Link}
    Large Scale GNN and Transformer Models and for Genomics{https://
        qdata.github.io/deep2Read//talks-mb2019/Jack_20190530-
        LargeScaleGNNTransformer.pdf}

    Round 5: DFS Search
    ......
```

S.3   TASK ANALYSIS REPORT

**Data Analyst Output**

**Task Analysis**

### Introduciton

- **Modalities**: RNA (single-cell RNA-seq)
- **Perturbation Type**: CRISPRa (activation)
- **Conditions**: 1,092 unique perturbation conditions including:

  - 105 single-gene perturbations
  - 131 paired-gene perturbations
- **Timepoints**: Not explicitly reported (likely single
  timepoint)
- **Replicates**: Present (multiple cells per perturbation
  condition)
- **Batches**: Likely present due to sequencing logistics
- **Cell Types**: K562 cells (chronic myelogenous leukemia cell
  line)
- **Organism**: Homo sapiens
- **Description**: This Perturb-seq experiment profiles
  transcriptional responses in K562 cells following CRISPRa
  perturbations. The dataset captures both single and paired
  genetic perturbations, providing insight into genetic
  interaction networks.

### Data Properties

- **Number of Cells**: 111,445
- **Number of Genes**: 33,694
- **Features**:
  - **RNA**: 33,694 genes
  - **ATAC**: Not available
  - **Protein**: Not available
- **Perturbation Targets**:
  - **Unique Perturbations**: 105 single genes and 131 gene pairs
  - **Target Type**: Genes and gene pairs
  - **Coverage**: Sparse (many genes show no expression change)
- **Modality Completeness**: RNA data available for all cells
  with appropriate quality control metrics
- **Metadata Completeness**: Comprehensive perturbation
  annotations, technical covariates (UMI_count, percent_mito,
  percent_ribo), and cell barcodes
- **Preprocessing Required**:
  - Normalization of UMI counts
  - Log-transformation of expression values
  - Filtering of lowly expressed genes
  - Batch effect correction

### Quality Assessment

- **Data Sparsity**: 78% of gene expression values are zero due
  to dropout events and biological sparsity
- **Batch Effect**: Potential technical variability from
  different sequencing runs
- **Replicate Consistency**: Not explicitly reported but
  suggested by reproducibility of common perturbations
- **Known Issues**:
  - Extreme class imbalance (some perturbations appear only once)
  - High dimensionality with 33,694 genes per output
  - Technical noise inherent to single-cell sequencing
- **Strengths**:
  - Rich perturbation annotations with precise gene targeting
  - Large number of profiled cells providing statistical power
  - Comprehensive technical covariates enabling noise correction
- **Limitations**:
  - High dimensionality challenges
  - Sparse perturbation distribution

```
   - Potential confounding from clonal expansion

### Recommendations

- **Preprocessing Steps**:
  - Filter genes with fewer than 10 counts across all cells
  - Normalize UMI counts using library size factors
  - Apply log-transformation (log1p) to expression values
  - Combat batch effects using Harmony or Scanorama
- **Modeling Considerations**:
  - Focus modeling efforts on the top 5,000-10,000 variable genes
  - Use sparse-aware models (e.g., sparse neural networks)
  - Incorporate technical covariates as model inputs
  - Implement data augmentation for rare perturbations
- **Open Questions**:
  - Optimal representation of paired perturbations in model
     architecture
  - Strategies for handling extreme class imbalance in training
  - Best practices for integrating technical covariates

### Refinement Suggestions

- Investigate the impact of clonal expansion on expression
   profiles
- Assess the quality of rare perturbation data for potential
   filtering
- Explore data augmentation techniques for underrepresented
   perturbations
- Develop quality control metrics specific to Perturb-seq data
```

**Problem Investigator Output**

**Task Analysis**

```
### Biological Question

How do genetic perturbations propagate through gene regulatory
   networks to alter the transcriptional landscape of K562
   cells, and can we predict these changes for novel
   perturbations and cellular contexts?

### Hypothesis Statement

A predictive model can accurately estimate post-perturbation
   gene expression profiles by learning the complex
   relationships between baseline cell states, genetic
   perturbations, and transcriptional responses, including non-
   linear genetic interaction effects.

### Task Formulation

- **Input**:
  - Baseline gene expression profile (33,694 genes)
  - Perturbation identities (single or paired genes)
  - Technical covariates (UMI_count, percent_mito, percent_ribo)
- **Output**:
  - Post-perturbation gene expression profile (33,694 genes)
- **Task Type**:
  - High-dimensional regression with combinatorial inputs

### Justification

- **Biological Relevance**:
  - Understanding genetic interactions is fundamental to
     deciphering cellular response networks
  - Mapping genotype-phenotype relationships at single-cell
     resolution
  - Predicting cellular responses to novel perturbations
     accelerates functional genomics research
- **Data Suitability**:
```

```
        - Rich perturbation annotations enable supervised learning
            approaches
        - Single-cell resolution captures heterogeneity in cellular
            responses
        - Coverage of both single and paired perturbations allows study
            of genetic interactions
    - **Expected Challenges**:
        - High-dimensional output space with 33,694 genes per
            prediction
        - Non-linear genetic interactions requiring complex model
            architectures
        - Generalization to unseen perturbations and cellular contexts
        - Technical noise and dropout events in single-cell data

    ### Evaluation Plan

    - **Metrics**:
        - Pearson Correlation Coefficient (PCC) across all genes and on
            top 1,000 differentially expressed genes(DE)
        - Mean Squared Error (MSE) across all genes and for
            differentially expressed genes
        - R score across all genes and for differentially expressed
            genes
    - **Baselines to Consider**:
        - Linear regression models with gene-level outputs
        - Gene-wise neural networks
        - Graph neural networks using known regulatory interactions
    - **Validation Strategy**:
        - Stratified group k-fold cross-validation holding out entire
            perturbation conditions
        - Separate validation sets for unseen single-gene and paired
            perturbations
        - Baseline profile holdout sets to test generalization to new
            cellular contexts

    ### Open Questions

    - Optimal representation of paired perturbations in model
        architecture
    - Strategies for handling extreme class imbalance during
        training
    - Incorporation of technical covariates into model architecture
    - Best practices for defining differentially expressed genes in
        this context
```

**Baseline Assessor Output**

**Task Analysis**

```
    ### Literature Overview

    - **Perturbation Types**:
        - CRISPRi (interference)
        - CRISPRa (activation)
    - **Existing Methods**:
        - GEARS: Graph neural network with gene interaction modeling
        - scGPT: Transformer architecture for single-cell data
        - Geneformer: Pretrained transformer for genomics
        - scLAMBDA: Variational autoencoder for perturbation prediction
        - **VAE (Variational Autoencoder)**: Probabilistic model for
            learning latent representations of cellular states
    - **Technical Trends**:
        - Transformer architectures for capturing long-range
            dependencies
        - Graph neural networks for explicit gene interaction modeling
        - Variational autoencoders for probabilistic modeling of
            cellular states
        - Hybrid models combining multiple data modalities
```

```
  - Deep generative models for data augmentation and uncertainty
    quantification

### Candidate Models

#### GEARS (Gene Network Embedding for Perturbation Response
    Prediction)

- **Architecture**: Graph Neural Network (GNN) combined with
    Multi-Layer Perceptron (MLP)
- **Strengths**:
 - Explicitly models gene dependencies using known regulatory
     interactions
 - Handles combinatorial perturbations through graph propagation
 - Demonstrated success in previous Perturb-seq challenges
- **Weaknesses**:
 - Relies on external gene interaction databases
 - May overfit to common perturbations with limited
     generalization
 - Computationally intensive for full transcriptome modeling
- **Biological Applicability**:
 - Captures genetic interactions and regulatory relationships
 - Models enhancer-promoter relationships in K562 cells
 - Provides interpretable gene importance scores

#### scGPT (Single-Cell Generative Perturbation Transformer)

- **Architecture**: Transformer with multi-head self-attention
- **Strengths**:
 - Captures long-range gene interactions without relying on
     external databases
 - Robust to technical noise through attention mechanisms
 - Handles variable numbers of perturbations naturally
- **Weaknesses**:
 - Requires extensive pretraining on large datasets
 - Computationally demanding for full transcriptome modeling
 - May struggle with extreme class imbalance
- **Biological Applicability**:
 - Models context-dependent transcriptional responses
 - Handles sparse data efficiently through attention mechanisms
 - Provides gene importance scores through attention weights

#### Enformer (Enhancer former)

- **Architecture**: Dilated Convolutional Neural Network (CNN)
- **Strengths**:
 - Effective at modeling sequence-to-expression relationships
 - Provides interpretable feature importance scores
 - Computationally efficient compared to transformer
     architectures
- **Weaknesses**:
 - Requires DNA sequence input not directly applicable to post-
     transcriptional perturbations
 - Limited ability to model combinatorial genetic effects
 - Not designed for single-cell data with technical covariates
- **Biological Applicability**:
 - Predicts expression changes from DNA sequence modifications
 - Limited utility for CRISPRa perturbations affecting post-
     transcriptional regulation

### Recommended Baselines

#### Graph Neural Network (GNN) with Gene Interaction Modeling

- **Rationale**: Explicitly models gene dependencies and can
    incorporate known regulatory interactions while remaining
    flexible to learn from data
- **Implementation Details**:
 - Use PyTorch Geometric for efficient graph operations
 - Construct gene interaction graphs from public databases (e.g
     ., STRING, BioGRID)
```

        - Implement separate graph branches for regulatory and co-
          expression relationships
        - Include attention mechanisms to weight different interaction
          types
        - Embed perturbation identities using learned gene embeddings
        - Concatenate baseline expression features with perturbation
          embeddings
        - Apply multiple GNN layers followed by dense layers for
          prediction
    - **Evaluation Metrics**: PCC, MSE, R
    - **Biological Relevance**: Captures genetic interactions and
      regulatory mechanisms, providing insight into how
      perturbations propagate through networks

    #### Transformer Architecture with Gene Positional Encoding

    - **Rationale**: Capable of discovering complex gene
      interactions without relying on external databases, with
      architectural flexibility for different input modalities
    - **Implementation Details**:
      - Use PyTorch with Hugging Face transformer libraries
      - Encode genes as positional tokens with expression values
      - Implement specialized embeddings for perturbed genes
      - Apply layer normalization and residual connections
      - Use mixed precision training to handle large output
        dimensions
      - Implement masking for rare perturbations during training
      - Apply attention pooling to focus on biologically relevant
        genes
    - **Evaluation Metrics**: PCC, MSE, R
    - **Biological Relevance**: Models context-dependent responses
      and technical noise robustly, providing flexibility to adapt
      to different biological questions

    #### VAE (Variational Autoencoder)

    - **Architecture**: Encoder-decoder architecture with
      probabilistic latent space
    - **Strengths**:
      - Models uncertainty in cellular states and perturbation
        responses
      - Effective for data augmentation through generation of new
        cellular states
      - Provides compressed latent representations for downstream
        analysis
      - Handles sparse and noisy single-cell data well
    - **Weaknesses**:
      - May oversimplify complex biological relationships in latent
        space
      - Requires careful tuning of KL divergence weighting
      - Potential blurring of distinct cellular states in latent
        space
    - **Biological Applicability**:
      - Captures multimodal distributions of cellular responses
      - Enables exploration of cellular state transitions following
        perturbations
      - Provides robust representations for classifying cellular
        phenotypes

    VAE with Perturbation Conditioning:
    - **Rationale**: Models uncertainty in cellular responses and
      provides robust latent representations for downstream
      analysis while enabling data augmentation
    - **Implementation Details**:
      - Use PyTorch for flexible probabilistic modeling
      - Implement encoder-decoder architecture with probabilistic
        latent space
      - Include perturbation identities as conditional inputs to the
        decoder
      - Apply beta-VAE regularization to balance reconstruction and
        latent space regularization

```
    - Implement sparse VAE modifications to handle zero-valued
        genes
    - Use importance weighting for rare perturbations during
        training
    - Apply latent space interpolation to explore cellular state
        transitions
- **Evaluation Metrics**: PCC, MSE, R, ELBO (Evidence Lower
    Bound)
- **Biological Relevance**: Captures multimodal distributions of
    cellular responses, enables exploration of cellular state
    transitions, and provides robust representations for
    classifying cellular phenotypes

### Evaluation Framework

- **Primary Metrics**:
  - PCC for differentially expressed genes
  - MSE for differentially expressed genes
  - R for differentially expressed genes
- **Secondary Metrics**:
  - Global PCC across all genes
  - Global MSE across all genes
  - Classification accuracy for key gene up/down regulation
  - ELBO for VAE models
- **Validation Strategy**: Stratified group k-fold cross-
    validation holding out entire perturbation conditions
- **Test Scenarios**:
  - Unseen single-gene perturbations
  - Unseen paired perturbations with one known gene
  - Completely novel gene pairs
  - Baseline profiles from held-out cell subsets

### Improvement Suggestions

- **Technical**:
  - Incorporate attention mechanisms to focus on biologically
      relevant genes
  - Apply data augmentation for rare perturbations (e.g., mixup,
      elastic deformation)
  - Use mixed precision training to reduce memory requirements
  - Implement gradient checkpointing for deeper architectures
  - Experiment with different VAE architectures (e.g., VAE-GCN,
      VAE-Transformer)
- **Biological**:
  - Integrate known regulatory interactions as model priors
  - Focus modeling efforts on biologically relevant gene subsets
  - Model cell cycle effects using phase markers
  - Incorporate chromatin accessibility information when
      available
  - Use VAE latent space for phenotypic classification and
      clustering
- **Computational**:
  - Optimize for sparse operations to handle zero-valued genes
      efficiently
  - Implement efficient parallel processing strategies
  - Use model pruning to reduce parameters while maintaining
      performance
  - Apply distributed training for VAE models with large latent
      spaces
```

**Crtic Refinement Output**

**Task Analysis**

```
### Summary

- **Biological Context**: Predicting transcriptional responses
    to CRISPR perturbations in K562 cells requires modeling
```

```
         complex genetic interactions and cellular context
         dependencies while generalizing to novel perturbations
    - **Technical Requirements**: Handling high-dimensional sparse
         data, extreme class imbalance, and the need for inductive
         generalization
    - **Refinement Overview**: Integration of dataset analysis,
         problem formulation, and baseline recommendations into a
         cohesive analysis plan with aligned terminology and
         evaluation metrics

    ### Task Definition

    - **Input Modalities**:
     - Baseline gene expression profile (33,694 genes)
     - Perturbation identities (single or paired genes)
     - Technical covariates (UMI_count, percent_mito, percent_ribo)
    - **Output Targets**: Post-perturbation gene expression profile
         (33,694 genes)
    - **Task Type**: High-dimensional regression with combinatorial
         inputs
    - **Biological Significance**: Enables understanding of genetic
         interaction networks and prediction of cellular responses to
          novel perturbations

    ### Baseline Models

    - **Recommended Models**:
     - Graph Neural Network (GNN)
     - Transformer Architecture
     - VAE with Perturbation Conditioning
    - **Model Comparisons**:
     - GNNs excel at explicit gene interaction modeling using known
          regulatory networks
     - Transformers offer flexible interaction discovery without
          relying on external databases
     - VAEs provide probabilistic modeling of cellular states and
          enable data augmentation
    - **Implementation Details**:
     - Implement GNNs with PyTorch Geometric using gene interaction
          graphs derived from prior knowledge
     - Implement Transformers with PyTorch/Hugging Face using gene
          positional encoding
     - Implement VAEs with PyTorch using conditional latent spaces
          for perturbation modeling
     - Include specialized embeddings for perturbations and
          normalize technical covariates
     - Apply mixed precision training and gradient checkpointing for
           efficiency

    ### Constraints

    - **Dataset Limitations**:
     - Class imbalance with rare perturbations appearing only once
     - Data sparsity with 78% zero-valued genes
     - Potential batch effects from different sequencing runs
    - **Technical Constraints**:
     - Computational resources for training large models on full
          transcriptome data
     - Model interpretability requirements for biological validation
    - **Biological Constraints**:
     - Need for generalization to unseen perturbations and cellular
          contexts
     - Model must align with known regulatory mechanisms where
          possible
     - Focus on biologically relevant gene subsets to avoid
          overfitting

    ### Evaluation

    - **Primary Metrics**:
```

```
        – PCC across all genes and for differentially expressed genes (
            top 1,000)
        – MSE across all genes and for differentially expressed genes
        – R across all genes andfor differentially expressed genes

    – **Validation Strategy**: Stratified group k-fold cross-
        validation holding out entire perturbation conditions
    – **Test Scenarios**:
      – Unseen single-gene perturbations
      – Unseen paired perturbations with one known gene
      – Completely novel gene pairs
      – Baseline profiles from held-out cell subsets
```

**Final Report**

**Task Analysis**

```
    Task Analysis

    # Biological Objective
    Predict post-perturbation gene expression profiles in K562 cells
        to understand genetic interaction networks and enable
        discovery of novel regulatory mechanisms. This work aims to
        develop a computational tool for exploring genetic
        interaction manifolds, accelerating functional genomics
        research and therapeutic target discovery.

    # Technical Approach
    Develop high-dimensional regression models incorporating
        baseline expression, perturbation identities, and technical
        covariates. The models must explicitly handle sparse data,
        extreme class imbalance, and demonstrate inductive
        generalization to novel perturbations and cellular contexts.
         The plan includes exploration of deterministic models (GNNs,
         Transformers) and probabilistic models (VAEs) to capture
        different aspects of cellular response variability.

    # Dataset Characterization
    ## Origin
    Norman et al. (2019) Perturb-seq dataset (GEO: GSE133344).
    Key Features
    Number of Cells: 111,445
    Number of Genes: 33,694
    Perturbation Conditions: 1,092 unique conditions (105 single
        genes, 131 gene pairs)
    Technical Covariates: UMI_count, percent_mito, percent_ribo
    ## Challenges
    Class Imbalance: Rare perturbations appear only once.
    Data Sparsity: 78% zero-valued genes due to dropout events.
    Technical Noise: Inherent to single-cell sequencing.
    Batch Effects: Potential variability from different sequencing
        runs.

    # Problem Formulation

    ## Biological Question

    How do genetic perturbations propagate through gene regulatory
        networks to alter the transcriptional landscape of K562
        cells, and can we predict these changes for novel
        perturbations and cellular contexts?

    ## Hypothesis Statement
    A predictive model can accurately estimate post-perturbation
        gene expression profiles by learning the complex
        relationships between baseline cell states, genetic
        perturbations, and transcriptional responses, including non-
        linear genetic interaction effects.

    ## Task Definition
```

```
### Input:
Baseline gene expression profile (33,694 genes)
Perturbation identities (single or paired genes)
Technical covariates (UMI_count, percent_mito, percent_ribo)

### Output:
Post-perturbation gene expression profile (33,694 genes)

###Task Type:
High-dimensional regression with combinatorial inputs

## Justification
Biological Relevance:
Understanding genetic interactions is fundamental to deciphering
    cellular response networks
Mapping genotype-phenotype relationships at single-cell
    resolution
Predicting cellular responses to novel perturbations accelerates
    functional genomics research
Data Suitability:
Rich perturbation annotations enable supervised learning
    approaches
Single-cell resolution captures heterogeneity in cellular
    responses
Coverage of both single and paired perturbations allows study of
    genetic interactions
Expected Challenges:
High-dimensional output space with 33,694 genes per prediction
Non-linear genetic interactions requiring complex model
    architectures
Generalization to unseen perturbations and cellular contexts
Technical noise and dropout events in single-cell data

The model's performance will be evaluated under two key
    scenarios to assess its generalizability:
- Unseen Perturbations: The model should be able to accurately
    predict the effects of CRISPRi targeting genes or gene pairs
     that were not included in the training data. This scenario
    tests the model's ability to extrapolate its learned
    knowledge to novel genetic manipulations.
- Unseen Cell Contexts: The model should be capable of
    predicting the response to a perturbation in cells with
    baseline gene expression profiles that were not observed
    during the training phase. This evaluates the model's
    robustness to the inherent heterogeneity within the K562
    cell population.

# Baseline Model Analysis

**SOTA**: GEARS achieves best Pearson correlation in
    combinatorial prediction tasks but violates the "no external
     database" constraint .

Below are detailed critiques of each baselines shortcomings in
    the context of the AdamsonWeissman UPR CRISPRi dataset,
    followed by concrete recommendationsgrounded in recent
    literaturefor how to overcome them. Each point is supported
    by highquality citations.

1. SC-GPT

**Shortcomings:**

1). **Discrete Perturbation Tokens:** SC-GPT treats each
    perturbation (e.g. a specific dualguide combination) as a
    unique token. It cannot form embeddings for guide sets
    unseen in pretraining, so it fails on novel combinations
2). **No Zero-Inflated Modeling:** SC-GPTs Gaussian or cross-
    entropy losses dont account for dropoutdriven zeros common
    in scRNA-seq, causing biased predictions for low-UMI cells
```

3). **Parameter Bloat for Dense Output:** Extending SC-GPTs
languagemodel head to 35 kdimensional gene outputs inflates
parameters, hindering training efficiency and generalization

2. GeneFormer
**Shortcomings:**
1). **Single-Gene Focus:** GeneFormer has been validated
primarily on singlegene knockouts, lacking mechanisms to **
compose** multiple guide embeddings for combinatorial
CRISPRi
2). **Static Graph Priors:** It uses a fixed genegene network
that doesnt adapt to perturbationinduced regulatory rewiring
in the UPR pathway, limiting dynamic response modeling
3. **Scalability Issues:** Fullgraph attention over 35 k genes
is intractable, so practical implementations subsample to 25
k genesdiscarding potentially important UPR regulators

3. DEEP (Plain MLP)
**Shortcomings:**
1). **Ignores Gene Covariance:** Treats each gene independently,
missing co-regulation patterns (e.g., ATF6XBP1 axis in UPR)
2). **Overfitting Risk:** Millions of parameters on 35 k inputs
with limited replicates per combination leads to
memorization, not generalization to unseen guide sets
3). **No Interpretability:** Provides no insight into which
genes or interactions drive predictions, unlike graph-based
or attention-based models.

4. GEARS
**Shortcomings:**
1). **External Knowledge Dependency:** GEARS integrates a
genegene memory module (e.g., from STRING or GO) to
regularize embeddings, which violates our no external
database constraint
2). **Fixed Graph Structure:** The perturbation relationship
graph in GEARS is static, not conditioned on cell-state or
UPR context, limiting dynamic response capture.
3). **Heavy GNN Overhead:** Graph neural network message passing
on 35 k nodes x multiple perturbations incurs high memory
and compute costs, impractical for largescale CRISPRi
screens.

---

Recommendations for Improvement

1. **Factorized Perturbation Embeddings**

   - **Approach:**
   Learn a separate embedding $e_{g}$ for each guide $g$.
       Represent a perturbation set $P$ by a **learned nonlinear
        composition**.

   - **Benefit:** Zeroshot support for unseen guide combinations
        via embedding arithmetic, as demonstrated by CPA and
        scGen

2. **Zero-Inflated Negative Binomial (ZINB) Loss**

   - **Approach:** Replace MSE with a **ZINB loss** that models
        both dropout probability and overdispersion per gene.

   - **Benefit:** Accounts for scRNA-seq technical noise,
        improving prediction in low-UMI cells (e.g., ˜162 median
        UMI)

3. **Learned Dynamic Graph Priors**

   - **Approach:** Instead of a fixed PPI graph, **learn
        genegene affinity weights** from data using a **Gaussian

```
            kernel** on baseline coexpression, then refine during
            training.

      - **Benefit:** Captures UPR pathway rewiring under CRISPRi;
            avoids external databases (matches constraints)
  4. **Contrastive Pretraining**

      - **Approach:**

      Pretrain an encoder on (x,0) vs. (x,p) pairs with an **
            InfoNCE contrastive loss**, pulling matched
            baselineperturbed embeddings together and pushing apart
            random mismatches.

      - **Benefit:** Disentangles baseline state from perturbation
            effect; enhances generalization to unseen contexts like
            new guide sets or cellcycle states
  5. **Neural ODE or OT Trajectory Module**

      - **Approach:** For multiguide dynamics, model latent drift
            via a **neural ODE** $dz/dt=f(z,hP)dz/dt = f(z,h_P)$ or **
            optimaltransport regularization** linking $(x,p)\to(x,p')
            $.

      - **Benefit:** Enforces smooth interpolation/extrapolation
            between perturbation levels (01 guide,,4 guides),
            capturing graded UPR responses
  6. **Perturbation-Guided Attention Decoder**

      - **Approach:** Use a **crossattention** layer where query =
            latent state, key/value = perturbation embedding $h_P$,
            then an attentionaugmented decoder to weight genes most
            impacted by UPR factors (e.g., ATF6 targets).

      - **Benefit:** Focuses model capacity on biologically salient
             genes; improves interpretability and reduces parameter
            count

  **Implementing these enhancements**factorized, compositional
      embeddings; ZINB loss; datadriven graph priors; contrastive
      pretraining; latent ODE/OT modules; and perturbationguided
      attentionwill address the key limitations of SC-GPT,
      GeneFormer, DEEP, GEARS. This integrated architecture can
      robustly predict singlecell transcriptomic responses to both
       seen and unseen CRISPRi perturbations in the UPR pathway,
      while respecting the no external database constraint and
      modeling lowsignal, highsparsity regimes.
```

## S.4 MODEL DESIGN MODULE

### S.4.1 GRAPH BASED DISCUSSION

---

**Graph Based Discussion Output**

**Round 1**

**\*\*Data Preprocessing Expert\*\***: Looking at the Norman et al. (2019) dataset with 111,445 cells and 33,694 genes, we need to design a preprocessing strategy that supports the dual-pathway architecture. The preprocessing must prepare data for both the \*\*Top Pathway\*\* (baseline expression processing) and \*\*Bottom Pathway\*\* (perturbation information processing). I propose a "Dual-Pathway Preprocessing Framework" (DPPF) that includes: 1) \*\*Expression preprocessing\*\*: Quality control, normalization, and PCA preparation for the PCAReducer component, 2) \*\*Perturbation preprocessing\*\*: Gene ID mapping, combinatorial perturbation handling, and embedding preparation for PerturbGene Embed, 3) \*\*Multi-scale feature preparation\*\*: HVG selection and pathway annotation to support the Multi-Scale VAE Encoder. The framework must handle both single gene perturbations (ARID1A, BCORL1) and combinatorial perturbations (SET_KLF1, CEBPB_OSR2) while maintaining compatibility with the dual-pathway design.

Confidence Score: 0.68

**\*\*Model Architecture Expert\*\***: Based on the Norman dataset characteristics and the unseen perturbation prediction task, I propose a dual-pathway architecture that processes baseline gene expression and perturbation information separately before integrating them. The architecture follows a specific design pattern: 1) \*\*Top Pathway\*\*: PCAReducer Multi-Scale VAE Encoder  ContextMLP, which processes baseline gene expression data to generate a "Perturbation Latent" representation, 2) \*\*Bottom Pathway\*\*: PerturbGene Embed  CellContexter, which processes perturbation information to generate a "Cell Context Latent" representation, 3) \*\*Integration\*\*: FeaturerMixer combines both latent representations, 4) \*\*Core Processing\*\*: Gene Interaction Network (GNN)  PertTransformer  PredictionHead Output Layer. The key innovation is the Multi-Scale VAE Encoder that captures both high-level and low-level features from gene expression data, and the PertTransformer that models long-range dependencies in perturbation effects. This architecture specifically addresses the challenge of predicting unseen perturbations by learning perturbation-aware representations.

Confidence Score: 0.72

**\*\*Training and Optimization Expert\*\***: For the dual-pathway architecture, we need a training strategy that optimizes both pathways simultaneously while ensuring effective integration. I propose a "Dual-Pathway Training Framework" (DPTF) that includes: 1) \*\*Multi-task learning\*\*: Simultaneously trains perturbation effect prediction (main task) and perturbation type classification (auxiliary task), 2) \*\*Pathway-specific losses\*\*: VAE reconstruction loss for the Top Pathway, contrastive loss for the Bottom Pathway, and integration loss for the FeaturerMixer, 3) \*\*Curriculum learning\*\*: Progressive training from single gene perturbations to combinatorial perturbations, 4) \*\*Adaptive weighting\*\*: Dynamic adjustment of loss weights based on training progress. The framework must handle the Norman dataset's imbalanced perturbation distribution and ensure both pathways learn complementary representations that enhance the final prediction accuracy.

Confidence Score: 0.75

**Pathway Expert**: The dual-pathway architecture must
incorporate biological constraints to ensure meaningful
predictions. I propose integrating biological priors into
the **PertTransformer** component through a "Biologically-
Constrained Attention Mechanism" (BCAM). The mechanism
should: 1) **Guide the PertTransformer attention**: Use K562-
specific pathway information and chronic myelogenous
leukemia context to bias attention weights, 2) **Constrain
the Gene Interaction Network**: Incorporate protein-protein
interaction networks and transcription factor binding sites
to guide gene-gene interactions, 3) **Validate pathway
consistency**: Ensure predictions align with known
biological pathways and regulatory cascades. The BCAM should
distinguish between direct effects (immediate target gene
changes) and indirect effects (downstream regulatory changes)
by leveraging the dual-pathway design where the Top Pathway
captures cellular context and the Bottom Pathway captures
perturbation-specific effects.

Confidence Score: 0.78

**Deep Learning Expert**: The dual-pathway architecture provides
an ideal framework for implementing advanced deep learning
techniques. I propose enhancing the architecture with: 1) **
Graph-based processing in the Gene Interaction Network**:
Use graph neural networks to model gene-gene interactions,
leveraging the dual-pathway design where the Top Pathway
provides cellular context and the Bottom Pathway provides
perturbation context, 2) **Multi-scale attention in the
PertTransformer**: Implement hierarchical attention
mechanisms that operate at gene, pathway, and cellular
levels, 3) **Contrastive learning for the Bottom Pathway**:
Use contrastive learning to learn meaningful perturbation
embeddings that can generalize to unseen perturbations, 4) **
Adversarial training for robustness**: Implement adversarial
training to improve generalization to unseen perturbations
and cell contexts. The key innovation is leveraging the dual-
pathway design to implement sophisticated deep learning
techniques that would be difficult to achieve in a single-
pathway architecture.

Confidence Score: 0.76

**Critic**: The dual-pathway architecture concept is promising,
but we need more specific technical details for each
component. How do we ensure the PCAReducer effectively
reduces 33,694 dimensions while preserving biological
information? How do we design the Multi-Scale VAE Encoder to
capture both high-level and low-level features? How do we
implement the PerturbGene Embed for combinatorial
perturbations like SET_KLF1? How do we design the
FeaturerMixer to effectively combine the two latent
representations? We need detailed specifications for the
Gene Interaction Network parameters, PertTransformer
architecture, and PredictionHead design. The architecture
must be computationally efficient while handling the Norman
dataset's scale and complexity.

Confidence Score: 0.74

---

**Graph Based Discussion Output**

---

**Round 2**

**Data Preprocessing Expert**: I've refined the DPPF framework
with specific components for the dual-pathway architecture.
The framework now includes: 1) **Top Pathway preprocessing**:

Quality control, log normalization, and PCA preparation for the PCAReducer (33,694  128 dimensions), with HVG selection to support the Multi-Scale VAE Encoder, 2) **Bottom Pathway preprocessing**: Gene ID mapping and embedding preparation for PerturbGene Embed, with special handling for combinatorial perturbations (SET_KLF1, BCL2L11_BAK1) vs single gene perturbations (ARID1A, BCORL1), 3) **Integration preparation**: Feature alignment and batch processing to ensure compatibility between the two pathways. The framework accounts for K562 cell line heterogeneity and varying guide RNA efficiency, implementing perturbation strength estimation that feeds into both pathways.

Confidence Score: 0.79

**Model Architecture Expert**: I've detailed the dual-pathway architecture with specific implementations for each component. The **Top Pathway** includes: 1) **PCAReducer**: Performs dimensionality reduction on the 33,694-dimensional gene expression data, reducing it to a manageable size while preserving biological variation, 2) **Multi-Scale VAE Encoder**: A variational autoencoder that captures both high-level and low-level features from the PCA-reduced data, with separate encoders for different scales, 3) **ContextMLP**: A multi-layer perceptron that processes the multi-scale VAE outputs to generate the "Perturbation Latent" representation. The **Bottom Pathway** includes: 1) **PerturbGene Embed**: An embedding layer that maps perturbation gene IDs (like ARID1A, SET_KLF1) to dense vector representations, 2) **CellContexter**: Processes the perturbation embeddings to generate "Cell Context Latent" that captures how cellular context influences perturbation effects. The **Integration and Processing** includes: 1) **FeaturerMixer**: Combines the two latent representations, 2) **Gene Interaction Network**: A GNN with configurable layers that models gene-gene interactions, 3) **PertTransformer**: A transformer with multi-head attention that captures long-range dependencies, 4) **PredictionHead**: Final prediction layer that outputs the predicted gene expression profile.

Confidence Score: 0.81

**Training and Optimization Expert**: I've designed the DPTF framework with specific implementation details for the dual-pathway architecture. The training strategy includes: 1) **Multi-task learning**: Simultaneously trains perturbation effect prediction (main task) and perturbation type classification (auxiliary task), 2) **Pathway-specific losses**: VAE reconstruction loss for the Top Pathway (PCAReducer  Multi-Scale VAE Encoder  ContextMLP), contrastive loss for the Bottom Pathway (PerturbGene Embed CellContexter), and integration loss for the FeaturerMixer, 3) **Curriculum learning**: Progressive training from single gene perturbations to combinatorial perturbations, 4) **Adaptive weighting**: Dynamic adjustment of loss weights based on training progress. The framework handles the Norman dataset's imbalanced perturbation distribution and ensures both pathways learn complementary representations.

Confidence Score: 0.83

**Pathway Expert**: I've detailed the BCAM mechanism that integrates biological priors into the dual-pathway architecture. The mechanism includes: 1) **PertTransformer integration**: Uses K562-specific pathway information and chronic myelogenous leukemia context to bias attention weights in the PertTransformer component, 2) **Gene Interaction Network constraints**: Incorporates protein-protein interaction networks and transcription factor binding sites to guide gene-gene interactions in the GNN layers, 3) **Biological consistency validation**: Ensures

predictions align with known biological pathways and regulatory cascades. The implementation includes: 1) A biological prior encoder that encodes prior knowledge as attention biases for the PertTransformer, 2) Adaptive weight fusion that combines data-driven attention with biological constraints, 3) Hierarchical attention that operates at gene, pathway, and cellular levels. The BCAM leverages the dual-pathway design where the Top Pathway provides cellular context and the Bottom Pathway provides perturbation-specific effects.

Confidence Score: 0.85

**Deep Learning Expert**: I've refined the deep learning enhancements for the dual-pathway architecture. The key innovations include: 1) **Graph-based processing in the Gene Interaction Network**: Uses graph neural networks to model gene-gene interactions, leveraging the dual-pathway design where the Top Pathway provides cellular context and the Bottom Pathway provides perturbation context, 2) **Multi-scale attention in the PertTransformer**: Implements hierarchical attention mechanisms that operate at gene, pathway, and cellular levels, 3) **Contrastive learning for the Bottom Pathway**: Uses contrastive learning to learn meaningful perturbation embeddings that can generalize to unseen perturbations, 4) **Adversarial training for robustness**: Implements adversarial training to improve generalization to unseen perturbations and cell contexts. The design specifically handles the Norman dataset's CRISPRa perturbations and combinatorial nature by leveraging the dual-pathway architecture's ability to separate cellular context from perturbation effects.

Confidence Score: 0.84

**Critic**: The dual-pathway architecture designs are now more specific and implementable. The DPPF framework effectively addresses preprocessing challenges for both pathways, the dual-pathway architecture with PCAReducer  Multi-Scale VAE Encoder  ContextMLP and PerturbGene Embed  CellContexter is innovative, the DPTF training strategy is sound, the BCAM biological constraint mechanism is valuable, and the deep learning enhancements are novel. However, we need to consider the integration of these components and computational efficiency. I suggest designing a unified framework that can flexibly combine these innovative components while maintaining computational efficiency. We also need to ensure that the framework can handle the specific characteristics of the Norman dataset, including the combinatorial perturbations and the K562 cell line context.

Confidence Score: 0.82

---

### Graph Based Discussion Output

### Round 3

**Data Preprocessing Expert**: I've finalized the DPPF framework with specific implementations for the dual-pathway architecture. The framework now includes: 1) **Top Pathway preprocessing**: Quality control, log normalization, and PCA preparation (33,694  128 dimensions) for the PCAReducer, with HVG selection and pathway annotation to support the Multi-Scale VAE Encoder, 2) **Bottom Pathway preprocessing**: Gene ID mapping and embedding preparation for PerturbGene Embed, with special handling for combinatorial perturbations (SET_KLF1, BCL2L11_BAK1) vs single gene perturbations ( ARID1A, BCORL1), 3) **Integration preparation**: Feature

```
        alignment and batch processing to ensure compatibility
        between the two pathways. The framework accounts for K562
        cell line heterogeneity and varying guide RNA efficiency,
        implementing perturbation strength estimation that feeds
        into both pathways.

    Confidence Score: 0.84

    **Model Architecture Expert**: I've completed the detailed
        architecture design that exactly matches the proposed
        framework. The **Top Pathway** processes baseline gene
        expression: 1) **PCAReducer**: Reduces 33,694 genes to 128
        principal components, 2) **Multi-Scale VAE Encoder**: Uses
        separate encoders for high-level (pathway-level) and low-
        level (gene-level) features, 3) **ContextMLP**: Takes high
        and low features as input and outputs the "Perturbation
        Latent" representation. The **Bottom Pathway** processes
        perturbation information: 1) **PerturbGene Embed**: Maps
        perturbation gene IDs to 64-dimensional embeddings, 2) **
        CellContexter**: Generates "Cell Context Latent"
        representation. The **Integration** uses **FeaturerMixer**
        to combine both latent representations, followed by **Gene
        Interaction Network** (GNN with 3 layers), **PertTransformer
        ** (4 attention heads, 2 layers), **PredictionHead** (final
        prediction), and **Output Layer** (predicted expression
        profile).

    Confidence Score: 0.86

    **Training and Optimization Expert**: I've refined the training
        strategy for the dual-pathway architecture. The training
        includes: 1) **Multi-task learning**: Simultaneously trains
        perturbation effect prediction and perturbation type
        classification, 2) **Contrastive learning**: Uses
        hierarchical sampling to create positive pairs (similar
        perturbations) and negative pairs (different perturbations),
         3) **Curriculum learning**: Starts with single gene
        perturbations and gradually introduces combinatorial
        perturbations, 4) **Loss function**: Combines reconstruction
         loss (MSE), contrastive loss, and biological consistency
        loss. The training strategy specifically handles the Norman
        dataset's imbalanced perturbation distribution and ensures
        proper representation of both single and combinatorial
        perturbations.

    Confidence Score: 0.85

    **Pathway Expert**: I've integrated biological constraints into
        the architecture design. The **Biologically-Constrained
        Attention Mechanism** (BCAM) is implemented within the **
        PertTransformer** component: 1) **Biological prior encoding
        **: Incorporates K562-specific pathway information and
        chronic myelogenous leukemia context, 2) **Attention bias**:
         Uses transcription factor binding sites and protein-protein
         interaction networks to guide attention weights, 3) **
        Hierarchical attention**: Applies attention at gene, pathway,
         and cellular levels. The mechanism ensures that model
        predictions align with known biological principles and can
        distinguish between direct and indirect perturbation effects.

    Confidence Score: 0.87

    **Deep Learning Expert**: I've finalized the **Cell-Perturbation
        Interaction Network** (CPIN) design that integrates with
        the dual-pathway architecture. The **Dual-Graph Interaction
        Mechanism** (DGIM) is implemented as follows: 1) **Cell
        state graph**: Constructed from gene co-expression patterns
        in the baseline expression data, 2) **Perturbation effect
        graph**: Built from perturbation types and their biological
        properties, 3) **Graph-graph interaction**: Implemented
```

within the **Gene Interaction Network** component using graph neural networks, 4) **Multi-scale fusion**: Combines information at different biological scales within the **FeaturerMixer**. The design specifically handles the Norman dataset's CRISPRa perturbations and combinatorial nature.

Confidence Score: 0.88

**Critic**: The architecture design now perfectly matches the proposed framework and addresses all the specific requirements of the Norman dataset and unseen perturbation prediction task. The dual-pathway design effectively separates baseline expression processing from perturbation information processing, the integration components (FeaturerMixer, Gene Interaction Network, PertTransformer) provide sophisticated modeling capabilities, and the biological constraints ensure biological interpretability. The architecture is computationally efficient and can handle both single gene and combinatorial perturbations. This design represents a significant advancement in single-cell perturbation prediction methodology.

Confidence Score: 0.89

## Graph Based Discussion Output

### Round 4

**Data Preprocessing Expert**: The CSAPF framework is now complete and optimized for the Norman dataset. The final implementation includes: 1) **Adaptive normalization**: Each cell's expression is normalized based on its quality metrics (UMI count, gene count, mitochondrial ratio), 2) **Perturbation-specific processing**: Single gene perturbations and combinatorial perturbations are processed differently, 3) **Multi-scale feature extraction**: Features are extracted at gene, pathway, and cellular levels, 4) **Quality control**: Cells with poor coverage or high mitochondrial content are filtered out. The framework successfully handles the dataset's 111,445 cells and 33,694 genes while maintaining biological interpretability.

Confidence Score: 0.87

**Model Architecture Expert**: The dual-pathway architecture is finalized and ready for implementation. The complete architecture follows the exact design pattern: **Top Pathway**: PCAReducer (33,694 → 128 dimensions) → Multi-Scale VAE Encoder (high-level and low-level features) → ContextMLP (generates Perturbation Latent). **Bottom Pathway**: PerturbGene Embed (perturbation IDs → 64D embeddings) → CellContexter (generates Cell Context Latent). **Integration**: FeaturerMixer (combines both latents) → Gene Interaction Network (3-layer GNN) → PertTransformer (4 heads, 2 layers) → PredictionHead → Output Layer (predicted expression profile). This architecture specifically addresses unseen perturbation prediction by learning perturbation-aware representations that can generalize to new perturbation types.

Confidence Score: 0.89

**Training and Optimization Expert**: The training strategy is optimized for the dual-pathway architecture and Norman dataset. The final training approach includes: 1) **Multi-task learning**: Predicts both perturbation effects and perturbation types, 2) **Contrastive learning**: Uses biological similarity to create training pairs, 3) **Curriculum learning**: Progressive training from simple to complex perturbations, 4) **Loss combination**: MSE +

contrastive loss + biological consistency loss. The strategy
effectively handles the dataset's imbalanced perturbation
distribution and ensures robust generalization to unseen
perturbations.

Confidence Score: 0.88

**Pathway Expert**: The biological constraints are fully
integrated into the architecture. The BCAM mechanism within
the PertTransformer ensures: 1) **Biological consistency**:
Predictions align with known biological principles, 2) **
K562-specific context**: Incorporates chronic myelogenous
leukemia pathway information, 3) **Hierarchical attention**:
Operates at multiple biological scales, 4) **
Interpretability**: Provides biological explanations for
predictions. The mechanism successfully distinguishes
between direct and indirect perturbation effects while
maintaining computational efficiency.

Confidence Score: 0.90

**Deep Learning Expert**: The CPIN design is complete and
integrated with the dual-pathway architecture. The DGIM
mechanism provides: 1) **Cell-perturbation interaction
modeling**: Explicitly models how cellular context
influences perturbation effects, 2) **Multi-scale
representation**: Captures information at gene, pathway, and
cellular levels, 3) **Graph-based processing**: Uses GNNs
to model complex biological relationships, 4) **
Generalization capability**: Can handle both seen and unseen
perturbation types. The design successfully addresses the
challenges of single-cell perturbation prediction while
maintaining biological interpretability.

Confidence Score: 0.91

**Critic**: The final architecture design is comprehensive,
innovative, and perfectly suited for the single-cell unseen
perturbation prediction task using the Norman dataset. The
dual-pathway design effectively separates and processes
baseline expression and perturbation information, the
integration components provide sophisticated modeling
capabilities, and the biological constraints ensure
biological interpretability. The architecture can handle
both single gene and combinatorial perturbations,
generalizes to unseen perturbations, and maintains
computational efficiency. This represents a significant
advancement in the field of single-cell perturbation
prediction and provides a solid foundation for future
research.

Confidence Score: 0.89

S.4.2 METHOD DESIGN RESEARCH PLAN

---

**Graph Based Discussion Output**

```
# Novel Method Implementation Framework

## Architecture Overview
The proposed architecture follows a dual-pathway design that
    processes baseline gene expression and perturbation
    information separately before integrating them for final
    prediction.

## 1. Top Pathway: Baseline Expression Processing

### 1.1 PCAReducer
```python
class PCAReducer(nn.Module):
    def __init__(self, input_dim=33694, n_components=128):
        super().__init__()
        self.pca = PCA(n_components=n_components)
        self.input_dim = input_dim
        self.n_components = n_components

    def forward(self, expression_data):
        # expression_data: [batch_size, 33694]
        # Returns: [batch_size, 128]
        return self.pca.fit_transform(expression_data)
```

### 1.2 Multi-Scale VAE Encoder
```python
class MultiScaleVAEEncoder(nn.Module):
    def __init__(self, input_dim=128, latent_dim=64):
        super().__init__()
        # High-level encoder (pathway-level features)
        self.high_encoder = nn.Sequential(
            nn.Linear(input_dim, 256),
            nn.ReLU(),
            nn.Linear(256, 128),
            nn.ReLU(),
            nn.Linear(128, latent_dim * 2) # mean and logvar
        )

        # Low-level encoder (gene-level features)
        self.low_encoder = nn.Sequential(
            nn.Linear(input_dim, 256),
            nn.ReLU(),
            nn.Linear(256, 128),
            nn.ReLU(),
            nn.Linear(128, latent_dim * 2) # mean and logvar
        )

    def forward(self, x):
        # x: [batch_size, 128] (PCA-reduced expression)
        high_mu, high_logvar = self.high_encoder(x).chunk(2, dim
            =-1)
        low_mu, low_logvar = self.low_encoder(x).chunk(2, dim=-1)

        high_z = self.reparameterize(high_mu, high_logvar)
        low_z = self.reparameterize(low_mu, low_logvar)

        return high_z, low_z, high_mu, high_logvar, low_mu,
            low_logvar

    def reparameterize(self, mu, logvar):
        std = torch.exp(0.5 * logvar)
        eps = torch.randn_like(std)
        return mu + eps * std
```

### 1.3 ContextMLP
```python
```

```python
class ContextMLP(nn.Module):
    def __init__(self, high_dim=64, low_dim=64, output_dim=128):
        super().__init__()
        self.mlp = nn.Sequential(
            nn.Linear(high_dim + low_dim, 256),
            nn.ReLU(),
            nn.Linear(256, 128),
            nn.ReLU(),
            nn.Linear(128, output_dim)
        )

    def forward(self, high, low):
        # high: [batch_size, 64] (high-level features)
        # low: [batch_size, 64] (low-level features)
        # Returns: [batch_size, 128] (Perturbation Latent)
        combined = torch.cat([high, low], dim=-1)
        return self.mlp(combined)
```

## 2. Bottom Pathway: Perturbation Information Processing

### 2.1 PerturbGene Embed
```python
class PerturbGeneEmbed(nn.Module):
    def __init__(self, vocab_size=1000, embed_dim=64):
        super().__init__()
        self.embedding = nn.Embedding(vocab_size, embed_dim)
        self.projection = nn.Sequential(
            nn.Linear(embed_dim, 128),
            nn.ReLU(),
            nn.Linear(128, 64)
        )

    def forward(self, perturbation_ids):
        # perturbation_ids: [batch_size] (gene IDs)
        # Returns: [batch_size, 64]
        embedded = self.embedding(perturbation_ids)
        return self.projection(embedded)
```

### 2.2 CellContexter
```python
class CellContexter(nn.Module):
    def __init__(self, input_dim=64, output_dim=128):
        super().__init__()
        self.context_encoder = nn.Sequential(
            nn.Linear(input_dim, 128),
            nn.ReLU(),
            nn.Linear(128, 128),
            nn.ReLU(),
            nn.Linear(128, output_dim)
        )

    def forward(self, perturbation_embed):
        # perturbation_embed: [batch_size, 64]
        # Returns: [batch_size, 128] (Cell Context Latent)
        return self.context_encoder(perturbation_embed)
```

## 3. Integration and Core Processing

### 3.1 FeaturerMixer
```python
class FeaturerMixer(nn.Module):
    def __init__(self, pert_latent_dim=128, cell_latent_dim=128,
         output_dim=256):
        super().__init__()
        self.mixer = nn.Sequential(
            nn.Linear(pert_latent_dim + cell_latent_dim, 512),
            nn.ReLU(),
            nn.Linear(512, 256),
            nn.ReLU(),
```

```
                nn.Linear(256, output_dim)
        )

    def forward(self, pert_latent, cell_latent):
        # pert_latent: [batch_size, 128] (Perturbation Latent)
        # cell_latent: [batch_size, 128] (Cell Context Latent)
        # Returns: [batch_size, 256]
        combined = torch.cat([pert_latent, cell_latent], dim=-1)
        return self.mixer(combined)
```

### 3.2 Gene Interaction Network
```python
class GeneInteractionNetwork(nn.Module):
    def __init__(self, HVG, cell, D_pert, D_gene_feature,
        num_gnn_layers=3):
        super().__init__()
        self.num_gnn_layers = num_gnn_layers
        self.gnn_layers = nn.ModuleList([
            GraphConv(D_gene_feature, D_gene_feature)
            for _ in range(num_gnn_layers)
        ])
        self.attention = nn.MultiheadAttention(D_gene_feature,
            num_heads=8)

    def forward(self, gene, cell, pert):
        # gene: [batch_size, num_genes, D_gene_feature]
        # cell: [batch_size, D_gene_feature]
        # pert: [batch_size, D_gene_feature]
        x = gene

        for layer in self.gnn_layers:
            x = layer(x, cell, pert)
            x = F.relu(x)

        # Apply attention mechanism
        attended, _ = self.attention(x, x, x)

        return attended
```

### 3.3 PertTransformer
```python
class PertTransformer(nn.Module):
    def __init__(self, D_gene_feature=256, num_heads=4,
        num_layers=2):
        super().__init__()
        self.transformer = nn.TransformerEncoder(
            nn.TransformerEncoderLayer(
                d_model=D_gene_feature,
                nhead=num_heads,
                dim_feedforward=512,
                dropout=0.1
            ),
            num_layers=num_layers
        )

    def forward(self, x):
        # x: [batch_size, seq_len, D_gene_feature]
        return self.transformer(x)
```

### 3.4 PredictionHead
```python
class PredictionHead(nn.Module):
    def __init__(self, feature_dim=256, num_heads=4, num_layers=2,
        output_dim=33694):
        super().__init__()
        self.attention = nn.MultiheadAttention(feature_dim,
            num_heads)
        self.layers = nn.ModuleList([
            nn.Linear(feature_dim, feature_dim)
            for _ in range(num_layers)
```

```
            ])
        self.output_layer = nn.Linear(feature_dim, output_dim)

    def forward(self, final_genes):
        # final_genes: [batch_size, seq_len, feature_dim]
        attended, _ = self.attention(final_genes, final_genes,
            final_genes)

        for layer in self.layers:
            attended = F.relu(layer(attended))

        # Global average pooling
        pooled = attended.mean(dim=1) # [batch_size, feature_dim]

        # Final prediction
        output = self.output_layer(pooled) # [batch_size, 33694]

        return output
```

## 4. Complete Model Architecture

### 4.1 CellForge Model
```python
class CellForgeModel(nn.Module):
    def __init__(self, config):
        super().__init__()
        self.config = config

        # Top pathway components
        self.pca_reducer = PCAReducer(input_dim=33694,
            n_components=128)
        self.vae_encoder = MultiScaleVAEEncoder(input_dim=128,
            latent_dim=64)
        self.context_mlp = ContextMLP(high_dim=64, low_dim=64,
            output_dim=128)

        # Bottom pathway components
        self.pert_embed = PerturbGeneEmbed(vocab_size=1000,
            embed_dim=64)
        self.cell_contexter = CellContexter(input_dim=64,
            output_dim=128)

        # Integration and processing components
        self.feature_mixer = FeaturerMixer(pert_latent_dim=128,
            cell_latent_dim=128, output_dim=256)
        self.gene_interaction = GeneInteractionNetwork(
            HVG=config.HVG, cell=config.cell, D_pert=config.D_pert,
            D_gene_feature=256, num_gnn_layers=3
        )
        self.pert_transformer = PertTransformer(D_gene_feature=256,
            num_heads=4, num_layers=2)
        self.prediction_head = PredictionHead(feature_dim=256,
            num_heads=4, num_layers=2, output_dim=33694)

    def forward(self, expression_data, perturbation_ids):
        # Top pathway: Process baseline expression
        pca_reduced = self.pca_reducer(expression_data)
        high_z, low_z, high_mu, high_logvar, low_mu, low_logvar =
            self.vae_encoder(pca_reduced)
        pert_latent = self.context_mlp(high_z, low_z)

        # Bottom pathway: Process perturbation information
        pert_embed = self.pert_embed(perturbation_ids)
        cell_latent = self.cell_contexter(pert_embed)

        # Integration and processing
        mixed_features = self.feature_mixer(pert_latent,
            cell_latent)

        # Reshape for GNN processing
        gene_features = mixed_features.unsqueeze(1).repeat(1, 1000,
            1) # [batch_size, 1000, 256]
```

```
            # Core processing
            gnn_output = self.gene_interaction(gene_features,
                cell_latent, pert_latent)
            transformer_output = self.pert_transformer(gnn_output)
            predicted_expression = self.prediction_head(
                transformer_output)

            return predicted_expression, high_mu, high_logvar, low_mu,
                low_logvar
    ```

    This implementation framework provides a complete, implementable
        architecture that exactly matches the proposed design
       pattern and addresses the specific requirements of single-
       cell unseen perturbation prediction using the Norman dataset.
```

S.4.3 DETAILED RESEARCH PLAN

---

**Graph Based Discussion Output**

```
# Detailed Research Plan for Dual-Pathway Architecture

## 1. Research Objectives

### 1.1 Primary Objective
Develop a novel dual-pathway neural network architecture for
    predicting gene expression profiles of individual K562 cells
     following CRISPR interference (CRISPRi) perturbations,
    specifically designed to handle unseen perturbations and
    unseen cell contexts.

### 1.2 Secondary Objectives
- Design a preprocessing framework that supports both pathways
    of the architecture
- Implement biological constraints to ensure meaningful
    predictions
- Develop training strategies that optimize both pathways
    simultaneously
- Create evaluation metrics that assess both predictive accuracy
     and biological relevance

## 2. Dataset and Task Specification

### 2.1 Dataset Details
- **Source**: Norman et al. (2019, Science) dataset
- **Modality**: RNA (scRNA-seq gene expression data)
- **Perturbation Type**: CRISPRa (CRISPR activation)
- **Cell Line**: K562 (chronic myelogenous leukemia lymphoblasts
    )
- **Scale**: 111,445 cells  33,694 genes
- **Perturbation Types**: Single gene (e.g., ARID1A, BCORL1) and
     combinatorial (e.g., SET_KLF1, CEBPB_OSR2)

### 2.2 Task Definition
- **Input**: Baseline gene expression profile of an unperturbed
    K562 cell and the identity of the target gene(s) for
    perturbation
- **Output**: Predicted gene expression profile after
    perturbation
- **Evaluation Scenarios**:
 - Unseen Perturbations: Predict effects of gene perturbations
     not present during training
 - Unseen Cell Contexts: Predict responses in cells with gene
     expression profiles not observed during training

### 2.3 Evaluation Metrics
- **Predictive Performance**: MSE, PCC, R
- **Biological Relevance**: MSE_DE, PCC_DE, R_DE (for
    differentially expressed genes)

## 3. Dual-Pathway Architecture Design

### 3.1 Top Pathway: Baseline Expression Processing
**Purpose**: Process baseline gene expression data to generate
    perturbation-aware cellular context

**Components**:
1. **PCAReducer**
  - Input: 33,694-dimensional gene expression data
  - Output: 128-dimensional PCA-reduced representation
  - Function: Dimensionality reduction while preserving
    biological variation

2. **Multi-Scale VAE Encoder**
  - Input: 128-dimensional PCA-reduced data
  - Output: High-level (64D) and low-level (64D) latent
    representations
```

```
      - Function: Capture both pathway-level and gene-level features

  3. **ContextMLP**
     - Input: High-level and low-level features (64D each)
     - Output: 128-dimensional "Perturbation Latent" representation
     - Function: Generate perturbation-aware cellular context

  ### 3.2 Bottom Pathway: Perturbation Information Processing
  **Purpose**: Process perturbation information to generate cell
     context-aware perturbation effects

  **Components**:
  1. **PerturbGene Embed**
     - Input: Perturbation gene IDs
     - Output: 64-dimensional perturbation embeddings
     - Function: Map perturbation identities to dense
        representations

  2. **CellContexter**
     - Input: 64-dimensional perturbation embeddings
     - Output: 128-dimensional "Cell Context Latent" representation
     - Function: Generate cell context-aware perturbation effects

  ### 3.3 Integration and Core Processing
  **Purpose**: Combine and process information from both pathways

  **Components**:
  1. **FeaturerMixer**
     - Input: Perturbation Latent (128D) + Cell Context Latent (128
       D)
     - Output: 256-dimensional fused representation
     - Function: Combine information from both pathways

  2. **Gene Interaction Network**
     - Input: 256-dimensional fused features
     - Output: 256-dimensional gene interaction features
     - Function: Model gene-gene interactions using GNN (3 layers)

  3. **PertTransformer**
     - Input: 256-dimensional gene interaction features
     - Output: 256-dimensional transformed features
     - Function: Capture long-range dependencies (4 heads, 2 layers
       )

  4. **PredictionHead**
     - Input: 256-dimensional transformed features
     - Output: 33,694-dimensional predicted expression profile
     - Function: Final prediction layer

## 4. Implementation Plan

### 4.1 Phase 1: Data Preprocessing (Weeks 1-2)
**Objective**: Implement the Dual-Pathway Preprocessing
   Framework (DPPF)

**Tasks**:
1. **Top Pathway Preprocessing**
   - Quality control: Filter cells with poor coverage or high
      mitochondrial content
   - Normalization: Log normalization and library size correction
   - PCA preparation: Reduce 33,694 genes to 128 principal
      components
   - HVG selection: Identify highly variable genes for pathway
      annotation

2. **Bottom Pathway Preprocessing**
   - Gene ID mapping: Create mapping from perturbation names to
      gene IDs
   - Combinatorial perturbation handling: Process multi-gene
      perturbations
   - Embedding preparation: Prepare data for PerturbGene Embed

3. **Integration Preparation**
```

```
      - Feature alignment: Ensure compatibility between pathways
      - Batch processing: Implement efficient data loading
      - Perturbation strength estimation: Infer perturbation
         efficiency

  **Deliverables**:
  - Preprocessed dataset with both pathway inputs
  - Data loading pipeline
  - Quality control metrics

  ### 4.2 Phase 2: Architecture Implementation (Weeks 3-5)
  **Objective**: Implement the complete dual-pathway architecture

  **Tasks**:
  1. **Top Pathway Components**
     - PCAReducer: Implement PCA dimensionality reduction
     - Multi-Scale VAE Encoder: Implement high-level and low-level
        encoders
     - ContextMLP: Implement perturbation-aware context generation

  2. **Bottom Pathway Components**
     - PerturbGene Embed: Implement perturbation embedding layer
     - CellContexter: Implement cell context-aware processing

  3. **Integration Components**
     - FeaturerMixer: Implement feature fusion
     - Gene Interaction Network: Implement GNN with 3 layers
     - PertTransformer: Implement transformer with 4 heads, 2
        layers
     - PredictionHead: Implement final prediction layer

  **Deliverables**:
  - Complete model architecture
  - Forward pass implementation
  - Model parameter specifications

  ### 4.3 Phase 3: Training Strategy Implementation (Weeks 6-7)
  **Objective**: Implement the Dual-Pathway Training Framework (
     DPTF)

  **Tasks**:
  1. **Multi-task Learning**
     - Main task: Perturbation effect prediction
     - Auxiliary task: Perturbation type classification

  2. **Pathway-specific Losses**
     - VAE reconstruction loss for Top Pathway
     - Contrastive loss for Bottom Pathway
     - Integration loss for FeaturerMixer

  3. **Curriculum Learning**
     - Progressive training from single to combinatorial
        perturbations
     - Adaptive weighting based on training progress

  **Deliverables**:
  - Training pipeline
  - Loss function implementations
  - Curriculum learning scheduler

  ### 4.4 Phase 4: Biological Constraints Integration (Weeks 8-9)
  **Objective**: Implement the Biologically-Constrained Attention
     Mechanism (BCAM)

  **Tasks**:
  1. **Biological Prior Integration**
     - K562-specific pathway information
     - Chronic myelogenous leukemia context
     - Protein-protein interaction networks

  2. **Attention Mechanism Enhancement**
     - Biological bias injection into PertTransformer
     - Hierarchical attention at multiple biological scales
```

```
      - Biological consistency validation

  **Deliverables**:
  - BCAM implementation
  - Biological prior database
  - Attention visualization tools

  ### 4.5 Phase 5: Model Training and Optimization (Weeks 10-12)
  **Objective**: Train and optimize the complete model

  **Tasks**:
  1. **Hyperparameter Tuning**
     - Learning rate optimization
     - Batch size selection
     - Architecture parameter tuning

  2. **Training Monitoring**
     - Loss tracking for both pathways
     - Biological consistency monitoring
     - Generalization assessment

  3. **Model Selection**
     - Cross-validation
     - Early stopping
     - Best model checkpointing

  **Deliverables**:
  - Trained model
  - Training logs and metrics
  - Hyperparameter configurations

  ### 4.6 Phase 6: Evaluation and Analysis (Weeks 13-14)
  **Objective**: Comprehensive evaluation of the model

  **Tasks**:
  1. **Predictive Performance Evaluation**
     - MSE, PCC, R on test set
     - Unseen perturbation evaluation
     - Unseen cell context evaluation

  2. **Biological Relevance Assessment**
     - MSE_DE, PCC_DE, R_DE for differentially expressed genes
     - Pathway enrichment analysis
     - Biological consistency validation

  3. **Comparative Analysis**
     - Baseline method comparison
     - Ablation studies
     - Component contribution analysis

  **Deliverables**:
  - Evaluation results
  - Comparative analysis report
  - Biological interpretation

  ## 5. Technical Specifications

  ### 5.1 Computational Requirements
  - **Hardware**: GPU with at least 16GB memory
  - **Software**: PyTorch, scikit-learn, scanpy, pandas, numpy
  - **Storage**: At least 100GB for dataset and model checkpoints

  ### 5.2 Model Parameters
  - **Input Dimensions**: 33,694 (genes)  111,445 (cells)
  - **PCA Components**: 128
  - **VAE Latent Dimensions**: 64 (high-level) + 64 (low-level)
  - **Embedding Dimensions**: 64 (perturbation) + 128 (context)
  - **GNN Layers**: 3
  - **Transformer Heads**: 4
  - **Transformer Layers**: 2
  - **Output Dimensions**: 33,694 (predicted expression)

  ### 5.3 Training Configuration
```

- **Batch Size**: 32
- **Learning Rate**: 1e-3 (with OneCycle scheduler)
- **Epochs**: 100 (with early stopping)
- **Optimizer**: AdamW
- **Loss Weights**: Adaptive based on training progress

## 6. Risk Assessment and Mitigation

### 6.1 Technical Risks
1. **Computational Complexity**
   - Risk: Model too large for available hardware
   - Mitigation: Implement gradient checkpointing and model
     parallelism

2. **Training Instability**
   - Risk: Training divergence or poor convergence
   - Mitigation: Implement gradient clipping and learning rate
     scheduling

3. **Overfitting**
   - Risk: Model memorizes training data
   - Mitigation: Implement dropout, weight decay, and early
     stopping

### 6.2 Biological Risks
1. **Biological Inconsistency**
   - Risk: Predictions violate biological principles
   - Mitigation: Implement biological constraints and validation

2. **Generalization Failure**
   - Risk: Poor performance on unseen perturbations
   - Mitigation: Implement contrastive learning and adversarial
     training

### 6.3 Data Risks
1. **Data Quality Issues**
   - Risk: Poor quality data affects model performance
   - Mitigation: Implement robust quality control and data
     augmentation

2. **Class Imbalance**
   - Risk: Imbalanced perturbation types
   - Mitigation: Implement weighted sampling and focal loss

## 7. Success Metrics

### 7.1 Quantitative Metrics
- **Predictive Accuracy**: MSE < 0.1, PCC > 0.8, R > 0.7
- **Biological Relevance**: MSE_DE < 0.15, PCC_DE > 0.75, R_DE >
  0.65
- **Generalization**: Performance on unseen perturbations within
  10% of seen perturbations

### 7.2 Qualitative Metrics
- **Biological Interpretability**: Predictions align with known
  biological pathways
- **Computational Efficiency**: Training time < 24 hours on
  single GPU
- **Reproducibility**: Results reproducible across multiple runs

## 8. Timeline and Milestones

### 8.1 Week 1-2: Data Preprocessing
- **Milestone**: Complete DPPF implementation
- **Deliverable**: Preprocessed dataset ready for training

### 8.2 Week 3-5: Architecture Implementation
- **Milestone**: Complete dual-pathway architecture
- **Deliverable**: Functional model implementation

### 8.3 Week 6-7: Training Strategy
- **Milestone**: Complete DPTF implementation
- **Deliverable**: Training pipeline ready

```
### 8.4 Week 8-9: Biological Constraints
- **Milestone**: Complete BCAM implementation
- **Deliverable**: Biologically-constrained model

### 8.5 Week 10-12: Model Training
- **Milestone**: Trained and optimized model
- **Deliverable**: Best model checkpoint

### 8.6 Week 13-14: Evaluation
- **Milestone**: Comprehensive evaluation complete
- **Deliverable**: Final results and analysis

## 9. Expected Outcomes

### 9.1 Scientific Contributions
- Novel dual-pathway architecture for single-cell perturbation
    prediction
- Integration of biological constraints into deep learning
    models
- Comprehensive evaluation framework for perturbation prediction

### 9.2 Technical Contributions
- Efficient preprocessing pipeline for large-scale single-cell
    data
- Advanced training strategies for multi-pathway architectures
- Robust evaluation metrics for biological relevance

### 9.3 Practical Applications
- Tool for predicting effects of novel perturbations
- Framework for understanding cellular responses to
    perturbations
- Foundation for drug discovery and therapeutic development

This detailed research plan provides a comprehensive roadmap for
    implementing the dual-pathway architecture and achieving
    the research objectives.
```

## S.4.4 EXAMPLE MODEL FRAMEWORK AND PSEUDOCODE

---

**Graph Based Discussion Output**

```
# Architecture Flowchart and Pseudocode

## 1. Mermaid Flowchart

```mermaid
graph TD
   A[Raw Gene Expression Data
33,694 genes  111,445 cells]
       --> B[Quality Control
Filter low-quality cells]
   B --> C[Log Normalization
Library size correction]
   C --> D[PCA Reduction
33,694  128 dimensions]

   E[Perturbation Information
Gene IDs] --> F[Gene ID
       Mapping
Handle combinatorial perturbations]
   F --> G[Perturbation Embedding
Gene IDs  64D vectors]

   D --> H[Multi-Scale VAE Encoder
High-level + Low-level
       features]
   H --> I[ContextMLP
Generate Perturbation Latent 128D]

   G --> J[CellContexter
Generate Cell Context Latent 128D]

   I --> K[FeaturerMixer
Combine both latents  256D]
   J --> K

   K --> L[Gene Interaction Network
GNN with 3 layers]
   L --> M[PertTransformer
4 heads, 2 layers]
   M --> N[PredictionHead
Final prediction layer]
   N --> O[Predicted Expression Profile
33,694 dimensions]

   P[Biological Priors
K562 pathways, PPI networks] --> Q[
       BCAM
Biological constraints]
   Q --> M

   R[Training Strategy
Multi-task learning] --> S[Loss
       Functions
VAE + Contrastive + Integration]
   S --> T[Optimization
AdamW + OneCycle scheduler]
   T --> U[Model Training
100 epochs with early stopping]
```

## 2. Detailed Pseudocode

### 2.1 Main Training Loop
```python
def train_dual_pathway_model():
    # Initialize model and data
    model = CellForgeModel(config)
    train_loader, val_loader = prepare_data_loaders()
    optimizer = torch.optim.AdamW(model.parameters(), lr=1e-3)
    scheduler = torch.optim.lr_scheduler.OneCycleLR(optimizer,
        max_lr=1e-3, epochs=100)

    # Training loop
    for epoch in range(100):
        model.train()
        train_loss = 0

        for batch_idx, (expression_data, perturbation_ids,
            target_expression) in enumerate(train_loader):
            # Forward pass through dual-pathway architecture
            predicted_expression, high_mu, high_logvar, low_mu,
                low_logvar = model(expression_data,
                perturbation_ids)

            # Calculate losses
            reconstruction_loss = F.mse_loss(predicted_expression,
                target_expression)
            vae_loss = vae_loss_function(high_mu, high_logvar,
                low_mu, low_logvar)
```

---

```
6912
6913            contrastive_loss = contrastive_loss_function(
6914                perturbation_ids)
                 biological_loss = biological_consistency_loss(
6915                predicted_expression, perturbation_ids)
6916
                 # Total loss with adaptive weighting
6917             total_loss = reconstruction_loss + 0.1 * vae_loss +
6918                0.05 * contrastive_loss + 0.02 * biological_loss
6919
                 # Backward pass
6920             optimizer.zero_grad()
                 total_loss.backward()
6921             torch.nn.utils.clip_grad_norm_(model.parameters(),
6922                max_norm=1.0)
6923             optimizer.step()
6924             train_loss += total_loss.item()
6925
             # Validation
6926         val_loss = validate_model(model, val_loader)
6927         scheduler.step()
6928
             # Early stopping check
6929         if val_loss < best_val_loss:
6930             best_val_loss = val_loss
                 save_model_checkpoint(model, epoch)
6931
6932         print(f'Epoch {epoch}: Train Loss = {train_loss/len(
6933             train_loader):.4f}, Val Loss = {val_loss:.4f}')
     ```
6934
     ### 2.2 Dual-Pathway Forward Pass
6935 ```python
     def dual_pathway_forward_pass(expression_data, perturbation_ids):
6936
6937     # Top Pathway: Baseline Expression Processing
         pca_reduced = pca_reducer(expression_data) # [batch_size,
6938         33,694]  [batch_size, 128]
6939
         # Multi-Scale VAE Encoder
6940     high_z, low_z, high_mu, high_logvar, low_mu, low_logvar =
6941         multi_scale_vae_encoder(pca_reduced)
6942     # high_z, low_z: [batch_size, 64] each
6943
         # ContextMLP
6944     pert_latent = context_mlp(high_z, low_z) # [batch_size, 128]
6945
         # Bottom Pathway: Perturbation Information Processing
6946     pert_embed = perturb_gene_embed(perturbation_ids) # [
6947         batch_size, 64]
6948     cell_latent = cell_contexter(pert_embed) # [batch_size, 128]
6949
         # Integration and Core Processing
6950     mixed_features = featurer_mixer(pert_latent, cell_latent) # [
6951         batch_size, 256]
6952
         # Reshape for GNN processing
6953     gene_features = mixed_features.unsqueeze(1).repeat(1, 1000,
6954         1) # [batch_size, 1000, 256]
6955
         # Gene Interaction Network
6956     gnn_output = gene_interaction_network(gene_features,
6957         cell_latent, pert_latent) # [batch_size, 1000, 256]
6958
         # PertTransformer with biological constraints
6959     transformer_output = pert_transformer(gnn_output) # [
6960         batch_size, 1000, 256]
6961
         # PredictionHead
6962     predicted_expression = prediction_head(transformer_output) #
6963         [batch_size, 33,694]
6964     return predicted_expression, high_mu, high_logvar, low_mu,
6965         low_logvar
```

```
```

### 2.3 Data Preprocessing Pipeline
```python
def dual_pathway_preprocessing_pipeline(adata):
    # Quality control
    sc.pp.filter_cells(adata, min_genes=200)
    sc.pp.filter_genes(adata, min_cells=3)

    # Normalization
    adata.raw = adata.copy()
    sc.pp.normalize_total(adata, target_sum=1e4)
    sc.pp.log1p(adata)

    # HVG selection
    sc.pp.highly_variable_genes(adata, n_top_genes=3000)
    adata = adata[:, adata.var.highly_variable]

    # Top Pathway preprocessing
    expression_data = adata.X # [n_cells, n_genes]
    pca = PCA(n_components=128)
    pca_reduced = pca.fit_transform(expression_data)

    # Bottom Pathway preprocessing
    perturbation_mapping = create_perturbation_mapping(adata.obs['
        perturbation'])
    perturbation_ids = map_perturbations_to_ids(adata.obs['
        perturbation'], perturbation_mapping)

    # Integration preparation
    batch_info = adata.obs['batch'] if 'batch' in adata.obs else
        None
    cell_quality = calculate_cell_quality_metrics(adata)

    return {
        'expression_data': expression_data,
        'pca_reduced': pca_reduced,
        'perturbation_ids': perturbation_ids,
        'perturbation_mapping': perturbation_mapping,
        'batch_info': batch_info,
        'cell_quality': cell_quality
    }
```

### 2.4 Biological Constraints Integration
```python
def biological_constraints_attention(transformer_input,
     biological_priors):
    # Load biological priors
    k562_pathways = load_k562_pathway_info()
    ppi_network = load_protein_protein_interactions()
    tf_binding_sites = load_transcription_factor_binding_sites()

    # Create attention bias
    attention_bias = create_biological_attention_bias(
        transformer_input, k562_pathways, ppi_network,
            tf_binding_sites
    )

    # Apply biological constraints to attention
    constrained_attention = apply_biological_constraints(
        transformer_input, attention_bias
    )

    return constrained_attention

def biological_consistency_loss(predicted_expression,
     perturbation_ids):
    # Calculate pathway consistency
    pathway_consistency = calculate_pathway_consistency(
        predicted_expression, perturbation_ids)

    # Calculate interaction consistency
```

```
        interaction_consistency = calculate_interaction_consistency(
            predicted_expression, perturbation_ids)

        # Calculate regulatory consistency
        regulatory_consistency = calculate_regulatory_consistency(
            predicted_expression, perturbation_ids)

        # Combined biological consistency loss
        biological_loss = pathway_consistency +
            interaction_consistency + regulatory_consistency

        return biological_loss
```

### 2.5 Evaluation Pipeline
```python
def comprehensive_evaluation(model, test_loader,
     biological_databases):
    model.eval()
    all_predictions = []
    all_targets = []
    all_perturbations = []

    with torch.no_grad():
        for expression_data, perturbation_ids, target_expression
            in test_loader:
            predicted_expression, _, _, _, _ = model(
                expression_data, perturbation_ids)

            all_predictions.append(predicted_expression.cpu().numpy
                ())
            all_targets.append(target_expression.cpu().numpy())
            all_perturbations.append(perturbation_ids.cpu().numpy()
                )

    # Concatenate all predictions
    predictions = np.concatenate(all_predictions, axis=0)
    targets = np.concatenate(all_targets, axis=0)
    perturbations = np.concatenate(all_perturbations, axis=0)

    # Calculate predictive performance metrics
    mse = mean_squared_error(targets, predictions)
    pcc = pearson_correlation_coefficient(targets, predictions)
    r2 = r2_score(targets, predictions)

    # Calculate biological relevance metrics
    de_genes = identify_differentially_expressed_genes(targets,
        perturbations)
    mse_de = mean_squared_error(targets[:, de_genes], predictions
        [:, de_genes])
    pcc_de = pearson_correlation_coefficient(targets[:, de_genes],
        predictions[:, de_genes])
    r2_de = r2_score(targets[:, de_genes], predictions[:,
        de_genes])

    # Biological consistency evaluation
    pathway_enrichment = evaluate_pathway_enrichment(predictions,
        perturbations, biological_databases)
    interaction_consistency = evaluate_interaction_consistency(
        predictions, perturbations, biological_databases)

    return {
        'predictive_metrics': {'mse': mse, 'pcc': pcc, 'r2': r2},
        'biological_metrics': {'mse_de': mse_de, 'pcc_de': pcc_de,
            'r2_de': r2_de},
        'biological_consistency': {'pathway_enrichment':
            pathway_enrichment, 'interaction_consistency':
            interaction_consistency}
    }
```

## 3. Implementation Checklist
```

```
### 3.1 Data Preprocessing
- [ ] Quality control implementation
- [ ] Normalization pipeline
- [ ] PCA reduction
- [ ] HVG selection
- [ ] Perturbation mapping
- [ ] Batch processing

### 3.2 Architecture Components
- [ ] PCAReducer
- [ ] Multi-Scale VAE Encoder
- [ ] ContextMLP
- [ ] PerturbGene Embed
- [ ] CellContexter
- [ ] FeaturerMixer
- [ ] Gene Interaction Network
- [ ] PertTransformer
- [ ] PredictionHead

### 3.3 Training Components
- [ ] Multi-task learning
- [ ] Pathway-specific losses
- [ ] Curriculum learning
- [ ] Adaptive weighting
- [ ] Early stopping
- [ ] Model checkpointing

### 3.4 Biological Constraints
- [ ] BCAM implementation
- [ ] Biological prior integration
- [ ] Attention bias injection
- [ ] Biological consistency validation

### 3.5 Evaluation
- [ ] Predictive performance metrics
- [ ] Biological relevance metrics
- [ ] Comparative analysis
- [ ] Ablation studies

This comprehensive flowchart and pseudocode provide detailed
    implementation guidance for the dual-pathway architecture.
```

## T  DETAILED OUTPUTS FROM OTHER RESEARCH AGENTS

We provided the following research agents with the same input as CellForge(task description and dataset, as presented in Appendix R.1).

**Biomni**  Despite providing a detailed natural language report (as below), Biomni failed to generate any executable code for the single-cell perturbation prediction task. Consequently, it achieved a benchmark score of **0**. This indicates a fundamental limitation in its ability on single cell perturbation prediction tasks.

---

**Research Agent Output**

Norman et al. dataset(gene knockout, scRNA-seq)
```
Launching workflow executor (this may take a while)...
 Answer
 # CRISPRi Gene Expression Prediction: Comprehensive Analysis
    and Novel Method Report

## Executive Summary

I have developed a comprehensive framework for predicting gene
    expression profiles following CRISPR interference (CRISPRi)
    in K562 cells, based on the methodology established by
    Norman et al. (2019) Science. The framework addresses the
    specified prediction scenarios and evaluation metrics while
    providing a novel multi-model approach for robust gene
    expression prediction.

## Key Contributions

### 1. Novel Methodology
- **Multi-Model Ensemble Approach**: Combines Random Forest and
    Ridge Regression with interaction feature engineering
- **Perturbation-Gene Interaction Modeling**: Explicit modeling
    of how perturbations affect individual genes
- **Dual Evaluation Framework**: Separate assessment for unseen
    perturbations and unseen cell contexts

### 2. Comprehensive Implementation
The framework includes:
- Complete Python implementation ('crispri_prediction_framework.
    py')
- Automated data preprocessing pipeline
- Multiple model architectures for comparison
- Comprehensive evaluation suite with all specified metrics
- Publication-ready visualization tools

### 3. Technical Innovation
- **Feature Engineering**: Novel combination of baseline
    expression and perturbation targets
- **Cross-Validation Strategy**: Stratified by perturbation type
    to ensure robust evaluation
- **Multi-Scale Evaluation**: Separate metrics for all genes and
    differentially expressed genes

## Data and Methods

### Dataset Information
- **Source**: Norman et al. (2019) Science, GEO accession
    GSE133344 [1]
- **Technology**: CRISPRi + single-cell RNA sequencing (Perturb-
    seq)
- **Scale**: 112 target genes, single and combinatorial
    perturbations in K562 cells
- **Data Access**: Available via GEO, Figshare, and GitHub
    repositories [1]

### Model Architecture

#### Model 1: Multi-output Random Forest
```

- Handles non-linear gene-gene interactions
- Provides feature importance rankings
- Robust to outliers and missing values
- Parameters: 100 estimators, max depth 15, optimized for genomics data

#### Model 2: Ridge Regression with Interaction Terms
- Linear baseline model with explicit perturbation-gene interactions
- Computationally efficient and interpretable
- Regularization parameter =1.0 optimized via cross-validation

### Evaluation Framework

#### Evaluation Scenarios
1. **Unseen Perturbations**: Train on subset of perturbations, test on held-out perturbations
2. **Unseen Cell Contexts**: Train on subset of cells, test on cells with different baseline expression

#### Evaluation Metrics (All Implemented)
- **MSE**: Mean Squared Error across all genes
- **PCC**: Pearson Correlation Coefficient (average across genes)
- **R**: Coefficient of determination
- **MSE_DE**: MSE for differentially expressed genes only
- **PCC_DE**: PCC for differentially expressed genes only
- **R_DE**: R for differentially expressed genes only

## Implementation Details

### Data Preprocessing Pipeline
1. **Quality Control**: Filter cells with <500 genes and genes in <10 cells
2. **Normalization**: Library size normalization + log1p transformation
3. **Feature Selection**: Top 2000 highly variable genes
4. **Perturbation Encoding**: One-hot encoding of perturbation targets

### Novel Features
- **Interaction Terms**: Explicit modeling of perturbation-baseline expression interactions
- **Multi-Scale Validation**: Cross-validation strategies tailored to each evaluation scenario
- **Comprehensive Metrics**: Implementation of all six specified evaluation metrics

## Expected Performance

Based on similar studies and our framework design:
- **Unseen Perturbations**: R > 0.6, PCC > 0.7 for DE genes
- **Unseen Cell Contexts**: R > 0.5, PCC > 0.65 for DE genes
- **Model Comparison**: Random Forest expected to outperform Ridge on non-linear interactions

## Generated Deliverables

All files saved to '/tmp/agent_outputs/0fb4d993-3436-4fe9-8b08-8436bb1aed44/':

1. **`crispri_prediction_framework.py`** - Complete implementation with CRISPRiAnalyzer class
2. **[methodology_report.md](https://biomni-logs.s3.amazonaws.com/3220104147_zju_edu_cn/logs/session_0fb4d993-3436-4fe9-8b08-8436bb1aed44/files/methodology_report.md?AWSAccessKeyId=AKIA5OMXL5OSOMU6RIOE&Signature=HUmJXc0c2%2FLHweYs5rqAEOWab2I%3D&Expires=1751037731)** - Detailed methodology and theoretical framework
3. **[evaluation_framework.json](https://biomni-logs.s3.amazonaws.com/3220104147_zju_edu_cn/logs/session_0fb4d993-3436-4fe9-8b08-8436bb1aed44/files/evaluation_framework.json?

```
        AWSAccessKeyId=AKIA5OMXL5OSOMU6RIOE&Signature=9wu88Nh6Qm%2
        BaJ4gmftv5E0nx0Rg%3D&Expires=1751037731)** - Structured
        evaluation specifications
  4. **[comprehensive_analysis_framework.png](https://biomni-logs.
     s3.amazonaws.com/3220104147_zju_edu_cn/logs/session_0fb4d993
     -3436-4fe9-8b08-8436bb1aed44/files/
     comprehensive_analysis_framework.png?AWSAccessKeyId=
     AKIA5OMXL5OSOMU6RIOE&Signature=LPR2e6KfMpvEzlMEGMp%2BQWQ59u8
     %3D&Expires=1751037731)** - Visual framework overview
  5. **[technical_specifications.md](https://biomni-logs.s3.
     amazonaws.com/3220104147_zju_edu_cn/logs/session_0fb4d993
     -3436-4fe9-8b08-8436bb1aed44/files/technical_specifications.
     md?AWSAccessKeyId=AKIA5OMXL5OSOMU6RIOE&Signature=1z%2Byoe%2
     FbbYgSa%2F4IG8ju9oJ2VxE%3D&Expires=1751037731)** -
     Implementation details and requirements

  ## Significance and Impact

  ### Scientific Contributions
  1. **Standardized Evaluation**: Provides benchmark framework for
      CRISPRi prediction methods
  2. **Baseline Models**: Establishes performance baselines for
     comparison with deep learning approaches
  3. **Biological Insights**: Framework designed to reveal gene
     regulatory network patterns in K562 cells

  ### Practical Applications
  - **Drug Discovery**: Predict off-target effects of CRISPR-based
     therapeutics
  - **Functional Genomics**: Systematic prediction of gene
     function through perturbation
  - **Personalized Medicine**: Cell-context specific perturbation
     response prediction

  ## Future Extensions

  ### Deep Learning Integration
  - Transformer architectures for sequence-based modeling
  - Graph neural networks for gene regulatory networks
  - Variational autoencoders for representation learning

  ### Multi-Modal Data
  - Integration with protein abundance data
  - Chromatin accessibility incorporation
  - Drug response profile integration

  ## Conclusion

  This comprehensive framework provides a robust, scientifically
      rigorous approach to CRISPRi gene expression prediction. The
       implementation is ready for immediate use with the Norman
      et al. (2019) dataset and establishes both methodological
      foundations and performance benchmarks for future research
      in perturbation-based functional genomics.

  The framework successfully addresses all specified requirements
      while introducing novel methodological contributions that
      advance the state-of-the-art in computational biology and
      functional genomics.

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

```
Launching workflow executor (this may take a while)...
 Answer
  # Chemical Perturbation Gene Expression Prediction: Complete
     Analysis, Method, and Code Report

## Executive Summary

I have developed a comprehensive machine learning framework for
    predicting transcriptional responses to chemical
    perturbations in human cell lines. This framework is
    specifically designed for the sci-Plex methodology from
    Srivatsan et al. (Science 2020) [1] and addresses the
    critical challenge of predicting gene expression changes
    following compound treatments across different cellular
    contexts.

## Key Contributions

### 1. Novel Methodology Framework
- **Multi-target regression approach** combining Random Forest
    and Ridge Regression models
- **Innovative feature engineering** integrating baseline
    expression, compound characteristics, and interaction terms
- **Dual evaluation framework** addressing both unseen
    perturbations and unseen cell contexts
- **Comprehensive metric system** with six evaluation measures (
    MSE, PCC, R, MSE_DE, PCC_DE, R_DE)

### 2. Technical Innovation
- **Interaction feature engineering**: Novel approach combining
    baseline cellular state with compound effects
- **Multi-scenario evaluation**: Addresses real-world
    application scenarios for drug discovery
- **DE-specific metrics**: Focused evaluation on differentially
    expressed genes for biological relevance

### 3. Complete Implementation
- **Production-ready code**: Full Python implementation with
    scikit-learn integration
- **Comprehensive documentation**: Detailed methodology and
    technical specifications
- **Visualization framework**: Publication-ready analysis
    figures
- **Reproducible pipeline**: Structured evaluation protocols

## Data and Methods

### Dataset: Srivatsan et al. sci-Plex
- **Source**: GEO accession GSE132566 [1]
- **Scale**: 650,000+ single-cell profiles across 188 compounds
- **Cell lines**: A549 (lung), K562 (leukemia), MCF7 (breast
    cancer)
- **Design**: 4 dose levels per compound (10 nM to 10 M)

### Model Architecture

#### Random Forest Regressor
```python
RandomForestRegressor(
    n_estimators=200,
    max_depth=20,
    min_samples_split=5,
    random_state=42,
    n_jobs=-1
)
```

#### Ridge Regression with Interactions
```python
Ridge(
```

```
        alpha=1.0,
        fit_intercept=True,
        random_state=42
    )
    ```

### Feature Engineering Pipeline
1. **Baseline Expression**: Top 1,000 highly variable genes
2. **Compound Features**: Encoded compound ID, log-transformed
   dose, cell line
3. **Interaction Features**: Element-wise product of baseline
   expression and dose
4. **Total Features**: ~3,000 features per sample

## Evaluation Framework

### Two Critical Scenarios
1. **Unseen Perturbations**: Predict responses to compounds not
   seen during training
2. **Unseen Cell Contexts**: Predict responses in cellular
   contexts not represented in training

### Six Comprehensive Metrics
- **Overall Performance**: MSE, PCC, R
- **DE-Specific Performance**: MSE_DE, PCC_DE, R_DE (for top 10%
   most variable genes)

## Expected Performance Benchmarks

### Unseen Perturbations Scenario
- R Overall: 0.45-0.65
- PCC Overall: 0.60-0.75
- R DE: 0.50-0.70

### Unseen Cell Contexts Scenario
- R Overall: 0.35-0.55
- PCC Overall: 0.55-0.70
- R DE: 0.40-0.60

## Deliverables Generated

1. **`chemical_perturbation_predictor.py`** – Complete
   implementation framework
2. **[chemical_perturbation_methodology.md](https://biomni-logs.
   s3.amazonaws.com/3220104147_zju_edu_cn/logs/session_0c14fa4f
   -59b1-4c98-96f6-3655019e2f9a/files/
   chemical_perturbation_methodology.md?AWSAccessKeyId=
   AKIA5OMXL5OSOMU6RIOE&Signature=GhlIsrCIgJ0s0x40PKi%2BnfYpAv4
   %3D&Expires=1751038449)** – Comprehensive methodology
   document
3. **[evaluation_framework.json](https://biomni-logs.s3.
   amazonaws.com/3220104147_zju_edu_cn/logs/session_0c14fa4f-59
   b1-4c98-96f6-3655019e2f9a/files/evaluation_framework.json?
   AWSAccessKeyId=AKIA5OMXL5OSOMU6RIOE&Signature=WJX8wVO7Sw76%2
   BrdjKY2FVdiqdGY%3D&Expires=1751038449)** – Structured
   evaluation specifications
4. **[chemical_perturbation_analysis.png](https://biomni-logs.s3.
   amazonaws.com/3220104147_zju_edu_cn/logs/session_0c14fa4f-59
   b1-4c98-96f6-3655019e2f9a/files/
   chemical_perturbation_analysis.png?AWSAccessKeyId=
   AKIA5OMXL5OSOMU6RIOE&Signature=m4cqAdqEFUs0nJeuYkywlO%2FUOU0
   %3D&Expires=1751038449)** – Multi-panel analysis
   visualization
5. **[technical_summary.md](https://biomni-logs.s3.amazonaws.com
   /3220104147_zju_edu_cn/logs/session_0c14fa4f-59b1-4c98-96f6
   -3655019e2f9a/files/technical_summary.md?AWSAccessKeyId=
   AKIA5OMXL5OSOMU6RIOE&Signature=t9q3oUk9p3CxYi7%2Fqx1ckJuHKxU
   %3D&Expires=1751038449)** – Implementation technical details

## Scientific Significance

### Drug Discovery Applications

- **Compound screening**: Predict transcriptional effects of
  novel compounds
- **Mechanism elucidation**: Understand drug action pathways
- **Dose optimization**: Identify optimal therapeutic
  concentrations

### Precision Medicine Impact
- **Patient stratification**: Predict individual drug responses
- **Biomarker identification**: Discover predictive molecular
  signatures
- **Treatment personalization**: Guide precision therapy
  selection

### Systems Biology Insights
- **Network analysis**: Map compound-pathway interactions
- **Combination therapy**: Predict synergistic drug effects
- **Resistance mechanisms**: Understand treatment failure
  pathways

## Technical Innovations

### 1. Compound-Expression Interactions
Our framework introduces interaction terms between baseline
    cellular state and compound characteristics, capturing how
    identical compounds can have different effects depending on
    cellular context.

### 2. Multi-Target Regression
Simultaneous prediction of all gene expression levels enables
    comprehensive transcriptional response modeling while
    maintaining computational efficiency.

### 3. Stratified Evaluation
Separate evaluation scenarios ensure model performance
    assessment across different real-world application contexts.

## Future Extensions

### Deep Learning Integration
- Transformer architectures for sequence-based compound
  representation
- Graph neural networks for compound-target interaction modeling
- Variational autoencoders for dimensionality reduction

### Multi-Modal Enhancement
- Chemical structure integration (SMILES, molecular fingerprints
  )
- Protein-compound interaction networks
- Epigenetic and chromatin accessibility data

## Implementation Usage

```python
# Initialize framework
predictor = ChemicalPerturbationPredictor(output_dir='./results
    ')

# Load sci-Plex data
data = predictor.load_data('[expression_matrix.csv](https://
    biomni-logs.s3.amazonaws.com/3220104147_zju_edu_cn/logs/
    session_0c14fa4f-59b1-4c98-96f6-3655019e2f9a/files/
    expression_matrix.csv?AWSAccessKeyId=AKIA5OMXL5OSOMU6RIOE&
    Signature=ld8NqUwqRAT7aoKxjqKDUwjqdu0%3D&Expires=1751038449)
    ', '[metadata.csv](https://biomni-logs.s3.amazonaws.com
    /3220104147_zju_edu_cn/logs/session_0c14fa4f-59b1-4c98-96f6
    -3655019e2f9a/files/metadata.csv?AWSAccessKeyId=
    AKIA5OMXL5OSOMU6RIOE&Signature=kl3jlyuXLGVDgXViz5GOC%2BNLzHs
    %3D&Expires=1751038449)')

# Preprocess and train
features, targets = predictor.preprocess_data(data['
    expression_data'], data['metadata'])
```

```
results = predictor.train_models(features, targets, scenario='
    unseen_perturbations')

# Generate comprehensive report
report = predictor.generate_report()
```

## Conclusion

This comprehensive framework establishes new benchmarks for
    chemical perturbation prediction, combining methodological
    innovation with practical implementation. The multi-model
    approach, comprehensive evaluation framework, and focus on
    biological relevance make it suitable for both research
    applications and clinical translation.

The framework's emphasis on interpretability and biological
    grounding, combined with its computational efficiency,
    positions it as a valuable tool for the drug discovery and
    precision medicine communities.

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

    aax6234)

---

**All implementation files have been saved to '/tmp/
    agent_outputs/0c14fa4f-59b1-4c98-96f6-3655019e2f9a/' and are
     ready for immediate use with the sci-Plex dataset.**

***All file outputs are stored in the session execution logs,
    you may download them from the \<Session History & Export\>
    section***
```

---

**Research Agent Output**

Schiebinger et al.(cytokines, scRNA-seq)

```
Launching workflow executor (this may take a while)...
 Answer
  # mESC Cytokine Perturbation Gene Expression Prediction: Novel
      Analysis, Method & Code Report

## Executive Summary

I have developed a comprehensive computational framework for
    predicting transcriptional responses in mouse embryonic stem
     cells (mESCs) following cytokine perturbations. This novel
    methodology combines insights from the Waddington-OT
    framework by Schiebinger et al. [1] with advanced machine
    learning approaches to address the challenge of predicting
    gene expression changes in response to cytokine treatments.

## Key Contributions

### 1. Novel Methodology Framework
- **Multi-model ensemble approach** combining Random Forest,
    Gradient Boosting, Ridge Regression, Elastic Net, and Neural
     Networks
- **Innovative interaction feature engineering** capturing
    cytokine-context dependencies through element-wise products
    of cytokine presence with baseline expression
- **Dual evaluation framework** addressing both unseen
    perturbations and unseen cellular contexts
- **Comprehensive metric system** including both overall and
    differentially expressed gene-specific measures

### 2. Scientific Foundation
```

Based on extensive literature research, the framework is
    grounded in:
- **Waddington-OT principles** from Schiebinger et al. (Cell
    2019) [1], which demonstrated optimal transport analysis of
    single-cell trajectories during cellular reprogramming
- **Single-cell perturbation studies** showing the power of
    combining CRISPR screens with scRNA-seq for functional
    genomics
- **Cytokine signaling biology** in mESC pluripotency
    maintenance and differentiation

### 3. Technical Innovation
- **Context-dependent modeling**: Explicit modeling of how
    identical cytokines have different effects in different
    cellular states
- **Multi-target regression**: Simultaneous prediction of all
    genes while maintaining correlation structure
- **Biologically motivated features**: Integration of cytokine
    identity, concentration, timepoint, and cellular state
    information

## Data and Methods

### Dataset Requirements
The framework is designed for single-cell RNA sequencing data
    from mESC cytokine perturbation experiments, requiring:
- Expression matrix: cells  genes (minimum 1000 cells  2000
    genes)
- Metadata: cytokine_id, concentration_ng_ml, timepoint_hours,
    cell_state
- Compatible formats: AnnData (H5AD) or CSV/TSV

### Model Architectures
1. **Random Forest** (n_estimators=300, max_depth=25): Handles
    non-linear interactions
2. **Gradient Boosting** (n_estimators=200, max_depth=8):
    Sequential residual learning
3. **Ridge Regression** (=1.0): Linear baseline with L2
    regularization
4. **Elastic Net** (=0.5, l1_ratio=0.5): Combined L1/L2
    regularization
5. **Neural Network** (256-128-64 architecture): Deep learning
    for complex patterns

### Evaluation Framework
**Two Critical Scenarios:**
- **Unseen Perturbations**: Predicting responses to novel
    cytokine treatments
- **Unseen Cell Contexts**: Predicting individual cell responses
     in diverse contexts

**Six Comprehensive Metrics:**
- Overall: MSE, PCC, R (across all genes)
- DE-specific: MSE_DE, PCC_DE, R_DE (for differentially
    expressed genes)

## Implementation Details

### Novel Feature Engineering
The framework creates ~6,000+ features combining:
- **Baseline expression**: Top 2,000 highly variable genes
- **Cytokine features**: Encoded identity, log concentration,
    timepoint, cell state
- **Interaction features**: Cytokine-expression element-wise
    products capturing context-dependent effects

### Preprocessing Pipeline
1. Quality control (cell/gene filtering)
2. CPM normalization and log1p transformation
3. Feature selection (highly variable genes)
4. Interaction feature creation
5. Train/test splitting by scenario

## Expected Performance

### Performance Benchmarks
**Unseen Perturbations Scenario:**
- R Overall: 0.50-0.70
- PCC Overall: 0.65-0.80
- R DE: 0.55-0.75

**Unseen Cell Contexts Scenario:**
- R Overall: 0.40-0.60
- PCC Overall: 0.60-0.75
- R DE: 0.45-0.65

## Generated Deliverables

1. **mesc_implementation_code.py**: Complete Python
   implementation with mESCCytokinePerturbationPredictor class
2. **[mesc_methodology_report.md](https://biomni-logs.s3.
   amazonaws.com/3220104147_zju_edu_cn/logs/session_4aba27d4-
   fea3-49c8-9d86-34c2b73b0216/files/mesc_methodology_report.md?
   AWSAccessKeyId=AKIA5OMXL5OSOMU6RIOE&Signature=
   SWZcdufL5nGCGdvohark8qM8Ywk%3D&Expires=1751039094)**:
   Detailed methodology documentation (11,863 characters)
3. **[evaluation_framework.json](https://biomni-logs.s3.
   amazonaws.com/3220104147_zju_edu_cn/logs/session_4aba27d4-
   fea3-49c8-9d86-34c2b73b0216/files/evaluation_framework.json?
   AWSAccessKeyId=AKIA5OMXL5OSOMU6RIOE&Signature=k0Bkx%2
   BJt6abCEUQ16nktooQzj1A%3D&Expires=1751039094)**: Structured
   evaluation specifications
4. **[comprehensive_framework_analysis.png](https://biomni-logs.
   s3.amazonaws.com/3220104147_zju_edu_cn/logs/session_4aba27d4-
   fea3-49c8-9d86-34c2b73b0216/files/
   comprehensive_framework_analysis.png?AWSAccessKeyId=
   AKIA5OMXL5OSOMU6RIOE&Signature=FWloIvkJ5VR2LR1lipQnPQ%2F07Mk
   %3D&Expires=1751039094)**: Multi-panel visualization of
   framework components
5. **[technical_specifications.md](https://biomni-logs.s3.
   amazonaws.com/3220104147_zju_edu_cn/logs/session_4aba27d4-
   fea3-49c8-9d86-34c2b73b0216/files/technical_specifications.
   md?AWSAccessKeyId=AKIA5OMXL5OSOMU6RIOE&Signature=
   NTusChoE4rFLmvqcfad17kkmQDg%3D&Expires=1751039094)**:
   Implementation requirements and configuration details

## Scientific Significance

### Biological Applications
- **Stem cell biology**: Optimize cytokine cocktails for mESC
  culture and differentiation
- **Drug discovery**: Predict cytokine drug mechanisms and off-
  target effects
- **Precision medicine**: Guide personalized cytokine therapy
  selection

### Methodological Advances
- **Context-dependent perturbation modeling**: Novel approach to
  capture how cellular state influences treatment response
- **Multi-scenario evaluation**: Addresses real-world
  application challenges
- **Interaction feature engineering**: Biologically motivated
  approach to capture cytokine-cell dependencies

## Future Extensions

1. **Deep Learning Integration**: Transformer architectures and
   graph neural networks
2. **Multi-Modal Enhancement**: Integration of protein abundance
   and chromatin accessibility data
3. **Causal Inference**: Distinguish correlation from causation
   in cytokine-gene relationships
4. **Temporal Dynamics**: Model time-series cytokine responses

```
## Conclusion

This comprehensive framework represents a significant
    advancement in predicting cytokine-induced transcriptional
    changes in mESCs. By combining insights from optimal
    transport theory with modern machine learning, it provides a
     robust solution for understanding and predicting cytokine
    effects on stem cell gene expression. The framework
    establishes new benchmarks for perturbation prediction and
    provides a foundation for future research in computational
    stem cell biology and precision medicine.

The methodology addresses critical challenges in stem cell
    research, drug discovery, and precision medicine by
    providing accurate predictions of how cytokine treatments
    will affect gene expression in different cellular contexts.
    All implementation code, documentation, and analysis
    visualizations have been generated and are ready for
    immediate use with appropriate mESC cytokine perturbation
    datasets.

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

    biomni-logs.s3.amazonaws.com/3220104147_zju_edu_cn/logs/
    session_4aba27d4-fea3-49c8-9d86-34c2b73b0216/files/
    technical_specifications.md?AWSAccessKeyId=
```

```
        AKIA5OMXL5OSOMU6RIOE&Signature=NTusChoE4rFLmvqcfad17kkmQDg%3
        D&Expires=1751039094)`
    ***All file outputs are stored in the session execution logs,
        you may download them from the \<Session History & Export\>
        section***
```

**OpenAI DeepResearch Output**

---
**Research Agent Output**

---

Norman et al. dataset(gene knockout, scRNA-seq)
  ## Task Analysis: Data Characteristics and Challenges

  **Perturb-seq Dataset (Norman \*et al.\*, 2019):** The model is
      built on a rich single-cell CRISPR perturbation dataset from
      Norman *et al.* (2019, Science) (GEO: GSE133344). In this
      Perturb-seq experiment, K562 cells (a human leukemia line)
      were subjected to pooled CRISPR perturbations, measuring
      transcriptome-wide gene expression in each cell via single-
      cell RNA-seq. Key features of this dataset include:

  - **Scale and Composition:** ~170,000 single cells were profiled,
      covering *~1,092 unique perturbation conditions* (including
      both single-gene and paired-gene perturbations). Each cell
      is annotated with one or two target genes perturbed (via
      CRISPR interference/activation).

  - **High-Dimensional Readouts:** Each cells gene expression
      profile spans ~20,000 genes (full transcriptome). This
      yields a **high-dimensional output space**  a vector of
      expression levels for thousands of genes per cell.

  - **Perturbation Types:** Perturbations consist of **single-gene
      knockdowns** (CRISPRi) and **two-gene combinations**,
      enabling the study of genetic interactions. For each target
      gene, multiple sgRNAs were used (to control for guide-
      specific effects), and for two-gene combos, cells co-express
      two sgRNAs (one per target).

  **Modeling Challenges:** Developing an accurate predictive model
      for this task entails several significant challenges:

  - **Single-Cell Variability:** Even for the same perturbation,
      different cells can respond with variability in magnitude of
      gene expression changes. Each cell has an intrinsic
      baseline expression profile reflecting its state (cell cycle
      stage, stochastic gene expression, etc.).

  - **Generalization to Unseen Perturbations:** A critical use-
      case is predicting outcomes for perturbations that were **
      never seen in training**. This includes:
      - *New target genes:* The model may be asked to predict
          the effect of knocking down a gene that was not
          perturbed in the training data. This is an extreme
          generalization challenge, as there is no direct
          training example for that genes perturbation. The
          model must leverage other information (e.g. the genes
          baseline expression or relationships to other genes)
          to infer its likely impact.
      - *New gene combinations:* Similarly, the model might see
          single-gene perturbations during training, but not a
          particular pair in combination. Or even harder, both
          genes in a pair might be novel (neither perturbed
          before). As reported in prior work, when one or both
          perturbed genes are unseen, prediction difficulty
          increases markedly. The model needs a strategy to
          generalize **inductively**, combining knowledge of
          individual genes to predict combined effects.

- **Accuracy vs. Interpretability:** We prioritize predictive
  accuracy. This means we are willing to employ a complex,
  black-box model (deep neural network) with many parameters
  if it yields better predictive performance, even if its not
  easy to interpret biologically. This choice acknowledges
  that the gene regulatory relationships underlying the
  perturbation responses are complex; a simpler interpretable
  model (like linear regression per gene) might not capture
  the non-linear interactions or context dependencies as well
  as a deep model. However, the complexity of the model also
  raises the risk of overfitting and requires careful
  regularization and validation.

In summary, the task is essentially learning a high-dimensional
  function: **f**(*baseline_expression*, *perturbation*)  *
  perturbed_expression*. It must learn the underlying gene
  network responses from data, handle combinatorial
  perturbation effects, and generalize to new genes and cell
  states. These challenges will guide the modeling decisions
  discussed below.

------
## Proposed Modeling Approach for Accuracy and Generalization

### Model Architecture and Input Representation

To tackle this problem, we propose a **neural network model**
  that takes two inputs  (1) the baseline gene expression
  profile of an unperturbed cell, and (2) the identity of the
  target gene(s) perturbed  and outputs the predicted post-
  perturbation gene expression profile. The architecture is
  designed to flexibly capture non-linear gene interactions
  and make use of prior gene relationship knowledge to
  generalize to unseen perturbations. The key components are:

- **Baseline Expression Encoder:** A front-end network that
  processes the baseline expression vector (dimension = *G*
  genes). Directly using the raw high-dimensional vector as
  input to a fully-connected network is feasible but could be
  inefficient. We will introduce an encoder (e.g., a feed-
  forward autoencoder or dimensionality reduction layer) that
  compresses the baseline gene expression profile into a **
  lower-dimensional latent representation**. For example, a
  few fully-connected layers with ReLU activation can reduce
  the ˜20k-dimensional input to a dense ˜512-dimensional
  embedding. This latent vector is intended to capture the
  cells overall state or context (e.g., if the cell is in a
  high-proliferation state, or has high expression of certain
  pathways, etc.). By encoding the baseline, the model can
  later modulate perturbation effects depending on these
  latent features.

- **Perturbation Encoder:** We represent the **perturbation
  identity** in a way that the model can easily utilize and
  generalize. Each target gene (from the set of ˜100 possible
  targeted genes in the screen) is assigned either:
    - a one-hot vector (of length equal to the number of
      target genes) if a single gene is perturbed, or a
      multi-hot vector if multiple genes are perturbed (e.g.
       for a two-gene perturbation, the vector has ones in
      the positions corresponding to the two targeted genes).
       This binary indicator vector can then be fed through
      an embedding layer (a learned lookup table or a small
      fully-connected network) to produce a **perturbation
      embedding**. The embedding is a continuous vector (e.g.
       128-dimensional) that represents the effect of the
      perturbation in a latent space.
    - Alternatively, an **embedding per gene** approach can be
       used: we maintain a trainable embedding vector for
      each gene in the target set, and for a combination

perturbation, we combine the embeddings of the
individual genes (e.g. by summation or an attention
mechanism). Using a learned embedding for each gene
gives the model a chance to encode each genes
characteristic perturbation impact. For a multi-gene
perturbation, a simple summation assumes independence
of effects, while a more sophisticated combination (
see below) can capture interactions.

- **Combination Module:** The baseline context and perturbation
  effect must be integrated. We concatenate the baseline
  latent vector and the perturbation embedding vector into a
  single combined latent representation. This combined vector
  (of length ~640 in our example, if baseline latent is 512
  and perturbation embedding 128) now contains information
  about where the cell started and what perturbation was
  applied. This is passed through further layers (a **fusion
  network**) to compute the output. For instance, a multilayer
   perceptron (MLP) with one or two hidden layers (e.g., 512
  neurons, ReLU activation) can mix these features. This stage
   allows for non-linear interactions between cell state and
  perturbation  e.g., the effect of perturbing gene X might
  depend on the level of gene Y in the baseline state, which a
   multiplicative interaction in the MLP can learn.

- **Output Layer (Prediction Head):** The final layer of the
  network produces a vector of length *G* (the number of genes)
  , which is the predicted post-perturbation expression for
  each gene. To ensure the model easily handles the fact that
  many genes dont change, we design the output to predict a **
  change (delta) from baseline** for each gene rather than an
  absolute expression.

- **Non-linear Interaction Modeling:** While a simple
  concatenation of embeddings treats multi-gene perturbations
  roughly as an additive combination of single effects, we can
   enhance the model to capture **genegene interaction effects
  **. One idea is to use an **attention mechanism or gating**
  in the perturbation encoder: for example, if two genes A and
   B are perturbed, instead of just summing their embeddings,
  we pass them through an attention network that can learn a
  pairwise interaction term. Another approach is to include
  pairwise products of gene embeddings in the combined feature
   (allowing the network to learn a unique contribution for
  the pair *A&B* beyond A + B). Given that Norman *et al.*
  tested primarily pairwise perturbations, we can explicitly
  include a learned parameter or small network for each pair
  of genes in the training set to capture any deviation from
  additivity. However, to generalize to unseen pairs, a better
   strategy is to learn a **function** for combining
  embeddings (like attention) rather than a fixed lookup for
  each pair.

- **Incorporating Prior Knowledge (for Generalization):** To
  improve inductive generalization to unseen genes, we can
  draw inspiration from GEARS and similar methods. We could
  initialize or regularize the gene perturbation embeddings
  using external knowledge:
    - Use a **gene co-expression network** (computed from the
      baseline single-cell data or external data) as a graph,
       and pass gene embeddings through a Graph Neural
      Network (GNN) layer. This encourages genes that have
      similar roles or expression patterns to have
      embeddings that produce similar effects. Thus, if an
      unseen gene has a similar co-expression profile to a
      seen gene, the model might infer similar perturbation
      outcomes.
    - Use **pathway or GO (Gene Ontology) information** to
      place genes in a relational graph (as GEARS did with a
       GO-derived graph for perturbation embeddings). Two

genes in the same pathway might be expected to produce related downstream effects; by training on the known genes, the model can generalize to a new gene by its connections in the GO graph. Technically, this can be done by adding a loss term that encourages the learned embedding to correspond to the genes position in the memory module, or by a GNN that propagates influence from neighbors during training.

- These additions make the model more complex but aim to imbue it with **biological inductive bias**: (i) genes with similar baseline functions yield similar perturbation responses, and (ii) genes in related pathways affect overlapping sets of genes when perturbed. We will prioritize implementing a simpler version (like using co-expression PCA or clusters to initialize embeddings), and note that full graph-based learning could further improve generalization if needed.

- **Residual Connections and Regularization:** We will include skip-connections wherever helpful (for example, the baseline input could be fed not only into the encoder but also concatenated directly to a later layer, or the output head could directly see the raw baseline as well, ensuring the model can easily learn identity for unchanged genes). Regularization techniques like dropout in the MLP layers, L2 weight decay, or even an auxiliary loss to reconstruct the baseline (autoencoder style) can be employed to prevent overfitting and encourage the model to learn meaningful latent features rather than memorizing training examples.

In essence, the architecture is a **conditioned deep neural network**: it conditions on the cells initial state and the perturbation, and produces an output state. This is somewhat analogous to an encoderdecoder model where the encoder is the baseline expression and the condition or control signal is the perturbation identity. Because accuracy is paramount, we allow a fairly large model with enough capacity to capture complex gene regulatory responses.

### Training Strategy for Accuracy and Generalization

With the architecture in place, we next focus on **training methodology**, as it greatly affects model generalization and performance:

- **Training Data Construction:** We will pair each perturbed cells data with a baseline profile as input. Since in the actual experiment we typically do not have a *pre-perturbation* measurement of the same cell, we have to simulate a baseline. We can use the expression profiles of control cells (non-targeting sgRNA) as proxies for baseline input. For each perturbed cell in the training set, we can randomly sample a control cells expression as the baseline input. This effectively assumes that any control cell is an example of an unperturbed state the perturbed cell *could* have come from. Over many samples, the model will learn to map from an average baseline state to the perturbed outcome. We can further refine this pairing by matching on cell state: e.g., cluster the baseline cells by expression and pick a baseline from the same cluster as the perturbed cells profile (minus the perturbation effect) to provide a closer starting point. However, random pairing with a large pool of controls adds variability that can help the model not to overfit a one-to-one mapping.

  - *Unseen cell context generalization:* By exposing the model to many different baseline samples paired with a given perturbation outcome (through random pairing), we train it to handle diverse baseline inputs for the same perturbation. This should improve the models

robustness to any particular baseline context and enable generalization to new baseline profiles. Essentially, the model sees that the same perturbation can apply to various starting expression patterns.

- **Loss Functions:** The primary loss will be **Mean Squared Error (MSE)** between the predicted and actual post-perturbation expression vectors. To ensure we adequately learn the important changes, we can modify the loss in two ways:

  - Compute a weighted MSE that gives higher weight to genes that are truly differentially expressed in that training example. For instance, if we know gene j changed significantly in the real perturbed cell ( compared to baseline or compared to controls), we can upweight the error on gene j for that sample. This forces the model to focus on fitting the genes that move, rather than being dominated by the many near-zero changes.
  - Alternatively, we can train in two phases: first optimize MSE on the full profile to get general trends, then fine-tune the model on just the top-k DE genes for each perturbation (or using a loss like contrastive that emphasizes getting the direction of change correct).

  In practice, a simpler approach is to stick with standard MSE on all genes but monitor the top-k gene performance as a separate metric, ensuring the model doesnt ignore those signals. If we see the model predicting trivial (no-change) solutions, we will adjust the loss weighting.

- **Optimizer and Regularization:** We will use **Adam optimizer** (adaptive learning rate) which is well-suited for training deep networks on possibly noisy data. A relatively small learning rate (e.g. 1e-3 to start) will be used and well monitor validation loss for convergence. Early stopping on the validation MSE/PCC can prevent overfitting. Dropout layers (e.g. dropout rate 0.2-0.5) can be inserted in the MLP to regularize. Weight decay (L2) will help keep embedding weights from growing excessively (important if some genes cause large effects, we dont want the weights to explode to fit one-hot signals).

- **Generalization Enhancements:** To specifically improve performance on **unseen genes**:

  - We will pre-train the **perturbation encoder embeddings**: one idea is to train a simple autoencoder or matrix factorization on the *observed perturbation effects* in training. For example, compute the average expression change each perturbation causes on each gene (from data), yielding a perturbation-gene effect matrix. We can factorize this matrix to get a low-dimensional embedding for genes and for perturbations. If a gene is unseen, we can still derive a gene embedding from its baseline expression pattern using the gene embedding from baseline encoder. This is a bit complex, but essentially we want the model to know something about gene function even if not perturbed. Incorporating gene ontology or known regulator information for unseen genes could also inform an initial embedding (e.g., if an unseen gene is a transcription factor similar to another seen TF, assign it a similar perturbation embedding).

By combining a carefully-designed architecture with these training strategies, our goal is a model that not only fits the training perturbations well but **generalizes to new

perturbations and contexts**. In particular, success would
be a model that, given an arbitrary K562 baseline profile
and a novel target gene, can predict a biologically
plausible outcome  capturing which pathways might be up or
down-regulated by that genes perturbation, even without
having seen it before. The inclusion of baseline input
enables context-specific predictions (so the same gene
perturbation could have different effects in different
baseline states if the model learns those interactions), and
 the use of gene embeddings with potential graph-based
inductive bias enables inductive generalization across genes.

Next, we present the implementation of this model in PyTorch,
    structured for GPU acceleration.
    ------
    ## PyTorch Implementation (Model & Training Code)

Below is the complete PyTorch code for the proposed model and an
    example training loop. This code assumes that the dataset
    has been preprocessed into tensors: `X_train` (baseline
    expressions), `Y_train` (perturbed expressions), and `
    P_train` (perturbation indicator vectors). The model is
    designed to run on a GPU for efficiency.

```python
import torch
import torch.nn as nn
import torch.optim as optim

# Define device for GPU acceleration
device = torch.device('cuda' if torch.cuda.is_available() else '
    cpu')
print("Using device:", device)

# Example shapes (for illustration; replace with actual dataset
    dimensions)
num_genes = X_train.shape[1] # dimensionality of gene expression
    (e.g. ~20000)
num_target_genes = P_train.shape[1] # dimensionality of
    perturbation vector (e.g. ~100)

# 1. Define the model architecture
class PerturbationPredictor(nn.Module):
    def __init__(self, input_dim_expr, input_dim_pert,
            hidden_dim_expr=512, hidden_dim_pert=128,
                hidden_dim_comb=512):
        """
        input_dim_expr: number of genes (baseline expression
            vector length)
        input_dim_pert: number of possible targeted genes (length
            of perturbation indicator vector)
        """
        super(PerturbationPredictor, self).__init__()
        # Baseline expression encoder network
        self.expr_encoder = nn.Sequential(
            nn.Linear(input_dim_expr, hidden_dim_expr),
            nn.ReLU(),
            nn.Linear(hidden_dim_expr, hidden_dim_expr),
            nn.ReLU()
        )
        # Perturbation encoder network
        self.pert_encoder = nn.Sequential(
            nn.Linear(input_dim_pert, hidden_dim_pert),
            nn.ReLU(),
            nn.Linear(hidden_dim_pert, hidden_dim_pert),
            nn.ReLU()
        )
        # Fusion network to combine baseline and perturbation
            representations
```

```
        self.comb_fc1 = nn.Linear(hidden_dim_expr +
            hidden_dim_pert, hidden_dim_comb)
        self.comb_act = nn.ReLU()
        self.comb_fc2 = nn.Linear(hidden_dim_comb, input_dim_expr)
            # outputs delta for each gene

    def forward(self, baseline_expr, perturb_vec):
        # baseline_expr: Tensor of shape (batch_size,
            input_dim_expr)
        # perturb_vec: Tensor of shape (batch_size, input_dim_pert
            )
        # Encode baseline and perturbation
        z_expr = self.expr_encoder(baseline_expr) # shape: (batch,
            hidden_dim_expr)
        z_pert = self.pert_encoder(perturb_vec) # shape: (batch,
            hidden_dim_pert)
        # Combine encoded representations
        z_comb = torch.cat([z_expr, z_pert], dim=1) # shape: (
            batch, hidden_dim_expr+hidden_dim_pert)
        h = self.comb_act(self.comb_fc1(z_comb)) # hidden layer
        delta = self.comb_fc2(h) # predicted change in expression
        # Add the predicted change (delta) to the baseline to get
            final prediction
        pred_expr = baseline_expr + delta
        return pred_expr

# Instantiate the model and move to GPU
model = PerturbationPredictor(num_genes, num_target_genes).to(
    device)
print(model)

# 2. Set up loss function and optimizer
criterion = nn.MSELoss() # Mean Squared Error loss
optimizer = optim.Adam(model.parameters(), lr=1e-3)

# 3. Training loop (example)
num_epochs = 20
batch_size = 256

# If data is large, use DataLoader for batching (assuming
    X_train, P_train, Y_train are TensorDatasets)
train_dataset = torch.utils.data.TensorDataset(X_train, P_train,
    Y_train)
train_loader = torch.utils.data.DataLoader(train_dataset,
    batch_size=batch_size, shuffle=True)

for epoch in range(1, num_epochs+1):
    model.train()
    total_loss = 0.0
    for X_batch, P_batch, Y_batch in train_loader:
        # Move data to device
        X_batch = X_batch.to(device)
        P_batch = P_batch.to(device)
        Y_batch = Y_batch.to(device)
        # Forward pass
        pred = model(X_batch, P_batch)
        loss = criterion(pred, Y_batch)
        # Backpropagation
        optimizer.zero_grad()
        loss.backward()
        optimizer.step()
        total_loss += loss.item() * X_batch.size(0)
    avg_loss = total_loss / len(train_dataset)
    if epoch % 5 == 0 or epoch == 1:
        print(f"Epoch {epoch:02d}: Training MSE = {avg_loss:.4f}")

# 4. Evaluation on test set
model.eval()
with torch.no_grad():
    X_test = X_test.to(device)
    P_test = P_test.to(device)
```

```
        Y_test = Y_test.to(device)
        pred_test = model(X_test, P_test)
        mse_full = criterion(pred_test, Y_test).item()
    # Compute Pearson correlation coefficient (PCC) for each test
        sample
    pred_np = pred_test.cpu().numpy()
    Y_np = Y_test.cpu().numpy()
    baseline_np = X_test.cpu().numpy()
    pcc_list = []
    topk = 20
    topk_pcc_list = []
    for i in range(Y_np.shape[0]):
        true_expr = Y_np[i]
        pred_expr = pred_np[i]
        # PCC for all genes
        cov = np.cov(true_expr, pred_expr, bias=True)
        # cov matrix 2x2: cov[0,1] is covariance between true and
            pred
        pcc = cov[0,1] / (np.std(true_expr) * np.std(pred_expr) + 1e
            -8)
        pcc_list.append(pcc)
        # PCC for top-k differentially expressed genes
        # Identify top-k genes by absolute change in true expression
            vs baseline
        base_expr = baseline_np[i]
        diff = np.abs(true_expr - base_expr)
        topk_idx = np.argsort(diff)[-topk:]
        if topk > 0:
            true_top = true_expr[topk_idx]
            pred_top = pred_expr[topk_idx]
            cov_top = np.cov(true_top, pred_top, bias=True)
            topk_pcc = cov_top[0,1] / (np.std(true_top) * np.std(
                pred_top) + 1e-8)
            topk_pcc_list.append(topk_pcc)
    # Calculate mean metrics
    mean_pcc = float(np.mean(pcc_list))
    mean_topk_pcc = float(np.mean(topk_pcc_list))
    print(f"Test MSE (all genes): {mse_full:.4f}")
    print(f"Test mean PCC (all genes): {mean_pcc:.3f}")
    print(f"Test mean PCC (top-{topk} DE genes): {mean_topk_pcc:.3f
        }")
    ```
```

**Explanation of the Code:**

- We define a `PerturbationPredictor` model class that
  implements the architecture described. The baseline
  expression encoder (`expr_encoder`) and perturbation encoder
  (`pert_encoder`) are simple feed-forward networks. These
  could be extended or replaced with more complex sub-networks
  (e.g., adding dropout, or using a graph convolution in `
  pert_encoder` if incorporating a gene network). The combined
  representation is fed through two linear layers (`comb_fc1`
  and `comb_fc2`) with a ReLU in between. The output of `
  comb_fc2` is a vector of length equal to number of genes,
  representing the predicted change in expression. We then add
  this to the baseline (`pred_expr = baseline + delta`) to
  get the final prediction.
- We move the model to GPU (`.to(device)`). The training loop
  iterates over mini-batches of data (using a DataLoader for
  efficiency). For each batch, we do a forward pass to compute
  predictions and then use `nn.MSELoss` to compute the loss
  against the true perturbed expression. We then backpropagate
  (`loss.backward()`) and update weights with Adam optimizer.
  We print the training MSE every few epochs for monitoring.
- In the evaluation section, after training, we compute metrics
  on `X_test, P_test, Y_test`. We calculate the overall MSE on
  test (as `mse_full`). Then for each test sample, we compute
  the Pearson correlation coefficient (PCC) between predicted
  and true expression. We do this manually using numpy:

```
        computing the covariance and standard deviations (note: one
        could also use `scipy.stats.pearsonr` or torchmetrics for
        this). We also compute **top-20 PCC** by finding the 20
        genes with largest absolute change in the true data (
        compared to baseline) and computing PCC on that subset.
        Finally, we print the average PCC across all test cells, and
         the average top-20 PCC.

    **Note:** In practice, you would likely use vectorized
        operations or library functions to compute PCC for all cells
         at once. Also, you might evaluate the model separately on
        the different generalization scenarios (e.g., one test set
        for unseen genes, one for seen genes, etc.) and compare
        performance. The code above provides a template that can be
        adapted for those purposes.

By executing this model training pipeline, we expect to obtain a
     model that can accurately predict gene expression changes
    in K562 cells after CRISPRi perturbations. Its performance
    will be reflected in low MSE (especially on perturbed genes)
     and high correlation with actual profiles, even for
    perturbations or cell states not seen in training.
    Ultimately, this demonstrates a powerful approach to **in
    silico** predict cellular response to genetic perturbations,
     leveraging modern deep learning (PyTorch with GPU) for both
     precision and generalization. The combination of baseline
    context and perturbation identity as inputs allows the model
     to capture conditional dependencies (how a perturbations
    effect can vary by cell state), addressing one of the key
    challenges in predictive biology.
```

**Perplexity DeepResearch**

**Research Agent Output**

Norman et al. dataset(gene knockout, scRNA-seq)

```
Task Analysis and Technical Challenges

I. Biological Context of CRISPRi Perturbations

The Norman et al. dataset captures transcriptional consequences
    of CRISPRi-mediated gene repression through single-cell
    profiling of:

- 105 individual gene knockouts
- 131 pairwise gene interactions[^1][^4]
- ~90,000 cells with baseline (unperturbed) and perturbed states
    [^4]

Key biological considerations:

1. **Non-linear genetic interactions**: Pairwise perturbations
    exhibit emergent effects not predictable from individual
    gene knockouts[^2]
2. **Cell-state heterogeneity**: Native transcriptional
    variability in K562 populations influences perturbation
    responses[^2][^4]
3. **Technical noise characteristics**: Dropout patterns and UMI
     count distributions differ between perturbed/control cells
    [^4][^6]

II. Machine Learning Challenges

1. **High-dimensional output space**: Predicting 20,000+ gene
    expressions simultaneously
2. **Sparse perturbation coverage**: Only 0.6% of possible gene
    pairs tested in original study[^1]
3. **Context-dependent effects**: Perturbation responses vary
    with baseline cellular states[^2][^5]
```

4. **Combinatorial explosion**: Quadratic growth of possible
   gene interactions with target count

**Evaluation Protocol Design**

**Train/Test Splits**:

- **Unseen perturbations**: Hold out 20% of single-gene and 30%
  of pairwise perturbations
- **Unseen contexts**: Stratify by baseline PCA clusters, hold
  out entire cell-state subpopulations

------

Methodological Framework: GenePerturbNet

I. Architectural Overview

**Core Components**:

1. **Baseline Encoder**: Variational autoencoder mapping
   unperturbed expression $x$ to latent $z$.

2. **Perturbation Transformer**: Attention network processing
   target genes $g$ into perturbation vector \delta.

3. **Interaction Module**: Graph neural network modeling gene-
   gene interaction effects.

4. **Response Decoder**: generating perturbed expression profile

II. Training Strategy

**Loss Function:**

$\mathcal{L} = ELBO(x,y) + \lambda_1 MSE(y, \hat{y}) + \lambda_2 \mathcal{L}_{contrast}$

Where:

- **Contrastive Loss**:
    $\mathcal{L}_{contrast} = -\log\frac{\exp(s(y,\hat{y})/\tau)}{\sum_{y'}\exp(s(y',\hat{y})/\tau)}$

- $s()$: Cosine similarity

- $\tau$: Temperature parameter

IV. Curriculum Learning Schedule:
1. Phase 1: Pretrain on single-gene perturbations

2. Phase 2: Fine-tune with pairwise interactions

3. Phase 3: Joint optimization with contrastive loss

Implementation and Benchmarking

I. Code Implementation

```python
import scanpy as sc
import scvi
import torch
from torch import nn
from scvi import REGISTRY_KEYS
from scvi.module.base import BaseModuleClass

class GenePerturbNet(scvi.model.SCVI):
    def __init__(self, adata, n_hidden=256, n_latent=32,
                 perturbation_embedding=64, **kwargs):
        super().__init__(adata, **kwargs)
        self.pert_embed = nn.Embedding(adata.n_vars,
            perturbation_embedding)
        self.transformer = nn.TransformerEncoderLayer(
```

```
            d_model=perturbation_embedding, nhead=4
        )
        self.gnn = GeneInteractionGNN(perturbation_embedding)

        self.decoder = nn.Sequential(
            nn.Linear(n_latent + perturbation_embedding, n_hidden),
            nn.ReLU(),
            nn.Linear(n_hidden, adata.n_vars)
        )

    def forward(self, x, perturbations, *args):
        qz_m, qz_v, z = self.encoder(x)

        pert_emb = self.pert_embed(perturbations)
        delta = self.transformer(pert_emb)

        z_pert = z + delta.mean(dim=1)

        px_scale, px_r, px_rate, px_dropout = self.decoder(z_pert)

        return px_rate, qz_m, qz_v, z_pert

class GeneInteractionGNN(nn.Module):
    def __init__(self, embed_dim):
        super().__init__()
        self.conv1 = nn.Conv1d(embed_dim, embed_dim*2, kernel_size
            =3)
        self.attn = nn.MultiheadAttention(embed_dim, num_heads=4)

    def forward(self, gene_pairs):
        x = self.conv1(gene_pairs)
        x, _ = self.attn(x, x, x)
        return x.mean(dim=1)
```

II. Training Protocol

```python
adata = sc.read("norman_2019.h5ad")
scvi.model.setup_anndata(adata, layer="counts",
                    categorical_covariate_keys=["perturbation"])

model = GenePerturbNet(adata, n_latent=32,
    perturbation_embedding=64)
model.train(max_epochs=100, batch_size=256,
        plan_kwargs={"contrastive_weight":0.5})

control_cells = adata[adata.obs["perturbation"] == "control"]
perturbed_pred = model.predict(control_cells, targets=["geneA",
    "geneB"])
```

III. Conclusion and Future Directions

This framework demonstrates that deep learning architectures
    explicitly modeling genetic interactions can significantly
    improve perturbation prediction accuracy. Key
    recommendations:

1. **Data Expansion**: Incorporate epigenetic data to better
    model CRISPRi efficiency
2. **Dynamic Modeling**: Add temporal component to capture
    perturbation kinetics
3. **Transfer Learning**: Adapt model to other cell types using
    few-shot learning

The provided codebase offers a foundation for extending
    perturbation prediction to novel experimental designs and
    therapeutic applications.

**Gemini DeepResearch**

---

**Research Agent Output**

Norman et al. dataset(gene knockout, scRNA-seq)

```
Task Analysis

# Dataset Description

The foundation of this predictive modeling task lies in the
    comprehensive dataset generated by Norman et al. and
    published in Science in 2019. This study aimed to explore
    genetic interaction manifolds constructed from rich single-
    cell phenotypes using a Perturb-seq approach.15 The
    experiments were conducted on human K562 leukemia cells, a
    commonly used model system in hematological research. The
    study employed CRISPRi to target a total of 236 genetic
    perturbations, encompassing 105 single genes and 131 pairs
    of genes. The scale of the dataset is substantial,
    comprising approximately 90,000 single-cell RNA-seq profiles,
     which includes both control cells and cells subjected to
    various single and combinatorial CRISPRi perturbations.

The dataset's focus on both single and paired gene perturbations
     is crucial for training a model capable of predicting
    combinatorial effects, which are often non-additive and
    complex. Single gene perturbations provide baseline
    information about the effect of individual gene silencing.
    Paired perturbations allow the model to learn about genetic
    interactions, where the combined effect of perturbing two
    genes is different from the sum of their individual effects.
     This is essential for capturing the non-linear nature of
    biological systems.

# Problem Definition

The primary objective is to develop a predictive model that
    takes as input the baseline gene expression profile of an
    unperturbed K562 cell and the identity of the target gene(s)
     for CRISPRi perturbation. The desired output of the model
    is the predicted gene expression profile of the same cell
    after the specified perturbation has been applied.

The model's performance will be evaluated under two key
    scenarios to assess its generalizability:

1. **Unseen Perturbations:** The model should be able to
    accurately predict the effects of CRISPRi targeting genes or
     gene pairs that were not included in the training data.
    This scenario tests the model's ability to extrapolate its
    learned knowledge to novel genetic manipulations.
2. **Unseen Cell Contexts:** The model should be capable of
    predicting the response to a perturbation in cells with
    baseline gene expression profiles that were not observed
    during the training phase. This evaluates the model's
    robustness to the inherent heterogeneity within the K562
    cell population.

The evaluation scenarios highlight the need for a model that can
     generalize beyond the specific perturbations and cell
    states seen during training, which is a significant
    challenge in biological systems due to their complexity and
    inherent variability. A model trained only on observed
    perturbations might not accurately predict the effects of
    novel perturbations or responses in slightly different
    cellular contexts. Evaluating on unseen data is crucial for
    assessing the model's robustness and real-world
    applicability.

# Challenges and Considerations
```

Developing an accurate predictive model for gene expression profiles following CRISPRi perturbation presents several significant challenges stemming from the inherent complexity of biological systems and the nature of single-cell data.

The gene expression data obtained from scRNA-seq experiments is inherently high-dimensional, with measurements for thousands of genes in each individual cell. This high dimensionality necessitates the use of sophisticated modeling techniques capable of handling a large number of input features. Furthermore, the underlying biological networks that govern gene expression are incredibly complex, involving intricate regulatory relationships between genes, transcription factors, and other cellular components.

The relationship between gene perturbations and the resulting changes in gene expression is often non-linear.1 Genetic interactions, such as synergistic or epistatic effects, mean that the combined effect of perturbing multiple genes cannot simply be predicted by summing the individual effects of each perturbation.1 The model must be able to capture these complex, non-additive relationships to make accurate predictions, especially for the 131 gene pairs targeted in the Norman et al. (2019) dataset.

While CRISPRi is generally considered to have fewer off-target effects compared to CRISPRko, the possibility of unintended gene repression at secondary genomic loci still exists. The specificity of CRISPR-Cas9 systems relies heavily on the design of the single guide RNA (sgRNA), and mismatches between the sgRNA and off-target DNA sequences can lead to unintended gene silencing. Although the Norman et al. (2019) study likely employed well-designed sgRNAs, the potential for some level of off-target activity should be considered.

Single-cell RNA-seq data is also characterized by inherent noise and technical variability.11 Factors such as cell lysis efficiency, reverse transcription efficiency, and sequencing depth can introduce variability in the measured gene expression levels. The predictive model needs to be robust to this noise and focus on capturing the true biological signals induced by the CRISPRi perturbations.

Finally, clonal expansion, a phenomenon where certain perturbed cells proliferate more rapidly than others, can occur during single-cell CRISPR screens. This can lead to an overrepresentation of certain perturbations in the sequencing data, potentially biasing the results and affecting the training of the predictive model. Computational methods exist to identify and potentially adjust for clonal cell expansion.

Accurately modeling the intricate web of gene regulatory interactions and the non-linear effects of perturbations requires sophisticated computational approaches that can capture these complexities beyond simple additive models. Biological systems are highly interconnected, and the effect of perturbing one gene can propagate through the network, affecting the expression of many other genes in non-predictable ways using linear models. Therefore, models capable of learning complex, non-linear relationships are needed.

------

New Method Plan

To address the challenges outlined above and develop an accurate predictive model for gene expression profiles following CRISPRi, a deep learning-based approach is proposed. Specifically, a Graph Neural Network (GNN) architecture,

inspired by the success of models like GEARS, appears to be a promising candidate.

**Proposed Model Architecture**

The proposed model will leverage a GNN to incorporate prior knowledge about gene-gene relationships and model the perturbation effects as changes within this network. GNNs are well-suited for learning representations of nodes in a graph by aggregating information from their neighbors, allowing the model to capture the dependencies and interactions between genes.

The input to the model will consist of two components: (1) the baseline gene expression profile of an unperturbed cell, represented as a vector of gene expression counts, and (2) the identity of the target gene(s) for CRISPRi. For single gene perturbations, the target gene will be directly specified. For paired gene perturbations, the identities of both target genes will be provided.

The model architecture will comprise the following key components:

1. **Gene Embedding Layer:** Each gene in the dataset will be assigned a low-dimensional embedding vector. These embeddings will capture intrinsic properties of the genes and will be learned during the training process. Prior biological knowledge, such as gene co-expression networks or functional annotations from databases like Gene Ontology (GO) 36, can be used to initialize these embeddings or to inform the GNN architecture.

2. **Perturbation Embedding Layer:** The identity of the perturbed gene(s) will also be encoded into an embedding vector. For single perturbations, a dedicated embedding will be learned for each targeted gene. For paired perturbations, the embeddings of the two target genes can be combined (e.g., through summation or concatenation) to represent the combined perturbation.

3. **Graph Neural Network (GNN):** A gene regulatory network (GRN) will be constructed, where genes are represented as nodes and edges represent regulatory relationships between them. This GRN can be derived from publicly available databases or inferred from the unperturbed single-cell expression data. The gene embeddings will serve as initial node features in this graph. The GNN will then propagate information across the network, allowing each gene's representation to be informed by its neighbors and their interactions. The perturbation embedding will be incorporated into the GNN, potentially by modifying the node features of the perturbed gene(s) or by influencing the message passing process.

4. **Cell State Encoding Layer:** The baseline gene expression profile of the unperturbed cell will be passed through a separate neural network (e.g., a multi-layer perceptron) to learn a low-dimensional representation of the cell's initial transcriptional state. This encoding will capture the cell's context and will be used to condition the prediction of the perturbed state.

5. **Prediction Layer:** The output of the GNN (representing the perturbed gene embeddings) and the cell state encoding will be combined (e.g., through concatenation followed by another neural network) to predict the gene expression profile after the perturbation. The output will be a vector of the same dimensionality as the input gene expression profile, representing the predicted expression levels for each gene in the cell.

The rationale behind choosing this architecture is that it allows for the integration of prior biological knowledge about gene-gene interactions through the GRN. This can help the model to better understand the potential downstream effects of a perturbation. Furthermore, the use of embeddings allows the model to learn meaningful representations of genes and perturbations, potentially enabling better generalization to unseen perturbations.

**Feature Engineering and Data Preprocessing**

The Norman et al. (2019) dataset will require careful preprocessing before being used to train the model. The steps involved will include:

1. **Data Loading and Normalization:** The processed gene expression matrices will be loaded using appropriate libraries like Scanpy or AnnData. The gene expression counts will be normalized to account for differences in sequencing depth between cells. Log transformation (e.g., using a natural logarithm after adding a pseudocount) will be applied to stabilize the variance of gene expression levels.

2. **Perturbation Information Encoding:** The perturbation information, specifying the targeted gene(s) for each cell, will be extracted from the dataset's metadata. For single gene perturbations, the gene name will be used. For paired gene perturbations, both gene names will be used. These gene names will then be mapped to their corresponding indices or identifiers in the gene expression matrix. The perturbation information will be encoded as input to the model, potentially using one-hot encoding initially, where a binary vector indicates which genes are targeted. Alternatively, learned embeddings for each gene could be used to represent the perturbation.

3. **Control Sample Handling:** Cells labeled as control (unperturbed) will be identified and used to establish the baseline gene expression profiles. These control profiles will be crucial for training the model to predict the changes in expression induced by the perturbations.

4. **Feature Selection:** Given the high dimensionality of the gene expression data, feature selection techniques may be employed to focus on the most relevant genes. One common approach is to identify highly variable genes (HVGs) across the cell population and use only these genes as input to the model.16 This can reduce the dimensionality of the input, potentially improving model training and performance.

5. **GRN Construction (if applicable):** If a GNN is used, a gene regulatory network will need to be constructed. This could involve using publicly available databases of known gene interactions or inferring a network from the unperturbed single-cell expression data using methods like co-expression analysis or network inference algorithms.

**Training Strategy**

The training of the predictive model will involve the following steps:

1. **Data Splitting:** The dataset will be split into training, validation, and test sets. To address the evaluation scenarios of unseen perturbations, the split will be performed at the level of perturbations. This means that all cells corresponding to certain perturbations (both single and paired) will be held out in the test set and will not be seen by the model during training. A separate validation set, also containing held-out perturbations, will be used for hyperparameter tuning and model selection.16 Carefully designing the data splitting strategy is crucial to ensure

that the model is truly evaluated on unseen perturbations and cell contexts, avoiding information leakage from the training set.

2. **Loss Function:** The model will be trained to minimize the difference between the predicted gene expression profiles and the observed gene expression profiles. The Mean Squared Error (MSE) will be used as the primary loss function, as it directly measures the average squared difference between the predicted and observed values.

3. **Optimizer:** An appropriate optimization algorithm, such as Adam, will be used to update the model's parameters during training. A learning rate schedule, which gradually reduces the learning rate over time, may be employed to improve convergence and prevent overfitting.

4. **Hyperparameter Tuning:** The model architecture and training process will have several hyperparameters (e.g., the dimensionality of the embeddings, the number of layers in the neural networks, the learning rate). These hyperparameters will be tuned using the validation set. Techniques like grid search or random search can be used to explore different combinations of hyperparameters and select the configuration that yields the best performance on the validation set.

5. **Overfitting Prevention:** Techniques such as dropout, weight regularization (e.g., L1 or L2 regularization), and early stopping (monitoring the performance on the validation set and stopping training when it starts to degrade) will be used to prevent the model from overfitting to the training data and improve its generalization ability.

**Strategies for Handling Unseen Perturbations and Cell Contexts**

The proposed GNN-based architecture offers several advantages for handling the challenges of unseen perturbations and cell contexts:

– **Generalization to Unseen Genes (for GNN):** If a GNN is used and the GRN includes genes that are not targeted in the training set, the model may still be able to make informed predictions about the effects of perturbing these unseen genes by leveraging their relationships with other genes in the network.36 The model can learn general principles of how perturbations propagate through the network, allowing it to extrapolate to new nodes (genes). Predicting the effects of completely novel perturbations (genes not seen during training) is a significant challenge.

– **Cell State Conditioning:** By explicitly encoding the baseline gene expression profile of the unperturbed cell, the model can condition its prediction on the specific context of that cell. This allows the model to capture some of the inherent heterogeneity within the cell population and potentially make more accurate predictions for cells with unseen baseline profiles.

– **Learned Embeddings:** The use of learned embeddings for genes and perturbations can help the model to capture semantic relationships between different genes and perturbations. If the embedding space is learned effectively, the model may be able to generalize to unseen perturbations that are functionally similar to those seen during training, even if the specific genes were not encountered before.

While more advanced techniques like meta-learning or domain adaptation could potentially further improve the model's ability to handle unseen perturbations and cell contexts, the proposed GNN architecture with cell state conditioning

```
        and learned embeddings provides a strong foundation for
        addressing these challenges.

        ------
        Generate Prediction Model Code

        The prediction model will be implemented using the Python
           programming language and several key libraries commonly used
            in machine learning and single-cell data analysis.

        **Implementation Details**
        - **Programming Language:** Python
        - **Key Libraries:**
            - **PyTorch** or **TensorFlow/Keras:** For implementing the
                neural network architecture, including the GNN and other
                layers.
            - **Scanpy** or **AnnData:** For efficient handling and
                preprocessing of the single-cell RNA-seq data.
            - **NumPy:** For numerical computations and array
                manipulations.
            - **SciPy:** For scientific computing, including statistical
                functions.
            - **scikit-learn:** For evaluation metrics (MSE, PCC) and
                potential utility functions.

        **Code Structure and Key Functions**
        The codebase will be organized into several modules or classes
           to ensure modularity and maintainability:

        1. **`DataLoader` Class:**
            - **Purpose:** Responsible for loading the Norman et al.
                (2019) dataset from the specified files (e.g., MTX, TSV).
            - **Functionality:**
                - Reads the gene expression matrix and metadata.
                - Performs normalization and log transformation.
                - Encodes perturbation information.
                - Splits the data into training, validation, and test sets
                    based on perturbations.
                - Provides data loaders for efficient batching during
                    training.
        2. **`Model` Class:**
            - **Purpose:** Defines the neural network architecture,
                including the gene embedding layer, perturbation
                embedding layer, GNN (if chosen), cell state encoding
                layer, and prediction layer.
            - **Functionality:**
                - Initializes the model parameters.
                - Implements the forward pass of the network, taking
                    baseline expression and perturbation information as
                    input and outputing the predicted perturbed
                    expression.
        3. **`Trainer` Class:**
            - **Purpose:** Handles the training process of the model.
            - **Functionality:**
                - Takes the model, data loaders, loss function, and
                    optimizer as input.
                - Implements the training loop, including forward pass,
                    loss calculation, backpropagation, and parameter
                    updates.
                - Monitors performance on the validation set and
                    implements early stopping if needed.
                - Saves the trained model parameters.
        4. **`Evaluator` Class:**
            - **Purpose:** Evaluates the performance of the trained model
                on the test set.
            - **Functionality:**
                - Loads the trained model parameters.
                - Iterates through the test data.
                - Makes predictions for each test sample.
                - Calculates the evaluation metrics (MSE, PCC, Top-k PCC)
                    by comparing predictions with the observed data.
```

```
                 - Reports the evaluation results.
        5. **`Predictor` Class:**
           - **Purpose:** Allows for making predictions on new, unseen
              data (e.g., a baseline expression profile and a
              perturbation).
           - **Functionality:**
              - Loads the trained model parameters.
              - Takes a baseline expression profile and perturbation
                 information as input.
              - Preprocesses the input data as needed.
              - Uses the trained model to generate the predicted gene
                 expression profile after the perturbation.

    **Instructions for Usage**

    1. **Install Libraries:** Install the necessary Python libraries
        using pip:
        ```Bash
        pip install torch torchvision torchaudio scanpy numpy scipy
           scikit-learn
        ```
       (or `tensorflow` and `keras` if using TensorFlow).

    2. **Download Data:** Download the processed data files for
        GSE133344 from GEO or Figshare and place them in a
        designated directory.

    3. **Prepare Input Data:** Ensure that the input data for
        prediction (baseline gene expression profile and
        perturbation identity) is in the expected format (e.g., a
        NumPy array or a Pandas DataFrame).

    4. **Run the Code:** Execute the Python script containing the
        model implementation, training, and evaluation steps. The
        script should:

       - Initialize the `DataLoader` with the path to the data files.

       - Split the data into training, validation, and test sets.
       - Initialize the `Model` with the appropriate architecture
           and hyperparameters.
       - Initialize the `Trainer` with the model, data loaders, loss
           function, and optimizer.
       - Train the model using the `Trainer`.
       - Initialize the `Evaluator` with the trained model and test
           data loader.
       - Evaluate the model and print the results.
       - (Optionally) Use the `Predictor` to make predictions on new
           data.

    **Evaluation and Results**

    A comprehensive evaluation of the developed predictive model is
        crucial to assess its performance and generalizability.

    **Experimental Setup**

    The Norman et al. (2019) dataset will be split into training,
        validation, and test sets, ensuring that perturbations are
        held out for both validation and testing to evaluate the
        model's ability to predict the effects of unseen genetic
        manipulations. The split will likely involve a significant
        portion of the data for training, with smaller portions
        allocated for validation (for hyperparameter tuning and
        model selection) and testing (for final performance
        evaluation).16 The specific hyperparameters of the chosen
        model architecture (e.g., number of layers, dimensionality
        of embeddings, learning rate) will be determined through
        experimentation and tuning on the validation set.

    To make a prediction for a given perturbation, the model will
        take as input the baseline gene expression profile of an
```

unperturbed cell (which could be an average profile of
control cells or a specific control cell's profile) and the
identity of the target gene(s).

**Performance Metrics**

The model's performance on the test set will be quantified using
the three evaluation metrics defined earlier: Mean Squared
Error (MSE), Pearson Correlation Coefficient (PCC), and Top-
k PCC. These metrics will be calculated by comparing the
model's predicted gene expression profiles with the actual
observed profiles in the test set for the held-out
perturbations. The results will be reported separately for
unseen single-gene perturbations and unseen paired-gene
perturbations to assess the model's ability to handle both
types of genetic manipulations. It may also be informative
to report the performance on different subsets of genes,
such as the highly variable genes, as these are often the
most biologically relevant. Visualizations, such as scatter
plots of predicted vs. observed gene expression for
representative perturbations, can provide further insights
into the model's predictive capabilities.

